# RESIST: Resilient Decentralized Learning Using Consensus Gradient Descent

## Abstract

*Empirical risk minimization* (ERM) is a cornerstone of modern *machine learning* (ML), supported by advances in optimization theory that ensure efficient solutions with provable algorithmic convergence rates, which measure the speed at which optimization algorithms approach a solution, and statistical learning rates, which characterize how well the solution generalizes to unseen data. Privacy, memory, computational, and communications constraints increasingly necessitate data collection, processing, and storage across network-connected devices. In many applications, these networks operate in decentralized settings where a central server cannot be assumed, requiring decentralized ML algorithms that are both efficient and resilient. Decentralized learning, however, faces significant challenges, including an increased attack surface for adversarial interference during decentralized learning processes. This paper focuses on the *man-in-the-middle* (MITM) attack, wherein adversaries exploit communication vulnerabilities between devices to inject malicious updates during training, potentially causing models to deviate significantly from their intended ERM solutions. To address this challenge, we propose RESIST (**R**esilient d**E**centralized learning using con**S**ensus grad**I**ent de**S**cen**T**), an optimization algorithm designed to be robust against adversarially compromised communication links, where transmitted information may be arbitrarily altered before being received. RESIST uses a multistep consensus gradient descent framework with robust-statistics-based screening of neighbor messages. It has a design parameter $J$ that controls the frequency of local gradient computation: *each local gradient update is preceded by $J-1$ communication and robust aggregation rounds*. At a fixed budget of algorithmic rounds, where each round is either communication and robust aggregation or a local gradient update, larger $J$ uses fewer local gradient updates, which is useful when local gradient computation is the dominant cost. We establish geometric algorithmic convergence guarantees for strongly convex and Polyak–Łojasiewicz ERM problems, and sublinear and finite-horizon guarantees for smooth nonconvex ERM problems. For heterogeneous local objectives, these guarantees quantify neighborhoods for the relevant iterate, objective-value, and stationarity errors. In the strongly convex homogeneous case, convergence becomes exact. We also establish statistical learning-rate guarantees under common-population independent and identically distributed sampling assumptions, including regimes in which the statistical error vanishes as the sample size grows. Experimental results demonstrate the robustness of RESIST across diverse attack strategies, screening methods, and loss functions. The $J$-ablation experiments further show that larger $J$ can improve convergence as a function of the total number of algorithmic rounds.

**Keywords:** Adversarial machine learning, decentralized gradient descent, distributed algorithms, empirical risk minimization, man-in-the-middle attack, nonconvex optimization, Polyak–Łojasiewicz functions, robust statistics.

## 1 Introduction

Learning a model from training data is foundational to modern *machine learning* (ML) applications. The performance of a learning algorithm is typically evaluated through the *statistical risk*, which measures the

expected loss on unseen data. A common approach to minimize statistical risk is *empirical risk minimization* (ERM) (Vapnik, 1999; Sebastiani, 2002; Kotsiantis et al., 2007; Bengio, 2009; Mohri et al., 2018), where a finite number of training samples are used to approximate the true risk. For convex loss functions, the ERM solution typically converges to the *Bayes optimal solution* as the number of samples grows to infinity (Vapnik, 1999), highlighting the interplay between data availability and model performance. Beyond statistical convergence, the efficiency of optimization algorithms in solving ERM problems—referred to as *algorithmic convergence*—is critical for practical applications. Strong guarantees, such as linear convergence for strongly convex problems and sublinear rates for nonconvex problems, ensure that optimization methods can efficiently approach the desired solution while scaling to the demands of modern ML systems. Together, statistical learning rates (characterizing generalization) and algorithmic convergence rates (quantifying optimization efficiency) define the practical feasibility of learning algorithms.

In many modern ML applications, data is inherently distributed across networked devices due to privacy constraints, bandwidth limitations, or sheer scale, as seen in multi-agent systems, Internet-of-Things (IoT) infrastructures, smart grids, and sensor networks. Traditional distributed learning approaches often assume the presence of a central server to coordinate the training process (Yang et al., 2020), as illustrated in Fig. 1(a). However, this assumption introduces potential single points of failure and also may not be practical in environments such as IoT systems and sensor networks. These limitations motivate *decentralized learning*, where learning is performed collaboratively across devices without centralized coordination (Predd et al., 2006; Boyd et al., 2011; Sayed, 2014; Nedić et al., 2018; Nokleby et al., 2020; Sun et al., 2023), as shown in Fig. 1(b). Decentralized learning systems, however, face unique challenges, including potentially non-independent and identically distributed data, changing network topologies, unreliable communication links, and adversarial attacks, which must be addressed to ensure scalability and resilience in practical settings.

Among the challenges faced by decentralized learning systems, adversarial attacks present a particularly critical problem, as they can significantly degrade both algorithmic convergence and generalization performance. While much of the existing literature on robust decentralized learning under adversarial attacks focuses on the Byzantine attack model (Driscoll et al., 2003; Sousa & Bessani, 2012; Vaidya & Garg, 2013; Su & Vaidya, 2016b; 2015; Yin et al., 2018; Lin et al., 2019; Yang et al., 2019; Kuwaranancharoen et al., 2020; Data & Diggavi, 2021; Peng et al., 2021; Wu et al., 2021; He et al., 2022a; Fang et al., 2022), which assumes some nodes are compromised by malicious actors and deliberately send arbitrary or corrupted values to their neighbors, this paper focuses on a different and less-explored threat: *man-in-the-middle* (MITM) attacks. Unlike Byzantine attacks, where the adversary operates at the node level (Fig. 1(c)), MITM attacks exploit vulnerabilities in communication links, as shown in Fig. 1(d). By compromising these communication links, adversaries can inject arbitrary noise or malicious updates into transmitted information. Such adversarially compromised communication links allow transmitted information to be arbitrarily altered before being received, potentially leading to significant errors in the learning process.

To address this threat, we propose and analyze a decentralized learning algorithm specifically designed to resist MITM attacks. Our work highlights the unique challenges posed by adversarially compromised communication links in decentralized learning systems and also shows how the standard node-level Byzantine attack model can be viewed as a static special case of the time-varying link-level MITM framework considered here (cf. Sec. 7). Our analysis encompasses both algorithmic and statistical perspectives, with a focus on strongly convex, Polyak–Łojasiewicz (Lojasiewicz, 1963), and nonconvex ERM problems.

## 1.1 Relation to prior works

The advent of large-scale ML tasks and the impracticality of consolidating data into a single location have driven significant interest in collaborative learning approaches (Nokleby et al., 2020). A key category in this field is distributed learning, which includes the parameter–server (Li et al., 2014) and federated learning (Konečný et al., 2016) settings, both relying on a central server to facilitate communication among network nodes. Algorithms for distributed and federated learning can be grouped into three main categories: first-order methods, such as distributed gradient descent and its stochastic variants (Blanchard et al., 2017; Chen et al., 2018; Cao & Lai, 2018; Damaskinos et al., 2018; Mhamdi et al., 2018; Xie et al., 2018a;b; Chen et al., 2020; Rajput et al., 2019; Jin et al., 2019; Data et al., 2021; El-Mhamdi et al., 2020a;b; He et al., 2022b), valued for their low computational complexity; augmented Lagrangian-based methods (Zhang &

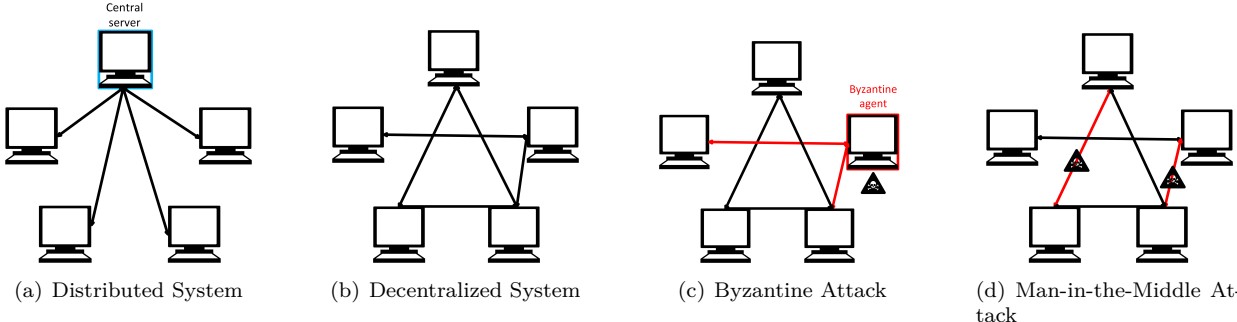

(a) Distributed System     (b) Decentralized System     (c) Byzantine Attack     (d) Man-in-the-Middle Attack

Figure 1: Illustrations of different system architectures and adversarial attack models: (a) A distributed system with centralized coordination, where a central server manages the training process. (b) A decentralized system, where nodes collaborate without central coordination. (c) A decentralized system under a Byzantine attack, where one of the five nodes is compromised (colored red) and sends arbitrary or corrupted values to its neighbors through red-colored links. (d) A decentralized system under a man-in-the-middle (MITM) attack, where two communication links are under attack (colored red), allowing the attacker to alter the transmitted information before it is received, even though no nodes are compromised. The attacked-link set may change over time, reflecting the time-varying link-level attack model studied in this paper. A discussion of how the standard node-level Byzantine attack model can be represented as a static MITM special case is provided in Sec. 7.

Kwok, 2014; Chang et al., 2016; Huang et al., 2020), which require solving local optimization subproblems—incurring higher computational complexity than gradient-based approaches—but can address challenging problems while preserving privacy (Chang et al., 2016; Huang et al., 2020); and second-order methods (Li et al., 2019b; Ghosh et al., 2020; Dinh et al., 2022; Liu et al., 2023), which, despite higher computational and communication costs, achieve second-order optimal convergence guarantees. Reliance on centralized coordination, however, introduces limitations such as single points of failure and system design constraints, prompting the development of decentralized learning systems (cf. Fig. 1(b)). But transitioning algorithmic techniques, along with the derivation of both algorithmic convergence guarantees and statistical learning rates, from distributed to decentralized settings poses unique challenges due to the lack of centralized coordination and fundamental architectural differences.

In decentralized learning, the absence of a central server is addressed by restricting communication to direct neighbors. While the grouping of decentralized algorithms into three main categories mirrors that of distributed learning—first-order methods, such as *decentralized gradient descent* (DGD) and its stochastic variants (Nedic & Ozdaglar, 2009; Ram et al., 2010; Nedić & Olshevsky, 2015; Pu & Nedić, 2021); augmented Lagrangian-based methods (Forero et al., 2010; Mota et al., 2013; Shi et al., 2014; Makhdoumi & Ozdaglar, 2017); and second-order methods (Jadbabaie et al., 2009; Wei et al., 2013; Mokhtari et al., 2016; 2017; Tutunov et al., 2019)—the methods themselves and their analysis differ significantly due to the lack of centralized coordination. Most existing works focus on achieving algorithmic convergence, often under idealized assumptions of trustworthy communication and faultless operations, while overlooking statistical learning rates that are essential for understanding how well solutions generalize to unseen data.

Adapting decentralized learning methods to adversarial environments is a relatively recent focus, with most efforts concentrating on the Byzantine attack model. First introduced in its general form in Dolev et al. (1987), the Byzantine attack refers to compromised nodes that deviate arbitrarily from expected behavior, making detection and defense particularly challenging. The rising prevalence of cybersecurity threats, vulnerabilities in communication channels, and the increasing reliance on ML in mission-critical applications have intensified the demand for robust defenses. Early research focused on detecting Byzantine nodes in distributed settings (Marano et al., 2009; Vempaty et al., 2013; Hashlamoun et al., 2018), followed by approaches leveraging centralized servers for resilient aggregation in the presence of Byzantine attacks (Cao & Lai, 2018; Su & Xu, 2018; Yin et al., 2019; 2018; Alistarh et al., 2018; Li et al., 2019a; Xie et al., 2018c; 2020).

In decentralized systems, initial efforts focused on Byzantine-resilient consensus averaging (LeBlanc et al., 2013; Vaidya et al., 2014), which were later extended to Byzantine-resilient learning for scalar-valued models (Su & Vaidya, 2016a; Sundaram & Gharesifard, 2019). However, these approaches do not directly apply to the vector-valued ML frameworks considered in this paper. While some works have addressed specific vector-valued problems, such as decentralized support vector machines (Yang & Bajwa, 2016) and decentralized estimation (Xu et al., 2018; Mitra et al., 2019; Su & Shahrampour, 2020; Ren et al., 2020; An & Yang, 2021), these solutions are not generalizable to the broader ERM framework.

Similar to the study of the ERM framework for centralized ML, the algorithmic and statistical guarantees of Byzantine-resilient decentralized learning methods for vector-valued models can be broadly categorized by specific loss function classes, typically divided into convex (strongly convex, strictly convex, and convex) and nonconvex (quasi-convex, semi-convex, and smooth nonconvex). The first work to address the vector-valued Byzantine-resilient learning problem with a general convex loss function was Yang & Bajwa (2019), which proposed a decentralized coordinate-descent-based learning algorithm termed ByRDiE. This algorithm demonstrated resilience to Byzantine attacks and convergence to the minimizer of a loss function comprising a convex differentiable term and a strictly convex, smooth regularizer. While Yang & Bajwa (2019) characterized both algorithmic convergence and statistical learning rates for ByRDiE, its focus on convex functions limited its scope. More critically, the coordinate-descent nature of ByRDiE leads to slow and inefficient computation for large-scale models, particularly for high-dimensional data in deep neural networks. Let $d$ denote the number of parameters in the ML model (e.g., the number of weights in a deep neural network). A single iteration of ByRDiE requires $d$ network-wide collaborative steps, with each step involving the computation of a $d$-dimensional gradient at every node, making it computationally expensive. In contrast, BRIDGE, proposed in Fang et al. (2022), requires only one round of updates per iteration for vector-valued models, offering a more efficient and scalable computational framework in decentralized settings. However, BRIDGE assumes loss functions are either strongly convex or locally strongly convex, restricting its applicability to a narrower class of problems.

In contrast to the focus on Byzantine attacks in ByRDiE and BRIDGE, this work addresses the MITM attack (cf. Fig. 1(d)), where adversaries exploit communication vulnerabilities to inject malicious updates during training, causing models to deviate significantly from their intended ERM solutions. The MITM attack model introduces unique challenges, as adversaries may target different communication links at different times. To tackle this, we propose RESIST (**R**esilient d**E**centralized learning using con**S**ensus grad**I**ent de**S**cen**T**). At a high level, RESIST combines one or more rounds of robust filtering of received model vectors with local gradient updates. This avoids the coordinate-wise network-wide gradient loop used by ByRDiE, although the filtering and communication costs still scale with the model dimension. It also connects RESIST to BRIDGE when the filtering stage consists of one *coordinate-wise trimmed mean* (CWTM) round before each local gradient update: in that case, the update structure matches BRIDGE with CWTM filtering, called BRIDGE-T in Fang et al. (2022), apart from the fixed stepsize used in RESIST. This connection is at the level of the update structure, not the full attack model, since BRIDGE studies node-level Byzantine attacks whereas RESIST studies link-level MITM attacks. The main distinctions from ByRDiE and BRIDGE are the link-level MITM attack model, the ability to use multiple filtering rounds between gradient updates, and the convergence analysis for a wider class of loss functions and heterogeneous local objectives. These distinctions require an analysis that goes beyond directly invoking BRIDGE or ByRDiE. Furthermore, within the framework of RESIST, we demonstrate that the Byzantine attack model can be viewed as a static special case of the MITM attack model (cf. Sec. 7), highlighting the broader applicability of the MITM framework in this context.

Given that the Byzantine attack model can be mapped to the MITM attack model within the framework of this paper (as detailed later in Sec. 7), we now discuss recent works beyond Yang & Bajwa (2019) and Fang et al. (2022) that focus on Byzantine-resilient vector-valued decentralized learning. These include Kuwaranancharoen et al. (2020); Peng et al. (2021); Guo et al. (2022); El-Mhamdi et al. (2021); Wu et al. (2023); Ghiasvand et al. (2024); Ghavamipour et al. (2024); Bakshi et al. (2024). Among these, Kuwaranancharoen et al. (2020) addresses only convex loss functions and does not provide algorithmic convergence rates or statistical learning rates. Additionally, the algorithm's robustness diminishes with increasing data dimensions, making it less effective for defending against Byzantine nodes in high-dimensional settings. Similarly, Peng et al. (2021) focuses on convex loss functions in heterogeneous data settings and time-varying

networks but also lacks statistical learning rate guarantees. The MOZI algorithm proposed in Guo et al. (2022) also targets convex loss functions but relies on an aggressive two-step filtering operation that limits the number of Byzantine nodes it can handle. Furthermore, its analysis assumes that faulty nodes send outlier messages relative to regular nodes, a condition often unmet under the Byzantine attack model. For nonconvex loss functions, El-Mhamdi et al. (2021) introduces three methods, including ICwTM, effectively a variant of BRIDGE from Fang et al. (2022). ICwTM incurs higher communication overhead as it requires nodes to exchange both local models and gradients, and assumes identical initialization across the network, which may be impractical in certain applications. Additionally, this work does not examine the impact of network topology on learning performance. The work Wu et al. (2023) proposes a stochastic gradient descent-based algorithm for nonconvex loss functions with heterogeneous data but does not extend to the MITM attack model and provides only bounds on the average gradient norm rather than guarantees on iterate values. Another approach, Ghiasvand et al. (2024), utilizes gradient tracking to manage heterogeneous data and improve communication efficiency but assumes attackers apply uniform perturbations, limiting its applicability to generalized Byzantine or MITM attack scenarios. Finally, Ghavamipour et al. (2024) and Bakshi et al. (2024) develop algorithms for privacy-preserving and validated decentralized learning under Byzantine attacks, respectively, but rely on secure private key or secret-sharing mechanisms among honest nodes, making them unsuitable for scenarios lacking secure communication links.

Next, we focus on the distinction between our work on the MITM attack model and related work in the Byzantine-resilient literature that aligns with our goal of deriving linear (geometric) convergence rates for strongly convex losses. The closest such work is Kuwaranancharoen & Sundaram (2025), which introduces REDGRAF, a framework for Byzantine-resilient decentralized optimization that generalizes BRIDGE and includes several filtering rules, such as CWTM and other vector- or distance-based variants. The REDGRAF analysis proves geometric convergence to a characterized neighborhood for strongly convex losses. Its CWTM instance is the relevant comparator to the version of RESIST that uses one CWTM filtering round before each local gradient update: under the same high-level update structure, both correspond to the fixed-stepsize BRIDGE-T case. Relative to REDGRAF-CWTM, this paper considers a link-level MITM model, allows multiple CWTM filtering rounds between local gradient updates, and analyzes strongly convex, Polyak–Łojasiewicz (PŁ), and smooth nonconvex losses. Under their respective theorem assumptions, both REDGRAF-CWTM and RESIST yield geometric convergence to neighborhoods in heterogeneous deterministic settings. For REDGRAF-CWTM, the radius is controlled by the spread of the regular nodes' local minimizers, and this radius can vanish when that spread is zero. This zero-radius specialization, however, is conditional on the regime where the REDGRAF-CWTM theorem applies; for high-dimensional models, that regime effectively requires the relevant condition number to be close to one, which can be highly restrictive. In RESIST, the corresponding residual terms are expressed through MITM-induced inexact averaging and heterogeneity quantities. These deterministic bounds are then combined with uniform concentration under common-population independent and identically distributed (i.i.d.) sampling to obtain finite-sample statistical learning-rate guarantees and, in the strongly convex large-sample regime, convergence in probability to the statistical-risk minimizer. REDGRAF also analyzes other variants, such as *Simultaneous Distance-MinMax Filtering Dynamics* (SDMMFD). SDMMFD employs three distinct filtering mechanisms per iteration, resulting in three times the redundancy requirements compared to the CWTM-based update. Here, redundancy refers to the minimum neighborhood size required at each node to tolerate a given number of attacks. Consequently, for a fixed network topology, the SDMMFD variant can defend against only one-third of the number of attacks that RESIST can handle in a given network. In addition, in the REDGRAF analysis, CWTM has $\mathcal{O}(r^\star)$ convergence-radius dependence, where $r^\star$ is the enclosing radius for the regular nodes' local minimizers, while SDMMFD has $\mathcal{O}(\sqrt{d}r^\star)$ dependence. For these reasons, when comparing RESIST numerically with the BRIDGE and REDGRAF line of work in Sec. 9 under the aligned update structure with one CWTM filtering round before each local gradient update, we use REDGRAF-CWTM, equivalently BRIDGE with fixed stepsize, as the comparison point in the experiments.

A summary of how our work relates to prior works is provided in Table 1. This table compares RESIST with various vector-valued decentralized learning and optimization methods in the literature across key dimensions: the attack model, whether an algorithmic convergence rate is provided, whether a statistical learning rate is provided, and whether the analysis includes nonconvex loss functions.

| Algorithm | Attack Model | ACR | SCR | Nonconvex |
|---|---|:---:|:---:|:---:|
| DGD (Nedić & Olshevsky, 2015) | None | √ | × | × |
| NEXT (Lorenzo & Scutari, 2016) | None | × | × | √ |
| Nonconvex DGD (Zeng & Yin, 2018) | None | √ | × | √ |
| D-GET (Sun et al., 2020) | None | √ | √ | √ |
| GT-SARAH (Xin et al., 2022) | None | √ | √ | √ |
| MOZI (Guo et al., 2022) | Non-Byzantine | √ | × | × |
| Dec-FedTrack (Ghiasvand et al., 2024) | Non-Byzantine | √ | × | √ |
| ByRDiE (Yang & Bajwa, 2019) | Byzantine | √ | √ | × |
| Kuwaranancharoen et. al (Kuwaranancharoen et al., 2020) | Byzantine | × | × | × |
| ICwTM (El-Mhamdi et al., 2021) | Byzantine | √ | × | √ |
| DRSA (Peng et al., 2021) | Byzantine | √ | × | × |
| BRIDGE (Fang et al., 2022) | Byzantine | √ | √ | △ |
| BASIL (Elkordy et al., 2022) | Byzantine | √ | × | × |
| IOS (Wu et al., 2023) | Byzantine | √ | × | √ |
| RedGraf (Kuwaranancharoen & Sundaram, 2025) | Byzantine | √ | × | √ |
| SecureDL (Ghavamipour et al., 2024) | Byzantine | √ | × | × |
| VALID (Bakshi et al., 2024) | Byzantine | √ | × | × |
| **RESIST (This work)** | MITM, Byzantine | √ | √ | √ |

*ACR:* Refers to Algorithmic Convergence Rate.
*SCR:* Refers to Statistical Convergence Rate.
*Non-Byzantine:* Refers to works with assumptions on attack behavior that limit generalizability to Byzantine attacks.
△: Refers to global nonconvex functions with local strong convexity around stationary points.

Table 1: Comparison of RESIST with various vector-valued decentralized learning and optimization methods in the literature.

## 1.2 Our contributions

The primary contribution of this work is the development and analysis of RESIST, a decentralized first-order method robust to MITM attacks in the network, with a comprehensive analysis addressing both algorithmic convergence and statistical learning rates across different classes of convex and nonconvex loss functions. The MITM attack model has been extensively studied in the communications literature, with Conti et al. (2016) providing a detailed survey of scenarios where MITM attacks occur in communication networks and potential defenses against them. However, to the best of our knowledge, the MITM attack model has not been studied in decentralized learning settings, though it has been investigated in distributed learning frameworks, as in Chiang et al. (2009); Nadendla et al. (2014); Zhang et al. (2018). Notably, Nadendla et al. (2014) considers the MITM attack as a subset of the Byzantine attack, but this is based on the assumption of a *static* attack model where the attacker cannot switch between links. In contrast, the MITM attack model considered in this work, detailed in Sec. 2, allows the compromised communication links to vary over time, subject to the per-round attack budget and filtered-graph conditions stated later. This captures attacks whose affected links change over time rather than remaining fixed throughout the attack (see also our discussion relating the MITM and Byzantine attack models in Sec. 7). In this sense, our work is the first to study this time-varying link-level MITM attack model in the context of decentralized learning.

Within this framing, RESIST makes several key contributions to address the challenges posed by time-varying link-level MITM attacks in decentralized learning systems. Specifically, compared with the original BRIDGE analysis, RESIST gives geometric residual-neighborhood guarantees in strongly convex regimes under the assumptions of the corresponding theorems. Algorithmically, RESIST extends the fixed-stepsize CWTM update corresponding to BRIDGE-T and RedGraf-CWTM under the aligned conventions discussed earlier by using multiple robust consensus steps before each local gradient update. For a budget of $T$ algorithmic rounds, where a round is either one communication round with CWTM filtering or one local gradient update, RESIST performs about $T/J$ local gradient updates and $T(J-1)/J$ communication/filtering steps. This design is most useful when local gradient computation is the dominant cost. A key similarity between BRIDGE and RESIST is the use of robust-statistics-based screening rules for filtering potentially malicious information. However, while BRIDGE's analysis relies on results concerning

the product of stochastic mixing matrices from Vaidya (2012) over "filtered" graphs corresponding to the screening of Byzantine attacks, the time-varying link-level MITM attack model in this work, combined with multiple consensus steps, necessitates the derivation of new variants of the results in Vaidya (2012). These results, which are crucial for establishing consensus guarantees for RESIST, are provided in Appendix A.

In terms of our results purely from the perspective of convergence rates in decentralized optimization under malicious attacks in the MITM model, this work makes three significant contributions. First, in the strongly convex setting, we establish the geometric convergence rate of the iterate and consensus error to a residual neighborhood (Theorem 5.5). The radius of this neighborhood is quantified by factors such as the inexact averaging operation, the algorithm's stepsize, heterogeneity across local objective functions, and the coordinate-wise trimmed mean screening method—a filtering approach widely employed in robust distributed (Yin et al., 2018) and decentralized frameworks (Su & Vaidya, 2016a; Sundaram & Gharesifard, 2019; Yang & Bajwa, 2019; Fang et al., 2022). When the local objectives are homogeneous, the residual terms in this bound vanish and the same theorem gives exact geometric convergence. For heterogeneous fixed empirical objectives, the finite-network algorithmic result should therefore be read as a residual-neighborhood guarantee.

Second, for loss functions satisfying the Polyak–Łojasiewicz (PŁ) property (Lojasiewicz, 1963), we establish geometric convergence rates of the consensus and function value to a residual neighborhood around the minimal function value (Theorem 6.4). The radius of this neighborhood is similarly influenced by the inexact averaging operation, the algorithm's stepsize, heterogeneity across local objective functions, and the screening method. To the best of our knowledge, this is the first work to analyze the PŁ function class in the context of MITM attacks over decentralized optimization networks.

Finally, for smooth nonconvex functions (Sec. 6.2), using a diminishing stepsize, we derive sublinear convergence rates for the stationarity and consensus errors up to residual terms (Theorem 6.6). This result matches the best-known convergence rates for centralized stochastic gradient descent methods (Imaizumi & Iiduka, 2024) under the same stepsize schedule. Importantly, under the common-population i.i.d. ERM setting analyzed in Sec. 8, the corresponding residual terms vanish as the number of data samples grows. Additionally, we provide a finite-horizon guarantee for the nonconvex setting with a constant stepsize (Theorem 6.7), extending prior work (Wu et al., 2023) to accommodate the time-varying link-level MITM attack model.

In terms of statistical learning rates for decentralized learning systems, our contributions in Sec. 8 include the derivation of sample complexity guarantees for the decentralized ERM problem under MITM attacks, covering strongly convex, PŁ, and general smooth nonconvex loss functions (Theorems 8.2, 8.3, and 8.4, respectively). These guarantees establish that, under the common-population i.i.d. sampling and compactness assumptions stated in Sec. 8, RESIST solves the ERM problem with a statistical learning rate that matches the rate derived for BRIDGE (Fang et al., 2022), while extending the results to both the PŁ and general smooth nonconvex function classes. Notably, as in the BRIDGE framework, our results demonstrate a speed-up in the learning rate due to collaboration, despite the presence of attacks within the network. This speed-up, given $M$ nodes and $N$ samples per node, is guaranteed to lie between the local statistical learning rate of $\mathcal{O}(1/\sqrt{N})$ and the ideal decentralized learning rate without any attacks of $\mathcal{O}(1/\sqrt{MN})$. To the best of our knowledge, this is the first work to provide such statistical learning rate guarantees for the decentralized ERM problem under adversarial attacks for PŁ and general smooth nonconvex functions.

Last but not least, the numerical experiments in Sec. 9 validate the theoretical findings using real-world datasets, specifically MNIST (LeCun et al., 1998) and CIFAR-10 (Krizhevsky & Hinton, 2009). For the MNIST dataset, the experiments demonstrate RESIST's effectiveness on strongly convex loss functions across various system and algorithm parameters, as shown in Sec. 9.1, achieving comparable accuracy to other algorithms under diverse settings. For the CIFAR-10 dataset, the experiments in Sec. 9.2 highlight RESIST's strong performance on nonconvex objective functions and its robustness across different system parameters, algorithmic design choices, and attack strategies. The experiments also include $J$-ablation studies and non-i.i.d. data-partition tests, showing that larger $J$ can improve convergence with respect to the total number of algorithmic rounds and that RESIST remains robust under heterogeneous data partitions.

### 1.3 Notation

We use the following notation in the paper. The symbol $\mathbb{R}_+$ denotes the set of non-negative real numbers, $\varnothing$ represents the empty set, and $\mathrm{diam}(\cdot)$ and $|\cdot|$ denote the diameter and cardinality of a set, respectively. The probability measure is written as $\mathbb{P}$, expectation as $\mathbb{E}$, and a.s. signifies "almost surely." The space $L^\infty(\Omega)$ refers to functions on the domain $\Omega$ with bounded essential supremum, and $\|\cdot\|_{L^\infty(\Omega)}$ denotes the $L$-infinity norm over $\Omega$. Graphs are represented as $\mathcal{G}(\mathcal{N}, \mathcal{E})$, where $\mathcal{N}$ is the set of nodes and $\mathcal{E}$ the set of edges. For two nodes $u$ and $v$, the edge $uv$ is considered an incoming edge to node $v$ from its neighbor $u$.

Scalars are denoted by regular-faced letters (e.g., $a, A$), vectors by bold-faced lowercase letters (e.g., $\mathbf{a}$), and matrices by bold-faced uppercase letters (e.g., $\mathbf{A}$). All vectors are column vectors. The identity matrix is $\mathbf{I}$, the vector of all ones is $\mathbf{1}$, and $(\cdot)^T$ denotes the transpose. For a vector $\mathbf{a}$, $[\mathbf{a}]_k$ denotes its $k$-th element. For a matrix $\mathbf{A}$, $[\mathbf{A}]_i$ refers to the $i$-th column, $[\mathbf{A}]_{ij}$ refers to the element in the $i$-th row and $j$-th column, and $[\mathbf{A}]_{[a:b]\times[c:d]}$ represents the sub-block spanning rows $a$ to $b$ and columns $c$ to $d$. Inner products between vectors $\mathbf{a}_1$ and $\mathbf{a}_2$ are written as $\langle \mathbf{a}_1, \mathbf{a}_2 \rangle$. The $\ell_2$-norm of a vector $\mathbf{a}$ is denoted by $\|\mathbf{a}\|$, while $\|\mathbf{A}\|$, $\|\mathbf{A}\|_F$, and $\|\mathbf{A}\|_\infty$ represent the operator, Frobenius, and infinity norms of a matrix $\mathbf{A}$, respectively.

For matrices $\mathbf{A}$ and $\mathbf{B}$ of identical size, $\mathbf{A} \leqslant \gamma \mathbf{B}$ (for scalar $\gamma$) implies entry-wise inequality: $[\mathbf{A}]_{ij} \leqslant \gamma [\mathbf{B}]_{ij}$ for all $i, j$. The notation $\mathbf{A} \succeq \mathbf{B}$ indicates that $\mathbf{A} - \mathbf{B}$ is positive semidefinite. Scalar comparisons may also depend on a matrix norm; $f \lesssim_\mathbf{M} g$ implies $f \leqslant C(\mathbf{M})g$, where $C(\mathbf{M})$ is a constant related to the matrix norm $\|\cdot\|_\mathbf{M}$. Similarly, $\mathbf{P}(h, J) = \Theta(h)$ means $\|\mathbf{P}(h, J)\|_F$ is bounded by a constant times $h$. The notation $a_k = \mathbf{o}(b)$ implies that for any $\epsilon > 0$, there exists $k_0$ such that $|a_k| \leqslant \epsilon b$ for all $k \geqslant k_0$.

Finally, $\nabla$ denotes the gradient of a function, and $\nabla_k$ is the partial derivative with respect to the $k$-th coordinate. For continuously differentiable functions $f$, the gradient Lipschitz constant $\mathrm{LIP}(f)$ is defined as $\mathrm{LIP}(f) = \sup_{\mathbf{x},\mathbf{y};\ \mathbf{x}\neq\mathbf{y}} \frac{\|\nabla f(\mathbf{x}) - \nabla f(\mathbf{y})\|}{\|\mathbf{x}-\mathbf{y}\|}$.

### 1.4 Organization

The rest of the paper is organized as follows. In Sec. 2, we formalize the risk minimization problem, describe the system model, present the decentralized ERM formulation, and define the MITM attack model. Sec. 3 introduces the RESIST algorithm, including the role of the parameter $J$ in scheduling communication/CWTM rounds before local gradient updates, states the graph connectivity assumptions required for analysis, and develops preliminary consensus results under the MITM attack model with coordinate-wise trimmed mean screening. Sec. 4 establishes additional consensus guarantees for RESIST that are used in the subsequent convergence analysis. In Sec. 5, we present algorithmic convergence guarantees for strongly convex loss functions under a two-time-scale framework, with one scale corresponding to algorithmic iterations (time-scale $s$) and the other to the total number of algorithmic rounds, where each round is either an inter-neighbor communication/CWTM round or a local gradient-update round, in the synchronous, slotted setting (time-scale $t$). Sec. 6 extends the algorithmic convergence analysis to PŁ and smooth nonconvex loss functions. Sec. 7 shows how the classical node-level Byzantine attack model can be represented as a static special case of the MITM attack model within our analytical framework. Sec. 8 establishes statistical learning rates for strongly convex, PŁ, and smooth nonconvex loss functions under the stated sampling and compactness assumptions. Numerical results on real-world datasets are presented in Sec. 9 to demonstrate the effectiveness of RESIST. Finally, Sec. 10 concludes the paper, with all proofs and supplementary discussions provided in Appendices A–G.

## 2 Problem Formulation

### 2.1 Background: Statistical and empirical risk minimization

Let $\ell : (\mathbf{w}, \mathbf{z}) \mapsto \ell(\mathbf{w}, \mathbf{z})$ be a non-negative-valued (and possibly regularized) differentiable *loss function* that maps a *model* $\mathbf{w}$ and a *data sample* $\mathbf{z}$ to the corresponding loss $\ell(\mathbf{w}, \mathbf{z})$. Without loss of much generality, we assume the model $\mathbf{w}$ to be parametric, i.e., $\mathbf{w} \in \mathbb{R}^d$, where $d$ denotes the dimensionality of the model $\mathbf{w}$, such as the number of parameters in a deep neural network. The data sample $\mathbf{z}$, on the other hand, is treated as a random variable defined on a probability space $(\Omega, \mathcal{F}, \mathbb{P})$, i.e., $\mathbf{z}$ is $\mathcal{F}$-measurable and drawn from

the sample space $\Omega$ according to the probability law $\mathbb{P}$. The main objective in *machine learning* (ML) is to obtain an optimal model $\mathbf{w}_{\text{SR}}^*$ that minimizes the expected loss, known as the *statistical risk* (Mohri et al., 2018; Golden, 2020):

$$\mathbf{w}_{\text{SR}}^* \in \arg\min_{\mathbf{w} \in \mathbb{R}^d} \mathbb{E}_{\mathbb{P}}[\ell(\mathbf{w}, \mathbf{z})]. \tag{1}$$

A model $\mathbf{w}_{\text{SR}}^*$ satisfying (1) is termed a *statistical risk minimizer* (also referred to as a *Bayes optimal model*). However, in most ML applications, the full distribution of $\mathbf{z}$ is rarely known, making the direct computation of $\mathbb{E}_{\mathbb{P}}[\ell(\mathbf{w}, \mathbf{z})]$ infeasible. Instead, a finite collection $\mathcal{Z} := \{\mathbf{z}_n\}_{n=1}^N$ of data samples is typically drawn according to $\mathbb{P}$, and an empirical approximation of (1) is solved:

$$\mathbf{w}_{\text{ERM}}^* \in \arg\min_{\mathbf{w} \in \mathbb{R}^d} \left( \frac{1}{N} \sum_{n=1}^N \ell(\mathbf{w}, \mathbf{z}_n) =: f(\mathbf{w}) \right). \tag{2}$$

This formulation, referred to as *empirical risk minimization* (ERM), is widely used to approximate $\mathbf{w}_{\text{SR}}^*$ when the data distribution is unavailable. Two primary goals of numerically solving the ERM problem (2) in centralized settings are: ($i$) ensuring that the iterative algorithms used for optimization achieve fast algorithmic convergence to a stationary point (e.g., $\mathbf{w}_{\text{ERM}}^*$) of the average empirical loss $\frac{1}{N} \sum_{n=1}^N \ell(\cdot, \mathbf{z}_n)$, and ($ii$) ensuring that the obtained stationary point $\mathbf{w}_{\text{ERM}}^*$ exhibits fast statistical convergence (i.e., lower sample complexity) to the statistical risk minimizer $\mathbf{w}_{\text{SR}}^*$.

In this paper, unlike several prior works (cf. Table 1), we focus on deriving both the algorithmic convergence rate and the statistical learning rate of the ERM solution in scenarios where data samples are not available in a centralized location, necessitating decentralized collaboration. The results are specific to the decentralized setting under malicious attacks and rely on several assumptions about the loss function $\ell(\mathbf{w}, \mathbf{z})$, including its classification into function classes such as convex, PŁ, and smooth nonconvex, which will be formally characterized in subsequent sections. We now describe our framework for decentralized learning.

## 2.2 System model for decentralized learning

Consider a network of $M$ nodes—representing agents, smartphones, computers, etc.—modeled as a directed, static, and connected graph $\mathcal{G}(\mathcal{N}, \mathcal{E})$, where $\mathcal{N} := \{1, \ldots, M\}$ is the set of nodes, and $\mathcal{E}$ represents the communication links or edges between them. A directed edge $(i, j) \in \mathcal{E}$ indicates that node $j$ can directly receive messages from node $i$, and vice versa for $(j, i)$. The neighborhood set of node $j$, denoted $\mathcal{N}_j$, includes all nodes with a direct link to $j$: $\mathcal{N}_j := \{i \in \mathcal{N} : (i, j) \in \mathcal{E}\}$. Each node $j$ has access only to its local training dataset, $\mathcal{Z}_j := \{\mathbf{z}_{jn}\}_{n=1}^{|\mathcal{Z}_j|}$, as the complete dataset $\mathcal{Z} = \bigcup_{j=1}^M \mathcal{Z}_j$ is never available at a single location. Without loss of generality, we assume that all nodes have the same number of data samples, i.e., $|\mathcal{Z}_j| = N$ for all $j \in \mathcal{N}$, resulting in a total of $NM$ samples across the network.

To estimate the statistical risk minimizer $\mathbf{w}_{\text{SR}}^*$ (cf. (1)) in the decentralized setting, the following ERM problem ideally needs to be solved:

$$\min_{\mathbf{w} \in \mathbb{R}^d} \frac{1}{MN} \sum_{j=1}^M \sum_{n=1}^N \ell(\mathbf{w}, \mathbf{z}_{jn}) = \min_{\mathbf{w} \in \mathbb{R}^d} \frac{1}{M} \sum_{j=1}^M f_j(\mathbf{w}), \tag{3}$$

where $f_j(\mathbf{w}) := \frac{1}{N} \sum_{n=1}^N \ell(\mathbf{w}, \mathbf{z}_{jn})$ represents the *local* empirical risk associated with the data samples $\{\mathbf{z}_{jn}\}_{n=1}^N$ in the local dataset at the $j$-th node. For the algorithmic convergence analysis in Secs. 5–6, the local datasets are assumed fixed before an algorithm is run, so the functions $f_j$ are treated as deterministic local objectives and may differ across nodes. Thus, heterogeneous or non-i.i.d. local datasets do not invalidate the deterministic convergence results of this paper; rather, they enter through the error terms in the stated results: larger discrepancies among the local objectives can lead to larger upper bounds on the distance to the relevant optimality target. The statistical learning-rate results in Sec. 8, on the other hand, use an additional sampling assumption: all samples across the network are drawn independently from the same population law $\mathbb{P}$, so the empirical gradient averages at each node are centered at the gradient of the

same population risk. If different nodes instead sample from different population laws, each node has its own population risk. A statistical guarantee would then have to specify which aggregate population objective is being studied, for example, the unweighted average of the node-level population risks, a sample-size-weighted average, or an average weighted by the mixing weights induced by the decentralized updates. We therefore leave such non-i.i.d. statistical learning-rate guarantees to future work.

In the statistical learning literature, under mild assumptions on the data distribution, it is well established that the minimizer of (3) converges to $\mathbf{w}_{\text{SR}}^*$ with high probability at a rate of $\mathcal{O}(1/\sqrt{MN})$ for strictly convex loss functions (Vapnik, 1999), provided the data is centralized at a single location. However, due to the decentralized nature of the dataset, the results in Vapnik (1999) cannot be directly applied in the decentralized setting. Instead, we assume that each node $j$ learns and updates a local version of the desired global model, denoted by $\mathbf{w}_j$, based on its local dataset $\mathcal{Z}_j$, and collaborates with other nodes in the network to solve the following *decentralized* ERM problem:

$$\min_{\{\mathbf{w}_1,\ldots,\mathbf{w}_M\}} \frac{1}{M} \sum_{j=1}^{M} f_j(\mathbf{w}_j) \quad \text{subject to} \quad \forall i \in \mathcal{N}, j \in \mathcal{N}, \ \mathbf{w}_i = \mathbf{w}_j. \tag{4}$$

Traditional first-order decentralized learning algorithms iteratively solve (4) to learn the desired global model (Predd et al., 2006; Forero et al., 2010; Boyd et al., 2011; Duchi et al., 2012; Sayed, 2014; Nedić et al., 2018; Sun et al., 2023). In each iteration, these algorithms typically require each node $j$ to perform two key tasks: ($i$) refine the local model $\mathbf{w}_j$ by performing a consensus update with its neighboring nodes through inter-neighbor communication; and ($ii$) update the local model using a local learning rate and gradient information, followed by broadcasting the updated information to its outgoing neighbors. This iterative process continues until certain convergence criteria are met, depending on the specific objectives of the algorithm. While this paper adopts the same general framework for decentralized learning, our focus is on scenarios where malicious actors may compromise the system. The attack model considered in this work is described next.

### 2.3 Man-in-the-middle attack model

In a decentralized system, malicious actors can compromise the system in two primary ways: by targeting nodes or by attacking the communication links between nodes. Node-level attacks, where an adversary overtakes a node and causes it to deviate arbitrarily from the agreed-upon algorithmic protocol without detection, are commonly referred to as the Byzantine attack model and have been extensively studied in the decentralized learning literature (e.g., see Fang et al. (2022) and references therein). In contrast, significantly less is known about attacks focused on network edges, or communication links. One such attack is the *man-in-the-middle* (MITM) attack. While the MITM attack model has a well-established history (cf. Sec. 1), this paper studies a link-level MITM model in which the compromised-link set is allowed to vary over time. At each time slot, the adversary may alter messages transmitted over a bounded number of incoming communication links at any node, and the compromised-link set may change from one time slot to the next. Fixed attacked-link patterns are included as special cases, but the model also captures attacks in which different links are compromised at different times. We do not require the attack to remain on a link for any minimum duration, nor do we bound how often the attacked set changes. The relevant conditions for the analysis are the per-time incoming-link attack budget introduced below and the filtered-graph connectivity condition in Assumption 3.3. As discussed in Sec. 7, the decentralized Byzantine model considered in prior works can be embedded as a static special case of this link-level framework.

Mathematically, we assume a synchronous, slotted model for the decentralized system, where each action (e.g., communication or computation) is executed within a predefined time slot, indexed by the iteration $t$ (referred to as time-scale $t$). Let $\mathcal{E}_b(t) \subset \mathcal{E}$ denote the set of edges compromised by malicious actors at a given iteration $t$, and let $\mathcal{B}(t) \subset \mathcal{N}$ represent the set of source nodes associated with these compromised edges—nodes that transmit information along edges targeted by the attack at time $t$. For a node $j$, define $\mathcal{N}_j^r(t)$ as the set of neighboring nodes with uncompromised outgoing edges to $j$. The set of neighbors whose information has been compromised during transmission to node $j$ can then be defined as $\mathcal{N}_j^b(t) := \mathcal{N}_j \backslash \mathcal{N}_j^r(t)$, where $\mathcal{N}_j$ is the set of all neighboring nodes of $j$. Note that $\mathcal{B}(t)$, the set of source nodes corresponding to compromised edges at time $t$, can be expressed as $\mathcal{B}(t) := \bigcup_{j \in \mathcal{N}} \mathcal{N}_j^b(t)$. The maximum

number of compromised edges incoming to any node in the network at any time instance is defined as $b :=$ $\sup_{0 \leqslant t < \infty} \sup_j |\mathcal{N}_j^b(t)|$, representing a parameter that quantifies the adversary's strength within the system.

**Example 2.1.** As an example of the MITM attack model above, consider the case of $b = 1$. For a representative node $j$, at time instance $t_1$, MITM attacks occur on its incoming edges, with the compromised source set being $\mathcal{N}_j^b(t_1) = \{u\}$, where node $u$ is a direct neighbor of $j$. The transmitted information from node $u$ to node $j$ may be altered to an arbitrary value, expressed as $m'_{uj}(t_1) = m_{uj}(t_1) + \zeta_{uj}(t_1)$, where $\zeta_{uj}(t_1)$ can be any value, either dependent or independent of $m_{uj}(t_1)$ (the original data transmitted from node $u$ to node $j$). At another time instance $t_2$, the attack may shift from edge $uj$ to edge $vj$, resulting in the compromised source set $\mathcal{N}_j^b(t_2) = \{v\}$. The transmitted information from node $v$ to node $j$ can then be altered as $m'_{vj}(t_2) = m_{vj}(t_2) + \zeta_{vj}(t_2)$, where $\zeta_{vj}(t_2)$ can again be any value, either dependent or independent of $m_{vj}(t_2)$ (the original data transmitted from node $v$ to node $j$). This example applies to every node $j$ in the network, with $j$ being used here as a representative example.

Note that the above example allows the compromised incoming link to change at every communication time slot while the per-time budget remains $b = 1$. Thus, the model includes the static case, where the same link is compromised throughout, and also allows moving-link sequences. We do not claim that moving-link attacks are necessarily more damaging than the worst admissible static attack; the point is that the analysis does not require the compromised-link set to be fixed. A static model that instead treats the union of all links ever attacked as permanently compromised can be overly conservative, since it treats attacks occurring at different times as if they occurred simultaneously.

## 2.4 Problem statement

MITM attacks present unique challenges for solving the decentralized ERM problem stated in (4). Such attacks can strategically alter messages transmitted over compromised edges, causing the learned models to deviate significantly from the desired solution. For instance, DGD (Zeng & Yin, 2018), which lacks mechanisms to screen or filter out compromised information, is particularly vulnerable to accumulating falsified data during consensus-based updates. This accumulation ultimately prevents convergence to the solution of (4). To address these challenges, robust statistics-based data aggregation methods, such as trimmed mean or median, are often employed in Byzantine-resilient decentralized learning frameworks to filter out potentially falsified information (Fang et al., 2022). These tools remain useful in the MITM setting, but the source of corruption differs from the node-level Byzantine model. A Byzantine node may choose arbitrary outgoing messages to its neighbors, whereas in the MITM model the transmitting nodes remain regular and the adversary alters selected messages while they traverse communication links. Since the compromised links can change across time subject to the per-time incoming-link attack budget, even with robust data aggregation, targeted attacks can significantly delay information mixing within the network. In extreme cases, without adequate assumptions on network connectivity, adversaries could compromise edges in a way that permanently isolates some nodes, preventing effective information exchange.

Similar to challenges faced in Byzantine-resilient decentralized learning (Fang et al., 2022), when the local empirical objectives are heterogeneous, achieving an exact solution to the decentralized ERM problem under MITM attacks is generally infeasible. Instead, a natural goal from an optimization perspective is to approximate the solution to (4) with controlled optimization error. This limitation arises because traditional consensus-based methods rely on doubly stochastic mixing matrices, which ensure exact averaging across the network by combining both incoming and outgoing information during the collaboration (i.e., consensus) phase. However, under MITM attacks, compromised edges and the necessary screening mechanisms disrupt proper information exchange, resulting in non-doubly stochastic mixing matrices. This deviation prevents exact averaging and, consequently, hinders convergence to the exact ERM solution, even when employing recent methods like push-pull approaches (Xin & Khan, 2018; Pu et al., 2021).

In this context, our primary goal is to develop an algorithm for the decentralized ERM problem in the presence of MITM attacks, and to establish two key guarantees from an optimization perspective, even when the local empirical risk functions $f_j$ are heterogeneous. First, we aim to establish approximate consensus guarantees, quantifying the extent to which the local models $\mathbf{w}_j$ agree with one another as a function of the number of algorithmic iterations (time-scale $s$). This addresses the consensus constraint $\forall i \in \mathcal{N}, j \in \mathcal{N}, \; \mathbf{w}_i = \mathbf{w}_j$

in (4). Second, we seek to derive convergence rates for approximate solutions to (4), where approximation refers to explicit consensus and optimization-error bounds, ensuring efficient convergence for various classes of local empirical risk functions $f_j$. These rates are analyzed as functions of both the time-scale $s$ (algorithmic iterations) and the time-scale $t$ (the total number of discrete actions in the system, including communications and updates), making the results broadly applicable from an optimization perspective.

Moreover, even when heterogeneous local objectives lead to an approximate solution of (4), our secondary goal is to demonstrate that the proposed algorithm can still generalize well to unseen data by reliably estimating the statistical risk minimizer. To connect this optimization guarantee to statistical learning, we later show that the proposed algorithm implicitly solves a weighted version of the decentralized ERM problem, formulated as:

$$\min_{\{\mathbf{w}_1,\ldots,\mathbf{w}_M\}} \sum_{j=1}^{M} c_j f_j(\mathbf{w}_j) \quad \text{subject to} \quad \forall i \in \mathcal{N}, j \in \mathcal{N}, \ \mathbf{w}_i = \mathbf{w}_j, \tag{5}$$

where $c_j \in [0,1]$ and $\sum_j c_j = 1$. Importantly, under the common-population i.i.d. sampling model used in the statistical analysis, the expected value of this weighted decentralized ERM problem aligns with that of the statistical risk minimization problem. Consequently, under the same common-population i.i.d. model, from a statistical learning theory perspective, we aim to establish the statistical learning rates at which the empirical solution obtained by the proposed algorithm approaches the statistical risk minimizer defined in (1).

## 3 RESIST: Resilient Decentralized Learning Using Consensus Gradient Descent

In this section, we formally introduce the proposed algorithm, RESIST (Algorithm 1), designed to enable efficient decentralized learning while remaining resilient to MITM attacks under the time-varying link-level model described in the previous section. To facilitate the subsequent analysis of the algorithmic convergence rates and statistical learning rates of RESIST, we describe the update schedule and the role of the parameter $J$, which controls how often local gradient updates are taken, and also present the main assumptions on the connectivity of the decentralized network in Sec. 3.1. Additionally, we establish preliminary results in Secs. 3.2 and 3.3, characterizing the resilience of RESIST in terms of consensus behavior under MITM attacks.

---

**Algorithm 1** RESIST (**R**esilient d**E**centralized learning using con**S**ensus grad**I**ent de**S**cen**T**)

---

**Input:** Local empirical loss functions $f_j$ for all $j \in \mathcal{N}$, maximum number of compromised edges across all iterations and neighborhoods $b$, parameter $J > 1$ controlling the frequency of gradient-based local model updates, stepsize $h$, and maximum number of iterations $T_{\max}$
1: **Initialize:** Set $s \leftarrow 0$ and initialize $\mathbf{w}_j(0)$ for all $j \in \mathcal{N}$
2: **for** $t = 0, 1, \ldots, T_{\max} - 1$ **do**
3:     **if** $(t+1) \mod J \neq 0$ **then**
4:         Broadcast $\mathbf{w}_j(t)$ for all $j \in \mathcal{N}$
5:         Receive $\mathbf{w}_i(t)$ at each node $j \in \mathcal{N}$ from all $i \in \mathcal{N}_j$
6:         $\mathbf{w}_j(t+1) \leftarrow \mathsf{CWTM}(\{\mathbf{w}_i(t)\}_{i \in \mathcal{N}_j \cup \{j\}}, b), \quad \forall j \in \mathcal{N}$     *// Coordinate-wise trimmed mean subroutine*
7:     **else**
8:         $\mathbf{w}_j(t+1) \leftarrow \mathbf{w}_j(t) - h\nabla f_j(\mathbf{w}_j(t)), \quad \forall j \in \mathcal{N}$     *// Local gradient-based model update step*
9:         $s \leftarrow s + 1$
10:    **end if**
11: **end for**
**Output:** Final local models $\mathbf{w}_j(T_{\max})$ for all $j \in \mathcal{N}$

---

RESIST is a fully decentralized algorithm, meaning it does not require knowledge of the global network topology, and nodes only communicate with their immediate neighbors. Additionally, each node has access only to its own local empirical loss function (i.e., local dataset) and does not access the local data of other nodes. RESIST is a first-order algorithm, as it updates the local models every few iteration indices $t$ using the local gradient information $\nabla f_j$ at that time. The primary parameters required for RESIST at each node include

the maximum number of edges within the neighborhood of any node expected to be under attack in any slot index $t$, denoted by $b$; the stepsize $h$; the maximum number of iterations $T_{\max}$ for which the algorithm should run; and a positive integer parameter $J > 1$, which determines how often the local gradient information is used to update the local models—specifically, a gradient step is taken every $J$-th iteration index $t$. Equivalently, one $J$-round block consists of $J - 1$ communication/filtering rounds followed by one local gradient update.

As described in Algorithm 1, RESIST updates local models through two primary mechanisms. First, in Steps 4–6, each node broadcasts its local model to its outgoing neighbors, receives models from its incoming neighbors, and then updates its own model using the *coordinate-wise trimmed mean* (CWTM) subroutine, described in Algorithm 2. This subroutine aggregates information using a coordinate-wise trimmed mean, helping mitigate the impact of MITM attacks on the communication links. This filtered aggregation process occurs over $J-1$ consecutive iterations $t$, forming the communication/filtering portion of each $J$-round block before the gradient-based update. Second, in Step 8, nodes update their models using local gradients. Since this gradient-based update is performed independently by each node without relying on information from neighbors, it remains secure against MITM attacks, even if network edges remain compromised.

Since RESIST takes a gradient step only at every $J$-th index $t$, while in the intervening indices nodes engage in local communication and update their local models without taking a gradient step, RESIST operates on two distinct time scales. The first time scale, denoted as $t$, represents the total number of discrete actions performed within the algorithm, encompassing both inter-neighbor communication-based updates and gradient-based updates of the local models. The second time scale, denoted as $s$, corresponds to algorithmic iterations—specifically, the number of updates to the local models based on local gradient information. We sometimes refer to $t$ as the *faster* time scale and $s$ as the *slower* time scale. Note that updates to the local model occur at both time scales; however, within time scale $s$, updates are exclusively based on local gradient information, and no inter-neighbor communication takes place at that time.

We now briefly discuss the CWTM filtering subroutine (Algorithm 2), which aggregates information from incoming edges along with the node's own information at a coordinate-wise level. The procedure involves removing the $b$ largest and $b$ smallest values in each coordinate before computing the average of the remaining values to update the model at a node. Mathematically, following prior works that use CWTM for filtering (Vaidya, 2012; Su & Vaidya, 2016b; Yang & Bajwa, 2019; Fang et al., 2022), for any iteration $t$, the $k$-th coordinate of the received models $\mathbf{w}_i(t)$ at node $j$, where $i \in \mathcal{N}_j$, defines the following sets:

$$\underline{\mathcal{N}}_j^k(t) := \underset{\mathcal{X}:\mathcal{X}\subset\mathcal{N}_j,|\mathcal{X}|=b}{\arg\min} \sum_{i\in\mathcal{X}}[\mathbf{w}_i(t)]_k, \tag{6}$$

$$\overline{\mathcal{N}}_j^k(t) := \underset{\mathcal{X}:\mathcal{X}\subset\mathcal{N}_j,|\mathcal{X}|=b}{\arg\max} \sum_{i\in\mathcal{X}}[\mathbf{w}_i(t)]_k, \quad \text{and} \tag{7}$$

$$\mathcal{C}_j^k(t) := \mathcal{N}_j \setminus \left\{ \underline{\mathcal{N}}_j^k(t) \bigcup \overline{\mathcal{N}}_j^k(t) \right\}. \tag{8}$$

Here, $\underline{\mathcal{N}}_j^k(t)$ is the *lower set* (nodes with incoming edges to $j$ that have the smallest $b$ values in the $k$-th coordinate at time $t$), $\overline{\mathcal{N}}_j^k(t)$ is the *upper set* (nodes with incoming edges to $j$ that have the largest $b$ values), and $\mathcal{C}_j^k(t)$ is the center set (remaining nodes with incoming edges after filtering the extreme values). If multiple sets satisfy the filtering criteria, a random selection is made. After filtering, the information from nodes in the center set is assigned equal weights, and the final average is computed in Step 5. To ensure that the center set is non-empty and the weights remain positive in Step 5 of Algorithm 2, the filtering parameter must satisfy $b < \frac{|\mathcal{N}_j|+1}{2}$.

The update structure above also makes the main communication, filtering, gradient-evaluation, and memory costs explicit. In one communication/CWTM round, node $j$ receives and processes $d$-dimensional model vectors from $|\mathcal{N}_j|$ incoming neighbors, for a received communication volume of $O(d|\mathcal{N}_j|)$ scalars. With direct sorting, CWTM costs $O(d|\mathcal{N}_j|\log|\mathcal{N}_j|)$ operations per communication/CWTM round; selection or partial sorting can reduce the dependence on $|\mathcal{N}_j|$ up to implementation constants. A direct implementation stores the incoming neighbor vectors for one communication round, requiring $O(d|\mathcal{N}_j|)$ memory for neighbor-vector buffers in addition to $O(d)$ memory for the local model and gradient-related quantities; this buffer requirement need not scale with $J$ if buffers are reused across communication rounds. Finally, after $T$

---

**Algorithm 2** Coordinate-wise Trimmed Mean (CWTM)

---

**Input:** Upper bound $b$ on the number of potentially compromised incoming edges per node, local models $\mathbf{w}_i(t)$ received by node $j$ from all $i \in \mathcal{N}_j$, and local model $\mathbf{w}_j(t)$ at node $j$

1: **for** $k = 1, \ldots, d$ **do**

2: $\quad \underline{\mathcal{N}}_j^k(t) \leftarrow \underset{\mathcal{X}:\mathcal{X}\subset\mathcal{N}_j,|\mathcal{X}|=b}{\arg\min} \sum_{i\in\mathcal{X}} [\mathbf{w}_i(t)]_k$ $\qquad\qquad$ // Identify nodes with the $b$ smallest values

3: $\quad \overline{\mathcal{N}}_j^k(t) \leftarrow \underset{\mathcal{X}:\mathcal{X}\subset\mathcal{N}_j,|\mathcal{X}|=b}{\arg\max} \sum_{i\in\mathcal{X}} [\mathbf{w}_i(t)]_k$ $\qquad\qquad$ // Identify nodes with the $b$ largest values

4: $\quad \mathcal{C}_j^k(t) \leftarrow \mathcal{N}_j \backslash \left\{ \underline{\mathcal{N}}_j^k(t) \cup \overline{\mathcal{N}}_j^k(t) \right\}$ $\qquad$ // Filter out nodes with the $b$ smallest and $b$ largest values

5: $\quad \left[\mathbf{w}_j^{\text{CWTM}}(t)\right]_k \leftarrow \frac{1}{|\mathcal{N}_j|-2b+1} \sum_{i\in\mathcal{C}_j^k(t)\cup\{j\}} [\mathbf{w}_i(t)]_k$ $\qquad$ // Compute trimmed mean

6: **end for**

**Output:** $\mathbf{w}_j^{\text{CWTM}}(t)$

---

algorithmic rounds, where each round is either a communication/CWTM round or a local gradient update, RESIST performs approximately $T/J$ local gradient evaluations and $T(J-1)/J$ communication/CWTM rounds, up to floor effects. Thus, for methods that reach a target accuracy within the same total number of algorithmic rounds, larger $J$ can substantially reduce the number of local gradient evaluations, which is useful when local gradient computation is the dominant cost.

Next, we highlight the parallels and distinctions between the BRIDGE algorithm (Fang et al., 2022), the REDGRAF framework (Kuwaranancharoen & Sundaram, 2025), and the proposed RESIST algorithm. When $J = 2$, RESIST uses one CWTM communication/filtering round before each local gradient update. Under the corresponding node-level Byzantine specialization, this is the fixed-stepsize CWTM update associated with BRIDGE-T and with the CWTM instance of REDGRAF. This comparison is at the level of the shared update structure, not the full attack model: the BRIDGE and REDGRAF analyses are for Byzantine node attacks, whereas RESIST is designed for link-level MITM attacks under a per-time incoming-link attack budget. Sec. 7 shows that static Byzantine attacks can be embedded into the MITM framework considered here. For $J > 2$, RESIST uses additional CWTM rounds between local gradient updates; this can improve information mixing before each gradient step and, as seen in the numerical results, can improve convergence behavior up to a problem-dependent choice of $J$, beyond the reduction in local computation cost from less frequent local gradient updates.

Finally, although analytical tools from the Byzantine-resilient literature suffice for analyzing decentralized methods robust to node-level attacks (Vaidya & Garg, 2013; Fang et al., 2022; He et al., 2022a; Kuwaranancharoen & Sundaram, 2025), they do not directly apply to MITM attacks within the RESIST framework. Instead, key techniques from Byzantine-resilient consensus and optimization must be carefully adapted to accommodate the time-varying link-level MITM attack model considered in this paper. Moreover, while standard methods exist for decentralized optimization over time-varying graphs (Nedić & Olshevsky, 2015), they break down in the presence of network attacks. To analyze the RESIST algorithm, we first extend relevant results from Byzantine-resilient consensus to the MITM attack setting in Secs. 3.2 and 3.3. Before presenting these results, we state the graph connectivity assumption that enables RESIST's resilience. This assumption is then used to show that the filtering subroutine CWTM (Algorithm 2) effectively protects nodes from falsified incoming information under MITM attacks, focusing exclusively on the consensus phase of the algorithm without considering gradient updates.

## 3.1 Graph connectivity assumption for RESIST

We begin with a couple of definitions that are essential for stating the graph connectivity assumption. The first definition introduces the concepts of *source node* and *source component* in a directed graph.

**Definition 3.1** (Source node and source component). A node in a directed graph $\mathcal{H}$, with node set $\mathcal{N}(\mathcal{H})$ and edge set $\mathcal{E}(\mathcal{H})$, is termed a *source node* if it has directed paths to all other nodes in the graph. A collection of source nodes forms a *source component* of the graph.

The next definition introduces the notion of *filtered graph topologies* associated with the original graph $\mathcal{G}(\mathcal{N}, \mathcal{E})$. This concept is inherently linked to the CWTM operation performed within RESIST (Algorithm 2) but applies more broadly to any variant of RESIST that filters out information arriving on $2b$ incoming edges of a node.

**Definition 3.2** (Filtered graph topology)**.** The set of *filtered graph topologies* of the graph $\mathcal{G}(\mathcal{N}, \mathcal{E})$ for a given parameter $b$ is defined as the set $\mathcal{T}_{\mathcal{F}}$ of all filtered graphs of $\mathcal{G}$, where each filtered graph $\mathcal{H} \in \mathcal{T}_{\mathcal{F}}$ is obtained by removing exactly $2b$ incoming edges at each node in $\mathcal{G}$. Formally,

$$\mathcal{T}_{\mathcal{F}} := \left\{ \mathcal{H} \mid \mathcal{N}(\mathcal{H}) = \mathcal{N}(\mathcal{G}),\ \mathcal{E}(\mathcal{H}) \subset \mathcal{E}(\mathcal{G}),\ \mathcal{H} \text{ is obtained by removing exactly } 2b \text{ incoming edges at each node,} \right.$$
$$\left. \text{where each } \mathcal{H} \text{ represents a specific instance of edge removals across all nodes.} \right\}.$$

Let $\tau$ denote the cardinality of $\mathcal{T}_{\mathcal{F}}$, i.e., $\tau := |\mathcal{T}_{\mathcal{F}}|$, which we refer to as the number of filtered graphs associated with the underlying graph $\mathcal{G}$ for a given parameter $b$.

Strictly speaking, we should write $\mathcal{T}_{\mathcal{F}}(\mathcal{G}, b)$ and $\tau(\mathcal{G}, b)$ to explicitly indicate their dependence on $\mathcal{G}$ and $b$, but we suppress this notation for simplicity. Additionally, while $\tau$ may be large depending on the topology of $\mathcal{G}$, it remains a finite quantity. In each iteration $t$ of RESIST where the CWTM operation is performed, the algorithm effectively operates on one of the filtered graphs $\mathcal{H} \in \mathcal{T}_{\mathcal{F}}$. The particular filtered graph selected in a given CWTM round need not be the same from one communication round to the next, since the messages retained by CWTM can change with the received values and the adversarial link sequence. Thus, $\mathcal{T}_{\mathcal{F}}$ should be read as the finite collection of all filtered graphs that can arise during the run, and the subsequent consensus analysis works with the sequence of selected filtered graphs rather than a single fixed topology. At the same time, the set of filtered graph topologies $\mathcal{T}_{\mathcal{F}}$ (and thus its cardinality $\tau$) depends only on the original graph $\mathcal{G}$ and the parameter $b$; it does not depend on $t$ or on which specific links are actually attacked during each iteration of the RESIST algorithm.

To ensure sufficient mixing of information within RESIST after the CWTM filtering operation—and, in particular, to guarantee that no node becomes isolated after filtering and that the weight assignments in Step 5 of Algorithm 2 remain non-negative—we require the following assumption on network connectivity:

**Assumption 3.3** (Sufficient network connectivity)**.** The graph $\mathcal{G}(\mathcal{N}, \mathcal{E})$ is assumed to be sufficiently connected, meaning every filtered graph in the set $\mathcal{T}_{\mathcal{F}}$ contains at least one source component with cardinality greater than one.

Assumption 3.3 is imposed on every graph in $\mathcal{T}_{\mathcal{F}}$, rather than on one fixed graph. Thus, RESIST can operate with different selected filtered graphs across communication rounds, provided each selected graph belongs to this family. The essential requirements are the per-time incoming-link attack budget and the source-component condition above, which ensure that enough reliable information remains after filtering.

Note that a network connectivity assumption similar to Assumption 3.3 also appears in the literature on Byzantine-resilient optimization and learning (Su & Vaidya, 2016b; Fang et al., 2022). However, since Byzantine attacks target nodes rather than edges, the corresponding assumptions in these works apply to subgraphs obtained by removing nodes along with their edges from the original graph. Specifically, the assumption in those works requires that each *reduced subgraph* contains a source component of cardinality at least $b + 1$, where $b$ is the maximum number of nodes under attack in the network. In contrast, the nature of MITM attacks necessitates the use of filtered graphs rather than reduced subgraphs. A filtered graph is obtained by removing only incoming edges into each node, whereas a reduced subgraph results from the removal of nodes along with their associated edges. Heuristically, for graphs with sufficiently high edge density (defined as the ratio of existing edges to the maximum possible edges in the graph), filtering edges rather than removing nodes generally results in a sparser structure compared to reduced subgraphs in Byzantine-resilient settings. This is because filtering edges alone leads to a lower edge density than removing both nodes and edges. Consequently, filtered graphs are, in general, less likely to contain a large number of source nodes compared to reduced subgraphs, where paths between nodes are more prevalent.

### 3.2 Supporting lemma for the information mixing step in RESIST

We now present a supporting lemma that establishes that the CWTM-based information mixing step (also referred to as the consensus step), Step 6 in Algorithm 1, ensures that the updated information at every node in the $k$-th coordinate is derived solely from information received through uncompromised edges.

To this end, consider an arbitrary iteration $t$ such that $(t+1) \mod J \neq 0$, and fix an arbitrary coordinate index $k \in \{1, \dots, d\}$. Define the vector $\mathbf{\Omega}(t) \in \mathbb{R}^M$, whose elements correspond to the $k$-th coordinate of the iterates $\mathbf{w}_j(t)$ for all nodes, stacked into the vector $\mathbf{\Omega}(t)$. Note that most quantities related to the $d$-dimensional optimization in this paper, including $\mathbf{\Omega}(t)$, inherently depend on the coordinate index $k$. However, since $k$ is chosen arbitrarily, we often omit this explicit dependence in this and subsequent sections to simplify notation.

In the following lemma, we establish that Steps 4–6 in Algorithm 1 ensure that the update at each node in the $k$-th coordinate is computed exclusively using uncompromised information. Specifically, we show that for $\mathbf{\Omega}(t) \in \mathbb{R}^M$, the update can be expressed as:

$$\mathbf{\Omega}(t+1) = \mathbf{Y}_k(t)\mathbf{\Omega}(t), \tag{9}$$

where $\mathbf{Y}_k(t)$ is a row-stochastic matrix that assigns zero weights to contributions from compromised incoming edges in this equivalent update. The explicit structure of $\mathbf{Y}_k(t)$, referred to as the *mixing matrix*, which depends on both the iteration index $t$ and the coordinate index $k$, is detailed in the following lemma.

**Lemma 3.4.** *Let* $\mathbf{W}(t) \in \mathbb{R}^{M \times d}$ *be the* iterate matrix *whose $i$-th row corresponds to the transpose of the local model (iterate) $\mathbf{w}_i(t) \in \mathbb{R}^d$ at node $i$, as given in Algorithm 1. Under Assumption 3.3, the mixing step (Step 6) in Algorithm 1, for any $k \in \{1, \dots, d\}$ and any iteration $t$ such that $(t+1) \mod J \neq 0$, can be equivalently expressed as:*

$$\big[\mathbf{W}(t+1)\big]_k = \mathbf{Y}_k(t)\big[\mathbf{W}(t)\big]_k, \tag{10}$$

*where the entries of $\mathbf{Y}_k(t)$, the row-stochastic mixing matrix with zero entries corresponding to compromised incoming edges, are given below (for notational convenience, the iteration index $t$ is omitted from various quantities in the following expression, though these quantities within the mixing matrix remain implicitly $t$-dependent). We split the construction of $\mathbf{Y}_k(t)$ into two cases using the quantity $q_j^k := b - b_j^* + b_j^k$. We first give the weight assignment for the case $q_j^k > 0$. This can occur when at least one node in $\mathcal{C}_j^k$ has a compromised link to $j$, when fewer than $b$ links to $j$ are compromised, or both. In this case, let $\overline{\mathcal{A}}_j^k$ and $\underline{\mathcal{A}}_j^k$ be arbitrary but fixed subsets, for the given $j, k, t$, satisfying*

$$\overline{\mathcal{A}}_j^k \subseteq \overline{\mathcal{N}}_j^k \cap \mathcal{N}_j^r, \qquad \underline{\mathcal{A}}_j^k \subseteq \underline{\mathcal{N}}_j^k \cap \mathcal{N}_j^r, \qquad |\overline{\mathcal{A}}_j^k| = |\underline{\mathcal{A}}_j^k| = q_j^k.$$

*Then one admissible choice of $\mathbf{Y}_k(t)$ is*

$$\big[\mathbf{Y}_k\big]_{ji} = \begin{cases} \frac{1}{2(|\mathcal{N}_j|-2b+1)}, & i \in \mathcal{N}_j^r \cap \mathcal{C}_j^k, \\ \frac{1}{|\mathcal{N}_j|-2b+1}, & i = j, \\ \sum\limits_{i' \in \mathcal{N}_j^b \cap \mathcal{C}_j^k} \frac{\theta_{i'}^k}{q_j^k(|\mathcal{N}_j|-2b+1)} \\ \quad + \sum\limits_{i' \in \mathcal{N}_j^r \cap \mathcal{C}_j^k} \frac{\theta_{i'}^k}{2q_j^k(|\mathcal{N}_j|-2b+1)}, & i \in \overline{\mathcal{A}}_j^k, \ \theta_{i'}^k \in [0,1], \\ \sum\limits_{i' \in \mathcal{N}_j^b \cap \mathcal{C}_j^k} \frac{1-\theta_{i'}^k}{q_j^k(|\mathcal{N}_j|-2b+1)} \\ \quad + \sum\limits_{i' \in \mathcal{N}_j^r \cap \mathcal{C}_j^k} \frac{1-\theta_{i'}^k}{2q_j^k(|\mathcal{N}_j|-2b+1)}, & i \in \underline{\mathcal{A}}_j^k, \ \theta_{i'}^k \in [0,1], \\ 0, & \text{otherwise,} \end{cases} \tag{11}$$

*Here, $b_j^* := |\mathcal{N}_j^b|$ denotes the actual (but unknown) number of nodes in the graph that have compromised outgoing edges to node $j$ in iteration $t$. The sets $|\mathcal{N}_j^b|$ and $|\mathcal{N}_j^r|$, both functions of $t$, are defined in Sec. 2.3,*

while $b_j^k$ represents the number of nodes with compromised outgoing edges to $j$ that remain in the filtered set $\mathcal{C}_j^k$ in iteration $t$. On the other hand, when $q_j^k := b - b_j^* + b_j^k = 0$, meaning that all nodes in $\mathcal{C}_j^k$ have uncompromised links to node $j$ in iteration $t$, the matrix $\mathbf{Y}_k(t)$ takes the following form:

$$\left[\mathbf{Y}_k\right]_{ji} = \begin{cases} \frac{1}{|\mathcal{N}_j|-2b+1}, & i \in \{j\} \cup \mathcal{C}_j^k, \\ 0, & otherwise. \end{cases} \tag{12}$$

The proof of this lemma is provided in Appendix B.1. To further clarify the weight assignments within the mixing matrix, we also present a simple illustrative example in Appendix B.2.

**Remark 3.5.** This lemma, along with the discussion in the next section and the analysis in Appendix A, parallels the corresponding discussion and analysis in Vaidya (2012) for Byzantine attacks. However, due to the nature of MITM attacks—which result in filtered graphs rather than reduced subgraphs—these results must be explicitly derived under the MITM attack model. Appendix A provides this necessary derivation. While not the primary contribution of this work, it is included for completeness and self-containment.

### 3.3 Geometric mixing rate for consensus along coordinates

In this section, we focus exclusively on the mixing-based updates in RESIST to analyze the role of the parameter $J$. Specifically, we consider the regime where $J$ is large enough that the condition $(t+1) \mod J = 0$ never applies, thereby isolating the effects of the consensus step from the gradient-based updates. Using the characterization of the coordinate-wise mixing matrix established in Lemma 3.4, we show that the product of mixing matrices, $\mathbf{Y}_k(t)\mathbf{Y}_k(t-1)\cdots\mathbf{Y}_k(0)$, converges geometrically to a rank-one stationary mixing matrix for each coordinate $k$. This geometric mixing property is a key ingredient in establishing the consensus guarantees of RESIST along individual coordinates and will be leveraged in the subsequent convergence analysis. The goal of this section is to outline the implications of Lemma 3.4 for geometric mixing behavior, while deferring the full technical details and proofs to Appendices A.1–A.3.

To formally express the geometric mixing behavior, we define a transition matrix $\mathbf{\Phi}(t,t_0)$ that captures the product of mixing matrices $\mathbf{Y}_k(t)$ from (11) and (12), omitting the subscript $k$ for notational simplicity. This transition matrix propagates information from time index $t_0 \leqslant t$ to $t$ and is given by:

$$\mathbf{\Phi}(t,t_0) := \mathbf{Y}(t)\mathbf{Y}(t-1)\cdots\mathbf{Y}(t_0). \tag{13}$$

Each factor in this product is the row-stochastic mixing matrix associated with a filtered graph induced by CWTM at the corresponding communication round. Since this filtered graph may vary with $t$, $\mathbf{\Phi}(t,t_0)$ is a product over a time-varying sequence of filtered graphs rather than a power of one fixed mixing matrix. If Assumption 3.3 on sufficient network connectivity of $\mathcal{G}$ holds, then from the discussion and analysis in Appendices A.1–A.3, it follows that:

$$\lim_{t\to\infty} \mathbf{\Phi}(t,0) = \mathbf{1}\mathbf{c}^T, \tag{14}$$

where the vector $\mathbf{c} \in \mathbb{R}^M$ satisfies $[\mathbf{c}]_j \geqslant 0$ and $\sum_{j=1}^M [\mathbf{c}]_j = 1$. The discussion and analysis in Appendix A further guarantee that this convergence is geometric. Specifically, removing the assumption that $J$ is very large and considering any $t_0 \leqslant t$ with $t_0$ and $t \in [lJ, (l+1)J - 2]$ for any $l = 0, 1, 2, \ldots$, it follows from Appendix A that:

$$\left|[\mathbf{\Phi}(t,t_0)]_{ji} - [\mathbf{c}]_i\right| \leqslant (1 - \beta^{\tau M})^{\lfloor \frac{t-t_0}{\tau M} \rfloor}, \tag{15}$$

where $\beta := \frac{\alpha}{4b}$ with $\alpha := \frac{1}{M-2b+1}$, and $\tau$ denotes the cardinality of the set of filtered graph topologies (see Definition 3.2).

The geometric mixing characterization in (15) of the mixing steps in RESIST is fundamental in determining the appropriate choice of the parameter $J$ in the algorithm. By selecting $J$ appropriately and substituting $t - t_0 = J - 2$ in (15), we ensure that the $k$-th coordinate of the local model parameter at each node reaches a state sufficiently close to a weighted agreement (consensus), where the weights are given by the entries of the vector $\mathbf{c}$ from (15), referred to as the consensus vector.

## 4    Preliminaries for Algorithmic Convergence Guarantees

In this section, we develop preliminary results that will be used to derive algorithmic convergence guarantees for RESIST applied to the decentralized optimization problem (4) under various classes of loss functions. As in the ERM formulation of (4), we fix an arbitrary realization of the local datasets $\{\mathcal{Z}_j\}_{j \in \mathcal{N}}$ (equivalently, we condition on the data). Accordingly, all statements in this section, as well as in Secs. 5 and 6, are understood to hold for any given fixed collection of samples. The focus in these sections is therefore exclusively on the algorithmic behavior of RESIST when optimizing the resulting empirical objectives. We return to the role of data randomness only when deriving statistical learning rates in Sec. 8. Under this convention, we suppress explicit data dependence and work with the induced local empirical risk functions $f_j(\cdot) := \frac{1}{N} \sum_{i=1}^N \ell(\cdot, \mathbf{z}_{ij})$, $j \in \mathcal{N}$, together with their full empirical gradients $\nabla f_j(\cdot) := \frac{1}{N} \sum_{i=1}^N \nabla \ell(\cdot, \mathbf{z}_{ij})$. Once the initialization is fixed, the RESIST updates are fully specified by the algorithmic rules and the fixed empirical functions $\{f_j\}$.[1]

Let $\mathbf{W}(t) \in \mathbb{R}^{M \times d}$ denote the iterate matrix at time $t$, as defined in Lemma 3.4, where the $i$-th row of $\mathbf{W}(t)$ corresponds to the local model $\mathbf{w}_i(t)$ at node $i$. For any coordinate index $k \in \{1, \dots, d\}$, let $[\mathbf{W}(t)]_k \in \mathbb{R}^M$ denote the $k$-th column of $\mathbf{W}(t)$. Define the separable aggregate function $F(\mathbf{W}) := \sum_{i=1}^M f_i(\mathbf{w}_i)$, where $\mathbf{W} = [\mathbf{w}_1, \dots, \mathbf{w}_M]^T$. The gradient of $F$ with respect to $\mathbf{W}$, denoted $\nabla F(\mathbf{W}) \in \mathbb{R}^{M \times d}$, is the matrix whose $i$-th row equals $[\nabla f_i(\mathbf{w}_i)]^T$; in particular, evaluated at $\mathbf{W}(t)$, the $i$-th row of $\nabla F(\mathbf{W}(t))$ is $[\nabla f_i(\mathbf{w}_i(t))]^T$. To facilitate the analysis, we introduce an auxiliary matrix sequence $\{\mathbf{T}(s)\}_{s \geqslant 0}$ that records the collection of local gradients evaluated at the iterates where gradient updates occur. Specifically, we define $\mathbf{T}(0) := \nabla F(\mathbf{W}(0))$, and update $\mathbf{T}(s)$ only at iterations where a gradient step is performed. Combining the coordinate-wise consensus update induced by the CWTM operation (Algorithm 2) with the local gradient update in RESIST (Algorithm 1), the evolution of the $k$-th coordinate of the iterates can be written as

$$[\mathbf{W}(t+1)]_k = \begin{cases} \mathbf{Y}_k(t)[\mathbf{W}(t)]_k, & (t+1) \bmod J \neq 0, \\ [\mathbf{W}(t)]_k - h[\mathbf{T}(s)]_k, & (t+1) \bmod J = 0, \end{cases} \tag{16}$$

where $s$ denotes the slow (algorithmic) time index that increments only when $(t+1) \bmod J = 0$. Moreover, whenever a gradient update is performed, the auxiliary variable $\mathbf{T}(s)$ is updated according to

$$[\mathbf{T}(s+1)]_k = [\nabla F(\mathbf{W}(t+1))]_k. \tag{17}$$

Next, we study the properties of products of the coordinate-wise mixing matrices $\{\mathbf{Y}_k(t)\}$. Define[2]

$$\mathbf{Q}_k(s) := \prod_{r=J\lfloor t/J \rfloor}^{J\lfloor t/J \rfloor + J - 2} \mathbf{Y}_k(r), \tag{18}$$

where $s := J\lfloor t/J \rfloor$ denotes the starting iteration of a block of $J - 1$ consecutive consensus updates between two gradient steps. Observe that $\mathbf{Q}_k(s)$ coincides with the transition matrix $\mathbf{\Phi}\big(J\lfloor t/J \rfloor + J - 2, \ J\lfloor t/J \rfloor\big)$, where $\mathbf{\Phi}(\cdot, \cdot)$ is the transition matrix defined in Sec. 3.3. Using this notation, the RESIST updates can be expressed on the $s$-time scale as

$$[\mathbf{W}(s+1)]_k = \mathbf{Q}_k(s)[\mathbf{W}(s)]_k - h[\mathbf{T}(s)]_k, \tag{19}$$

$$[\mathbf{T}(s+1)]_k = [\nabla F(\mathbf{W}(s+1))]_k. \tag{20}$$

The transition from iteration $s$ to $s + 1$ corresponds to iteration $t = sJ + J - 1$ on the original $t$-time scale. Although the update (20) involving the auxiliary variable $\mathbf{T}(s)$ may appear redundant, it significantly simplifies the subsequent analysis by allowing the algorithmic evolution to be written compactly on the $s$-time scale. We next present a corollary characterizing how the sequence of matrix products $\{\mathbf{Q}_k(s)\}_{s \geqslant 0}$ approaches consensus, which will be used to establish rates of consensus and convergence for the RESIST algorithm.

---

[1] The algorithmic convergence analysis does not require that all nodes use the same underlying loss function. In particular, each node $j$ may employ a different loss $\ell_j$, leading to local empirical risks of the form $f_j(\cdot) := \frac{1}{N} \sum_{i=1}^N \ell_j(\cdot, \mathbf{z}_{ij})$. What is essential for the analysis are the structural assumptions imposed on the local empirical loss functions $f_j$, rather than the identity of the underlying sample-level losses.

[2] In the product notation $\prod_i^j$, the matrix indexed by the upper limit $j$ appears on the left of the product. This is commonly referred to as a "backward product" (Leizarowitz, 1992).

**Corollary 4.1.** *Under Assumption 3.3 and for $J > 1$, the sequence of matrices $\{\mathbf{Q}_k(s)\}_{s=0}^{\infty}$ satisfies the following bound for any $i, j \in \{1, \cdots, M\}$:*

$$\left| \left[ \prod_{s=0}^{S} \mathbf{Q}_k(s) \right]_{ji} - [\mathbf{c}_k]_i \right| \leqslant \left(1 - \beta^{\tau M}\right)^{\left\lfloor \frac{S(J-1)-1}{\tau M} \right\rfloor}, \tag{21}$$

*for any $S > \frac{\tau M}{J-1}$, where $\mathbf{c}_k \in \mathbb{R}^M$ is the transpose of the row vector associated with the infinite backward product $\prod_{s=0}^{\infty} \mathbf{Q}_k(s)$, i.e.,*

$$\prod_{s=0}^{\infty} \mathbf{Q}_k(s) = \prod_{t=0}^{\infty} \mathbf{\Phi}(t, 0) = \mathbf{1}\mathbf{c}_k^T =: \mathbf{Q}_k^{\pi},$$

*where $\mathbf{Q}_k^{\pi}$ is a rank-one mixing matrix with generally non-uniform weights.*

*Furthermore, for any $J > \tau M + 1$ and any $s \geqslant 0$, we have*

$$\left| \left[ \mathbf{Q}_k(s) \right]_{ji} - [\mathbf{c}_k(s)]_i \right| \leqslant \left(1 - \beta^{\tau M}\right)^{\left\lfloor \frac{J-2}{\tau M} \right\rfloor}, \tag{22}$$

*where $\mathbf{c}_k(s)$ is the transpose of the row vector associated with the infinite backward product $\prod_{i=s}^{\infty} \mathbf{Q}_k(i)$, i.e.,*

$$\prod_{i=s}^{\infty} \mathbf{Q}_k(i) = \mathbf{1}\mathbf{c}_k(s)^T =: \mathbf{Q}_k^{\pi}(s),$$

*and $\mathbf{Q}_k^{\pi}(s)$ satisfies*

$$\mathbf{Q}_k^{\pi}(s) = \mathbf{Q}_k^{\pi}(s+1)\mathbf{Q}_k(s), \tag{23}$$

*for all $s \geqslant 0$, with $\mathbf{Q}_k^{\pi}(0) := \mathbf{Q}_k^{\pi}$.*

*Proof.* By construction of the mixing matrix $\mathbf{Y}_k(t)$ from (11) and (12) in Lemma 3.4, we get that $\mathbf{Q}_k(s)$ from (18) for any $s$ is a scrambling matrix for $J > \tau M + 1$; intuitively, this means that $\mathbf{Q}_k(s)$ is row stochastic and that every pair of its rows shares at least one column with positive entries, ensuring sufficient mixing. A formal definition and equivalent characterizations of scrambling matrices are provided in Appendix A. Then, for $S > \frac{\tau M}{J-1}$, the bound (21) follows from (15) and the proof of Lemma A.10 in Sec. A.4.

For obtaining the second inequality (22), fix any $s \geqslant 0$ and consider the tail sequence $\{\mathbf{Q}_k(i)\}_{i=s}^{\infty}$. Applying the same argument to this shifted sequence implies that the infinite backward product $\prod_{i=s}^{\infty} \mathbf{Q}_k(i)$ exists and converges to a rank-one row-stochastic matrix with identical rows, say $\mathbf{1}\mathbf{c}_k(s)^T$. Using Lemma A.10, (22) follows. Finally, (23) follows directly from the definition of the infinite backward product of matrices. ∎

Observe that the infinite product $\prod_{i=s}^{\infty} \mathbf{Q}_k(i)$ in Corollary 4.1 is equal to the transition matrix given by $\lim_{t\to\infty} \mathbf{\Phi}(t, sJ)$ along the $k$-th coordinate. This infinite product can be viewed as a stationary mixing matrix $\mathbf{Q}_k^{\pi}(s)$ with generally non-uniform weights. Due to the time-varying nature of the row-stochastic weight matrices $\mathbf{Y}_k(t)$ in the RESIST algorithm, it is difficult to directly derive a recursion for the *exact consensus error*, owing to both the uncertainty of the attacker's behavior and the screening mechanism. By the exact consensus error, we mean the quantity $\left\| \frac{\mathbf{1}\mathbf{1}^T}{M}[\mathbf{W}(s)]_k - [\mathbf{W}(s)]_k \right\|$, where $\mathbf{1} \in \mathbb{R}^M$. By a recursion, we mean a bound of the form

$$\left\| \frac{\mathbf{1}\mathbf{1}^T}{M}[\mathbf{W}(s+1)]_k - [\mathbf{W}(s+1)]_k \right\| \leqslant \rho \left\| \frac{\mathbf{1}\mathbf{1}^T}{M}[\mathbf{W}(s)]_k - [\mathbf{W}(s)]_k \right\| + e(s),$$

for some $\rho \geqslant 0$ and some bounded error term $e(s)$. The difficulty stems from the fact that, if one averages the update in (19), the right-hand side does not recover $\frac{\mathbf{1}\mathbf{1}^T}{M}[\mathbf{W}(s)]_k$, since the matrices $\mathbf{Q}_k(s)$ and $\frac{\mathbf{1}\mathbf{1}^T}{M}$ do not generally commute. As a result, the RESIST dynamics do not preserve the exact network average. Instead, the consensus process induces a *weighted agreement* characterized by the stationary mixing matrix

$\mathbf{Q}_k^\pi(s)$. This motivates analyzing an *inexact averaging error*, defined relative to $\mathbf{Q}_k^\pi(s)$ rather than the exact averaging operator. Using (23) from Corollary 4.1, we obtain the following recursive bound:

$$\left\| \mathbf{Q}_k^\pi(s+1)[\mathbf{W}(s+1)]_k - [\mathbf{W}(s+1)]_k \right\| \leqslant \rho \left\| \mathbf{Q}_k^\pi(s)[\mathbf{W}(s)]_k - [\mathbf{W}(s)]_k \right\| + e(s),$$

for some $\rho \geqslant 0$ and some bounded error term $e(s)$.

To make the above idea of inexact averaging concrete, we define averaging operators that will be instrumental in the convergence analysis of the RESIST algorithm.

**Definition 4.2.** For any $\mathbf{A} \in \mathbb{R}^{M \times d}$, where $d \geqslant 1$, the *inexact (approximate) averaging operator* $\widehat{(\cdot)}^{k,s}$ and the *exact averaging operator* $\overline{(\cdot)}$ are defined as

- $\widehat{(\cdot)}^{k,s} : \mathbf{A} \mapsto \mathbf{Q}_k^\pi(s)\mathbf{A}$

- $\overline{(\cdot)} : \mathbf{A} \mapsto \frac{\mathbf{1}\mathbf{1}^T}{M}\mathbf{A}$

These operators commute[3] with the $\nabla(\cdot)$ and $[\cdot]_k$ operators.

We note that if a matrix $\mathbf{A}(s)$ depends on $s$, then applying the operator $\widehat{(\cdot)}^{k,s}$ or the operator $\overline{(\cdot)}$ results in the matrices $\widehat{\mathbf{A}}^{k,s}(s)$ or $\overline{\mathbf{A}}(s)$, respectively. Similarly, when the gradient matrix $\nabla F(\mathbf{W}(s))$ is acted upon by the operator $\widehat{(\cdot)}^{k,s}$ or the operator $\overline{(\cdot)}$, the resulting matrices are denoted by $\nabla \widehat{F}^{k,s}(\mathbf{W}(s))$ or $\nabla \overline{F}(\mathbf{W}(s))$, respectively. Next, we define error sequences that capture the discrepancy between exact averaging (corresponding to the ideal case without attacks) and inexact (approximate) averaging induced by the uncertainty of the attackers and the screening mechanism in the RESIST algorithm. These sequences will be instrumental in establishing convergence guarantees for RESIST.

**Definition 4.3.** Let $\{\xi_k^1(s)\}_s$, $\{\xi_k^2(s)\}_s$, $\{\xi_k^3(s)\}_s$, $\{\xi_k^4(s)\}_s$, $\{\xi_k^5(s)\}_s$, and $\{\xi_{\mathbf{w}*}^6(s)\}_s$ be error sequences defined for all $k$ and $s$ as follows:

$$\xi_k^1(s) := \left\| [\widehat{\mathbf{W}}^{k,s}(s)]_k - [\mathbf{W}(s)]_k \right\|, \tag{24}$$

$$\xi_k^2(s) := \left\| [\widehat{\mathbf{T}}^{k,s}(s)]_k - [\mathbf{T}(s)]_k \right\|, \tag{25}$$

$$\xi_k^3(s) := \left\| [\widehat{\mathbf{W}}^{k,s}(s)]_k - [\overline{\mathbf{W}}(s)]_k \right\|, \tag{26}$$

$$\xi_k^4(s) := \left\| [\widehat{\mathbf{T}}^{k,s}(s)]_k - [\overline{\mathbf{T}}(s)]_k \right\|, \tag{27}$$

$$\xi_k^5(s) := \left\| [\mathbf{W}(s)]_k - [\overline{\mathbf{W}}(s)]_k \right\|, \tag{28}$$

$$\xi_{\mathbf{w}*}^6(s) := \left\| \mathbf{w}^* - \widehat{\mathbf{w}}^s(s) \right\|, \tag{29}$$

where $\mathbf{w}^* \in \arg\min_{\mathbf{w}} \frac{1}{M} \sum_{j=1}^M f_j(\mathbf{w})$. For strongly convex loss functions, $\mathbf{w}^*$ is unique, whereas for nonconvex loss functions, $\mathbf{w}^*$ denotes any stationary point satisfying Assumption 6.2. Moreover, for any $s \geqslant 0$,

$$\widehat{\mathbf{w}}^s(s) = \begin{bmatrix} \sum_{j=1}^M [\mathbf{c}_1(s)]_j [\mathbf{w}_j(s)]_1 \\ \sum_{j=1}^M [\mathbf{c}_2(s)]_j [\mathbf{w}_j(s)]_2 \\ \vdots \\ \sum_{j=1}^M [\mathbf{c}_k(s)]_j [\mathbf{w}_j(s)]_k \\ \vdots \\ \sum_{j=1}^M [\mathbf{c}_d(s)]_j [\mathbf{w}_j(s)]_d \end{bmatrix}, \tag{30}$$

where the weights $[\mathbf{c}_k(s)]_j$ for any $k$ and $j$ are defined in Corollary 4.1.

---

[3]The operators commute due to the linearity of the $\nabla$ operator. By linearity of $\nabla$, we mean that $\nabla(c_1 f_1 + c_2 f_2) = c_1 \nabla f_1 + c_2 \nabla f_2$ for any scalars $c_1, c_2$ and differentiable functions $f_1, f_2$.

The sequences in Definition 4.3 are referred to as error sequences since they quantify either the deviation of the $k$-th coordinate from its consensus value (both exact and inexact) or the distance between the coordinate-wise inexactly averaged iterate $\hat{\mathbf{w}}^s(s)$ and an optimal point $\mathbf{w}^*$. In particular, $\xi_k^1(s)$ and $\xi_k^5(s)$ are termed *consensus errors*, while $\xi_{\mathbf{w}^*}^6(s)$ is referred to as the *averaged iterate error*. For the remaining sequences, we use $\xi_k^2(s)$ to measure an *auxiliary-gradient discrepancy* involving the auxiliary matrix $\mathbf{T}(s)$, $\xi_k^4(s)$ to measure the corresponding *auxiliary-gradient averaging gap*, and $\xi_k^3(s)$ to measure the *iterate averaging gap* between applying the inexact averaging operator $\widehat{(\cdot)}^{k,s}$ and the exact averaging operator $\overline{(\cdot)}$ to $\mathbf{W}(s)$. The subsequent analysis uses these quantities in two stages. In Sec. 4.1, Lemma 4.6 gives the recursion for the exact consensus error $\xi_k^5(s)$, Lemma 4.7 gives the recursion for the inexact consensus error $\xi_k^1(s)$, Lemma 4.9 bounds $\xi_k^2(s)$, and Corollary 4.10 bounds $\xi_k^4(s)$. Lemma 4.11 then rewrites the update for $\hat{\mathbf{w}}^s(s)$ as an inexact gradient-descent recursion with additive errors $\mathbf{e}_1(s)$ and $\mathbf{e}_2(s)$: the term $\mathbf{e}_2(s)$ is bounded through $\xi_k^1(s)$, while $\mathbf{e}_1(s)$ captures the gradient error induced by coordinate-wise inexact averaging under filtering and attacks and is summarized later by the residual constant $C_0$ in Theorem 5.5. In Sec. 5, Lemma 5.3 controls the averaged iterate error $\xi_{\mathbf{w}^*}^6(s)$, and Lemma 5.4 introduces a state vector collecting $\xi_k^1(s)$, $\xi_k^5(s)$, and $\xi_{\mathbf{w}^*}^6(s)$ to prove the $s$-time residual-neighborhood bound in Theorem 5.5 and the corresponding $t$-time bound in Theorem 5.8.

**Remark 4.4** (Exact vs. inexact consensus)**.** The error sequences introduced above quantify disagreement among local iterates using different averaging operators. Throughout the remainder of the paper, we refer to consensus with respect to the operator that defines the corresponding error, with a slight abuse of language. In particular, vanishing *exact averaging error*, defined relative to the uniform averaging operator, is termed *exact consensus.* Algorithms with doubly stochastic averaging, such as DGD, are known to achieve this form of consensus. Likewise, vanishing *inexact averaging error*, defined relative to the weighted averaging operator induced by the mixing dynamics, is termed *inexact consensus.* In this case, the local iterates asymptotically agree on a convex combination of the local iterates rather than on the exact network average. Algorithms based on row-stochastic averaging, such as RESIST, generally exhibit this form of consensus.

We are now ready to develop the consensus guarantees for RESIST.

## 4.1 Exact and inexact consensus dynamics of RESIST on the $s$-time scale

Throughout this section, we assume that the local functions $f_i$ for all $i \in \mathcal{N}$ are continuously differentiable; no additional assumptions (such as convexity or strong convexity) are imposed at this stage. Recall that we introduced an auxiliary matrix-valued variable $\mathbf{T}(s)$ in the previous section to store gradient information across the network. We refer to this auxiliary variable as the *tracker*. We begin by presenting a lemma that characterizes the asymptotic behavior of the tracker update.

**Lemma 4.5.** *The average tracking vector $[\overline{\mathbf{T}}(s)]_k$ tracks the average gradient $[\nabla \overline{F}(\mathbf{W}(s))]_k$ along any dimension $k$, i.e., $[\overline{\mathbf{T}}(s)]_k = [\nabla \overline{F}(\mathbf{W}(s))]_k$. Further, suppose the sequence $\{\mathbf{W}(s)\}_s$ converges to some limit $\mathbf{W}^*$. Then we have that $[\overline{\mathbf{T}}(s)]_k \xrightarrow{s \to \infty} [\nabla \overline{F}(\mathbf{W}^*)]_k$ for any dimension $k$.*

*Proof.* Applying the operator $\overline{(\cdot)}$ to $[\mathbf{T}(s)]_k$ yields

$$[\overline{\mathbf{T}}(s)]_k = [\nabla \overline{F}(\mathbf{W}(s))]_k. \tag{31}$$

Taking the limit $s \to \infty$ and using the continuity of $\nabla f_i$ completes the proof. ∎

**Lemma 4.6.** *Under Assumption 3.3, the sequence $\{[\mathbf{W}(s)]_k\}_s$ for any $k$ satisfies the following bound:*

$$\xi_k^5(s+1) \leqslant M^{\frac{3}{2}}(1 - \beta^{\tau M})^{\left\lfloor \frac{(J-2)}{\tau M} \right\rfloor} \xi_k^5(s) + h \left\| [\overline{\mathbf{T}}(s)]_k - [\mathbf{T}(s)]_k \right\|,$$

*where $\beta = \frac{\alpha}{4b}$ with $\alpha = \frac{1}{M - 2b + 1}$.*

The proof of this lemma is provided in Appendix C.1. In addition, the reasons why existing algorithms designed to handle Byzantine attacks cannot be directly adapted to our setting are discussed in Remark C.1.

**Lemma 4.7.** *Under Assumption 3.3, the sequence $\{\xi_k^1(s)\}_s$ satisfies the following recursion for any $s \geqslant 0$:*

$$\xi_k^1(s+1) \leqslant M^{\frac{3}{2}}(\sqrt{M} + 1)(1 - \beta^{\tau M})^{\left\lfloor \frac{(J-2)}{\tau M} \right\rfloor} \xi_k^1(s) + h(\sqrt{M} + 1)\xi_k^2(s).$$

The proof of this lemma is in Appendix C.2. Observe that by carefully choosing $J$ in the inequalities of Lemmas 4.6 and 4.7, one can obtain geometric decay of the exact and inexact consensus errors up to residual terms. In particular, for geometric decay of $\xi_k^1(s)$ and $\xi_k^5(s)$, it suffices that $M^{\frac{3}{2}}(\sqrt{M}+1)\big(1 - \beta^{\tau M}\big)^{\left\lfloor \frac{J-2}{\tau M} \right\rfloor} < 1$, and hence $M^{\frac{3}{2}}\big(1 - \beta^{\tau M}\big)^{\left\lfloor \frac{J-2}{\tau M} \right\rfloor} < 1$ in Lemmas 4.7 and 4.6, respectively. Thus, for these two recursions, a sufficiently large choice of $J$ makes the coefficients multiplying $\xi_k^1(s)$ and $\xi_k^5(s)$ smaller than one. The additional terms in these recursions are handled separately: Lemma 4.7 contains the auxiliary-gradient discrepancy $\xi_k^2(s)$, while Lemma 4.6 contains the tracker-averaging gap $\big\|[\overline{\mathbf{T}}(s)]_k - [\mathbf{T}(s)]_k\big\|$. After introducing smoothness, Lemma 4.9 bounds both quantities by terms involving $\sum_{k=1}^d \xi_k^1(s)$, $\xi_{\mathbf{w}*}^6(s)$, and $\sum_{j=1}^M \big\|\mathbf{w}^* - \mathbf{w}_j^*\big\|$.

Before stating the smoothness condition, we recall that, throughout the algorithmic convergence analysis, we suppress explicit data dependence and work with the induced local empirical risk functions $f_j(\cdot) := \frac{1}{N}\sum_{i=1}^N \ell(\cdot, \mathbf{z}_{ij})$, where $f_j(\cdot) : \mathbb{R}^d \to \mathbb{R}$ maps the $d$-dimensional model space to the reals. Accordingly, any assumption on $f_j$ pertains only to its first argument, i.e., the model variable. We return to assumptions involving both the model parameters and the data samples when deriving statistical learning rates in Sec. 8.

**Assumption 4.8.** For all $j \in \{1, \ldots, M\}$, the function $f_j : \mathbb{R}^d \to \mathbb{R}$ is $L$-gradient Lipschitz continuous and lower bounded, i.e., $\inf_{\mathbf{w}} f_j(\mathbf{w}) > -\infty$.

As a direct consequence of Assumption 4.8, each $f_j$ is coordinate-wise $L$-gradient Lipschitz continuous. The lower boundedness assumption further implies that $\arg\min f_j \neq \varnothing$ for all $j \in \{1, \ldots, M\}$.

**Lemma 4.9.** *Let* $\mathbf{w}_j^* \in \arg\min_{\mathbf{w}} f_j(\mathbf{w}) \quad \forall \quad j \in \{1, 2, \ldots, M\}$, $\mathbf{w}^* \in \arg\min_{\mathbf{w}} f(\mathbf{w})$, *where* $f(\cdot) := \frac{1}{M}\sum_{j=1}^M f_j(\cdot)$. *Then under Assumptions 3.3 and 4.8, the sequence* $\{[\mathbf{T}(s)]_k\}_s$ *for any* $k$ *satisfies the following bounds:*

$$\xi_k^2(s) \leqslant (\sqrt{M}+1)L\sqrt{M}\sum_{k=1}^d \xi_k^1(s) + (\sqrt{M}+1)LM\xi_{\mathbf{w}*}^6(s) + (\sqrt{M}+1)L\sum_{j=1}^M \big\|\mathbf{w}^* - \mathbf{w}_j^*\big\|, \tag{32}$$

$$\big\|[\overline{\mathbf{T}}(s)]_k - [\mathbf{T}(s)]_k\big\| \leqslant L\sqrt{M}\sum_{k=1}^d \xi_k^1(s) + LM\xi_{\mathbf{w}*}^6(s) + L\sum_{j=1}^M \big\|\mathbf{w}^* - \mathbf{w}_j^*\big\|. \tag{33}$$

The proof of this lemma is given in Appendix C.3. As a direct consequence of Lemma 4.9, we have the following corollary.

**Corollary 4.10.** *Under Assumptions 3.3 and 4.8, the sequence* $\{\xi_k^4(s)\}_s$ *for any* $k$ *satisfies the following bound:*

$$\xi_k^4(s) \leqslant (\sqrt{M}+2)L\sqrt{2}\sum_{k=1}^d \xi_k^1(s) + (\sqrt{M}+2)LM\xi_{\mathbf{w}*}^6(s) + (\sqrt{M}+2)L\sum_{j=1}^M \big\|\mathbf{w}^* - \mathbf{w}_j^*\big\|. \tag{34}$$

In order to establish convergence guarantees for the RESIST algorithm, we require an update rule on the coordinate-wise inexact averaged vector $\widehat{\mathbf{w}}^s(s)$. The next lemma provides this update rule.

**Lemma 4.11.** *Under Assumptions 3.3 and 4.8, the sequence* $\{\widehat{\mathbf{w}}^s(s)\}_s$ *satisfies the following inexact gradient descent update[4] for any* $s \geqslant 0$:

$$\widehat{\mathbf{w}}^{s+1}(s+1) = \widehat{\mathbf{w}}^s(s) - h\nabla f(\widehat{\mathbf{w}}^s(s)) + \mathbf{e}_1(s) + \mathbf{e}_2(s), \tag{35}$$

---

[4]An inexact gradient descent update refers to the standard gradient descent with some additive error term.

where $f(\cdot) := \frac{1}{M}\sum_{j=1}^{M} f_j(\cdot)$,

$$
\mathbf{e}_1(s) = h\left(\begin{bmatrix}\nabla_1 f(\hat{\mathbf{w}}^s(s)) \\ \nabla_2 f(\hat{\mathbf{w}}^s(s)) \\ \vdots \\ \vdots \\ \vdots \\ \nabla_k f(\hat{\mathbf{w}}^s(s)) \\ \vdots \\ \vdots \\ \nabla_d f(\hat{\mathbf{w}}^s(s))\end{bmatrix} - \begin{bmatrix}\nabla_1 f^{1,s+1}(\hat{\mathbf{w}}^s(s)) \\ \nabla_2 f^{2,s+1}(\hat{\mathbf{w}}^s(s)) \\ \vdots \\ \vdots \\ \vdots \\ \nabla_k f^{k,s+1}(\hat{\mathbf{w}}^s(s)) \\ \vdots \\ \vdots \\ \nabla_d f^{d,s+1}(\hat{\mathbf{w}}^s(s))\end{bmatrix}\right) \tag{36}
$$

and[5]

$$
\mathbf{e}_2(s) = h\left(\begin{bmatrix}\sum_{j=1}^{M}[\mathbf{c}_1(s+1)]_j\nabla_1 f_j(\hat{\mathbf{w}}^s(s)) \\ \sum_{j=1}^{M}[\mathbf{c}_2(s+1)]_j\nabla_2 f_j(\hat{\mathbf{w}}^s(s)) \\ \vdots \\ \vdots \\ \sum_{j=1}^{M}[\mathbf{c}_k(s+1)]_j\nabla_k f_j(\hat{\mathbf{w}}^s(s)) \\ \vdots \\ \vdots \\ \sum_{j=1}^{M}[\mathbf{c}_d(s+1)]_j\nabla_d f_j(\hat{\mathbf{w}}^s(s))\end{bmatrix} - \begin{bmatrix}\sum_{j=1}^{M}[\mathbf{c}_1(s+1)]_j\nabla_1 f_j(\mathbf{w}_j(s)) \\ \sum_{j=1}^{M}[\mathbf{c}_2(s+1)]_j\nabla_2 f_j(\mathbf{w}_j(s)) \\ \vdots \\ \vdots \\ \sum_{j=1}^{M}[\mathbf{c}_k(s+1)]_j\nabla_k f_j(\mathbf{w}_j(s)) \\ \vdots \\ \vdots \\ \sum_{j=1}^{M}[\mathbf{c}_d(s+1)]_j\nabla_d f_j(\mathbf{w}_j(s))\end{bmatrix}\right), \tag{37}
$$

$$
\|\mathbf{e}_2(s)\| \leqslant Lh\sqrt{Md}\sum_{k=1}^{d}\xi_k^1(s), \tag{38}
$$

with $f^{k,s+1}(\cdot) := \sum_{j=1}^{M}[\mathbf{c}_k(s+1)]_j f_j(\cdot)$ for any $k,s$.

The proof of this lemma is given in Appendix D.1. Lemma 4.11 rewrites the coordinate-wise inexact averaged update as an inexact gradient-descent recursion for the averaged objective $f$. This separates two error sources. The term $\mathbf{e}_2(s)$ captures the effect of local disagreement and is bounded through the inexact consensus errors $\xi_k^1(s)$. The term $\mathbf{e}_1(s)$ captures the gradient mismatch induced by coordinate-wise inexact averaging under filtering and attacks. Thus, the subsequent convergence analysis studies an inexact gradient-descent recursion whose two perturbations have distinct origins: $\mathbf{e}_2(s)$ comes from local disagreement, while $\mathbf{e}_1(s)$ comes from inexact coordinate-wise averaging under filtering and attacks.

With the inexact recursion above in place, in order to develop convergence rates for RESIST in Algorithm 1 under different classes of loss functions, we will need the following assumption on the boundedness of iterates.

**Assumption 4.12.** The iterate sequence $\{\mathbf{w}_j(t)\}_t$ at any node $j$ generated by RESIST in Algorithm 1 stays uniformly bounded by some sufficiently large compact set $\mathcal{K}$ for any given bounded initialization of RESIST, where this compact set depends only on the initialization of RESIST.

---

[5]Here $\nabla_k$ is the partial derivative with respect to the $k$-th coordinate.

We include several remarks to clarify the role of this assumption. Boundedness assumptions of this type are common in decentralized optimization (Nedic & Ozdaglar, 2009; Duchi et al., 2012; Jakovetić et al., 2014; Sundhar Ram et al., 2010; Xin et al., 2019). In the present MITM attack setting, however, this assumption needs to be discussed carefully, because compromised links may transmit arbitrary values and may therefore appear capable of driving the iterates outside any bounded region. Even in this setting, Assumption 4.12 should be read as a compactness condition on the regular-node iterates, rather than as an automatic consequence of the attack model. At the communication step, CWTM provides a separate control mechanism for adversarially corrupted messages: by Lemma 3.4, each coordinate-wise CWTM update admits a row-stochastic equivalent matrix that assigns zero weight to compromised incoming information, so arbitrary corrupted magnitudes do not directly enter the retained coordinate-wise average. This communication-step property, however, does not by itself prove boundedness of the full optimization trajectory, since RESIST also applies local gradient updates. Assumption 4.12 is therefore used to keep the iterates in a common compact set where the gradient, perturbation, and residual quantities used in the analysis remain controlled. Appendix E.1 gives a restricted construction under additional structural conditions in which this assumption is satisfied; that construction should be read as a non-vacuity example rather than as a verification for every time-varying MITM attack sequence.

Finally, one possible way to enforce boundedness by construction is to modify RESIST so that each post-gradient iterate is projected onto a prescribed compact convex set that contains the intended operating region. Such a projected variant would change the algorithm, and the theorems below do not cover its convergence analysis because projection and the time-varying CWTM averaging operators need not commute. Replacing Assumption 4.12 by such an explicit safeguard, or removing it without an explicit safeguard, would therefore require separate analysis and is left for future work.

We now derive the convergence rates for RESIST under different classes of loss functions.

## 5 Convergence Analysis of RESIST Under Strong Convexity

We start this section by formally stating the strong convexity assumption on the local functions.

**Assumption 5.1.** For all $j \in \{1, \ldots, M\}$, the function $f_j : \mathbb{R}^d \to \mathbb{R}$ is $\mu$-strongly convex; i.e., the function $\mathbf{w} \mapsto f_j(\mathbf{w}) - \frac{\mu}{2}\|\mathbf{w}\|^2$ is convex on $\mathbb{R}^d$.

Although Assumption 5.1 of strong convexity is stronger than the usual convexity assumption with $\mu = 0$, we would like to emphasize that the loss functions in the ERM problem (4) under consideration are often strongly convex due to some form of added regularity (e.g., ridge regression). Also, in practice, while training the model over convex losses, one can easily add an $\ell_2$ regularization to satisfy the strong convexity assumption.

We now state an important property of strongly convex smooth functions.

**Lemma 5.2** ((Boyd & Vandenberghe, 2004))**.** *For any function $g$ on a finite dimensional Euclidean space that is $\mu$-strongly convex and $L$-gradient Lipschitz continuous, we have that for any $\mathbf{x}, \mathbf{y} \in \mathbb{R}^d$:*

$$\langle \nabla g(\mathbf{x}) - \nabla g(\mathbf{y}), \mathbf{x} - \mathbf{y} \rangle \geqslant \frac{\mu L}{\mu + L} \|\mathbf{x} - \mathbf{y}\|^2 + \frac{1}{\mu + L} \|\nabla g(\mathbf{x}) - \nabla g(\mathbf{y})\|^2. \tag{39}$$

Using Lemma 5.2, we can obtain the following contraction type bound on the error $\xi_{\mathbf{w}*}^6(s)$.

**Lemma 5.3.** *Under Assumptions 3.3, 4.8 and 5.1, the sequence $\{\hat{\mathbf{w}}^s(s)\}_s$ for any $h \in (0, \frac{2}{\mu+L})$ satisfies:*

$$\xi_{\mathbf{w}*}^6(s+1) \leqslant (1 - \mu h)\xi_{\mathbf{w}*}^6(s) + \|\mathbf{e}_1(s)\| + Lh\sqrt{Md} \sum_{k=1}^{d} \xi_k^1(s), \tag{40}$$

*where $\mathbf{e}_1(s)$ is defined in Lemma 4.11.*

The proof of this lemma is in Appendix E.2. Observe that using Lemma 5.3 recursively for all $s$, we can obtain geometric decay rates for the error $\xi_{\mathbf{w}*}^6(s)$ but up to some residual error terms that depend on $\sup_s \|\mathbf{e}_1(s)\|$ and also a series sum involving $\xi_k^1(s)$. Also, from Lemmas 4.6 and 4.7, we will have geometric decay of $\xi_k^1(s)$

and $\xi_k^5(s)$, respectively, up to some error terms involving $\xi_k^2(s)$, which again is controlled by Lemma 4.9. Now our goal is to derive a geometric decay rate that is uniform across $\xi_k^1(s), \xi_k^5(s), \xi_{\mathbf{w}*}^6(s)$ and for which the residual error terms only involve $\sup_s \|\mathbf{e}_1(s)\|$. To do so, we make use of tools from linear control systems theory and construct a vector recursion of the form

$$\mathbf{g}(s+1) \leqslant \mathbf{M}\,\mathbf{g}(s) + \boldsymbol{\epsilon}(s),$$

where the entries of the vector $\mathbf{g}(s)$ would comprise of $\xi_k^1(s), \xi_k^5(s), \xi_{\mathbf{w}*}^6(s)$ and the residual error vector $\boldsymbol{\epsilon}(s)$ depends only on $\|\mathbf{e}_1(s)\|$. The entries of matrix $\mathbf{M}$ are determined from Lemmas 4.6, 4.7, 4.9 and 5.3. Then, with a spectral radius of the matrix $\mathbf{M}$ less than 1, we obtain geometric decay of $\mathbf{g}(s)$ with respect to some norm and a residual error that depends on $\sup_s \|\mathbf{e}_1(s)\|$. The next lemma describes this recursion:

**Lemma 5.4.** *Under Assumptions 3.3, 4.8 and 5.1, the vectors $\mathbf{g}(s), \boldsymbol{\epsilon}(s)$ satisfy the following inexact recursion:*

$$\mathbf{g}(s+1) \leqslant \mathbf{M}(h, J)\mathbf{g}(s) + \boldsymbol{\epsilon}(s), \tag{41}$$

*where $\mathbf{M}(h, J) = \mathbf{M}_0 + \mathbf{P}(h, J)$ for some diagonal matrix $\mathbf{M}_0$ and a perturbation matrix $\mathbf{P}(h, J)$ whose entries depend linearly on $h$ which is given explicitly in Appendix E.3, and vectors $\mathbf{g}(s), \boldsymbol{\epsilon}(s)$ are defined as:*

$$\mathbf{g}(s)^T := \begin{bmatrix} \xi_1^1(s) & \xi_1^5(s) & \xi_2^1(s) & \xi_2^5(s) & \cdots & \cdots & \cdots & \xi_d^1(s) & \xi_d^5(s) & \xi_{\mathbf{w}*}^6(s) \end{bmatrix}, \tag{42}$$

$$\boldsymbol{\epsilon}(s)^T := \begin{bmatrix} a_2 h\Delta & a_4 h\Delta & a_2 h\Delta & a_4 h\Delta & \cdots & \cdots & \cdots & a_2 h\Delta & a_4 h\Delta & h\gamma(s) \end{bmatrix}, \tag{43}$$

*where $a_2 := (\sqrt{M} + 1)^2 L$, $a_4 := L$, $\Delta := \sum_{i=1}^M \|\mathbf{w}^* - \mathbf{w}_i^*\|$ with $\mathbf{w}^*, \mathbf{w}_i^*$ defined from Lemma 4.9 and $\gamma(s)$ satisfies the bound:*

$$\|\mathbf{e}_1(s)\| \leqslant h \sum_{k=1}^d |\nabla_k f(\hat{\mathbf{w}}^s(s)) - \nabla_k f^{k,s+1}(\hat{\mathbf{w}}^s(s))| = h\gamma(s), \tag{44}$$

*where the inexact averaged function $f^{k,s+1}(\cdot)$ is defined from Lemma 4.11.*

The proof of Lemma 5.4 and the exact expressions for the matrices $\mathbf{M}_0, \mathbf{P}(h, J)$ are given in Appendix E.3. Note that the matrix $\mathbf{M}(h, J)$ is expressed as a sum of a diagonal matrix $\mathbf{M}_0$ and a perturbation matrix $\mathbf{P}(h, J)$ so as to approximate the spectral radius of matrix $\mathbf{M}(h, J)$ in terms of the spectral radius of $\mathbf{M}_0$.

## 5.1 Convergence analysis of RESIST in $s$-time scale

We now present the convergence rates in $s$-time scale for RESIST in Algorithm 1 on strongly convex loss functions. When the local objectives are heterogeneous, these bounds are residual-neighborhood guarantees; exact convergence is recovered only when the residual quantities appearing below vanish, as in the homogeneous-objective case.

**Theorem 5.5.** *Under Assumptions 3.3, 4.8, 4.12 and 5.1, for any sufficiently small $h > 0$ and for any $J > \frac{\tau M \log(2M^{\frac{3}{2}}(\sqrt{M}+1))}{\log(1-\beta^{\tau M})^{-1}} + \tau M + 2$ :*

- *The inexact recursion from Lemma 5.4 has the following geometric rate to a $\mathcal{O}(C_0 + \Delta)$ ball for any $S > 1$ and a positive constant $C_0$:*

$$\|\mathbf{g}(S)\|_{\mathbf{M}(h,J)} \lesssim_{\mathbf{M}(h,J)} \left(\rho(\mathbf{M}(h, J))\right)^S \|\mathbf{g}(0)\| + \frac{(C_0 + \Delta)}{\mu - \epsilon}, \tag{45}$$

*where $C_0 := \sup_{s \geqslant 0} \sum_{k=1}^d |\nabla_k f(\hat{\mathbf{w}}^s(s)) - \nabla_k f^{k,s+1}(\hat{\mathbf{w}}^s(s))|$, $\Delta$ is defined in Lemma 5.4, $0 < \epsilon < \mu$, $\rho(\mathbf{M}(h, J)) \leqslant 1 - (\mu - \epsilon)h$ and $\|\cdot\|_{\mathbf{M}(h,J)}$ is a vector norm compatible to the matrix norm $\|\|\cdot\|\|_{\mathbf{M}(h,J)}$ for matrix $\mathbf{M}(h, J)$ such that $\|\|\mathbf{M}(h, J)\|\|_{\mathbf{M}(h,J)} = \rho(\mathbf{M}(h, J)) < 1$.*

- *Further, with the aid of Assumption 4.12, for any sufficiently small $h$ and some constant $C_1 > 0$, the consensus error sequences $\{\xi_k^1(s)\}_s, \{\xi_k^5(s)\}_s$ for any $k$ have the following geometric rates to a $\mathcal{O}(h)$ ball for any $S > 1$:*

$$\xi_k^1(S) \leqslant (a_1)^S \xi_k^1(0) + \frac{h}{1-a_1}\left(a_2\sqrt{M}(\sqrt{M}+1)C_1\,diam(\mathcal{K}) + a_2\Delta\right), \tag{46}$$

$$\xi_k^5(S) \leqslant (a_3)^S \xi_k^5(0) + \frac{h}{1-a_3}\left(a_4\sqrt{M}(\sqrt{M}+1)C_1\,diam(\mathcal{K}) + a_4\Delta\right), \tag{47}$$

*where $a_1 = M^{\frac{3}{2}}(\sqrt{M}+1)(1-\beta^{\tau M})^{\left\lfloor\frac{(J-2)}{\tau M}\right\rfloor}$ and $a_3 = M^{\frac{3}{2}}(1-\beta^{\tau M})^{\left\lfloor\frac{(J-2)}{\tau M}\right\rfloor}$ with $a_1 < 1$, $a_3 < 1$. Also, the averaged iterate error sequence $\{\xi_{\mathbf{w}*}^6(s)\}_s$ has the following geometric rate to a $\mathcal{O}(C_0 + h)$ ball for any $S > S_0$ where $S_0 \geqslant 1$ :*

$$\xi_{\mathbf{w}*}^6(S) \leqslant (1-\mu h)^{S-S_0}\xi_{\mathbf{w}*}^6(S_0) + \frac{C_0}{\mu} + \frac{L\sqrt{M}d}{\mu}\bigg((a_1)^{S_0}\xi_k^1(0)$$
$$+ \frac{h}{1-a_1}\left(a_2\sqrt{M}(\sqrt{M}+1)C_1\,diam(\mathcal{K}) + a_2\Delta\right)\bigg). \tag{48}$$

The proof of this theorem is in Appendix E.4. Note that the constants resulting from the "$\lesssim_{\mathbf{M}(h,J)}$" symbol are uniformly bounded for any sufficiently small $h \in [0, \frac{2}{\mu+L}]$. In particular, these constant terms are equal to the product $\|\mathbf{U}^{-1}\|\|\mathbf{U}\|$ where $\mathbf{M} = \mathbf{U}\Lambda\mathbf{U}^{-1}$ is the eigendecomposition of $\mathbf{M}(h, J)$. Since the matrix $\mathbf{U}$ is an $\mathcal{O}(h)$ perturbation of the eigenbasis for $\mathbf{M}_0$ from matrix perturbation theory, the uniform boundedness of the constants follows. In Theorem 5.5, for $\rho(\mathbf{M}(h,J)) \leqslant 1 - (\mu - \epsilon)h$, one usually doesn't have the control of $\mu$ but only has control of the stepsize $h$. To make this quantity small for faster convergence, one can only choose a large stepsize $h$. However, $h$ has a strict upper bound of $\frac{2}{L}$ to achieve convergence. On the other hand, in (46) and (47), when $M$ is large, we can always choose a large enough $J$ such that the quantity $a_1$ can be made small enough for faster convergence. This explains that the second part of Theorem 5.5 provides an improved geometric rate over the first part. Additionally, (46) and (47) give the guarantee of convergence to a ball of arbitrarily small radius by choosing small enough $h$ while in (45), the size of the ball is a constant with respect to $h$. The $C_0$ term measures the gradient gaps between exact and inexact averaging of local functions, and the $\Delta$ term captures the sum of the gaps between the minima of local functions and the minima of the averaged functions across the nodes. Both terms are controlled by discrepancies among the local objectives on the relevant compact set; they become small when the local functions are close in the corresponding $L^\infty$ sense, but need not be small without such structure.

**Corollary 5.6.** *Under the assumptions of Theorem 5.5, for any sufficiently small $h$ and for any $J > \frac{\tau M \log(2M^{\frac{3}{2}}(\sqrt{M}+1))}{\log(1-\beta^{\tau M})^{-1}} + \tau M + 2$, the vector $\mathbf{g}(s)$ satisfies:*

$$\limsup_{S\to\infty} \|\mathbf{g}(S)\| \lesssim_{\mathbf{M}(h,J)} \frac{(C_0 + \Delta)}{\mu - \epsilon}, \tag{49}$$

*for $0 < \epsilon < \mu$. Moreover, the consensus errors $\xi_k^1(S)$, $\xi_k^5(S)$ for any $k$ satisfy:*

$$\limsup_{S\to\infty} \xi_k^1(S) \leqslant \frac{h}{1-a_1}\left(a_2\sqrt{M}(\sqrt{M}+1)C_1\,diam(\mathcal{K}) + a_2\Delta\right), \tag{50}$$

$$\limsup_{S\to\infty} \xi_k^5(S) \leqslant \frac{h}{1-a_3}\left(a_4\sqrt{M}(\sqrt{M}+1)C_1\,diam(\mathcal{K}) + a_4\Delta\right), \tag{51}$$

*and the averaged iterate error $\xi_{\mathbf{w}*}^6(s)$ satisfies:*

$$\limsup_{S\to\infty} \xi_{\mathbf{w}*}^6(S) \leqslant \frac{C_0}{\mu} + \frac{L\sqrt{M}d}{\mu}\left(\frac{h}{1-a_1}\left(a_2\sqrt{M}(\sqrt{M}+1)C_1\,diam(\mathcal{K}) + a_2\Delta\right)\right). \tag{52}$$

The proof of this corollary is in Appendix E.5. From Theorem 5.5 and Corollary 5.6, we get that the consensus errors $\xi_k^1(s)$ and $\xi_k^5(s)$ converge to balls of radii $\frac{h}{1-a_1}\left(a_2\sqrt{M}(\sqrt{M}+1)C_1\text{diam}(\mathcal{K}) + a_2\Delta\right)$ and

$\frac{h}{1-a_3}\left(a_4\sqrt{M}(\sqrt{M}+1)C_1\mathrm{diam}(\mathcal{K})+a_4\Delta\right)$, respectively, at a geometric rate. Also, the averaged iterate error $\xi^6_{\mathbf{w}*}(s)$ converges to a ball of radius $\frac{C_0}{\mu}+\frac{L\sqrt{Md}}{\mu}\left(\frac{h}{1-a_1}\left(a_2\sqrt{M}(\sqrt{M}+1)C_1\mathrm{diam}(\mathcal{K})+a_2\Delta\right)\right)$ with a geometric rate. Though the radii of these balls may appear to be large, we note that the radii of the first two balls for the consensus error are controlled by $h$, which can be made sufficiently small by choosing a corresponding small $h$. In the case of averaged iterate error $\xi^6_{\mathbf{w}*}(s)$, the radius of the ball is controlled by $C_0$ and $h$, where the $h$ dependent term can also be made sufficiently small by choosing a corresponding small $h$.

If the local functions are identical, i.e., $f_i = f_j$ for all $i, j \in \mathcal{N}, i \neq j$, then from the definition of $C_0, \Delta$ in Theorem 5.5, the exact and inexact averaging coincide and hence $C_0 = \Delta = 0$. Then as a direct consequence of the first part of Corollary 5.6, we have $\lim_{S\to\infty}\|\mathbf{g}(S)\| = 0$. Therefore, for any $k$, from the definition of the state vector $\mathbf{g}(s)$ in (180), the consensus errors vanish asymptotically, i.e., $\lim_{S\to\infty}\xi^1_k(S) = 0$ and $\lim_{S\to\infty}\xi^5_k(S) = 0$, and the averaged iterate error also vanishes asymptotically, i.e., $\lim_{S\to\infty}\xi^6_{\mathbf{w}*}(S) = 0$. In the more realistic case of heterogeneous local loss functions, where $f_i \neq f_j$ for some $i \neq j$, the quantities $C_0$ and $\Delta$ capture the discrepancy among local objectives. In Sec. 5.3, we make this dependence explicit by bounding $C_0 + \Delta$ in terms of local gradient discrepancies and other problem constants.

In contrast to (45), Corollary 5.6 together with (46), (47), and (48) provide refined bounds on the consensus and averaged iterate errors. The vector recursion in Theorem 5.5 guarantees geometric convergence of $\|\mathbf{g}(s)\|$ to a ball whose radius depends on $C_0 + \Delta$, reflecting the residual bias induced by heterogeneity of the local loss functions. In contrast, the component-wise analysis shows that $\xi^1_k(s)$ and $\xi^5_k(s)$ converge geometrically up to a $\mathcal{O}(h)$ ball, while $\xi^6_{\mathbf{w}*}(s)$ converges geometrically up to a $\mathcal{O}(h + C_0)$ ball. The quantities $C_0$ and $\Delta$ depend explicitly on discrepancies among local objectives and therefore cannot generally be reduced without additional structural assumptions. Consequently, the limiting neighborhood in the bound for $\|\mathbf{g}(s)\|$ may be bounded away from zero in practice. However, since $h$ can be chosen arbitrarily small, the $\mathcal{O}(h)$ contribution can be controlled, and thus the consensus errors $\xi^1_k(s), \xi^5_k(s)$ can still be made arbitrarily small even when the averaged iterate error $\xi^6_{\mathbf{w}*}(s)$ remains influenced by the heterogeneity term $C_0$.

## 5.2 Convergence analysis of RESIST in $t$-time scale

We now present the $t$-time scale convergence analysis of RESIST. As in the $s$-time scale, the guarantees in this section are residual-neighborhood guarantees under heterogeneous local objectives; exact convergence is recovered when the residual quantities in the corresponding bounds vanish. We first define the coordinate-wise inexact averaged vector used in this time scale.

**Definition 5.7.** The coordinate-wise inexact averaged vector for the $t$-time scale, where $sJ \leqslant t < sJ+J-2$, is defined as

$$\hat{\mathbf{w}}^s(t) = \begin{bmatrix} \sum_{j=1}^M [\mathbf{c}_1(s)]_j[\mathbf{w}_j(t)]_1 \\ \sum_{j=1}^M [\mathbf{c}_2(s)]_j[\mathbf{w}_j(t)]_2 \\ \vdots \\ \sum_{j=1}^M [\mathbf{c}_k(s)]_j[\mathbf{w}_j(t)]_k \\ \vdots \\ \sum_{j=1}^M [\mathbf{c}_d(s)]_j[\mathbf{w}_j(t)]_d \end{bmatrix}, \tag{53}$$

where the weights $[\mathbf{c}_k(s)]_j$ for any $k, j$ follow from Corollary 4.1, and we have $\widehat{\mathbf{W}}^s(t) = \mathbf{1}(\hat{\mathbf{w}}^s(t))^T$. Also, $\mathbf{W}^* = \mathbf{1}(\mathbf{w}^*)^T$, where $\mathbf{w}^* := \arg\min_{\mathbf{w}} \frac{1}{M}\sum_{j=1}^M f_j(\mathbf{w})$.

**Theorem 5.8.** *Under Assumptions 3.3, 4.8, 4.12 and 5.1, if $J > \frac{\tau M \log(2M^{\frac{3}{2}}(\sqrt{M}+1))}{\log(1-\beta^{\tau M})^{-1}} + \tau M + 2$ then using Definitions 5.7 :*

- *RESIST (Algorithm 1) for $S = \lfloor \frac{t}{J} \rfloor$ has the following geometric convergence rate (with contraction factor $\rho(h,J)$) to a $\mathcal{O}(C_0 + \Delta)$ radius ball around $\mathbf{W}^*$:*

$$\left\| \mathbf{W}(t) - \overline{\mathbf{W}}(t) \right\|_F + \left\| \mathbf{W}^* - \widehat{\mathbf{W}}^S(t) \right\|_F + \left\| \mathbf{W}(t) - \widehat{\mathbf{W}}^S(t) \right\|_F \lesssim_{\mathbf{M}(h,J)}$$
$$\sqrt{3d}(\sqrt{M}+1)M\left( \left( \rho(\mathbf{M}(h,J)) \right)^{\frac{t}{J}-1} \|\mathbf{g}(0)\| + \frac{h(C_0+\Delta)}{1-\rho(\mathbf{M}(h,J))} \right), \tag{54}$$

*where $\rho(\mathbf{M}(h,J)) \leqslant 1 - (\mu-\epsilon)h < 1$ for any sufficiently small $h$, and $\epsilon = o(\mu) > 0$. Asymptotically, we have that*

$$\limsup_{t\to\infty} \left( \left\| \mathbf{W}(t) - \overline{\mathbf{W}}(t) \right\|_F + \left\| \mathbf{W}^* - \widehat{\mathbf{W}}^S(t) \right\|_F + \left\| \mathbf{W}(t) - \widehat{\mathbf{W}}^S(t) \right\|_F \right) \lesssim_{\mathbf{M}(h,J)}$$
$$\frac{\sqrt{3d}(\sqrt{M}+1)M(C_0+\Delta)}{\mu-\epsilon}. \tag{55}$$

- *RESIST (Algorithm 1), for any $S > S_0$ where $S_0 > 0$, has a faster geometric convergence rate (with contraction factor strictly smaller than $\rho(h,J)$) to a $\mathcal{O}(C_0 + h)$ radius ball around $\mathbf{W}^*$:*

$$\left\| \mathbf{W}(t) - \overline{\mathbf{W}}(t) \right\|_F + \left\| \mathbf{W}^* - \widehat{\mathbf{W}}^S(t) \right\|_F + \left\| \mathbf{W}(t) - \widehat{\mathbf{W}}^S(t) \right\|_F \leqslant$$
$$\sqrt{3d}(\sqrt{M}+1)M\left( d\left( (a_1)^{\frac{t}{J}-1}\xi_k^1(0) + \frac{h}{1-a_1}\left( a_2\sqrt{M}(\sqrt{M}+1)C_1\,diam(\mathcal{K}) + a_2\Delta \right) + \right.\right.$$
$$(a_3)^{\frac{t}{J}-1}\xi_k^5(0) + \frac{h}{1-a_3}\left( a_4\sqrt{M}(\sqrt{M}+1)C_1\,diam(\mathcal{K}) + a_4\Delta \right) \right) + (1-\mu h)^{\frac{t}{J}-1-S_0}\xi_{\mathbf{w}^*}^6(S_0)+$$
$$\left.\frac{C_0}{\mu} + \frac{L\sqrt{Md}}{\mu}\left( (a_1)^{S_0}\xi_k^1(0) + \frac{h}{1-a_1}\left( a_2\sqrt{M}(\sqrt{M}+1)C_1\,diam(\mathcal{K}) + a_2\Delta \right) \right) \right), \tag{56}$$

*where $a_1 < 1$, $a_3 < 1$, and $C_1$ is a constant specified in the proof that depends only on the dimension of the model parameter.*

The proof of this theorem is in Appendix E.6. Note that, from the second bullet of Theorem 5.8, the exact radius of the $\mathcal{O}(C_0 + h)$ ball is given by:

$$\limsup_{t\to\infty} \left( \left\| \mathbf{W}(t) - \overline{\mathbf{W}}(t) \right\|_F + \left\| \mathbf{W}^* - \widehat{\mathbf{W}}^S(t) \right\|_F + \left\| \mathbf{W}(t) - \widehat{\mathbf{W}}^S(t) \right\|_F \right) \leqslant$$
$$\sqrt{3d}(\sqrt{M}+1)M\left( \frac{hd}{1-a_1}\left( a_2\sqrt{M}(\sqrt{M}+1)C_1\mathrm{diam}(\mathcal{K}) + a_2\Delta \right) + \frac{hd}{1-a_3}\left( a_4\sqrt{M}(\sqrt{M}+1)C_1\mathrm{diam}(\mathcal{K}) + a_4\Delta \right) + \right.$$
$$\left. + \frac{C_0}{\mu} + \left( \frac{L\sqrt{Md}}{\mu}\frac{h}{1-a_1}\left( a_2\sqrt{M}(\sqrt{M}+1)C_1\mathrm{diam}(\mathcal{K}) + a_2\Delta \right) \right) \right). \tag{57}$$

### 5.3 Implications of Theorems 5.5 and 5.8

In this section, we interpret the convergence guarantees established in Theorems 5.5 and 5.8 by providing explicit bounds on the residual term $C_0 + \Delta$. In particular, we relate this quantity to the dissimilarity of local gradients and show how the convergence radius depends on the heterogeneity of the local loss functions. Beyond these heterogeneity terms, the derived convergence radius also reflects dependence on the dimension $d$, the number of nodes $M$, the stepsize, and the filtered-graph quantities entering the theorems. We first state a general bound on the distance between minimizers of two strongly convex functions in terms of their gradient discrepancy.

**Lemma 5.9.** *For a pair of $\mu$-strongly convex, continuously differentiable functions $f, g : \mathbb{R}^d \to \mathbb{R}$ with minima at $\mathbf{y}_f^*, \mathbf{y}_g^*$, respectively, let $\Omega \subset \mathbb{R}^d$ be a compact set containing both minimizers; for instance, $\Omega$ can be chosen as a closed ball of sufficiently large radius $\theta$. Then we have that $\left\| \mathbf{y}_f^* - \mathbf{y}_g^* \right\| \leqslant \frac{1}{\mu} \left\| \nabla(f - g) \right\|_{L^\infty(\Omega)}$.*[6]

*Proof.* From the fact that $\mathbf{y}_f^*, \mathbf{y}_g^* \in \Omega$ and $\nabla f(\mathbf{y}_f^*) = \nabla g(\mathbf{y}_g^*) = 0$, and by strong convexity, we have:

$$\mu \left\| \mathbf{y}_g^* - \mathbf{y}_f^* \right\| \leqslant \left\| \nabla f(\mathbf{y}_g^*) - \nabla f(\mathbf{y}_f^*) \right\| = \left\| \nabla f(\mathbf{y}_g^*) - \nabla g(\mathbf{y}_g^*) \right\| \leqslant \left\| \nabla(f - g) \right\|_{L^\infty(\Omega)}, \tag{58}$$

which completes the proof. ∎

**Corollary 5.10.** *Under Assumptions 3.3, 4.8, 4.12 and 5.1, suppose there exists a compact set $\Omega_0 \subset \mathbb{R}^d$, chosen as a closed ball of sufficiently large radius, such that $\Omega_0$ contains the global minimizer $\mathbf{w}^*$, the local minimizers $\{\mathbf{w}_j^*\}_{j=1}^M$, and the averaged iterates $\{\hat{\mathbf{w}}^s(s)\}_{s=0}^\infty$. Then we have that:*

$$C_0 + \Delta \leqslant \left( 2d(M-1) + \frac{M}{\mu} \right) \max_{\substack{i,j \in \mathcal{N}; \\ i \neq j}} \left\| \nabla(f_i - f_j) \right\|_{L^\infty(\Omega_0)}, \tag{59}$$

*and the iterate sequence $\{\mathbf{w}_j(t)\}_t$ for any $j \in \mathcal{N}$ from RESIST converges to an $\mathcal{O}(\max_{\substack{i,j \in \mathcal{N}; \\ i \neq j}} \left\| \nabla(f_i - f_j) \right\|_{L^\infty(\Omega_0)})$ neighborhood of $\mathbf{w}^*$ with a geometric rate in $t$ according to Theorem 5.8.*

*Proof.* From the definition of $C_0 = \sup_{s \geqslant 0} \sum_{k=1}^d |\nabla_k f(\hat{\mathbf{w}}^s(s)) - \nabla_k f^{k,s+1}(\hat{\mathbf{w}}^s(s))|$ and $\Delta = \sum_{i=1}^M \left\| \mathbf{w}^* - \mathbf{w}_i^* \right\|$ we can see that:

$$C_0 = \sup_{s \geqslant 0} \sum_{k=1}^d \left| \frac{1}{M} \sum_{i=1}^M \nabla_k f_i(\hat{\mathbf{w}}^s(s)) - \frac{1}{M} \sum_{i=1}^M [\mathbf{c}_k(s+1)]_i \nabla_k f_i(\hat{\mathbf{w}}^s(s)) \right| \tag{60}$$

$$= \sup_{s \geqslant 0} \sum_{k=1}^d \left| \sum_{i=1}^M \left( \frac{1}{M} - [\mathbf{c}_k(s+1)]_i \right) \left( \nabla_k f_i(\hat{\mathbf{w}}^s(s)) - \nabla_k f(\hat{\mathbf{w}}^s(s)) \right) \right| \tag{61}$$

$$= \sup_{s \geqslant 0} \sum_{k=1}^d \left| \sum_{i=1}^M \left( \frac{1}{M} - [\mathbf{c}_k(s+1)]_i \right) \left( \frac{1}{M} \sum_{l=1}^M \left( \nabla_k f_i(\hat{\mathbf{w}}^s(s)) - \nabla_k f_l(\hat{\mathbf{w}}^s(s)) \right) \right) \right| \tag{62}$$

$$\leqslant \frac{2}{M} \sup_{s \geqslant 0} \sum_{k=1}^d \sum_{i=1}^M \sum_{l=1}^M \left| \nabla_k f_i(\hat{\mathbf{w}}^s(s)) - \nabla_k f_l(\hat{\mathbf{w}}^s(s)) \right| \tag{63}$$

$$\leqslant \frac{2}{M} \sup_{s \geqslant 0} \sum_{k=1}^d \sum_{i=1}^M \sum_{l=1}^M \left\| \nabla f_i(\hat{\mathbf{w}}^s(s)) - \nabla f_l(\hat{\mathbf{w}}^s(s)) \right\| \leqslant 2d(M-1) \max_{\substack{i,j \in \mathcal{N}; \\ i \neq j}} \left\| \nabla(f_i - f_j) \right\|_{L^\infty(\Omega_0)}. \tag{64}$$

Next, we have that:

$$\left\| \nabla(f_i - f) \right\|_{L^\infty(\Omega_0)} = \left\| \nabla \left( f_i - \frac{1}{M} \sum_{l=1}^M f_l \right) \right\|_{L^\infty(\Omega_0)} = \left\| \frac{1}{M} \sum_{l=1}^M \nabla(f_i - f_l) \right\|_{L^\infty(\Omega_0)}$$

$$\leqslant \frac{1}{M} \sum_{l=1}^M \left\| \nabla(f_i - f_l) \right\|_{L^\infty(\Omega_0)}, \tag{65}$$

and thus by Lemma 5.9 applied with $\Omega = \Omega_0$ we have that $\left\| \mathbf{w}^* - \mathbf{w}_i^* \right\| \leqslant \frac{1}{\mu} \max_{\substack{i,j \in \mathcal{N}; \\ i \neq j}} \left\| \nabla(f_i - f_j) \right\|_{L^\infty(\Omega_0)}$ for any $i \in \mathcal{N}$ and hence we have $\Delta \leqslant \frac{M}{\mu} \max_{\substack{i,j \in \mathcal{N}; \\ i \neq j}} \left\| \nabla(f_i - f_j) \right\|_{L^\infty(\Omega_0)}$. Then by substituting $C_0 + \Delta \leqslant \left( 2d(M-1) + \frac{M}{\mu} \right) \max_{\substack{i,j \in \mathcal{N}; \\ i \neq j}} \left\| \nabla(f_i - f_j) \right\|_{L^\infty(\Omega_0)}$ in the bound (54) from Theorem 5.8, the proof is complete. ∎

---

[6]Note that the $\Omega$ used here is different from the $\Omega$ mentioned in Sec. 3.3.

From Corollary 5.10 we can see that an upper bound of $C_0 + \Delta$ is a function of the dissimilarity of local gradients $\|\nabla(f_i - f_j)\|_{L^\infty(\Omega_0)}$. To give an upper bound on the dissimilarity of local gradients $\|\nabla(f_i - f_j)\|_{L^\infty(\Omega_0)}$ and implicitly provide an upper bound for the term $C_0 + \Delta$, we now state an assumption of gradient similarity between the local functions that is often used in the decentralized literature.

**Assumption 5.11** (Bounded gradient similarity (Tyou et al., 2023))**.** We have $\frac{1}{M} \sum_{j=1}^{M} \|\nabla f_j(\mathbf{w})\|^2 \leqslant G^2 + D^2 \|\nabla f(\mathbf{w})\|^2$ for every $\mathbf{w} \in \mathbb{R}^d$ for some $G, D \geqslant 0$, where $f(\mathbf{w}) := \frac{1}{M} \sum_{j=1}^{M} f_j(\mathbf{w})$ denotes the average function.

Assumption 5.11 controls the dissimilarity between local gradients and the averaged gradient. This assumption does not require the local datasets to be i.i.d.; rather, it quantifies the degree of heterogeneity through the constants $G$ and $D$. In particular, when the local datasets are sampled i.i.d. from a common distribution, the gradient dissimilarity is naturally small, leading to smaller values of $G$ and $D$ and hence tighter convergence bounds. Under this assumption with $D < 1$, Corollary 5.10 follows, as shown in the next lemma.

**Lemma 5.12.** *Under Assumptions 3.3, 4.8, 4.12, 5.1 and 5.11 with $D < 1$, Corollary 5.10 is implied with $\Omega_0$ chosen as a sufficiently large closed ball containing the compact set from Assumption 4.12, the minimizers $\mathbf{w}^*, \{\mathbf{w}_j^*\}_{j=1}^{M}$, and the sequence $\{\hat{\mathbf{w}}^s(s)\}_{s=0}^{\infty}$.*

*Proof.* Note that for $D < 1$, by Jensen's inequality, we have the following bound for any $\mathbf{w} \in \mathbb{R}^d$:

$$\|\nabla f(\mathbf{w})\| \leqslant \frac{1}{M} \sum_{j=1}^{M} \|\nabla f_j(\mathbf{w})\| \leqslant \sqrt{\frac{1}{M} \sum_{j=1}^{M} \|\nabla f_j(\mathbf{w})\|^2} \leqslant \sqrt{G^2 + D^2 \|\nabla f(\mathbf{w})\|^2} \leqslant G + D \|\nabla f(\mathbf{w})\| \quad (66)$$

$$\implies \|\nabla f(\mathbf{w})\| \leqslant \frac{G}{1 - D} \quad (67)$$

$$\implies \|\nabla(f_i - f_j)(\mathbf{w})\| \leqslant 2MG\left(1 + \frac{D}{1 - D}\right), \quad (68)$$

where we used $\frac{1}{M} \sum_{j=1}^{M} \|\nabla f_j(\mathbf{w})\| \leqslant G + D \|\nabla f(\mathbf{w})\|$ in the last step. Hence, $\nabla(f_i - f_j) \in L^\infty(\mathbb{R}^d)$ for any $i, j \in \mathcal{N}, i \neq j$, and therefore this bound holds on any compact set. We next note that $\{\hat{\mathbf{w}}^s(s)\}_{s=0}^{\infty}$ is bounded. By Definition 4.3, each coordinate of $\hat{\mathbf{w}}^s(s)$ is a stochastic weighted average of the corresponding coordinates of the regular-node iterates, and those iterates are uniformly bounded by Assumption 4.12. Hence, we may choose a sufficiently large closed ball $\Omega_0$ containing the compact set from Assumption 4.12, the minimizers $\mathbf{w}^*, \{\mathbf{w}_j^*\}_{j=1}^{M}$, and the sequence $\{\hat{\mathbf{w}}^s(s)\}_{s=0}^{\infty}$. Applying Corollary 5.10 with this $\Omega_0$, we obtain

$$C_0 + \Delta \leqslant \left(2d(M - 1) + \frac{M}{\mu}\right) \max_{\substack{i,j \in \mathcal{N}; \\ i \neq j}} \|\nabla(f_i - f_j)\|_{L^\infty(\Omega_0)} \leqslant 2MG\left(2d(M - 1) + \frac{M}{\mu}\right)\left(1 + \frac{D}{1 - D}\right). \quad (69)$$

$\blacksquare$

We now discuss the geometric convergence guarantee in Theorem 5.8 to a $\mathcal{O}(C_0 + \Delta)$ ball around $\mathbf{w}^*$: the theorem does not guarantee convergence to the exact global minimizer $\mathbf{w}^*$, nor does it guarantee asymptotic consensus. Moreover, as $t \to \infty$, the iterate matrix $\mathbf{W}(t)$ can only be within a $\mathcal{O}(C_0 + \Delta)$ ball around $\mathbf{W}^*$, whose radius is upper bounded as in Lemma 5.12. The preceding corollary and lemma separate two related points. Corollary 5.10 first bounds $C_0 + \Delta$ by the local-gradient discrepancy on the common compact set $\Omega_0$, namely by $\max_{i \neq j} \|\nabla(f_i - f_j)\|_{L^\infty(\Omega_0)}$, together with the explicit dimension and network-size factors in the corollary. Lemma 5.12 then gives a sufficient bounded-gradient-similarity condition, through the constants $G, D$, under which Corollary 5.10 holds and yields an explicit bound on $C_0 + \Delta$. Substituting these bounds into Theorem 5.8 gives a residual neighborhood whose size also depends on the stepsize and filtered-graph constants appearing in that theorem. Thus, in the strongly convex deterministic setting, RESIST converges geometrically to a residual neighborhood whose size is controlled by these quantities. The radius becomes

small when the local gradients are close on $\Omega_0$, and it vanishes when the local gradients coincide on $\Omega_0$, in particular when all local objectives are the same.

Next, we relate this RESIST residual bound to the corresponding REDGRAF analysis of Kuwaranancharoen & Sundaram (2025), discussed in the related-work subsection of Sec. 1. That analysis establishes geometric convergence to characterized neighborhoods for Byzantine-resilient decentralized optimization algorithms under its stated assumptions. Its CWTM instance is the fixed-stepsize comparison point for RESIST with $J = 2$: in this specialization, both perform one CWTM filtering round followed by one local gradient update, although REDGRAF studies Byzantine-node attacks whereas RESIST studies link-level MITM attacks. For REDGRAF-CWTM, the neighborhood radius is controlled by the spread of the regular nodes' local minimizers, and the zero-radius specialization occurs in the theorem regime where the corresponding conditioning and stepsize requirements hold; as discussed in Sec. 1, this regime effectively requires the relevant condition number to be close to one in high-dimensional models. In the RESIST analysis, Corollary 5.10, with the sufficient condition in Lemma 5.12, instead expresses the residual through the local-gradient discrepancy on $\Omega_0$ and the MITM-induced inexact averaging terms. The distinctions relevant to the present strongly convex discussion are therefore the link-level MITM model, the use of $J > 2$ CWTM rounds between local gradient updates, and the residual formulation through local-gradient discrepancies and MITM-induced inexact averaging. Additional distinctions from REDGRAF, namely the Polyak–Łojasiewicz, smooth nonconvex, and statistical learning-rate guarantees, are developed in the subsequent sections.

Note that, up to this point, the convergence analysis in this section relies on Assumption 5.1, which requires the local loss functions to be strongly convex. However, this assumption may fail in modern machine learning applications that employ deep neural networks on complex datasets such as CIFAR-10, CIFAR-100, and ImageNet. In the next section, we provide convergence guarantees for RESIST without Assumption 5.1, covering certain classes of nonconvex loss functions.

# 6 Convergence Analysis of RESIST Under Nonconvexity

We now move beyond the strongly convex setting of Sec. 5. In this section, Assumption 5.1 is no longer imposed; instead, the loss functions satisfy the smoothness condition in Assumption 4.8, and any additional structural conditions are introduced only where they are needed. We analyze two cases: averaged objectives satisfying the Polyak–Łojasiewicz condition and general smooth nonconvex objectives. As before, the results are stated on the $s$-time scale; the corresponding $t$-time statements can be recovered by applying the same conversion used in Theorem 5.8.

## 6.1 Convergence analysis of RESIST for Polyak–Łojasiewicz (PŁ) functions

One common class of nonconvex loss functions is the Polyak-Łojasiewicz (PŁ) class, which includes widely used objectives under appropriate regularity or regularization conditions, such as least-squares objectives and regularized logistic losses. Functions satisfying the PŁ inequality have the property that the gradient norm grows proportionally to the square root of the function suboptimality, as described in the following assumption.

**Assumption 6.1.** The averaged function $f := \frac{1}{M} \sum_{j=1}^{M} f_j$ satisfies the Polyak-Łojasiewicz (PŁ) inequality (Lojasiewicz, 1963) with parameter $\mu \in (0, L)$, i.e., for any $\mathbf{w} \in \mathbb{R}^d$ we have:

$$\frac{1}{2\mu} \left\| \nabla f(\mathbf{w}) \right\|^2 \geqslant f(\mathbf{w}) - f^* \tag{70}$$

where $f^* := \min_{\mathbf{w} \in \mathbb{R}^d} f(\mathbf{w})$.

Note that in Assumption 6.1, the PŁ inequality is required only for the averaged function $f$, rather than for each local function $f_i$. This is consistent with the Kurdyka-Łojasiewicz (KŁ) assumption (a more general form of the PŁ condition) imposed on the averaged loss function in Zeng & Yin (2018), where DGD is used for decentralized optimization. Moreover, individual PŁ inequalities for the local functions $f_i$ do not necessarily imply a PŁ inequality for the averaged function $f$, in contrast to convexity, where the average of convex functions remains convex (see Appendix F.1 for an illustrative example).

To proceed with the analysis, we additionally impose the following assumption.

**Assumption 6.2.** Let $\mathcal{K}$ be the compact set from Assumption 4.12. We assume that $\mathcal{K}$ is sufficiently large such that $\arg\min_{\mathbf{w}} f_i(\mathbf{w}) \cap \mathcal{K} \neq \varnothing$ for all $i \in \{1, \cdots, M\}$ and $\arg\min_{\mathbf{w}} f(\mathbf{w}) \cap \mathcal{K} \neq \varnothing$.

Assumption 6.2 specifies an additional requirement on the compact set from Assumption 4.12. Assumption 4.12 bounds the regular-node iterates in a compact set; Assumption 6.2 further requires this set to contain at least one minimizer of each local objective and one minimizer of the averaged objective. Lemma 6.3 and Theorem 6.4 use this common set through its diameter. In addition, the consensus-error bounds in Theorem 6.4 involve distances of the form $\|\mathbf{w}^* - \mathbf{w}_i^*\|$ between a minimizer of the averaged objective and minimizers of the local objectives. Thus, Assumption 6.2 fixes the compact domain used in this PŁ subsection, while the boundedness of the iterates remains the condition imposed by Assumption 4.12.

**Lemma 6.3.** *Under Assumptions 3.3, 4.8, 4.12, 6.1, and 6.2, with the compact set $\mathcal{K}$ from Assumption 4.12 having diameter $diam(\mathcal{K})$, the function sequence $\{f(\widehat{\mathbf{w}}^s(s))\}_s$, for any $h \in (0, \frac{2}{L})$, satisfies:*

$$f(\widehat{\mathbf{w}}^{s+1}(s+1)) - f^* \leqslant \left(1 - \mu h(2 - Lh)\right)\left(f(\widehat{\mathbf{w}}^s(s)) - f^*\right) +$$
$$L \ diam(\mathcal{K})\left(\|\mathbf{e}_1(s)\| + Lh\sqrt{Md}\sum_{k=1}^{d}\left\|[\widehat{\mathbf{W}}^{k,s}(s)]_k - [\mathbf{W}(s)]_k\right\|\right),$$
(71)

*where $\mathbf{e}_1(s)$ is defined in Lemma 4.11.*

The proof of this lemma is given in Appendix F.2.

**Theorem 6.4.** *Under Assumptions 3.3, 4.8, 4.12, 6.1, and 6.2, for the compact set $\mathcal{K}$ from Assumptions 4.12 and 6.2 with diameter $diam(\mathcal{K})$, and for any $h \in (0, \frac{2}{L})$, there exists a constant $C_1$ depending only on d such that, for any*

$$J > \frac{\tau M \log\left(2M^{\frac{3}{2}}(\sqrt{M} + 1)\right)}{\log\left((1 - \beta^{\tau M})^{-1}\right)} + \tau M + 2,$$

*the consensus error sequences $\{\xi_k^1(s)\}_s$ and $\{\xi_k^5(s)\}_s$, for each coordinate k, converge geometrically to an $\mathcal{O}(h)$ neighborhood for any $S > 1$:*

$$\xi_k^1(S) \leqslant (a_1)^S \xi_k^1(0) + \frac{h}{1 - a_1}\left(a_2\sqrt{M}(\sqrt{M} + 1)C_1 diam(\mathcal{K}) + a_2\Delta\right),$$
(72)

$$\xi_k^5(S) \leqslant (a_3)^S \xi_k^5(0) + \frac{h}{1 - a_3}\left(a_4\sqrt{M}(\sqrt{M} + 1)C_1 diam(\mathcal{K}) + a_4\Delta\right),$$
(73)

*where $a_1 < 1$ and $a_3 < 1$.*

*Moreover, the function error sequence $\{f(\widehat{\mathbf{w}}^s(s)) - f^*\}_s$ converges geometrically to an $\mathcal{O}(C_0 + h)$ neighborhood:*

$$f(\widehat{\mathbf{w}}^S(S)) - f^* \leqslant \left(1 - \mu h(2 - Lh)\right)^S\left(f(\widehat{\mathbf{w}}^0(0)) - f^*\right) + L \ diam(\mathcal{K})\frac{C_0}{\mu(2 - Lh)}$$
$$+ \frac{L^2hd\sqrt{Md}}{1 - a_1}\left(diam(\mathcal{K})\right)^2\left(\frac{(\sqrt{M} + 1)^2}{\mu(2 - Lh)}LM(\sqrt{d} + 2) + M\right),$$
(74)

*where $C_0$ is the constant defined in Theorem 5.5.*

The proof of this theorem is provided in Appendix F.3. Unlike Theorem 5.5 for the strongly convex case, where the convergence rates are expressed in terms of iterate distances, Theorem 6.4 retains norm bounds for the consensus-error sequences but states the optimization guarantee in function value. Specifically, the consensus-error sequences converge geometrically to $\mathcal{O}(h)$ neighborhoods, while the function-value error converges geometrically to an $\mathcal{O}(C_0 + h)$ neighborhood. To the best of our knowledge, this is the first work establishing such a geometric residual-neighborhood guarantee for the PŁ function class in a decentralized setting under adversarial link-level MITM attacks.

## 6.2 Convergence Analysis of RESIST for Smooth Nonconvex Functions

Functions satisfying the PŁ inequality form a broad class that includes several common learning objectives under appropriate regularity or regularization conditions, such as least-squares objectives and regularized logistic losses. However, many modern models, including convolutional neural networks (CNNs) and deep neural networks (DNNs), lead to smooth nonconvex loss functions that do not necessarily satisfy the PŁ inequality. In such settings, the gradient norm no longer directly controls the function suboptimality, which makes optimization substantially more challenging. Consequently, to apply RESIST in these cases, we require convergence guarantees for general smooth nonconvex objectives. To establish these rates, we first state the following lemma.

**Lemma 6.5** (Hölder inequality for sums (Rudin, 1987)). *Let $\{a_s\}$ and $\{b_s\}$ be sequences of complex numbers indexed by $s \in E$, where $E$ is a finite or infinite index set. Then the following Hölder inequality holds:*

$$\left| \sum_{s \in E} a_s b_s \right| \leqslant \left( \sum_{s \in E} |a_s|^v \right)^{\frac{1}{v}} \left( \sum_{s \in E} |b_s|^q \right)^{\frac{1}{q}}, \tag{75}$$

*where $v > 1$ and $\frac{1}{v} + \frac{1}{q} = 1$.*

**Theorem 6.6.** *Under Assumptions 3.3, 4.8, 4.12, and 6.2, where $\mathcal{K}$ is the compact set in Assumptions 4.12 and 6.2, let $h = h(s) = \frac{p}{(s+1)^\omega}$ be a decaying stepsize with $p, \omega > 0$. For any $J > \frac{\tau M \log(2M^{\frac{3}{2}}(\sqrt{M}+1))}{\log((1-\beta^{\tau M})^{-1})} + \tau M + 2$, the consensus error sequences $\{\xi_k^1(s)\}_s, \{\xi_k^5(s)\}_s$ for any $k$ converge to $0$ at the rate*

$$\xi_k^1(S) = \mathcal{O}\left( \frac{1}{S^\omega} \right), \tag{76}$$

$$\xi_k^5(S) = \mathcal{O}\left( \frac{1}{S^\omega} \right). \tag{77}$$

*Moreover, if $h(s) = \frac{p}{(s+1)^\omega}$ with $\omega = \frac{1}{2} + \epsilon$ for any $0 < \epsilon < 1/2$ and $0 < p \leqslant \frac{1}{2L}$, then*

$$\min_{0 \leqslant s \leqslant S-1} \left\| \nabla f(\hat{\mathbf{w}}^s(s)) \right\|^2 \leqslant \frac{f(\hat{\mathbf{w}}^0(0)) - \inf_{\mathbf{w}} f(\mathbf{w})}{p S^{\frac{1}{2}-\epsilon}} + \frac{C_6}{S^{\frac{1}{2}-\epsilon}}$$
$$+ 2L\,diam(\mathcal{K})C_0 + \frac{2C_4 L^2 d\sqrt{M}d(diam(\mathcal{K}))^2}{S^{\frac{1}{2}-\epsilon}}, \tag{78}$$

*and*

$$\limsup_{S \to \infty} \min_{0 \leqslant s \leqslant S-1} \left\| \nabla f(\hat{\mathbf{w}}^s(s)) \right\|^2 \leqslant 2L\,diam(\mathcal{K})C_0, \tag{79}$$

*where $C_0 = \sup_{s \geqslant 0} \sum_{k=1}^d |\nabla_k f(\hat{\mathbf{w}}^s(s)) - \nabla_k f^{k,s+1}(\hat{\mathbf{w}}^s(s))|$, $C_4 = \mathcal{O}\big(M^2(1+p)(Ld\,diam(\mathcal{K}))^3\big)$, and $C_6 = \mathcal{O}\big(pL^3(Md\,diam(\mathcal{K}))^2\big)$.*

The proof of this theorem is given in Appendix F.4. For the decaying stepsize choice $h(s) = \frac{p}{(s+1)^{0.5+\epsilon}}$, Theorem 6.6 yields the sub-linear rate $\mathcal{O}(S^{-0.5+\epsilon})$ for the stationarity measure $\min_{0 \leqslant s \leqslant S-1} \|\nabla f(\hat{\mathbf{w}}^s(s))\|^2$ up to a residual $\mathcal{O}(C_0)$ term. This scaling matches the classical $S^{-1/2}$ rate for first-order methods in smooth nonconvex optimization under decaying stepsizes, including centralized stochastic gradient methods (Imaizumi & Iiduka, 2024). Such rates are known to be optimal (up to constants) in the standard first-order oracle model. We emphasize that the use of decaying stepsizes in this general smooth nonconvex setting is not due to stochasticity of the algorithm itself, but rather to the presence of persistent inexactness in the gradient updates induced by attacks and coordinate-wise screening. In the absence of structural conditions such as strong convexity or the PŁ inequality, decaying stepsizes ensure that the accumulated error terms remain controlled and that convergence to a neighborhood of a first-order stationary point is achieved. In particular, when $C_0 = \mathcal{O}(\delta)$, Theorem 6.6 guarantees convergence of the iterates to a $\delta$-neighborhood of a first-order stationary point. In the ERM formulation (3), we later show in Theorem 8.4 that $C_0 = \mathcal{O}(\frac{1}{\sqrt{N}})$

with high probability when each node has $N$ local samples. Hence, with sufficiently large $N$, near first-order stationarity is achieved with high probability. Establishing second-order optimality guarantees in the nonconvex setting is substantially more challenging, as it requires controlling escape from saddle points (Dixit et al., 2022; 2023), and is therefore left for future work.

The above analysis provides asymptotic convergence guarantees under decaying stepsizes. In practice, however, optimization algorithms are often run for a finite time horizon with a fixed stepsize. Motivated by this perspective, and in line with recent finite-time analyses such as (Wu et al., 2023), we now provide a non-asymptotic convergence guarantee under smooth nonconvex loss functions with a constant stepsize.

**Theorem 6.7.** *Under Assumptions 3.3, 4.8, 4.12, and 6.2, where $\mathcal{K}$ is the compact set in Assumptions 4.12 and 6.2, suppose RESIST is iterated for $S$ gradient steps with constant stepsize $h = \frac{1}{\sqrt{S}}$, and assume that $S > L^6(Md\,diam(\mathcal{K}))^4$. Then for any $J > \frac{\tau M \log(2M^{\frac{3}{2}}(\sqrt{M}+1))}{\log((1-\beta^{\tau M})^{-1})} + \tau M + 2$, the consensus errors $\xi_k^1(s), \xi_k^5(s)$ for any $k$ and any $s \leqslant S$ satisfy*

$$\xi_k^1(s) = \mathcal{O}\left((a_1)^s + \frac{1}{\sqrt{S}}\right), \tag{80}$$

$$\xi_k^5(s) = \mathcal{O}\left((a_3)^s + \frac{1}{\sqrt{S}}\right), \tag{81}$$

*where $a_1 < 1$ and $a_3 < 1$. Moreover, the gradient sequence $\{\nabla f(\hat{\mathbf{w}}^s(s))\}_{s=0}^{S-1}$ satisfies*

$$\frac{1}{S}\sum_{s=0}^{S-1}\|\nabla f(\hat{\mathbf{w}}^s(s))\|^2 \leqslant \left(1 - \frac{L}{\sqrt{S}}\right)^{-1}\frac{f(\hat{\mathbf{w}}^0(0)) - \inf_{\mathbf{w}} f(\mathbf{w})}{\sqrt{S}} + \frac{C_9}{\sqrt{S}}$$
$$+ \left(1 - \frac{L}{\sqrt{S}}\right)^{-1} L\,diam(\mathcal{K})\,C_0, \tag{82}$$

*where $C_9 = \mathcal{O}\left(L^3(Md\,diam(\mathcal{K}))^2\right)$.*

The proof of this theorem is given in Appendix F.5. Observe that the metric $\frac{1}{S}\sum_{s=0}^{S-1}\|\nabla f(\hat{\mathbf{w}}^s(s))\|^2$ used in Theorem 6.7 may appear non-standard; however, it has recently been employed in Wu et al. (2023) for decentralized SGD under Byzantine attacks. For sufficiently large $S$ and sufficiently small $C_0$, Theorem 6.7 implies near first-order stationarity. Specifically, this fixed-stepsize finite-horizon result says that, for a prescribed horizon $S$ and stepsize $h = 1/\sqrt{S}$, the average stationarity measure $\frac{1}{S}\sum_{s=0}^{S-1}\|\nabla f(\hat{\mathbf{w}}^s(s))\|^2$ is controlled over that run. The theorem still relies on compact-set control in Assumptions 4.12 and 6.2, as reflected in the constants involving $diam(\mathcal{K})$.

Having established convergence guarantees for RESIST under the MITM attack model, we now clarify its relation to decentralized Byzantine attacks. The next section shows that the Byzantine-resilient ERM problem over the regular nodes can be embedded into a static MITM formulation by representing messages sent from a fixed faulty-node set as compromised link transmissions. Under this embedding, the coordinate-wise averaging matrices have a block form in which the regular-node entries evolve independently of the faulty-node entries. Thus, the regular-node recursion has the same form as the MITM recursion analyzed above, so the convergence results apply to the regular-node Byzantine objective through this static construction.

# 7 Reduction of Decentralized Byzantine Attacks to the MITM Attack Model

A common conclusion in Byzantine-resilient decentralized learning is that, under arbitrary node failures, one cannot guarantee solving the full decentralized ERM problem in (4) over all nodes. Instead, the strongest achievable target is the ERM restricted to the regular (nonfaulty) nodes, as in the Byzantine consensus and learning literature (e.g., Su & Vaidya (2016b; 2015); Yang & Bajwa (2019); Fang et al. (2022)):

$$\min_{\{\mathbf{w}_j : j \in \mathcal{R}\}} \frac{1}{r}\sum_{j \in \mathcal{R}} f_j(\mathbf{w}_j) \text{ subject to } \forall i, j \in \mathcal{R}, \ \mathbf{w}_i = \mathbf{w}_j. \tag{83}$$

Here, $\mathcal{N}$ denotes the full set of nodes in the network, while $\mathcal{B} \subseteq \mathcal{N}$ and $\mathcal{R} := \mathcal{N}\backslash\mathcal{B}$ denote the (static) sets of faulty and regular nodes, respectively, under the classical Byzantine node-failure model. Let $r = |\mathcal{R}|$. The design parameter $b$ represents an upper bound on the number of Byzantine nodes, so that $0 \leqslant |\mathcal{B}| \leqslant b$ and consequently $r \geqslant M - b$. Without loss of generality, we relabel the regular nodes as $\mathcal{R} = \{1, \ldots, r\}$.

To connect this Byzantine objective to our MITM framework, we embed (83) into an $M$-node ERM problem in which the regular nodes are required to reach consensus, while the faulty nodes contribute no meaningful optimization signal. Concretely, (83) is equivalent (up to the harmless scaling $r/M$ in the objective) to the following static MITM ERM problem over all nodes:

$$\min_{\{\mathbf{w}_j : j \in \{1,\ldots,M\}\}} \frac{1}{M} \sum_{j=1}^{M} f_j(\mathbf{w}_j) \text{ subject to } \forall i,j \in \{1,\ldots,r\}, \ \mathbf{w}_i = \mathbf{w}_j; \quad f_j := \text{constant } \forall \ r < j \leqslant M. \quad (84)$$

We interpret this as a static MITM instance in which only the outgoing edges associated with nodes in $\mathcal{N}\backslash\mathcal{R}$ may be compromised for all time, while edges between regular nodes remain uncompromised.

Under this construction, the coordinate-wise RESIST iteration inherits the same update recursion previously derived for the MITM model (cf. (19) and (20)):

$$[\mathbf{W}(s+1)]_k = \mathbf{Q}_k(s)[\mathbf{W}(s)]_k - h[\nabla F(\mathbf{W}(s))]_k, \quad (85)$$

where $\mathbf{Q}_k(s) := \prod_{l=J\lfloor \frac{t}{J} \rfloor}^{J\lfloor \frac{t}{J} \rfloor + J - 2} \mathbf{Y}_k(l)$ and

$$\mathbf{Y}_k(l) = \begin{bmatrix} [\mathbf{Y}_k(l)]_{[1:r]\times[1:r]} & \mathbf{0}_{[1:r]\times[r+1:M]} \\ [\mathbf{Y}_k(l)]_{[r+1:M]\times[1:r]} & [\mathbf{Y}_k(l)]_{[r+1:M]\times[r+1:M]} \end{bmatrix} \quad (86)$$

from Corollary A.1 in Appendix A. Note that Corollary A.1 applies here because, from the viewpoint of a regular node and its local neighborhood, a Byzantine attack affecting at most $b$ nodes induces at most $b$ compromised incoming links into that neighborhood, provided $b < \min_{j \in \mathcal{N}} \frac{|\mathcal{N}_j|+1}{2}$. Consequently, $\mathbf{Q}_k(s)$ inherits the same block structure:

$$\mathbf{Q}_k(s) = \begin{bmatrix} \prod_{l=J\lfloor \frac{t}{J} \rfloor}^{J\lfloor \frac{t}{J} \rfloor + J - 2} [\mathbf{Y}_k(l)]_{[1:r]\times[1:r]} & \mathbf{0}_{[1:r]\times[r+1:M]} \\ \mathbf{A}_1(s) & \mathbf{A}_2(s) \end{bmatrix} \quad (87)$$

for some block matrices $\mathbf{A}_1(s), \mathbf{A}_2(s)$. In particular, the update in (85) for the first $r$ entries depends only on those same entries and is unaffected by the remaining $M - r$ components:

$$[\mathbf{W}(s+1)]_{k,1:r} = [\mathbf{Q}_k(s)]_{[1:r]\times[1:r]}[\mathbf{W}(s)]_{k,1:r} - h[\nabla F(\mathbf{W}(s))]_{k,1:r},$$

whereas the bottom $M - r$ entries may behave arbitrarily under the influence of the adversary and do not affect the regular-node dynamics through the zero block in the upper-right corner.

It is important to emphasize that the above embedding is purely analytical. Byzantine attacks operate at the node level, whereas MITM attacks act on communication links, so a direct graph-level identification between the two models is generally nontrivial. In particular, the construction above corresponds to a static MITM instance induced by a fixed faulty-node set $\mathcal{B}$. This distinction motivates the modified $\mathcal{T}_{\mathcal{F}}$ definition in Definition 3.2, adapted from standard Byzantine constructions in Su & Vaidya (2016b); Fang et al. (2022), together with the corresponding constant $\tau := |\mathcal{T}_{\mathcal{F}}|$. With these definitions in place, the consensus and geometric convergence analysis developed for the MITM formulation (84) applies directly to the evolution of the $r$ regular nodes under Byzantine attacks. Consequently, when restricted to $\mathcal{R}$, RESIST guarantees the same convergence properties for the Byzantine-resilient ERM problem (83) as those established under the static MITM construction in (84). Thus, this section should be read as a static Byzantine-to-MITM embedding; the time-varying link-level MITM model itself is the one introduced in Sec. 2.3 and analyzed through the filtered-matrix products in Secs. 3.1–3.3.

## 8 Statistical Learning Rates for RESIST

### 8.1 Preliminaries

In Sec. 2, we introduced the sample-level loss $\ell(\mathbf{w}, \mathbf{z})$, the statistical risk $\mathcal{R}(\mathbf{w}) := \mathbb{E}_{\mathbb{P}}[\ell(\mathbf{w}, \mathbf{z})]$, and the decentralized ERM objective

$$f(\mathbf{w}) := \frac{1}{MN} \sum_{j=1}^{M} \sum_{n=1}^{N} \ell(\mathbf{w}, \mathbf{z}_{jn}) = \frac{1}{M} \sum_{j=1}^{M} f_j(\mathbf{w}), \qquad f_j(\mathbf{w}) := \frac{1}{N} \sum_{n=1}^{N} \ell(\mathbf{w}, \mathbf{z}_{jn}). \tag{88}$$

In the algorithmic convergence analysis (Secs. 5–6), we fixed an arbitrary realization of the data (equivalently, we conditioned on the samples) and treated the induced empirical functions $\{f_j\}$ as deterministic, suppressing explicit dependence on $\{\mathbf{z}_{jn}\}$. In this section (and the associated proofs in Appendix G), we restore the statistical viewpoint and make the dependence on the underlying probability law $\mathbb{P}$ explicit in order to derive statistical learning rates.

Concretely, recall that each node $j$ holds a local dataset $\mathcal{Z}_j = \{\mathbf{z}_{jn}\}_{n=1}^{N}$, where $\mathbf{z}_{jn} \overset{\text{i.i.d.}}{\sim} \mathbb{P}$, and the collections $\{\mathcal{Z}_j\}_{j=1}^{M}$ are i.i.d. across nodes. The corresponding global and local empirical objectives are defined above. This common-population sampling model is used in the uniform concentration step behind the statistical rates: since all samples are drawn from the same population law, the empirical gradient averages are centered at the same population gradient $\nabla\mathcal{R}(\mathbf{w})$. If nodes instead sampled from node-specific laws, the centered population quantities would generally differ across nodes, and a theorem characterizing the statistical rates would first need to specify the target population risk and the resulting population-heterogeneity terms. We leave such an extension for future work. The statistical rates in this section are therefore common-population i.i.d. guarantees. Before proceeding further, we also recall the statistical risk minimizer and the ERM minimizer:

$$\mathbf{w}_{\text{SR}}^* \in \underset{\mathbf{w} \in \mathbb{R}^d}{\arg\min} \, \mathcal{R}(\mathbf{w}) = \underset{\mathbf{w} \in \mathbb{R}^d}{\arg\min} \, \mathbb{E}_{\mathbb{P}}[\ell(\mathbf{w}, \mathbf{z})], \quad \mathbf{w}^* \equiv \mathbf{w}_{\text{ERM}}^* \in \underset{\mathbf{w} \in \mathbb{R}^d}{\arg\min} \, f(\mathbf{w}) = \underset{\mathbf{w} \in \mathbb{R}^d}{\arg\min} \, \frac{1}{M} \sum_{j=1}^{M} f_j(\mathbf{w}). \tag{89}$$

We additionally denote any local empirical minimizer by $\mathbf{w}_j^* \in \arg\min_{\mathbf{w} \in \mathbb{R}^d} f_j(\mathbf{w}), j \in \{1, \ldots, M\}$. We further denote the optimal statistical and empirical risks by

$$\mathcal{R}_{\text{SR}}^* := \mathcal{R}(\mathbf{w}_{\text{SR}}^*), \qquad f_{\text{ERM}}^* := f(\mathbf{w}^*). \tag{90}$$

Under differentiability and standard interchange conditions, the statistical risk satisfies $\nabla\mathcal{R}(\mathbf{w}) = \mathbb{E}_{\mathbb{P}}[\nabla\ell(\mathbf{w}, \mathbf{z})]$. The statistical and empirical minimizers satisfy the first-order optimality conditions

$$\nabla\mathcal{R}(\mathbf{w}_{\text{SR}}^*) = \mathbf{0}, \qquad \nabla f_j(\mathbf{w}_j^*) = \mathbf{0}, \ j \in \{1, \ldots, M\}, \qquad \nabla f(\mathbf{w}^*) = \mathbf{0}. \tag{91}$$

Taking expectation with respect to the data yields[7]

$$\mathbb{E}[\nabla f_j(\mathbf{w}_j^*)] = \mathbf{0}, \ j \in \{1, \ldots, M\}, \qquad \mathbb{E}[\nabla f(\mathbf{w}^*)] = \mathbf{0}. \tag{92}$$

We also note that in this section, Assumptions 4.8, 5.1, and 6.1 are understood to hold almost surely with respect to the probability law $\mathbb{P}$ whenever they are invoked. In particular, whenever Assumptions 4.8 or 5.1 are assumed for the empirical objectives, the same property extends to the statistical risk function $\mathcal{R}(\mathbf{w})$ through the expectation operator. Finally, we introduce a statistical analogue of Assumption 4.12 to control the boundedness of the iterates uniformly over random sample realizations.

**Assumption 8.1** (Statistical uniform boundedness). Consider the ERM problem (3) with $N$ i.i.d. samples at each node and a fixed network size $M$. Assume the initialization is uniformly bounded across nodes, i.e., $\max_{1 \leqslant j \leqslant M} \|\mathbf{w}_j(0)\|$ is bounded by a deterministic constant independent of $N$ and the sample realization. Then, for each realization of the datasets $\{\mathcal{Z}_j\}_{j=1}^{M}$, there exists a compact set $\mathcal{K}_N(\{\mathcal{Z}_j\}_{j=1}^{M}) \subset \mathbb{R}^d$ such that the RESIST iterates satisfy $\mathbf{w}_j(t) \in \mathcal{K}_N(\{\mathcal{Z}_j\}_{j=1}^{M}), \ \forall t \geqslant 0, \ \forall j \in \{1, \ldots, M\}$, almost surely with respect to $\mathbb{P}$. Moreover, there exists a deterministic compact set $\mathcal{K} \subset \mathbb{R}^d$ with diameter $\text{diam}(\mathcal{K})$, which may depend on $M$ but is independent of $N$ and the sample realization, such that $\mathcal{K}_N(\{\mathcal{Z}_j\}_{j=1}^{M}) \subset \mathcal{K}$ $\mathbb{P}$-a.s.

---

[7]From here onward, we drop the subscript $\mathbb{P}$ whenever the underlying probability law is clear from context.

Assumption 8.1 is the statistical analogue of Assumption 4.12. The main difference is that, for each realization of the datasets, the realization-dependent compact set $\mathcal{K}_N\left(\{\mathcal{Z}_j\}_{j=1}^M\right)$ may depend on $N$ and on the specific sample draw. However, to establish sample-complexity guarantees for RESIST, we require a deterministic compact domain on which the finite-net and covering-number bounds used in the uniform concentration argument can be applied; this is the statistical reason for imposing a uniform, data-independent bound on the iterates. This is ensured by the existence of a deterministic compact set $\mathcal{K}$ that contains all realization-dependent sets $\mathcal{K}_N\left(\{\mathcal{Z}_j\}_{j=1}^M\right)$ almost surely and is independent of $N$. The construction in Appendix G.4 shows that Assumption 8.1 is not vacuous under additional structural conditions on the loss functions and the network dynamics. As with the deterministic boundedness discussion after Assumption 4.12, this construction should be read as a restricted non-vacuity example rather than as a verification of Assumption 8.1 for arbitrary time-varying MITM attack sequences. The projected variant mentioned there is one possible way to enforce compactness by construction, but it changes the algorithm and therefore would require a separate statistical analysis as well.

In the next three subsections, we derive statistical learning rates for RESIST under the strongly convex, PŁ, and smooth nonconvex settings, corresponding to the cases analyzed in Secs. 5 and 6.

## 8.2 Statistical learning rate of RESIST under strong convexity

Theorem 5.5 in Sec. 5 established the geometric convergence of RESIST for fixed data realizations, where the error bounds relied on the data-dependent constants $C_0$ and $\Delta$. In this section, we refine that analysis for the statistical setting by bounding these quantities as explicit functions of the sample size $N$. The following theorem provides high-probability bounds on the consensus and optimization errors, thereby characterizing the resulting statistical learning rate (i.e., the sample complexity) of RESIST.

**Theorem 8.2.** *Consider the ERM formulation in* (3) *with $N$ i.i.d. training samples at each node, all drawn from the common population law $\mathbb{P}$. Under Assumptions 3.3, 4.8, 5.1, and 8.1, suppose the parameter $J$ satisfies $J > \frac{\tau M \log\left(2M^{\frac{3}{2}}(\sqrt{M}+1)\right)}{\log\left((1-\beta^{\tau M})^{-1}\right)} + \tau M + 2$. Then, for any $i \in \mathcal{N}$, the iterate sequence $\{\mathbf{w}_i(s)\}_s$ generated by RESIST converges geometrically in the $s$-time scale to a neighborhood of the statistical risk minimizer $\mathbf{w}_{\mathrm{SR}}^*$ whose radius scales as $\mathcal{O}\left(\frac{1}{\sqrt{N}} + h\right)$ with high probability. In particular:*

- *For any $\epsilon' \in (0,1)$, the consensus errors $\xi_k^1(s)$ and $\xi_k^5(s)$ (cf. Definition 4.3), for any coordinate $k$, satisfy*

$$\limsup_{s \to \infty} \xi_k^1(s) \leqslant \mathcal{O}\left(hM \, diam(\mathcal{K})\right) + \mathcal{O}\left(\frac{2Mh}{\mu}\sqrt{\log\left(\frac{4d}{\delta}\right)}\frac{L'd}{\sqrt{2N}}\right), \tag{93}$$

$$\limsup_{s \to \infty} \xi_k^5(s) \leqslant \mathcal{O}\left(hM \, diam(\mathcal{K})\right) + \mathcal{O}\left(\frac{2Mh}{\mu}\sqrt{\log\left(\frac{4d}{\delta}\right)}\frac{L'd}{\sqrt{2N}}\right), \tag{94}$$

*with probability at least $1 - \delta$, where*

$$\delta = 2d\exp\left(-\frac{2(\epsilon')^2 MN}{(L'd)^2}\right) + 2d\exp\left(-\frac{2(\epsilon')^2 N}{(L'd)^2}\right), \tag{95}$$

*and $L'$ is a constant that satisfies $L' = \max\left\{\mathcal{O}(Ld \, diam(\mathcal{K})), \ \mathcal{O}(L(diam(\mathcal{K}))^2)\right\}$.*

- *For any $\epsilon' \in (0,1)$, for any sufficiently large $N$, and any stepsize $h < \min\left\{\frac{1}{M^2\sqrt{d}}, \frac{2}{\mu+L}\right\}$, the averaged iterate error satisfies*

$$\limsup_{s \to \infty} \|\mathbf{w}_{\mathrm{SR}}^* - \hat{\mathbf{w}}^s(s)\| \leqslant \mathcal{O}\left(\frac{6}{\mu}\sqrt{\frac{L'^2 d^2\|\boldsymbol{\alpha}\|^2 \log\frac{12}{\delta}}{N}}\right) + \mathcal{O}\left(hM\sqrt{Md} \, diam(\mathcal{K})\right), \tag{96}$$

*with probability at least $1 - \delta$, where*

$$\delta = 6d\exp\left(-\frac{(\epsilon')^2 MN}{4(L'd)^2}\right) + 2d\exp\left(-\frac{2(\epsilon')^2 N}{(L'd)^2}\right)$$

$$+ 2\exp\left(-\frac{4MN(\epsilon')^2}{16(L')^2Md^2\|\boldsymbol{\alpha}\|^2 + (\epsilon')^2} + M\log\left(\frac{12L'd\sqrt{M}}{\epsilon'}\right) + d\log\left(\frac{12L'\Gamma_0 d}{\epsilon'}\right)\right), \quad (97)$$

with $\Gamma_0 := diam(\mathcal{K})$ and a stochastic vector $\boldsymbol{\alpha} \in \mathbb{R}^M$ representing the effective mixing weights, satisfying $\|\boldsymbol{\alpha}\|^2 \in \left[\frac{1}{M}, 1\right]$.

- As $N \to \infty$, the averaged iterates converge in probability to the exact statistical risk minimizer:

$$\lim_{N\to\infty}\limsup_{s\to\infty}\left(\|\mathbf{W}(s) - \overline{\mathbf{W}}(s)\|_F + \|\mathbf{W}_{\text{SR}}^* - \widehat{\mathbf{W}}^s(s)\|_F + \|\mathbf{W}(s) - \widehat{\mathbf{W}}^s(s)\|_F\right) \xrightarrow{P} 0. \quad (98)$$

Note that $X_N \xrightarrow{P} 0$ denotes convergence in probability, and the proof of this theorem is in Appendix G.2. Theorem 8.2 consists of three parts. The first establishes asymptotic consensus of the local iterates to an $\mathcal{O}\left(h + \frac{h}{\sqrt{N}}\right)$ neighborhood with high probability; this neighborhood can be made arbitrarily small by selecting a sufficiently small stepsize $h$. The second establishes asymptotic convergence of the averaged iterates to an $\mathcal{O}\left(\frac{1}{\sqrt{N}} + h\right)$ neighborhood of the statistical risk minimizer $\mathbf{w}_{\text{SR}}^*$ with high probability; this neighborhood can likewise be reduced by choosing $h$ sufficiently small when the sample size $N$ is large. The third shows that, as $N \to \infty$, the averaged iterates converge in probability to the exact statistical risk minimizer $\mathbf{w}_{\text{SR}}^*$.

Note that in the bound for the averaged iterate error, the factor $\|\boldsymbol{\alpha}\|^2$ determines the effective statistical rate. The vector $\boldsymbol{\alpha}$ represents the mixing weights induced by the screening mechanism of RESIST and the presence of attacks, consistent with Yang & Bajwa (2019) and Fang et al. (2022). In the absence of screening or adversarial behavior, one recovers uniform weights equal to $1/M$, yielding a statistical rate of order $\mathcal{O}(1/\sqrt{MN})$, which matches the centralized learning rate. In general, however, the exact value of $\boldsymbol{\alpha}$ depends on the attack pattern and cannot be characterized explicitly; only the bounds $\|\boldsymbol{\alpha}\|^2 \in \left[\frac{1}{M}, 1\right]$ can be guaranteed. Consequently, the learning rate interpolates between the centralized rate $\mathcal{O}(1/\sqrt{MN})$ and the local rate $\mathcal{O}(1/\sqrt{N})$, depending on the impact of the attacks.

It is also useful to clarify the distinction between the first two bullet points in Theorem 8.2, which contain residual $\mathcal{O}(h)$ terms, and the third bullet point, which asserts exact convergence in probability. The first two statements provide high-probability guarantees for fixed and finite $N$, where the local empirical risks $f_j$ remain distinct due to sampling variability. This heterogeneity induces a non-vanishing error floor proportional to the stepsize $h$. The third statement concerns the limit in which $N \to \infty$. As the sample size grows, the local empirical risks converge to the common statistical risk $\mathcal{R}$, and the heterogeneity across nodes vanishes. In this asymptotic regime, the problem becomes effectively homogeneous, and RESIST achieves consensus and convergence to $\mathbf{w}_{\text{SR}}^*$ in probability.

## 8.3 Statistical learning rate of RESIST for Polyak–Łojasiewicz (PŁ) functions

We now extend the statistical refinement developed for the strongly convex case to the Polyak–Łojasiewicz (PŁ) function class. Theorem 6.4 in Sec. 6 established geometric convergence of RESIST in function value under the PŁ condition for fixed data realizations. As in the strongly convex setting, the constants $C_0$ and $\Delta$ appearing in that result depend on the particular data realization. The following theorem makes their dependence on the sample size $N$ explicit and characterizes the corresponding statistical learning rate (i.e., the sample complexity) under the PŁ condition.

**Theorem 8.3.** *Consider the ERM formulation in* (3) *with $N$ i.i.d. training samples at each node, all drawn from the common population law $\mathbb{P}$. Under Assumptions 3.3, 4.8, 6.1, and 8.1, suppose the stepsize satisfies $h \in (0, \frac{2}{L})$ and the parameter $J$ satisfies $J > \frac{\tau M \log\left(2M^{\frac{3}{2}}(\sqrt{M}+1)\right)}{\log\left((1-\beta^{\tau M})^{-1}\right)} + \tau M + 2$. Then the function value sequence $\{f(\widehat{\mathbf{w}}^s(s))\}_s$ generated by RESIST converges geometrically in the $s$-time scale to a neighborhood of the minimum statistical risk $\mathcal{R}_{\text{SR}}^*$ whose radius scales as $\mathcal{O}\left(h + \frac{1}{\sqrt{N}}\right)$ with high probability. In particular, for any $\epsilon' \in (0,1)$, for sufficiently large $N$ and $\sqrt{M} > \mu$, we have*

$$\limsup_{s\to\infty}|\mathcal{R}_{\text{SR}}^* - f(\widehat{\mathbf{w}}^s(s))| \leqslant \mathcal{O}\left(\frac{L\,diam(\mathcal{K})}{\mu(2-Lh)}\sqrt{\frac{L'^2d^2\|\boldsymbol{\alpha}\|^2\log\frac{12}{\delta}}{N}}\right) + \mathcal{O}\left(\frac{hL^3M^{\frac{5}{2}}(d\,diam(\mathcal{K}))^2}{\mu}\right), \quad (99)$$

*with probability at least $1 - \delta$, where*

$$\delta = 2\exp\left(-\frac{4MN(\epsilon')^2}{16(L')^2Md^2\|\boldsymbol{\alpha}\|^2 + (\epsilon')^2} + M\log\left(\frac{12L'd\sqrt{M}}{\epsilon'}\right) + d\log\left(\frac{12L'\Gamma_0 d}{\epsilon'}\right)\right)$$
$$+ 4d\exp\left(-\frac{(\epsilon')^2MN}{4(L'd)^2}\right) + 2\exp\left(-\frac{2(\epsilon')^2MN}{(L')^2}\right), \tag{100}$$

*for constants $L', \Gamma_0$ defined in Theorem 8.2 and a stochastic vector $\boldsymbol{\alpha} \in \mathbb{R}^M$ representing the effective mixing weights, satisfying $\|\boldsymbol{\alpha}\|^2 \in \left[\frac{1}{M}, 1\right]$.*

The proof of this theorem is provided in Appendix G.3. Unlike Theorem 8.2 for the strongly convex case, Theorem 8.3 does not provide explicit statistical rates for the consensus error terms $\xi_k^1(s)$ and $\xi_k^5(s)$. This distinction stems from the nature of the PŁ condition, which controls suboptimality in function value but does not directly yield bounds on iterate distances. As a result, while we establish high-probability convergence of the function values to an $\mathcal{O}\left(h + \frac{1}{\sqrt{N}}\right)$ neighborhood of the minimum statistical risk $\mathcal{R}_{\mathrm{SR}}^*$, we do not obtain corresponding high-probability guarantees for consensus of the iterates in this setting.

As in the strongly convex case, the bound in the theorem decomposes into a statistical estimation term of order $\mathcal{O}(1/\sqrt{N})$ and a residual algorithmic bias of order $\mathcal{O}(h)$. The statistical term depends on the factor $\|\boldsymbol{\alpha}\|^2$, which captures the effective mixing weights induced by the screening mechanism and the presence of attacks. Since $\|\boldsymbol{\alpha}\|^2 \in \left[\frac{1}{M}, 1\right]$, the resulting statistical rate ranges between the centralized rate $\mathcal{O}(1/\sqrt{MN})$ and the local rate $\mathcal{O}(1/\sqrt{N})$. Under a constant stepsize $h$, the residual $\mathcal{O}(h)$ term does not vanish as $N$ increases. Although the statistical estimation component shrinks with larger sample sizes, the bound retains a nonzero bias proportional to $h$, reflecting the fixed-stepsize dynamics and the screening mechanism of RESIST. Consequently, exact convergence of the function values to $\mathcal{R}_{\mathrm{SR}}^*$ is not guaranteed under constant stepsizes in the PŁ setting. Further discussion is provided in Appendix G.

## 8.4 Statistical learning rate of RESIST for smooth nonconvex functions

We now extend the statistical refinement to the class of smooth nonconvex objectives. Theorem 6.6 in Sec. 6 established a sublinear convergence guarantee for RESIST in terms of the minimum squared gradient norm under fixed data realizations. The following theorem quantifies how this convergence behavior scales with the sample size $N$, thereby characterizing the statistical learning rate for smooth nonconvex functions under a diminishing stepsize.

**Theorem 8.4.** *Consider the ERM formulation in (3) with $N$ i.i.d. training samples at each node, all drawn from the common population law $\mathbb{P}$. Under Assumptions 3.3, 4.8, and 8.1, suppose RESIST is run with diminishing stepsize $h(s) = \frac{p}{(s+1)^\omega}$, where $\omega = \frac{1}{2} + \epsilon$, $0 < \epsilon < \frac{1}{2}$, and $0 < p \leqslant \frac{1}{2L}$, and let $J$ satisfy $J > \frac{\tau M \log\left(2M^{\frac{3}{2}}(\sqrt{M}+1)\right)}{\log\left((1-\beta^{\tau M})^{-1}\right)} + \tau M + 2$. Then the minimum squared gradient norm over $S$ iterations, $\min_{0 \leqslant s \leqslant S-1}\|\nabla f(\hat{\mathbf{w}}^s(s))\|^2$, converges at a sublinear rate of order $\mathcal{O}(S^{-0.5+\epsilon})$ to a neighborhood of zero whose radius scales as $\mathcal{O}(1/\sqrt{N})$ with high probability. In particular, for any $\epsilon' \in (0, 1)$ and sufficiently large $N$, we have*

$$\limsup_{S \to \infty} \min_{0 \leqslant s \leqslant S-1}\|\nabla f(\hat{\mathbf{w}}^s(s))\|^2 \leqslant \mathcal{O}\left(L\,diam(\mathcal{K})\sqrt{\frac{L'^2d^2\|\boldsymbol{\alpha}\|^2\log\frac{4}{\delta}}{N}}\right), \tag{101}$$

*with probability at least $1 - \delta$, where*

$$\delta = 2\exp\left(-\frac{4MN(\epsilon')^2}{16(L')^2Md^2\|\boldsymbol{\alpha}\|^2 + (\epsilon')^2} + M\log\left(\frac{12L'd\sqrt{M}}{\epsilon'}\right) + d\log\left(\frac{12L'\Gamma_0 d}{\epsilon'}\right)\right)$$
$$+ 2d\exp\left(-\frac{(\epsilon')^2MN}{4(L'd)^2}\right), \tag{102}$$

*and $L', \Gamma_0$ are the same constants as in Theorem 8.2, while $\boldsymbol{\alpha} \in \mathbb{R}^M$ satisfies $\|\boldsymbol{\alpha}\|^2 \in \left[\frac{1}{M}, 1\right]$. Moreover,*

$$\lim_{N \to \infty} \limsup_{S \to \infty} \min_{0 \leqslant s \leqslant S-1} \|\nabla f(\hat{\mathbf{w}}^s(s))\|^2 \xrightarrow{P} 0. \tag{103}$$

The proof of Theorem 8.4 combines the results of Theorem 6.6 and Lemma G.1 in both finite- and infinite-sample regimes and is therefore omitted here. The above theorem serves as the statistical counterpart to the diminishing-stepsize result in Theorem 6.6. We now turn to the constant-stepsize setting. In particular, the following theorem makes explicit how the convergence guarantee in Theorem 6.7 extends to the statistical regime as a function of the sample size $N$, thereby characterizing the statistical learning rate under a fixed stepsize.

**Theorem 8.5.** *With the ERM formulation* (3) *and $N$ i.i.d. training samples at each node $i$,* all drawn from the common population law $\mathbb{P}$*, under Assumptions 3.3, 4.8, and 8.1, suppose RESIST is iterated for $S$ gradient steps with a constant stepsize $h = \frac{1}{\sqrt{S}}$, where $S > L^6(Md\, diam(\mathcal{K}))^4$, and $J > \frac{\tau M \log(2M^{\frac{3}{2}}(\sqrt{M}+1))}{\log(1-\beta^{\tau M})^{-1}} + \tau M + 2$. Then, for any $\epsilon' \in (0,1)$ and sufficiently large $N$, the following holds:*

$$\frac{1}{S} \sum_{s=0}^{S-1} \|\nabla f(\hat{\mathbf{w}}^s(s))\|^2 \leqslant \left(1 - \frac{L}{\sqrt{S}}\right)^{-1} \frac{f(\hat{\mathbf{w}}^0(0)) - \inf_{\mathbf{w}} f(\mathbf{w})}{\sqrt{S}} + \frac{C_9}{\sqrt{S}} + \mathcal{O}\left(L\, diam(\mathcal{K})\sqrt{\frac{L'^2 d^2 \|\boldsymbol{\alpha}\|^2 \log \frac{4}{\delta}}{N}}\right) \tag{104}$$

*with probability at least $1 - \delta$, where*

$$\delta = 2 \exp\left(-\frac{4MN(\epsilon')^2}{16(L')^2 Md^2 \|\boldsymbol{\alpha}\|^2 + (\epsilon')^2} + M \log\left(\frac{12L'd\sqrt{M}}{\epsilon'}\right) + d \log\left(\frac{12L'\Gamma_0 d}{\epsilon'}\right)\right) + 2d \exp\left(-\frac{(\epsilon')^2 MN}{4(L'd)^2}\right). \tag{105}$$

*Here $C_9 = \mathcal{O}\left(L^3(Md\, diam(\mathcal{K}))^4\right)$, $L'$ and $\Gamma_0$ are the same constants as in Theorem 8.2, and $\boldsymbol{\alpha} \in \mathbb{R}^M$ satisfies $\|\boldsymbol{\alpha}\|^2 \in \left[\frac{1}{M}, 1\right]$. Moreover, in the infinite-sample regime, we have*

$$\limsup_{N \to \infty} \frac{1}{S} \sum_{s=0}^{S-1} \|\nabla f(\hat{\mathbf{w}}^s(s))\|^2 \overset{P}{\leqslant} \left(1 - \frac{L}{\sqrt{S}}\right)^{-1} \frac{f(\hat{\mathbf{w}}^0(0)) - \inf_{\mathbf{w}} f(\mathbf{w})}{\sqrt{S}} + \frac{C_9}{\sqrt{S}}, \tag{106}$$

*where $X \overset{P}{\leqslant} A$ denotes $\Pr\{X \leqslant A\} = 1$.*

The proof of Theorem 8.5 follows directly from Theorem 6.7 and Lemma G.1. As in Theorem 6.7, the fixed-stepsize statement in this theorem is a finite-horizon statistical guarantee for the prescribed run of $S$ gradient steps and remains conditional on the compact-set control in Assumption 8.1 during that run.

### 8.5 Discussion

We summarize the asymptotic statistical guarantees across the three function classes under the common-population i.i.d. sampling model and the compact-domain condition in Assumption 8.1. In the smooth strongly convex regime with constant stepsize (Theorem 8.2), the iterate error converges to zero in probability as $N \to \infty$, yielding exact recovery of the statistical minimizer in the large-sample limit. In the smooth PŁ regime with constant stepsize $h$ (Theorem 8.3), the averaged function error converges to an $\mathcal{O}(h)$ neighborhood of zero as $N \to \infty$, reflecting a residual algorithmic bias that persists even with infinite data. In the smooth nonconvex setting, the behavior depends on the stepsize schedule: with a diminishing stepsize (Theorem 8.4), the minimum gradient norm converges to zero in probability, establishing asymptotic first-order stationarity; with a fixed stepsize over a finite horizon of $S$ iterations (Theorem 8.5), the guarantee is a finite-time bound in which the optimization term scales as $\mathcal{O}(1/\sqrt{S})$, and therefore decreases only as the number of iterations increases, rather than vanishing solely through larger sample size $N$.

Across all three function classes, the finite-sample statistical term scales with $\|\boldsymbol{\alpha}\|/\sqrt{N}$, interpolating between the centralized rate $\mathcal{O}(1/\sqrt{MN})$ and the local rate $\mathcal{O}(1/\sqrt{N})$. The influence of adversarial behavior is therefore reflected through this same interpolation mechanism.

Having established the algorithmic convergence in Secs. 4–6, including geometric consensus-error guarantees with explicit residual bounds, geometric convergence rates to residual neighborhoods for strongly convex and PŁ objectives, and smooth nonconvex guarantees that control consensus errors and stationarity measures, with residual terms under diminishing stepsizes and finite-horizon bounds under constant stepsizes, together with the complementary statistical learning rates developed in this section, we now turn to numerical experiments. Section 9 complements the theory with experiments that illustrate RESIST's resilience and scaling behavior under the data partitions and attack scenarios considered below.

## 9 Numerical Results

The numerical experiments are organized into two parts corresponding to the strongly convex and nonconvex regimes. First, we consider a strongly convex learning problem on the MNIST dataset (LeCun et al., 1998). We train an $\ell_2$-regularized linear classifier using cross-entropy loss, so that the resulting objective is smooth and strongly convex, satisfying Assumptions 4.8 and 5.1. This setting is used to illustrate the geometric convergence behavior established for the strongly convex case.

Second, we evaluate RESIST on the CIFAR-10 dataset (Krizhevsky & Hinton, 2009) using a convolutional neural network, which yields a smooth nonconvex learning objective. This setup illustrates the behavior predicted by the nonconvex analysis in Sec. 6. Since PŁ functions are a subclass of smooth nonconvex objectives, the performance of RESIST in this setting also provides insight into the regime considered in Sec. 6.1, although the PŁ condition is not explicitly verified for the neural network model.

In all experiments, the communication network is modeled as an Erdős–Rényi graph with $M$ nodes and connection probability $\rho$, meaning that an edge exists independently between any two distinct nodes with probability $\rho$. To simulate the time-varying link-level MITM attack model defined in Sec. 2.3, we fix an attack budget parameter $b$, representing the maximum number of compromised incoming edges per node anticipated by the algorithm. At each iteration $t$, the adversary randomly selects a subset of edges to compromise (subject to the budget $b$), and the information transmitted over these links is replaced with corrupted vectors determined by the specific attack strategy, as detailed in the corresponding subsections.

The network topology is generated to provide sufficient neighborhood redundancy for the CWTM step by ensuring that the minimum node degree is at least $2b+1$. For larger attack budgets (e.g., $b=8$ and $b=16$), the connection probability $\rho$ is increased accordingly. Although the random selection of compromised edges typically results in $|\mathcal{N}_j^b(t)| < b$ for many nodes and time instances, this degree condition ensures that the

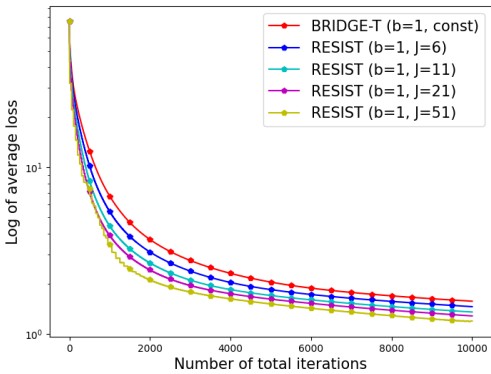

Figure 2: Logarithm of the average training loss versus total iterations (algorithmic rounds) for different choices of $J$ on MNIST ($M = 50$, $\rho = 0.5$, $b = 1$). The behavior in the later iterations aligns with the geometric convergence predicted by the strongly convex analysis.

trimming step remains well defined even when the realized attack reaches the budget at a node. Note that the convergence analysis, in contrast, relies on the filtered-graph connectivity condition stated in Assumption 3.3, which is not merely a minimum-degree requirement.

Throughout this section, iteration counts refer to algorithmic rounds rather than wall-clock time. Each algorithmic round is either a communication round or a local gradient-update round; for RESIST and its variants, a communication round includes the specified screening rule. In the $J$-ablation experiments, changing $J$ changes how a fixed round budget is divided between communication/filtering and gradient computation. In the MNIST and CIFAR-10 $J$-ablation experiments, the fixed-stepsize $J = 2$ case provides the closest comparison with BRIDGE-T and REDGRAF-CWTM under the aligned conventions discussed in Sec. 1. Finally, except for the dedicated non-i.i.d. MNIST experiments in Sec. 9.1.5, the experiments use i.i.d. data partitions across nodes. These non-i.i.d. experiments evaluate robustness under heterogeneous partitions and are meant to be read alongside the deterministic residual-neighborhood theory, where Assumption 5.11 captures local-gradient heterogeneity through a bounded-gradient-similarity condition, rather than as a separate non-i.i.d. statistical learning-rate guarantee.

## 9.1 Strongly convex setting: Linear classifier on MNIST

We evaluate the performance of RESIST in the strongly convex regime under MITM attacks using the MNIST dataset. The dataset contains 60,000 training images and 10,000 test images of handwritten digits ('0'–'9'), where each image is flattened into a 784-dimensional feature vector. The 60,000 training samples are distributed equally among the $M$ nodes. Unless otherwise specified (e.g., in the non-i.i.d. experiments in Sec. 9.1.5), the data are partitioned in an i.i.d. manner across nodes.

We benchmark RESIST against decentralized gradient descent (DGD) (Nedic & Ozdaglar, 2009) with multi-step consensus, which is known to fail under adversarial communication. In addition, we compare RESIST equipped with robust screening rules inherited from federated learning, including coordinate-wise median (Yin et al., 2018), Krum (Blanchard et al., 2017), and Bulyan (Mhamdi et al., 2018). In the non-i.i.d. setting, we compare with the *Byzantine-robust decentralized stochastic optimization* (DRSA) algorithm (Peng et al., 2021), which we evaluate under the MITM attack model through random link compromises. The relation of RESIST to BRIDGE-T, REDGRAF-CWTM, and ByRDiE is discussed in Sec. 1. In the $J$-ablation experiments below, we return to this comparison when interpreting the $J = 2$ curve.

We conduct five sets of experiments: (*i*) RESIST with different choices of the scheduling parameter $J$ to illustrate its effect on convergence in the strongly convex setting; (*ii*) RESIST under MITM attacks with varying attack budgets, compared with DGD using multi-step consensus; (*iii*) RESIST with varying network sizes $M \in \{10, 20, 50, 100\}$; (*iv*) RESIST with different screening rules (coordinate-wise median, Krum, and Bulyan); and (*v*) a comparison between RESIST and DRSA under extreme and moderate non-i.i.d. data distributions (Sec. 9.1.5). All attacks in these experiments correspond to the *random attack* strategy (Young & Yung, 1997; Bellare et al., 2014), in which the adversary replaces the transmitted vector on a compromised link with values randomly sampled from a Gaussian distribution with zero mean and unit variance.

Performance is evaluated using two metrics: the *average training loss* and the *average classification accuracy* on the 10,000 test images, both averaged across the $M$ nodes. In all reported results, the horizontal axis represents the *total number of iterations* (algorithmic rounds), accounting for both communication rounds and gradient updates. Unless explicitly marked as *faultless*, each communication round contains $b$ randomly compromised communication links, with the same value of $b$ used as the robustness parameter in RESIST.

### 9.1.1 Linear convergence for varying $J$

In this set of experiments, we fix $M = 50$, $\rho = 0.5$, and $b = 1$. The training data are partitioned i.i.d. across the nodes. We vary the parameter $J$ over the values $\{2, 6, 11, 21, 51\}$. When $J = 2$, RESIST performs one CWTM communication/filtering round before each local gradient update. With the constant stepsize used in these experiments, this is the fixed-step CWTM case corresponding to BRIDGE-T and REDGRAF-CWTM in the comparison described in Sec. 1 (note, however, that BRIDGE-T's convergence guarantees use diminishing stepsizes). This comparison is at the level of the CWTM update pattern, since BRIDGE-T and RED-

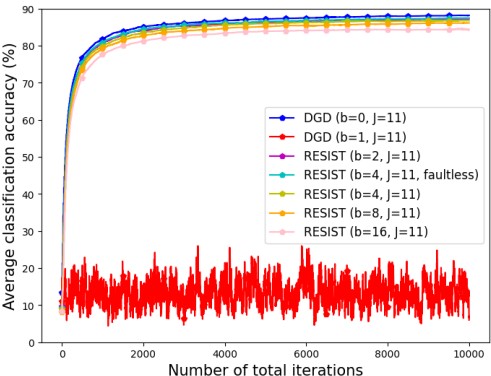

Figure 3: Comparison of RESIST and DGD with multi-step consensus under varying attack budgets $b$ on MNIST ($M = 50$, $J = 11$). For each curve, except the "faultless" setup, $b$ denotes both the attack budget anticipated by RESIST and the actual number of compromised links per communication round (i.e., $B = b$); "faultless" corresponds to $B = 0$.

GRAF-CWTM are formulated for Byzantine-node attacks, whereas the experiments here use the link-level MITM model. ByRDiE is not the corresponding comparison point because, as discussed in Sec. 1, it uses a coordinate-wise network-wide gradient loop that is substantially more expensive for high-dimensional models.

To illustrate the convergence behavior, we plot $\ln\left(\frac{1}{M} \sum_{j=1}^{M} f_j(\mathbf{w})\right)$ versus the total number of iterations (algorithmic rounds). The communication graph and attack configuration are kept fixed across all choices of $J$. For a fixed number of plotted iterations, larger $J$ uses fewer local gradient computations than the $J = 2$ case because more of the algorithmic rounds are communication/filtering rounds. Fig. 2 shows that larger $J$ can nevertheless improve the loss trajectory with respect to this iteration count, which can be especially useful when local gradient computation is substantially more expensive than communication and filtering. The plot reports iteration counts rather than wall-clock time, so this should be read as a computational interpretation rather than a direct timing measurement. In particular, after approximately 4000 iterations, the near-linear trend on the logarithmic scale is consistent with the geometric convergence rate established in the strongly convex analysis.

### 9.1.2 RESIST versus DGD with multi-step consensus under varying $b$

In this set of experiments, we fix $M = 50$ and $J = 11$ with i.i.d. data distribution. We vary the attack budget parameter $b \in \{0, 2, 4, 8, 16\}$, which represents the maximum number of compromised links that RESIST is designed to tolerate. To ensure that the connectivity requirements of Assumption 3.3 are satisfied for each attack level, we set the connection probability to $\rho = 0.5$ for $b \in \{0, 2, 4\}$, $\rho = 0.75$ for $b = 8$, and $\rho = 1$ for $b = 16$. At each communication round, $B$ communication links are randomly selected to undergo MITM attacks. In all experiments except those explicitly marked as "faultless," we set $B = b$, so that the realized number of compromised links matches the prescribed attack level. In the faultless case, we set $B = 0$, thereby evaluating the algorithm in the absence of attacks while keeping the same design budget $b$. For DGD with multi-step consensus, we report results only for $B = 0$ and $B = 1$.

As shown in Fig. 3, DGD achieves an accuracy of 88.16% when $B = 0$, which serves as the benchmark in this setting. This value is consistent with standard performance for a linear classifier on MNIST without data preprocessing. However, its performance deteriorates sharply even when $B = 1$, highlighting its vulnerability to adversarial communication and its inability to tolerate higher attack levels. In contrast, the accuracy of RESIST decreases gradually as the attack level increases. For example, the performance gap between $b = 2$ and $b = 8$ is approximately 2.5%, reflecting the trade-off between robustness and accuracy when selecting $b$. Moreover, comparing the faulty and faultless settings for the same $b$, the accuracy difference is about 0.5%, indicating that the impact of MITM attacks remains controlled and does not destabilize the learning dynamics. Overall, these results demonstrate that RESIST maintains stable performance under substantial adversarial interference, whereas classical decentralized gradient methods break down even under minimal attack.

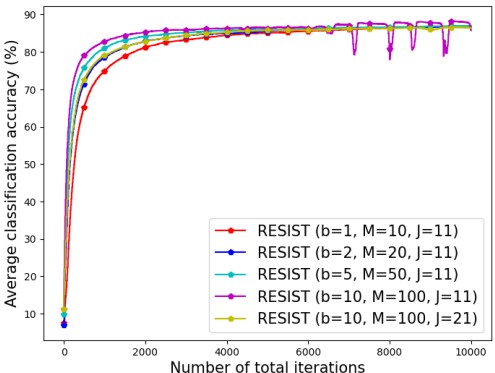

Figure 4: Performance of RESIST for different network sizes $M$ with $b = 0.1M$ and $\rho = 0.5$. Results are shown for $J = 11$ and $J = 21$.

### 9.1.3 RESIST under varying network sizes

These experiments evaluate how the convergence behavior of RESIST changes as the network size increases. To this end, we fix $\rho = 0.5$ and consider i.i.d. data distribution. We vary the network size $M \in \{10, 20, 50, 100\}$ and set the attack budget to $b = 0.1M$, so that the number of compromised links scales proportionally with the network size. At each communication round, exactly $b$ links are randomly selected to undergo MITM attacks. We consider $J \in \{11, 21\}$ to examine how the choice of $J$ interacts with increasing network size.

As shown in Fig. 4, the convergence behavior and final accuracy remain largely stable as $M$ increases up to 50 when $J = 11$. However, when $M = 100$ and $J = 11$, oscillatory behavior appears after approximately 7000 iterations, affecting the convergence dynamics. This observation is consistent with Theorem 5.5, which indicates that larger values of $J$ are required as the network size $M$ increases to maintain stability.

Although the theoretical lower bound on $J$ in Theorem 5.5 is conservative and need not be enforced exactly in practice, these experiments are consistent with the theoretical guidance that larger networks may require larger values of $J$ for stable convergence. Indeed, when $M = 100$ and $J = 21$, RESIST regains stable convergence and achieves final accuracy comparable to that observed for smaller networks. These results show that RESIST scales effectively with network size when $J$ is chosen in a way that reflects the network size.

### 9.1.4 RESIST with alternative screening rules under varying $b$

These experiments evaluate how the performance of RESIST depends on the choice of screening rule under adversarial attacks. In particular, we examine whether the robustness properties of RESIST are specific to the coordinate-wise trimmed mean or extend to other screening mechanisms developed for distributed and federated learning. We fix $M = 50$, $J = 11$, and $\rho = 0.5$ with i.i.d. data distribution, and consider attack budgets $b \in \{2, 4\}$. At each communication round, exactly $b$ communication links are randomly selected to undergo MITM attacks. We compare the original RESIST algorithm (with coordinate-wise trimmed mean screening) against three variants: RESIST-M, RESIST-K, and RESIST-B, obtained by replacing the trimmed mean screening rule with coordinate-wise median (Yin et al., 2018), Krum (Blanchard et al., 2017), and Bulyan (Mhamdi et al., 2018), respectively.

Fig. 5 shows that RESIST achieves stable convergence with all four screening rules, with only minor differences in average validation accuracy. When the number of compromised links increases from $b = 2$ to $b = 4$, the performance of each variant degrades slightly, as expected since a larger fraction of communication links is adversarially perturbed. Overall, these results indicate that the RESIST framework is not tied to a specific screening rule and maintains robustness across multiple screening strategies.

### 9.1.5 RESIST versus DRSA under non-i.i.d. data distributions

So far, our experiments have focused on i.i.d. data distributions across nodes. While the algorithmic convergence analysis in this paper accommodates heterogeneous local objectives through residual terms involving quantities such as $C_0 + \Delta$, the empirical results presented thus far correspond to identically distributed data. Here, we use explicitly non-i.i.d. label partitions to test how RESIST behaves when the local empirical objectives are more heterogeneous. These experiments are empirical robustness evaluations, not non-i.i.d. statistical learning-rate guarantees.

Among prior works reporting non-i.i.d. experiments are DRSA (Peng et al., 2021) and BRIDGE (Fang et al., 2022). As discussed earlier, the relevant comparison with BRIDGE-T is the fixed-stepsize $J = 2$ CWTM update, where one CWTM filtering round precedes each gradient step. We have already used that case as the BRIDGE-T reference in the $J$-ablation experiments, so we do not add a separate BRIDGE curve here. Instead, we compare RESIST directly with DRSA, using the same random link-level MITM compromise protocol for both methods.

We consider two non-i.i.d. data partitions: an extreme label-skew setting and a moderate label-skew setting. In both settings, we fix $M = 50$, $b \in \{2, 4\}$, $J = 11$, and $\rho = 0.5$.

**Extreme non-i.i.d. setting:** We partition the dataset by labels. For a network with 50 nodes, all samples labeled "0" are assigned to the first five nodes, all samples labeled "1" to the next five nodes, and so on, so that each group of five nodes contains data from only a single class. This construction induces substantial heterogeneity across nodes.

As shown in the left panel of Fig. 6, both algorithms perform well in the faultless setting. When the number of compromised links increases to two, the accuracy of both methods decreases by approximately 1%, and when it increases to four, the accuracy drops by roughly 3%. Despite the strong heterogeneity, RESIST maintains high classification accuracy.

The degradation is less pronounced than that reported in Fang et al. (2022), where the gap between faultless and faulty extreme non-i.i.d. settings under Byzantine node attacks is approximately 8%. This difference arises from the attack model: Byzantine node attacks can corrupt local datasets, which is particularly detrimental in extreme non-i.i.d. settings where entire classes may be concentrated on a small subset of nodes. In contrast, communication-level MITM attacks affect only transmitted messages and do not modify the underlying local datasets.

**Moderate non-i.i.d. setting:** We again partition the dataset by labels but distribute the samples of each label evenly across ten nodes so that each node receives data from two different classes. As shown in the right panel of Fig. 6, both algorithms maintain strong performance under zero, two, and four compromised links. Compared to the extreme case, the performance degradation is milder, indicating that the closer the data distribution is to i.i.d., the smaller the impact of adversarial communication.

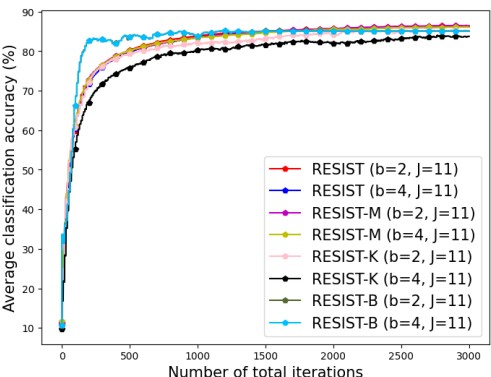

Figure 5: Comparison of RESIST with different screening rules under $b = 2$ and $b = 4$ compromised links on MNIST ($M = 50$, $J = 11$, $\rho = 0.5$).

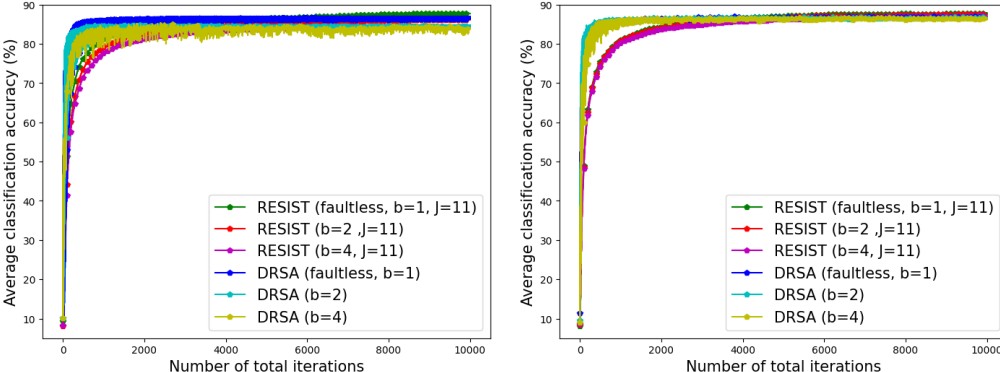

Figure 6: Comparison of RESIST and DRSA under zero, two, and four compromised links in the extreme non-i.i.d. setting (left) and moderate non-i.i.d. setting (right) on MNIST.

Overall, these experiments demonstrate that RESIST remains empirically robust under heterogeneous data distributions and adversarial communication. This behavior is consistent with the deterministic residual-neighborhood theory, where stronger heterogeneity can enlarge the residual terms, while the statistical guarantees in Sec. 8 remain stated for the common-population i.i.d. setting. A non-i.i.d. statistical extension is a natural direction, but, as discussed at the start of Sec. 8, deriving such rates would require specifying the sampling model, the target population risk, and the resulting population-heterogeneity terms.

### 9.2   Nonconvex setting: Convolutional neural networks on CIFAR-10

We next evaluate the performance of RESIST in the nonconvex regime using the CIFAR-10 dataset. We train a convolutional neural network (CNN) consisting of four convolutional layers, each followed by a max-pooling layer, and two fully connected layers. This architecture yields a smooth nonconvex objective function. The dataset contains 50,000 training images and 10,000 test images across 10 classes, where each image is represented as a 3,072-dimensional vector. The 50,000 training samples are distributed in an i.i.d. manner across the $M$ nodes (set to $M = 50$ unless otherwise specified).

We conduct six groups of experiments, varying one or two experimental factors at a time while fixing the others: ($i$) RESIST with different choices of the update-schedule parameter $J$; ($ii$) RESIST under MITM attacks with varying attack budgets, compared with DGD with multi-step consensus; ($iii$) RESIST with alternative screening rules inherited from distributed and federated learning, including coordinate-wise median (Yin et al., 2018) and Krum (Blanchard et al., 2017); ($iv$) RESIST under different types of MITM attacks; ($v$) RESIST under varying network sizes; and ($vi$) constant versus diminishing stepsizes.

Performance is evaluated using the average classification accuracy on the 10,000 test images, averaged across the $M$ local models. In all reported results, the horizontal axis represents the total number of iterations (algorithmic rounds), accounting for both communication rounds and local gradient-update rounds. Unless explicitly marked as *faultless*, each communication round contains $b$ randomly compromised communication links, with the same value of $b$ used as the robustness parameter in RESIST.

#### 9.2.1   Effect of the update-schedule parameter $J$

In this set of experiments, we fix $M = 50$, $\rho = 0.5$, and $b = 1$, with i.i.d. data distribution across nodes. We vary the update-schedule parameter $J \in \{2, 3, 6, 9\}$. As in the MNIST $J$-ablation discussion in Sec. 9.1.1, the $J = 2$ case is the fixed-step CWTM comparison point corresponding to BRIDGE-T and RedGraf-CWTM under the aligned update conventions discussed in Sec. 1.

As shown in Fig. 7, when the network topology and the number of compromised links are fixed, increasing $J$ improves the classification accuracy up to $J = 6$. Both $J = 3$ and $J = 6$ achieve higher accuracy than the fixed-step $J = 2$ CWTM comparison point while maintaining a comparable convergence speed. For a fixed

number of plotted iterations, these larger values of $J$ also use fewer local gradient computations than $J = 2$, because more of the algorithmic rounds are communication/filtering rounds. When $J = 9$, the convergence becomes slower, although the final accuracy remains higher than the same $J = 2$ comparison point. Although the theoretical analysis (e.g., Theorem 6.6) provides a lower bound on $J$ to guarantee stability, this bound is conservative in practice. Moreover, because the algorithmic-round budget in the experiments is finite, increasing $J$ does not necessarily improve convergence behavior, as illustrated here in Fig. 7. Consequently, $J$ should be treated as a tunable hyperparameter in practice rather than selected strictly according to the theoretical lower bound. Because the plot reports iterations rather than wall-clock time, this is not a timing comparison; rather, it shows that the larger-$J$ runs achieve these accuracy/convergence profiles while using fewer local gradient computations at the same plotted iteration count.

### 9.2.2 RESIST versus DGD with multi-step consensus under varying $b$

In this set of experiments, we fix $M = 50$, $J = 6$, and $\rho = 0.5$, with i.i.d. data distribution across nodes. We vary the attack budget parameter $b \in \{0, 1, 2, 4\}$, which represents the maximum number of communication links that RESIST is designed to tolerate per communication round. In all experiments except those explicitly marked as *faultless*, the actual number of compromised links per communication round is equal to $b$. For DGD with multi-step consensus, we report results only for $b = 0$ and $b = 1$, since the method fails to converge even when a single link is compromised.

As shown in Fig. 8, DGD with multi-step consensus achieves an accuracy of 59.16% in the absence of attacks, which is consistent with the centralized baseline for this architecture and serves as a reference point for comparison. However, its performance deteriorates substantially when a single link is compromised, indicating its sensitivity to adversarial interference. In contrast, RESIST maintains stable behavior under increasing attack intensity. Although accuracy decreases as $b$ grows, the degradation is gradual: the difference between $b = 0$ and $b = 4$ is approximately 1.3%. For $b = 1$, the faulty configuration differs from the faultless one by only about 0.4%, indicating that MITM attacks introduce only minor perturbations to the optimization trajectory. These observations highlight the algorithm's robustness in the nonconvex regime.

### 9.2.3 RESIST under varying network sizes

In this set of experiments, we evaluate the scalability of RESIST by varying the network size $M \in \{10, 20, 50, 100\}$. We fix $\rho = 0.5$ and use an i.i.d. data distribution across nodes. To maintain a comparable proportion of adversarially compromised communication links as the network grows, we set the attack budget to $b = 0.1M$, so that 10% of the links are randomly attacked at each communication round. We examine the performance for different values of $J \in \{3, 6, 11\}$.

As shown in Fig. 9, when $J = 6$ is fixed, increasing the network size while maintaining the same proportion of compromised links improves the accuracy up to $M = 50$. For larger networks, the performance becomes

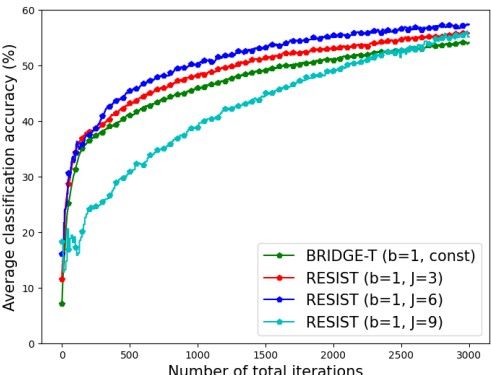

Figure 7: Performance of RESIST for different choices of the update-schedule parameter $J$ on CIFAR-10 ($M = 50$, $\rho = 0.5$, $b = 1$). "Const" indicates that a constant stepsize is used.

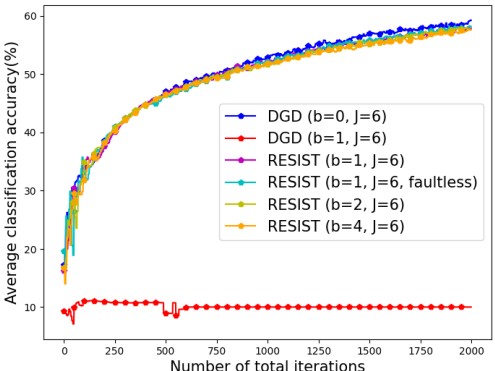

Figure 8: Comparison of RESIST and DGD with multi-step consensus under varying attack budgets on CIFAR-10 ($M = 50$, $J = 6$, $\rho = 0.5$).

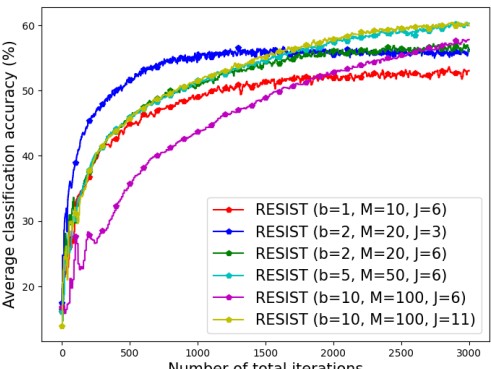

Figure 9: Performance of RESIST under varying network sizes $M$ and values of $J$ on CIFAR-10, with attack budget scaled as $b = 0.1M$.

more sensitive to the choice of $J$. To further illustrate this dependence, we compare different values of $J$ at fixed network sizes. When $M = 20$, both $J = 3$ and $J = 6$ achieve similar performance, indicating that moderate communication is sufficient at smaller scales. However, when $M = 100$, larger values of $J$ yield improved performance, which is consistent with the theoretical insights suggesting that stronger consensus (larger $J$) becomes increasingly important as the network grows.

This observation reinforces the earlier discussion: while the theoretical analysis (Theorem 6.6) indicates that $J$ should increase with $M$ to ensure stability, the derived bounds are conservative in practice. Consequently, $J$ serves as a tunable hyperparameter that may need to grow with network size to maintain stable performance. For a fixed algorithmic-round budget, larger $J$ also reduces the number of local gradient computations, which can be computationally useful when local gradient computation is the dominant runtime cost relative to communication and filtering.

### 9.2.4 RESIST with alternative screening rules under varying $b$

In this set of experiments, we examine the sensitivity of RESIST to the choice of screening rule in the nonconvex setting. We fix $M = 50$, $J = 6$, and $\rho = 0.5$ with an i.i.d. data distribution across nodes. We vary the attack budget $b \in \{1, 2, 4\}$, and at each communication round exactly $b$ communication links are randomly selected to undergo MITM attacks.

We compare the standard RESIST algorithm (which employs coordinate-wise trimmed mean) with two variants: RESIST-M, which replaces trimmed mean with coordinate-wise median (Yin et al., 2018), and

RESIST-K, which uses Krum (Blanchard et al., 2017). We exclude Bulyan (Mhamdi et al., 2018) in this setting due to its higher computational complexity for high-dimensional models such as CIFAR-10.

As shown in Fig. 10, RESIST and RESIST-M achieve comparable accuracy across all tested attack budgets, with only modest degradation as $b$ increases. In contrast, RESIST-K exhibits a more noticeable decline in accuracy as the number of compromised links grows, although it remains substantially more stable than DGD with multi-step consensus discussed in Sec. 9.2.2. Overall, these results indicate that coordinate-wise screening strategies such as trimmed mean and median provide stronger robustness in this setting, while the RESIST framework remains compatible with different screening mechanisms.

### 9.2.5 RESIST under different MITM attack strategies

In this set of experiments, we fix $M = 50$, $J = 6$, and $\rho = 0.5$ with an i.i.d. data distribution across nodes. We vary the attack budget $b \in \{2, 4\}$ and evaluate the robustness of RESIST under different MITM attack strategies. In addition to the *random attacks* described previously (Young & Yung, 1997; Bellare et al., 2014), we consider three structured attack models: (*i*) *sign-flipping attacks* (Taran et al., 2019; Xu et al., 2023; Park & Lee, 2024), where the transmitted vector is replaced by its negation; (*ii*) *label-flipping (data poisoning) attacks* (Alfeld et al., 2016; Tolpegin et al., 2020; Yerlikaya & Şerif Bahtiyar, 2022), where the adversary replaces the transmitted update with one generated as if half of the local training data were mislabeled; and (*iii*) *constant attacks* (Tilborg & Jajodia, 2011; Anderson, 2020), where the transmitted vector on a compromised link is replaced by the zero vector. At each communication round, exactly $b$ communication links are randomly selected to be compromised according to the specified attack model.

As shown in Fig. 11, RESIST exhibits strong robustness across all considered attack types. The degradation in accuracy when increasing the attack budget from $b = 2$ to $b = 4$ is modest (approximately 0.5%) for each attack strategy, indicating stable behavior as the number of compromised links grows. Across different attack types, the variation in performance is more pronounced, with accuracy differences ranging from approximately 1% to 3%. In particular, structured attacks such as sign-flipping and constant attacks tend to induce larger degradation than purely random attacks. This behavior is consistent with the nature of coordinate-wise screening methods, which more readily filter unstructured perturbations than adversarially aligned updates. Nevertheless, RESIST maintains stable training dynamics across all tested scenarios.

### 9.2.6 Effect of constant versus diminishing stepsizes

In this set of experiments, we examine the effect of the stepsize schedule on the performance of RESIST. We fix $M = 50$, $J = 6$, and $\rho = 0.5$ with an i.i.d. data distribution across nodes, and vary the attack budget $b \in \{1, 2, 4\}$. We compare a constant stepsize with a diminishing stepsize schedule. In the diminishing case, the stepsize decays proportionally to $1/t$.

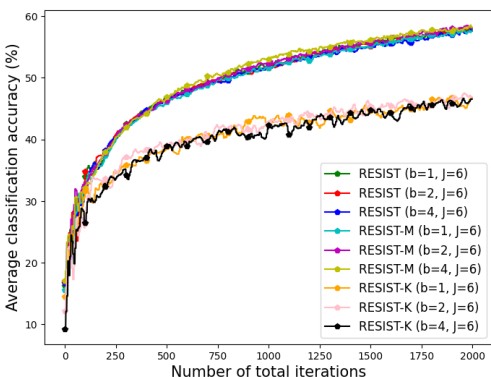

Figure 10: Comparison of RESIST (Trimmed Mean), RESIST-M (Median), and RESIST-K (Krum) under $b \in \{1, 2, 4\}$ compromised links on CIFAR-10 ($M = 50$, $J = 6$, $\rho = 0.5$).

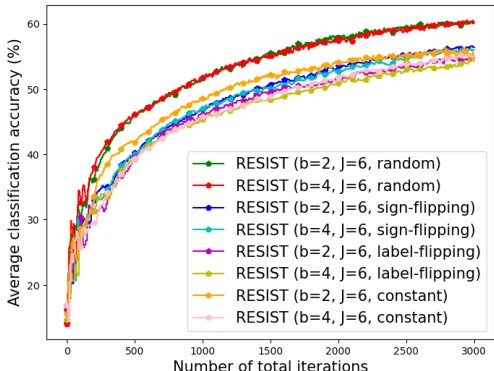

Figure 11: Performance of RESIST under different MITM attack strategies with $b = 2$ and $b = 4$ on CIFAR-10 ($M = 50$, $J = 6$, $\rho = 0.5$).

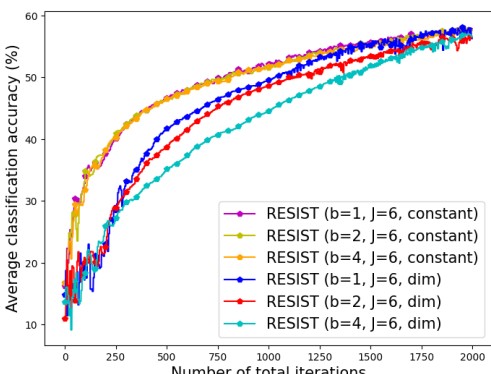

Figure 12: Comparison of RESIST with constant and diminishing stepsizes on CIFAR-10 ($M = 50$, $J = 6$, $\rho = 0.5$).

As shown in Fig. 12, the use of a constant stepsize leads to faster convergence in the early stages of training compared to the diminishing stepsize regime. But the final accuracy achieved by the diminishing schedule is comparable to that obtained with a properly tuned constant stepsize. These observations are consistent with the theoretical results. Theorem 8.4 establishes asymptotic stationarity under an appropriate diminishing stepsize, while Theorem 8.5 provides a finite-horizon bound on the average stationarity measure under a constant stepsize. In practice, a constant stepsize may be preferable when rapid convergence within a limited algorithmic-round budget is desired, whereas diminishing stepsizes remain theoretically attractive for asymptotic guarantees.

## 10  Conclusion

In this work, we introduced a novel algorithm termed Robust dEcentralized learning with conSensus gradIent deScenT (RESIST), designed to solve optimization and machine learning problems with data distributed across a decentralized communication network. We established algorithmic convergence guarantees with explicit residual-neighborhood characterizations under heterogeneous local objectives, together with statistical learning guarantees under the common-population i.i.d. setting for three classes of loss functions: strongly convex, PŁ, and smooth nonconvex objectives. To the best of our knowledge, this is the first work to formally model link-level Man-in-the-middle (MITM) attacks in decentralized optimization while providing rigorous convergence guarantees in strongly convex, PŁ, and smooth nonconvex settings. Numerical experiments on MNIST and CIFAR-10 further demonstrate the robustness of RESIST under varying attack budgets, values of the update-schedule parameter $J$, screening rules, network sizes, data partitions, and attack strategies.

Several directions remain for future work. These include formal statistical learning guarantees under explicitly non-i.i.d. sampling models, extensions to asynchronous communication protocols, improvements of convergence and statistical rates, and a deeper analysis of alternative screening mechanisms within decentralized learning frameworks.

**Broader Impact Statement**

This work advances the theoretical foundations of resilient decentralized machine learning, with potential benefits for safety-critical applications such as IoT systems, sensor networks, smart grids, and multi-agent systems that must operate without centralized coordination in adversarial environments. By providing provable guarantees against MITM attacks, RESIST can help practitioners build more trustworthy collaborative learning systems in settings where communication infrastructure may be compromised.

From a societal perspective, stronger defenses against adversarial interference in decentralized learning can reduce the risk of model manipulation in high-stakes deployments, such as autonomous systems, medical monitoring networks, and distributed infrastructure. As with any work that formalizes attack models and defenses, however, there is a potential dual-use concern: a detailed treatment of the MITM attack model could inform adversarial strategies in addition to defenses. We note that the MITM attack model is well-established in the communications and cybersecurity literature, and the primary contribution of this work is on the defensive side, providing robust-statistics-based screening methods and convergence guarantees that are resilient to such attacks. Practitioners deploying RESIST should remain aware that the robustness guarantees depend on network connectivity assumptions and the fraction of compromised communication links, and that exceeding these bounds may degrade the algorithm's resilience.

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

# A Supporting Preliminaries on the Connectivity of the Network

In this appendix, we will provide some preliminaries regarding the network connectivity and its associated lemmas, corollaries, and definitions, which will help us derive the consensus and convergence rates of the RESIST algorithm that are provided in Sections 5 and 6.

## A.1 Adaptation of Claim 2 from Vaidya (2012) for the coordinate-wise geometric mixing in Sec. 3.3

Recall from Lemma 3.4 that the mixing matrix $\mathbf{Y}_k(t)$ depends on the coordinate $k$. For simplicity of notation, we omit the $k$-dependency for the remainder of this appendix. Furthermore, since the mixing operations in Step 5 of the subroutine in Algorithm 2 occur independently across all $k \in \{1, \ldots, d\}$, we may, without loss of generality, take $d = 1$. In this case, the state matrix $\mathbf{W}(t)$ from Lemma 3.4 reduces to an $M$-dimensional vector.

Let $\mathbf{v}(0)$ denote the column vector of initial model parameters across all nodes. For $t \geqslant 1$, let $\mathbf{v}(t)$ denote the $M$-dimensional column vector consisting of the model parameters of all nodes at the end of iteration $t$. Note that when $d = 1$, the matrix $\mathbf{W}(t)$ in Lemma 3.4 coincides with the vector $\mathbf{v}(t)$. The $i$-th entry of $\mathbf{v}(t)$ is denoted by $v_i(t)$. Finally, let $\mathbf{y}_i(t)$ denote the $i$-th row of the matrix $\mathbf{Y}(t)$, where $i \in \mathcal{N}$.

**Corollary A.1.** *We can express the iterative update of the model parameters for any node $i \in \{1, \cdots, M\}$ performed in the CWTM step of Algorithm 1 in the following linear (matrix–vector) form:*[8]

$$v_i(t) = \mathbf{y}_i(t)\mathbf{v}(t - 1). \tag{107}$$

*The $i$-th row vector $\mathbf{y}_i(t)$ of the matrix $\mathbf{Y}(t)$ further satisfies the following four conditions:*

1. *$\mathbf{y}_i(t)$ is a stochastic row vector of size $M$. Thus, $[\mathbf{Y}(t)]_{ij} \geqslant 0$ for $1 \leqslant j \leqslant M$, and $\sum_{j=1}^M [\mathbf{Y}(t)]_{ij} = 1$.*

2. *$[\mathbf{Y}(t)]_{ii}$ equals $a_i$, where $a_i = \frac{1}{|\mathcal{N}_i| - 2b + 1}$, which is the weight that node $i$ assigns to itself.*

3. *$[\mathbf{Y}(t)]_{ij}$ is nonzero only if $(j, i) \in \mathcal{E}$ or $j = i$.*

4. *At least $|\mathcal{N}_i \backslash \mathcal{N}_i^b| - b + 1$ elements of $[\mathbf{Y}(t)]_i$ are lower bounded by some constant $\beta > 0$, where $\mathcal{N}_i^b$ denotes the time-dependent set, with the time index suppressed, of neighboring nodes whose links to node $i$ are compromised during the communication step associated with the update $v_i(t) = \mathbf{y}_i(t)\mathbf{v}(t - 1)$, and $b$ is the design parameter of the algorithm representing the upper bound on the number of compromised links the algorithm can defend against within each neighborhood. The constant $\beta$ is independent of $i$ and $t$, and its explicit value will be specified later in Sec. A.2.*

*Consequently, for $b < \min_j \frac{|\mathcal{N}_j|}{2}$, the scalar $v_i(t)$ is a convex combination of the entries of the vector $\mathbf{v}(t-1)$.*

The proof of this corollary follows that of Claim 2 in Vaidya (2012), with the distinction that we consider compromised links rather than compromised nodes, and is therefore omitted.

## A.2 Assumption on graph connectivity and its implications for the geometric mixing rate along coordinates in Sec. 3.3

From Vaidya (2012), we derive some basic results to establish the geometric mixing rate along coordinates. Recalling the filtered graph topology $\mathcal{T}_{\mathcal{F}}$ from Definition 3.2, let $\mathbf{H}$ denote the connectivity matrix for graph $\mathcal{H} \in \mathcal{T}_{\mathcal{F}}$, where $\mathbf{H}$ has entries 1 corresponding to an incoming edge and 0 otherwise.

**Lemma A.2** (Adaptation of Lemma 1 from Vaidya (2012)). *For any $\mathcal{H} \in \mathcal{T}_{\mathcal{F}}$, the matrix power $\mathbf{H}^M$ has at least one non-zero column.*[9]

---

[8]Recall that $\mathbf{y}_i(t)$ is the vector corresponding to the $i$-th row of the matrix $\mathbf{Y}(t)$. In addition to $t$, $\mathbf{y}_i(t)$ may depend on the vector $\mathbf{v}(t - 1)$ as well as the behavior of the compromised links incident to node $i$ that are under attack at time $t - 1$. For simplicity, this dependence is not explicitly reflected in the notation.

[9]The lemma continues to hold for any matrix power greater than $M$.

The proof is provided in Vaidya (2012).

**Definition A.3.** An element of a matrix is said to be "non-trivial" if it is lower bounded by a positive constant $\beta$.

Recall from Corollary A.1 that $a_i = \frac{1}{|\mathcal{N}_i| - 2b + 1}$. To establish a uniform lower bound applicable to both cases in Lemma 3.4 for all $i \in \{1, \ldots, M\}$, we define a section-specific constant $\alpha = \frac{1}{M - 2b + 1}$, which serves as a uniform lower bound on $a_i$. Applying this constant $\alpha$ to the corresponding formulations of $\mathbf{y}_i(t)$ in (11) and (12), we choose $\beta$ along similar lines as in Vaidya (2012):

$$\beta = \min_{k,i} \frac{\alpha}{4q_i^k} = \frac{\alpha}{4b}. \tag{108}$$

**Lemma A.4** (Adaptation of Lemma 2 from Vaidya (2012)). *For any $t \geqslant 1$, the screening in Algorithm 2 leads to a filtered graph $\mathcal{H}(t)$ that coincides with one of the filtered graphs $\mathcal{H} \in \mathcal{T}_\mathcal{F}$, and $\beta \mathbf{H}(t) \leqslant \mathbf{Y}(t)$, where $\mathbf{H}(t)$ is the connectivity matrix associated with $\mathcal{H}(t)$ at time $t$ and $\beta$ is defined above.*

*Proof.* The proof follows along similar lines as in Vaidya (2012). Observe that the $i$-th row of the weight matrix $\mathbf{Y}(t)$ corresponds to the update of $\mathbf{v}(t)$ performed at node $i$. Recall that $[\mathbf{Y}(t)]_{ij}$ is non-zero only if $(j, i) \in \mathcal{E}$. Also, by Corollary A.1, $\mathbf{y}_i(t)$ (i.e., the $i$-th row of $\mathbf{Y}(t)$) contains at least $|\mathcal{N}_i \backslash \mathcal{N}_i^b| - b + 1$ non-trivial elements corresponding to uncompromised incoming edges of node $i$ and itself (i.e., the diagonal element).

Now observe that, since $|\mathcal{N}_i^b| \leqslant b$, the number of non-trivial entries in $\mathbf{y}_i(t)$ identified above is at least $|\mathcal{N}_i| - 2b + 1$. We may therefore choose $|\mathcal{N}_i| - 2b + 1$ of these non-trivial positions, including the diagonal position, as the non-zero positions in the $i$-th row of $\mathbf{H}(t)$. Applying this choice row by row gives a filtered graph $\mathcal{H}(t) \in \mathcal{T}_\mathcal{F}$. By construction, every non-zero entry of $\mathbf{H}(t)$ corresponds to an entry of $\mathbf{Y}(t)$ that is at least $\beta$, and hence $\beta \mathbf{H}(t) \leqslant \mathbf{Y}(t)$. ∎

### A.3 Stochastic matrix properties for the geometric mixing rate along coordinates in Sec. 3.3

We note that this subsection corresponds to the presentation in Vaidya (2012), but we provide the details here to clarify the definitions and properties used in our analysis. For a row stochastic matrix $\mathbf{A}$, the coefficients of ergodicity $\delta(\mathbf{A})$ and $\lambda(\mathbf{A})$ are defined as in Wolfowitz (1963):

$$\delta(\mathbf{A}) := \max_j \max_{i_1, i_2} \left| [\mathbf{A}]_{i_1 j} - [\mathbf{A}]_{i_2 j} \right|,$$

$$\lambda(\mathbf{A}) := 1 - \min_{i_1, i_2} \sum_j \min \left( [\mathbf{A}]_{i_1 j}, [\mathbf{A}]_{i_2 j} \right).$$

It is easy to see that $0 \leqslant \delta(\mathbf{A}) \leqslant 1$ and $0 \leqslant \lambda(\mathbf{A}) \leqslant 1$, and that the rows are identical if and only if $\delta(\mathbf{A}) = 0$. Additionally, $\lambda(\mathbf{A}) = 0$ if and only if $\delta(\mathbf{A}) = 0$.

The next result from Hajnal & Bartlett (1958) establishes a relation between the coefficient of ergodicity $\delta(\cdot)$ of a product of row stochastic matrices and the coefficients of ergodicity $\lambda(\cdot)$ of the individual matrices defining the product.

**Proposition A.5** ((Hajnal & Bartlett, 1958)). *Let $\mathbf{Q}(1), \mathbf{Q}(2), \ldots, \mathbf{Q}(p)$ be square row-stochastic matrices with the same dimensions and $p \geqslant 1$. Then, $\delta(\mathbf{Q}(1)\mathbf{Q}(2) \cdots \mathbf{Q}(p)) \leqslant \prod_{i=1}^p \lambda(\mathbf{Q}(i))$.*

Proposition A.5 implies that if, for all $i$, $\lambda(\mathbf{Q}(i)) \leqslant 1 - \gamma$ for some $\gamma > 0$, then $\delta(\mathbf{Q}(1)\mathbf{Q}(2) \cdots \mathbf{Q}(p))$ converges to zero as $p \to \infty$. We next consider the notion of a scrambling matrix, which has also been studied in the literature (Hajnal & Bartlett, 1958; Wolfowitz, 1963).

**Definition A.6.** A row-stochastic matrix $\mathbf{H}$ is said to be a scrambling matrix if $\lambda(\mathbf{H}) < 1$.

**Remark A.7.** In a scrambling matrix $\mathbf{H}$, since $\lambda(\mathbf{H}) < 1$, for each pair of rows $i_1$ and $i_2$, there exists a column $j$ (which may depend on $i_1$ and $i_2$) such that $[\mathbf{H}]_{i_1 j} > 0$ and $[\mathbf{H}]_{i_2 j} > 0$ (Hajnal & Bartlett, 1958; Wolfowitz, 1963). As a special case, if any one column of a row-stochastic matrix $\mathbf{H}$ contains only nonzero elements that are lower bounded by some constant $\gamma > 0$, then $\mathbf{H}$ must be scrambling, and $\lambda(\mathbf{H}) \leqslant 1 - \gamma$.

### A.4 Consensus guarantees with geometric convergence

To show that consensus is achieved at a geometric rate, we again follow the proof techniques from Vaidya (2012).

**Lemma A.8** (Adaptation of Lemma 3 from Vaidya (2012)). *In the product $\prod_{t=z}^{z+\tau M-1} \mathbf{H}(t)$ of $\mathbf{H}(t)$ matrices over $\tau M$ consecutive iterations for any $z \geqslant 0$, at least one column is non-zero.*

*Proof.* Since the product $\prod_{t=z}^{z+\tau M-1} \mathbf{H}(t)$ consists of $\tau M$ matrices in $\mathcal{T}_{\mathcal{F}}$, at least one of the $\tau$ distinct connectivity matrices in $\mathcal{T}_{\mathcal{F}}$, say $\mathbf{H}_*$, must appear in the above product at least $M$ times by the pigeonhole principle. Now observe that: $(i)$ by Lemma A.2, $\mathbf{H}_*^M$ contains a non-zero column (say the $k$-th column), and $(ii)$ all the $\mathbf{H}(t)$ matrices in the product have non-zero diagonal entries. These two observations together imply that the $k$-th column in the above product is non-zero. ∎

Recall the sequence of matrices $\mathbf{Q}(i)$ used in Sec. 5, where each $\mathbf{Q}(i)$ is defined as a product of $\tau M$ consecutive $\mathbf{Y}(t)$ matrices. Specifically, $\mathbf{Q}(i) = \prod_{t=(i-1)\tau M+1}^{i\tau M} \mathbf{Y}(t)$. Combining this definition with (107), we have $\mathbf{v}(k\tau M) = \left(\prod_{i=1}^{k} \mathbf{Q}(i)\right)\mathbf{v}(0)$.

**Lemma A.9** (Adaptation of Lemma 4 from Vaidya (2012)). *For $i \geqslant 1$, $\mathbf{Q}(i)$ is a scrambling row-stochastic matrix, and $\lambda(\mathbf{Q}(i))$ is bounded above by $1 - \beta^{\tau M}$.*

*Proof.* Since $\mathbf{Q}(i)$ is a product of row-stochastic matrices $\{\mathbf{Y}(t)\}$, it is row stochastic. From Lemma A.4, for each $t$, $\beta\mathbf{H}(t) \leqslant \mathbf{Y}(t)$. Therefore, $\beta^{\tau M} \prod_{t=(i-1)\tau M+1}^{i\tau M} \mathbf{H}(t) \leqslant \mathbf{Q}(i)$.

Using $z = (i-1)\tau M + 1$ in Lemma A.8, we conclude that the matrix product on the left-hand side of the above inequality contains a non-zero column. Therefore, $\mathbf{Q}(i)$ also contains a non-zero column and is thus a scrambling matrix by Remark A.7.

Observe that $\tau M$ is finite; hence $\beta^{\tau M}$ is non-zero. Since the non-zero entries in the $\mathbf{H}(t)$ matrices are all equal to 1, the non-zero elements in $\prod_{t=(i-1)\tau M+1}^{i\tau M} \mathbf{H}(t)$ must each be greater than or equal to 1. Therefore, there exists a non-zero column in $\mathbf{Q}(i)$ whose entries are all greater than or equal to $\beta^{\tau M}$, and consequently $\lambda(\mathbf{Q}(i)) \leqslant 1 - \beta^{\tau M}$. ∎

**Lemma A.10.** *For the update $\mathbf{v}(t) = \mathbf{Y}(t)\mathbf{v}(t-1)$ and any time index $t_0$, we have the following geometric rate for $t > t_0$ and every $i$ and $j$:*

$$\left|[\boldsymbol{\Phi}(t, t_0)]_{ji} - [\mathbf{c}]_i\right| \leqslant (1 - \beta^{\tau M})^{\left\lfloor \frac{t-t_0}{\tau M} \right\rfloor} \tag{109}$$

*for some stochastic vector $\mathbf{c}$ and $\boldsymbol{\Phi}(t, t_0) := \mathbf{Y}(t)\mathbf{Y}(t-1)\cdots\mathbf{Y}(t_0)$. Also, for some vector $\boldsymbol{\alpha} = \alpha\mathbf{1}$ with a scalar $\alpha$, we have*

$$\lim_{t \to \infty} \mathbf{v}(t) = \boldsymbol{\alpha}.$$

*Proof.* By Proposition A.5,

$$\lim_{t \to \infty} \delta\left(\Pi_{i=t_0}^{t} \mathbf{Y}(i)\right) \leqslant \lim_{t \to \infty} \Pi_{i=t_0}^{t} \lambda(\mathbf{Y}(i)) \tag{110}$$

$$\leqslant \lim_{t \to \infty} \Pi_{i=t_0}^{\left\lfloor \frac{t}{\tau M} \right\rfloor} \lambda(\mathbf{Q}(i)) \tag{111}$$

$$= 0. \tag{112}$$

The above argument uses the facts that $\lambda(\mathbf{Y}(t)) \leqslant 1$ and $\lambda(\mathbf{Q}(i)) \leqslant (1 - \beta^{\tau M}) < 1$ from Lemma A.9. Thus, the rows of the matrix $\prod_{i=t_0}^{t} \mathbf{Y}(i)$ become identical as $t \to \infty$.

So far, we have only deduced weak ergodicity (which indicates that the limit $\prod_{i=t_0}^{\infty} \mathbf{Y}(i)$ is independent of the initial time $t_0$) of the infinite product $\prod_{i=t_0}^{\infty} \mathbf{Y}(i)$. However, Theorem A in Leizarowitz (1992) states that weak ergodicity is equivalent to strong ergodicity (which indicates that the matrices are uniformly mixing

and all trajectories converge to the same stationary distribution) in the case of backward products. Since the product under any arbitrary permutation[10] of $\{\mathbf{Y}(t)\}_t$ contains a non-zero column, by Lemmas A.8 and A.9, we conclude that the infinite product $\prod_{i=t_0}^{\infty} \mathbf{Y}(i)$ is a scrambling matrix and hence converges.

Suppose the rows of this infinite product converge to a stochastic vector $\mathbf{c}$, and thus $\mathbf{\Phi}(t, t_0) \to \mathbf{C}$ as $t \to \infty$, where the rows of $\mathbf{C}$ are identical and equal to the transpose of $\mathbf{c}$. Together with the fact that $\mathbf{v}(t) = (\Pi_{i=1}^{t} \mathbf{Y}(i)) \mathbf{v}(0)$, this implies that the nodes achieve consensus to some vector $\boldsymbol{\alpha} = \mathbf{Cv}(0)$ with $\boldsymbol{\alpha} = \alpha \mathbf{1}$, i.e.,

$$\lim_{t \to \infty} \mathbf{v}(t) = \lim_{t \to \infty} \left( \Pi_{i=1}^{t} \mathbf{Y}(i) \right) \mathbf{v}(0) = \boldsymbol{\alpha}.$$

Finally, using the ergodicity property in Leizarowitz (1992), we have $\delta(\mathbf{\Phi}(t, t_0)) = \delta(\mathbf{\Phi}(t, t_0) - \mathbf{C})$, which yields the rate

$$\left| [\mathbf{\Phi}(t, t_0)]_{ji} - [\mathbf{c}]_i \right| \leqslant \delta(\mathbf{\Phi}(t, t_0) - \mathbf{C}) \leqslant (1 - \beta^{\tau M})^{\left\lfloor \frac{t - t_0}{\tau M} \right\rfloor}. \tag{113}$$

This completes the proof. ∎

# B  Weight Assignment for the Mixing Matrix

In this appendix, we provide a choice of the weight assignment used in the analysis of the RESIST algorithm along with an associated example to demonstrate that our screening method guarantees that the update only involves information that is not compromised.

## B.1  Proof of Lemma 3.4

*Proof.* Let us define the notation $b_j^*(t) := |\mathcal{N}_j^b(t)|$ as the actual (unknown) number of nodes in the graph that have compromised outgoing edges to node $j$. Then we must have that $b_j^*(t) \leqslant b$ for all $t$ and $j$. To make the rest of the expressions clearer, we drop the iteration index $t$ for the remainder of this discussion wherever appropriate, even though the variables remain $t$-dependent. We will, however, occasionally retain $k$-dependency where the variables depend on the $k$-th coordinate.

Next, suppose $b_j^k$ is the number of nodes with compromised edges to $j$ that remain in the filtered set $\mathcal{C}_j^k$, and define $q_j^k := b - b_j^* + b_j^k$. Since by definition $b - b_j^* \geqslant 0$ and $b_j^k \geqslant 0$, only one of two cases can occur during each iteration for every coordinate $k$: $(i)$ $q_j^k > 0$ or $(ii)$ $q_j^k = 0$.

For case $(i)$, we either have $b - b_j^* > 0$, or $b_j^k > 0$, or both. Since at most $b_j^* \leqslant b$ incoming edges to node $j$ are compromised, and exactly $b$ largest and $b$ smallest entries are removed in the screening step, it follows that neither $\overline{\mathcal{N}}_j^k$ nor $\underline{\mathcal{N}}_j^k$ can consist entirely of compromised nodes. More precisely, let $\bar{b}_j^k := |\overline{\mathcal{N}}_j^k \cap \mathcal{N}_j^b|$ and $\underline{b}_j^k := |\underline{\mathcal{N}}_j^k \cap \mathcal{N}_j^b|$. Then $b_j^* = \bar{b}_j^k + \underline{b}_j^k + b_j^k$ and $q_j^k = b - b_j^* + b_j^k = b - \bar{b}_j^k - \underline{b}_j^k$. Thus $|\overline{\mathcal{N}}_j^k \cap \mathcal{N}_j^r| = q_j^k + \underline{b}_j^k \geqslant q_j^k$ and $|\underline{\mathcal{N}}_j^k \cap \mathcal{N}_j^r| = q_j^k + \bar{b}_j^k \geqslant q_j^k$. Choose subsets $\overline{\mathcal{A}}_j^k \subseteq \overline{\mathcal{N}}_j^k \cap \mathcal{N}_j^r$ and $\underline{\mathcal{A}}_j^k \subseteq \underline{\mathcal{N}}_j^k \cap \mathcal{N}_j^r$ with $|\overline{\mathcal{A}}_j^k| = |\underline{\mathcal{A}}_j^k| = q_j^k$.

For every $i' \in \mathcal{C}_j^k$, the ordering after the screening step gives $[\mathbf{w}_m]_k \leqslant [\mathbf{w}_{i'}]_k \leqslant [\mathbf{w}_{m'}]_k$ for all $m \in \underline{\mathcal{A}}_j^k$ and $m' \in \overline{\mathcal{A}}_j^k$. Averaging over the selected sets gives $\frac{1}{q_j^k} \sum_{m \in \underline{\mathcal{A}}_j^k} [\mathbf{w}_m]_k \leqslant [\mathbf{w}_{i'}]_k \leqslant \frac{1}{q_j^k} \sum_{m \in \overline{\mathcal{A}}_j^k} [\mathbf{w}_m]_k$, and hence there exists $\theta_{i'}^k \in [0, 1]$ such that

$$[\mathbf{w}_{i'}]_k = \theta_{i'}^k \frac{1}{q_j^k} \sum_{m \in \overline{\mathcal{A}}_j^k} [\mathbf{w}_m]_k + (1 - \theta_{i'}^k) \frac{1}{q_j^k} \sum_{m \in \underline{\mathcal{A}}_j^k} [\mathbf{w}_m]_k.$$

If these two averages are equal, choose any $\theta_{i'}^k \in [0, 1]$. For each regular node $i' \in \mathcal{C}_j^k \cap \mathcal{N}_j^r$, the weight associated with its center value is split: half remains on $i'$, and the other half is assigned according to the representation above. For each compromised center value indexed by $i' \in \mathcal{C}_j^k \cap \mathcal{N}_j^b$, the full weight is assigned through this representation. Consequently, the elements of the matrix $\mathbf{Y}_k$ can then be written as in (11).

---

[10]The conclusion of Lemma A.8 still holds for any arbitrary order of multiplication due to strong ergodicity.

For case $(ii)$, we must have $b - b_j^* = 0$ and $b_j^k = 0$. Thus, all nodes remaining in $\mathcal{C}_j^k$ have uncompromised edges to $j$. Therefore, we can describe $\mathbf{Y}_k$ in this case as in (12).

Combining the expressions of $\mathbf{Y}_k$ in the two cases above allows us to express the update in (9) exclusively in terms of uncompromised information. ∎

### B.2 An illustrative example of the weight assignment

Consider the network shown in Fig. 13, where each node broadcasts a two-dimensional model vector $\mathbf{w}_i(t) \in \mathbb{R}^2$ to all of its neighbors. We set $b = 1$, so that each node can tolerate at most one compromised incoming link per iteration. Gray directed edges deliver the true broadcast vector, whereas the two red directed edges represent compromised transmissions. In particular, node $\mathbf{A}$ receives $\tilde{\mathbf{w}}_{\mathbf{B} \to \mathbf{A}}(t) = [3, 8]^\top$ instead of $\mathbf{w}_{\mathbf{B}}(t)$, and node $\mathbf{E}$ receives $\tilde{\mathbf{w}}_{\mathbf{C} \to \mathbf{E}}(t) = [6, 7]^\top$. All other received messages coincide with the broadcast vectors shown in the figure. For simplicity, we omit the time index $t$ in the discussion below and break ties deterministically. To make the construction explicit, we derive the first-coordinate mixing matrix $\mathbf{Y}_1(t)$ induced by the screening procedure. The broadcast first-coordinate values are $7, 5, 4, 2, 2$ for nodes $\mathbf{A}, \mathbf{B}, \mathbf{C}, \mathbf{D}, \mathbf{E}$, respectively.

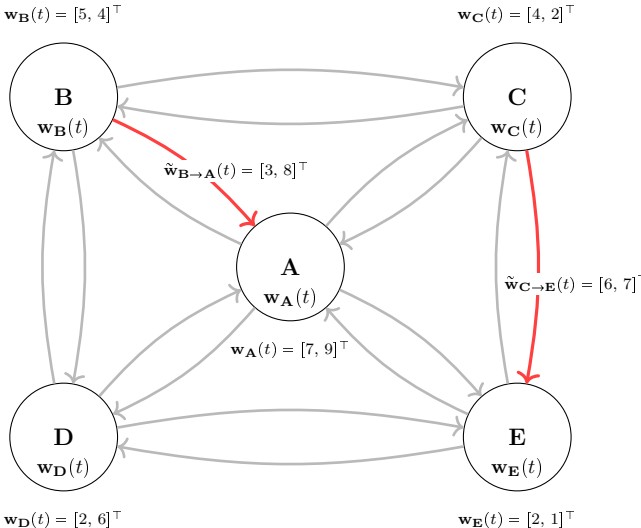

Figure 13: Coordinate-wise screening and weight assignment with $b = 1$. Each node broadcasts $\mathbf{w}_i(t)$ to all neighbors; on the two red directed links, the receiver obtains the corrupted vectors $\tilde{\mathbf{w}}_{\mathbf{B} \to \mathbf{A}}(t)$ and $\tilde{\mathbf{w}}_{\mathbf{C} \to \mathbf{E}}(t)$ shown above.

Consider node $\mathbf{A}$. According to Algorithm 2, filtering is performed only over its incoming neighbors $\mathcal{N}_{\mathbf{A}} = \{\mathbf{B}, \mathbf{C}, \mathbf{D}, \mathbf{E}\}$, from which it receives first-coordinate values $\{3, 4, 2, 2\}$. After sorting and removing the largest and smallest values ($b = 1$), the upper set is $\{\mathbf{C}\}$ with value 4, the lower set is $\{\mathbf{D}\}$ with value 2 (by tie-breaking), and the center set is $\{\mathbf{B}, \mathbf{E}\}$ with values $\{3, 2\}$. Node $\mathbf{A}$ retains its own value 7 unconditionally. Since $|\mathcal{N}_{\mathbf{A}}| - 2b + 1 = 3$, the baseline weight is $1/3$. Because the actual number of compromised incoming links is $b_{\mathbf{A}}^* = 1$ and one compromised link remains in the center set ($b_{\mathbf{A}}^1 = 1$), we have $q_{\mathbf{A}}^1 = b - b_{\mathbf{A}}^* + b_{\mathbf{A}}^1 = 1$, so (11) applies. Writing the center-set values as convex combinations of the upper and lower sets gives $3 = 0.5 \cdot 4 + 0.5 \cdot 2$ (for $\mathbf{B}$) and $2 = 0 \cdot 4 + 1 \cdot 2$ (for $\mathbf{E}$). Redistributing the corresponding baseline weights yields contributions $\frac{1}{6}$ to $\mathbf{C}$ and $\frac{1}{6}$ to $\mathbf{D}$ from $\mathbf{B}$, and an additional $\frac{1}{6}$ to $\mathbf{D}$ from $\mathbf{E}$ (with $\frac{1}{6}$ retained by $\mathbf{E}$). Consequently, $[\mathbf{Y}_1]_{\mathbf{A}\mathbf{A}} = \frac{1}{3}$, $[\mathbf{Y}_1]_{\mathbf{A}\mathbf{E}} = \frac{1}{6}$, $[\mathbf{Y}_1]_{\mathbf{A}\mathbf{C}} = \frac{1}{6}$, $[\mathbf{Y}_1]_{\mathbf{A}\mathbf{D}} = \frac{1}{3}$, and $[\mathbf{Y}_1]_{\mathbf{A}\mathbf{B}} = 0$.

Next consider node $\mathbf{B}$. It receives first-coordinate values $\{7, 4, 2\}$ from neighbors $\{\mathbf{A}, \mathbf{C}, \mathbf{D}\}$. After removing the largest value 7 and the smallest value 2 ($b = 1$), the center set is $\{\mathbf{C}\}$ with value 4. Since node $\mathbf{B}$ has no compromised incoming links, $b_{\mathbf{B}}^* = 0$ and $b_{\mathbf{B}}^1 = 0$, hence $q_{\mathbf{B}}^1 = b - b_{\mathbf{B}}^* + b_{\mathbf{B}}^1 = 1$, so (11) applies. The center value 4 is written as a convex combination of the upper and lower sets, $4 = \frac{2}{5} \cdot 7 + \frac{3}{5} \cdot 2$. Because $|\mathcal{N}_{\mathbf{B}}| - 2b + 1 = 2$, the baseline weight is $1/2$. According to (11), node $\mathbf{C}$ retains half of this baseline weight,

namely 1/4, and the remaining 1/4 is redistributed, yielding $\frac{2}{5} \cdot \frac{1}{4} = \frac{1}{10}$ to $\mathbf{A}$ and $\frac{3}{5} \cdot \frac{1}{4} = \frac{3}{20}$ to $\mathbf{D}$. Node $\mathbf{B}$ retains its own weight 1/2. Thus, $[\mathbf{Y}_1]_{\mathbf{BB}} = \frac{1}{2}$, $[\mathbf{Y}_1]_{\mathbf{BC}} = \frac{1}{4}$, $[\mathbf{Y}_1]_{\mathbf{BA}} = \frac{1}{10}$, and $[\mathbf{Y}_1]_{\mathbf{BD}} = \frac{3}{20}$.

Nodes $\mathbf{C}$ and $\mathbf{D}$ are treated analogously. Each receives $\{7, 5, 2\}$ from $\{\mathbf{A}, \mathbf{B}, \mathbf{E}\}$, removes the largest value 7 and the smallest value 2, and retains the center set $\{\mathbf{B}\}$ with value 5. Since neither node has compromised incoming links, $q_{\mathbf{C}}^1 = q_{\mathbf{D}}^1 = 1$, and (11) applies. Writing $5 = \frac{3}{5} \cdot 7 + \frac{2}{5} \cdot 2$, and noting that the baseline weight is again 1/2, each node assigns 1/4 to $\mathbf{B}$ and redistributes the remaining 1/4, yielding $\frac{3}{20}$ to $\mathbf{A}$ and $\frac{1}{10}$ to $\mathbf{E}$. Consequently, $[\mathbf{Y}_1]_{\mathbf{CC}} = \frac{1}{2}$, $[\mathbf{Y}_1]_{\mathbf{CB}} = \frac{1}{4}$, $[\mathbf{Y}_1]_{\mathbf{CA}} = \frac{3}{20}$, $[\mathbf{Y}_1]_{\mathbf{CE}} = \frac{1}{10}$, and similarly $[\mathbf{Y}_1]_{\mathbf{DD}} = \frac{1}{2}$, $[\mathbf{Y}_1]_{\mathbf{DB}} = \frac{1}{4}$, $[\mathbf{Y}_1]_{\mathbf{DA}} = \frac{3}{20}$, $[\mathbf{Y}_1]_{\mathbf{DE}} = \frac{1}{10}$.

Finally, consider node $\mathbf{E}$. It receives first-coordinate values $\{7, 6, 2\}$ from neighbors $\{\mathbf{A}, \mathbf{C}, \mathbf{D}\}$, where the value 6 corresponds to the compromised transmission on the link $\mathbf{C} \to \mathbf{E}$. After removing the largest value 7 and the smallest value 2 ($b = 1$), the center set is $\{\mathbf{C}\}$ with value 6. Since $|\mathcal{N}_{\mathbf{E}}| - 2b + 1 = 2$, the baseline weight is 1/2. Moreover, the compromised link remains in the center set, so $b_{\mathbf{E}}^* = 1$ and $b_{\mathbf{E}}^1 = 1$, which gives $q_{\mathbf{E}}^1 = 1$ and places $\mathbf{E}$ in the case (11). Thus, $\mathbf{E}$ keeps its self-weight $[\mathbf{Y}_1]_{\mathbf{EE}} = 1/2$, while the entire baseline weight 1/2 associated with the compromised center value is redistributed to the upper and lower sets. Writing 6 as a convex combination of the upper and lower values yields $6 = \frac{4}{5} \cdot 7 + \frac{1}{5} \cdot 2$, so the redistribution contributes $[\mathbf{Y}_1]_{\mathbf{EA}} = \frac{4}{5} \cdot \frac{1}{2} = \frac{2}{5}$ and $[\mathbf{Y}_1]_{\mathbf{ED}} = \frac{1}{5} \cdot \frac{1}{2} = \frac{1}{10}$, with $[\mathbf{Y}_1]_{\mathbf{EC}} = 0$.

Collecting the rows in the order $\mathbf{A}, \mathbf{B}, \mathbf{C}, \mathbf{D}, \mathbf{E}$, the first-coordinate mixing matrix is

$$
\mathbf{Y}_1(t) = \begin{pmatrix} \frac{1}{3} & 0 & \frac{1}{6} & \frac{1}{3} & \frac{1}{6} \\ \frac{1}{10} & \frac{1}{2} & \frac{1}{4} & \frac{3}{20} & 0 \\ \frac{3}{20} & \frac{1}{4} & \frac{1}{2} & 0 & \frac{1}{10} \\ \frac{3}{20} & \frac{1}{4} & 0 & \frac{1}{2} & \frac{1}{10} \\ \frac{2}{5} & 0 & 0 & \frac{1}{10} & \frac{1}{2} \end{pmatrix}.
$$

Each row is stochastic and, after redistribution, depends only on uncompromised information from the upper and lower sets, in agreement with Lemma 3.4. The second-coordinate matrix $\mathbf{Y}_2(t)$ is obtained analogously from the second-coordinate received values, and the same construction applies coordinate-wise in higher dimensions.

## C   Proofs of Supporting Lemmas Used to Derive the Consensus Guarantee

### C.1   Proof of Lemma 4.6

Applying the $\overline{(\cdot)}$ operator to both sides of (19) we get the following update:

$$
[\overline{\mathbf{W}}(s+1)]_k = \frac{\mathbf{1}\mathbf{1}^T}{M} \mathbf{Q}_k(s) [\mathbf{W}(s)]_k - h[\overline{\mathbf{T}}(s)]_k. \tag{114}
$$

Next, subtracting (114) from (19) we obtain:

$$
[\overline{\mathbf{W}}(s+1)]_k - [\mathbf{W}(s+1)]_k = (\frac{\mathbf{1}\mathbf{1}^T}{M} - \mathbf{I})\mathbf{Q}_k(s)[\mathbf{W}(s)]_k - h([\overline{\mathbf{T}}(s)]_k - [\mathbf{T}(s)]_k) \tag{115}
$$

$$
= (\frac{\mathbf{1}\mathbf{1}^T}{M} - \mathbf{I})\mathbf{Q}_k(s)([\mathbf{W}(s)]_k - [\overline{\mathbf{W}}(s)]_k) - h([\overline{\mathbf{T}}(s)]_k - [\mathbf{T}(s)]_k) \tag{116}
$$

$$
= (\frac{\mathbf{1}\mathbf{1}^T}{M} - \mathbf{I})(\mathbf{Q}_k(s) - \mathbf{1}\mathbf{c}_k(s)^T)([\mathbf{W}(s)]_k - [\overline{\mathbf{W}}(s)]_k)
$$
$$
- h([\overline{\mathbf{T}}(s)]_k - [\mathbf{T}(s)]_k), \tag{117}
$$

where in the second step we used the fact that the vector $[\overline{\mathbf{W}}(s)]_k$ has identical entries and hence lies in the null space of $(\frac{\mathbf{1}\mathbf{1}^T}{M} - \mathbf{I})\mathbf{Q}_k(s)$ and in the last step we used the fact that the vector $\mathbf{1}\mathbf{c}_k(s)^T([\mathbf{W}(s)]_k - [\overline{\mathbf{W}}(s)]_k)$ has identical entries and hence lies in the null space of $\frac{\mathbf{1}\mathbf{1}^T}{M} - \mathbf{I}$. Taking norm on both sides of (117), using

the property $\|\mathbf{A}\| \leqslant \sqrt{M}\|\mathbf{A}\|_\infty$ for any $\mathbf{A} \in \mathbf{R}^{M \times M}$ and Corollary 4.1 then yields:

$$\left\|[\overline{\mathbf{W}}(s+1)]_k - [\mathbf{W}(s+1)]_k\right\| \leqslant \left\|\frac{\mathbf{1}\mathbf{1}^T}{M} - \mathbf{I}\right\|\left\|\mathbf{Q}_k(s) - \mathbf{1}\mathbf{c}_k(s)^T\right\|\left\|[\mathbf{W}(s)]_k - [\overline{\mathbf{W}}(s)]_k\right\| + h\left\|[\overline{\mathbf{T}}(s)]_k - [\mathbf{T}(s)]_k\right\| \tag{118}$$

$$\leqslant M^{\frac{1}{2}}\left\|\mathbf{Q}_k(s) - \mathbf{1}\mathbf{c}_k(s)^T\right\|_\infty\left\|[\mathbf{W}(s)]_k - [\overline{\mathbf{W}}(s)]_k\right\| + h\left\|[\overline{\mathbf{T}}(s)]_k - [\mathbf{T}(s)]_k\right\| \tag{119}$$

$$\leqslant M^{\frac{3}{2}}(1 - \beta^{\tau M})^{\left\lfloor\frac{(J-2)}{\tau M}\right\rfloor}\left\|[\mathbf{W}(s)]_k - [\overline{\mathbf{W}}(s)]_k\right\| + h\left\|[\overline{\mathbf{T}}(s)]_k - [\mathbf{T}(s)]_k\right\|, \tag{120}$$

which completes the proof. ∎

## C.2  Proof of Lemma 4.7

We first apply the $\widehat{(\cdot)}^{k,s+1}$ operator to both sides of (19) to get the following update:

$$[\widehat{\mathbf{W}}^{k,s+1}(s+1)]_k = \mathbf{Q}_k^\pi(s+1)\mathbf{Q}_k(s)[\mathbf{W}(s)]_k - h[\widehat{\mathbf{T}}^{k,s+1}(s)]_k. \tag{121}$$

Subtracting (19) from (121) yields:

$$[\widehat{\mathbf{W}}^{k,s+1}(s+1)]_k - [\mathbf{W}(s+1)]_k = (\mathbf{Q}_k^\pi(s+1)\mathbf{Q}_k(s) - \mathbf{Q}_k(s))[\mathbf{W}(s)]_k - h([\widehat{\mathbf{T}}^{k,s+1}(s)]_k - [\mathbf{T}(s)]_k) \tag{122}$$

$$= (\mathbf{Q}_k^\pi(s+1) - \mathbf{I})(\mathbf{Q}_k(s) - \mathbf{1}\mathbf{c}_k(s)^T)[\mathbf{W}(s)]_k - h([\widehat{\mathbf{T}}^{k,s+1}(s)]_k - [\mathbf{T}(s)]_k) \tag{123}$$

$$= (\mathbf{Q}_k^\pi(s+1) - \mathbf{I})(\mathbf{Q}_k(s) - \mathbf{1}\mathbf{c}_k(s)^T)([\mathbf{W}(s)]_k - [\widehat{\mathbf{W}}^{k,s}(s)]_k)$$
$$+ h(\mathbf{Q}_k^\pi(s+1) - \mathbf{I})([\widehat{\mathbf{T}}^{k,s}(s)]_k - [\mathbf{T}(s)]_k), \tag{124}$$

where in the second last step, we introduced the vector $\mathbf{c}_k(s)$ from Corollary 4.1 and used the fact that the matrix $\mathbf{1}\mathbf{c}_k(s)^T$ lies in the null space of $(\mathbf{Q}_k^\pi(s+1) - \mathbf{I})$. In the last step, we used the facts that the vector $[\widehat{\mathbf{W}}^{k,s}(s)]_k = \mathbf{Q}_k^\pi(s)[\mathbf{W}(s)]_k$ has all identical entries since $\mathbf{Q}_k^\pi(s)$ has identical rows, $\mathbf{Q}_k(s)$ is row stochastic and thus $\mathbf{Q}_k(s)[\widehat{\mathbf{W}}^{k,s}(s)]_k = [\widehat{\mathbf{W}}^{k,s}(s)]_k$, which has identical entries, and finally the vector $[\widehat{\mathbf{W}}^{k,s}(s)]_k$ lies in the null space of $(\mathbf{Q}_k^\pi(s+1) - \mathbf{I})$ and $(\mathbf{Q}_k(s) - \mathbf{1}\mathbf{c}_k(s)^T)$. Along similar lines we also have that $([\widehat{\mathbf{T}}^{k,s+1}(s)]_k - [\mathbf{T}(s)]_k) = -(\mathbf{Q}_k^\pi(s+1) - \mathbf{I})([\widehat{\mathbf{T}}^{k,s}(s)]_k - [\mathbf{T}(s)]_k)$.

Finally, taking operator norm on both sides of (124), using Cauchy-Schwarz inequality, the bound $\|\mathbf{Q}_k^\pi(s)\| = \|\mathbf{1}\mathbf{c}_k(s)^T\| \leqslant \sqrt{M}$ for any $s$, $\|\mathbf{A}\| \leqslant \sqrt{M}\|\mathbf{A}\|_\infty$ for any $\mathbf{A} \in \mathbf{R}^{M \times M}$ and Corollary 4.1 yields:

$$\left\|[\widehat{\mathbf{W}}^{k,s+1}(s+1)]_k - [\mathbf{W}(s+1)]_k\right\| \leqslant \|\mathbf{Q}_k^\pi(s+1) - \mathbf{I}\|\left\|\mathbf{Q}_k(s) - \mathbf{1}\mathbf{c}_k(s)^T\right\|\left\|[\widehat{\mathbf{W}}^{k,s}(s)]_k - [\mathbf{W}(s)]_k\right\|$$
$$+ h\|\mathbf{Q}_k^\pi(s+1) - \mathbf{I}\|\left\|[\widehat{\mathbf{T}}^{k,s}(s)]_k - [\mathbf{T}(s)]_k\right\| \tag{125}$$

$$\leqslant \sqrt{M}(\sqrt{M}+1)\left\|\mathbf{Q}_k(s) - \mathbf{1}\mathbf{c}_k(s)^T\right\|_\infty\left\|[\widehat{\mathbf{W}}^{k,s}(s)]_k - [\mathbf{W}(s)]_k\right\|$$
$$+ h(\sqrt{M}+1)\left\|[\widehat{\mathbf{T}}^{k,s}(s)]_k - [\mathbf{T}(s)]_k\right\| \tag{126}$$

$$\leqslant M^{\frac{3}{2}}(\sqrt{M}+1)(1 - \beta^{\tau M})^{\left\lfloor\frac{(J-2)}{\tau M}\right\rfloor}\left\|[\widehat{\mathbf{W}}^{k,s}(s)]_k - [\mathbf{W}(s)]_k\right\|$$
$$+ h(\sqrt{M}+1)\left\|[\widehat{\mathbf{T}}^{k,s}(s)]_k - [\mathbf{T}(s)]_k\right\|. \tag{127}$$

This completes the proof. ∎

**Remark C.1.** Note that in the steps leading up to (124) in the proof of Lemma 4.7, we cannot simply use the technique of one-step contraction from Lemma 1 in Xin & Khan (2018) because of the fact that matrix $\mathbf{Q}_k(s)$ in our case is time varying. Now, even though the spectral radius of the matrix $\mathbf{Q}_k(s) - \mathbf{1}(\mathbf{c}_k(s))^T$ is

strictly less than 1 when $\mathbf{Q}_k(s)$ is irreducible, its operator norm may not be less than 1. Also, no two matrices from the sequence $\{\mathbf{Q}_k(s) - \mathbf{1}(\mathbf{c}_k(s))^T\}_s$ may be simultaneously diagonalizable with the same eigenvectors, and hence we cannot simply apply some $s$-independent matrix norm on both sides of (124) so as to replace the operator norm with spectral radius. However, the time-invariant mixing matrix in Xin & Khan (2018) makes it possible to apply a compatible matrix norm on both sides of their inequality, something which is not possible in our case.

### C.3 Proof of Lemma 4.9

Let $\widetilde{\mathbf{W}}^* \in \mathbb{R}^{M \times d}$ be a matrix whose $i^{th}$ row is $\mathbf{w}_i^*$. Then, we get $\nabla F(\widetilde{\mathbf{W}}^*) = \mathbf{0}$. Further define $\widehat{\mathbf{W}}^s(s) := \mathbf{1}(\hat{\mathbf{w}}^s(s))^T$. Using the definition of $\hat{\mathbf{w}}^s(s)$ we also get:

$$Lh\sqrt{d}\sum_{j=1}^{M}\|\hat{\mathbf{w}}^s(s) - \mathbf{w}_j(s)\| = Lh\sqrt{d}\sum_{j=1}^{M}\sqrt{\sum_{k=1}^{d}\left(\sum_{l=1}^{M}[\mathbf{c}_k(s)]_l[\mathbf{w}_l(s)]_k - [\mathbf{w}_j(s)]_k\right)^2} \tag{128}$$

$$\leqslant Lh\sqrt{d}\sum_{j=1}^{M}\sum_{k=1}^{d}\left|\sum_{l=1}^{M}[\mathbf{c}_k(s)]_l[\mathbf{w}_l(s)]_k - [\mathbf{w}_j(s)]_k\right| \tag{129}$$

$$= Lh\sqrt{d}\sum_{k=1}^{d}\sum_{j=1}^{M}\left|\sum_{l=1}^{M}[\mathbf{c}_k(s)]_l[\mathbf{w}_l(s)]_k - [\mathbf{w}_j(s)]_k\right| \tag{130}$$

$$\leqslant Lh\sqrt{Md}\sum_{k=1}^{d}\sqrt{\sum_{j=1}^{M}\left|\sum_{l=1}^{M}[\mathbf{c}_k(s)]_l[\mathbf{w}_l(s)]_k - [\mathbf{w}_j(s)]_k\right|^2} \tag{131}$$

$$= Lh\sqrt{Md}\sum_{k=1}^{d}\left\|[\widehat{\mathbf{W}}^{k,s}(s)]_k - [\mathbf{W}(s)]_k\right\|. \tag{132}$$

Then, as a consequence of (132) we get the following bound:

$$\sum_{j=1}^{M}\|\hat{\mathbf{w}}^s(s) - \mathbf{w}_j(s)\| \leqslant \sqrt{M}\sum_{k=1}^{d}\left\|[\widehat{\mathbf{W}}^{k,s}(s)]_k - [\mathbf{W}(s)]_k\right\|. \tag{133}$$

Taking norm of $[\hat{\mathbf{T}}^{k,s}(s)]_k - [\mathbf{T}(s)]_k$, using the fact that $\|\mathbf{Q}_k^\pi\| = \|\mathbf{1}\mathbf{c}_k^T\| \leqslant \sqrt{M}$ and simplifying using Assumption 4.8, Jensen's inequality and (133) yield:

$$\left\|[\hat{\mathbf{T}}^{k,s}(s)]_k - [\mathbf{T}(s)]_k\right\| = \left\|[\nabla\hat{F}^{k,s}(\mathbf{W}(s))]_k - [\nabla F(\mathbf{W}(s))]_k\right\| \tag{134}$$

$$\leqslant \|\mathbf{Q}_k^\pi(s) - \mathbf{I}\|\,\|[\nabla F(\mathbf{W}(s))]_k\| \tag{135}$$

$$\leqslant (\sqrt{M} + 1)\left(\left\|\nabla F(\mathbf{W}(s)) - \nabla F(\widehat{\mathbf{W}}^s(s))\right\|_F + \left\|\nabla F(\widehat{\mathbf{W}}^s(s)) - \nabla F(\widetilde{\mathbf{W}}^*)\right\|_F\right) \tag{136}$$

$$\leqslant (\sqrt{M} + 1)L\left(\sqrt{\sum_{i=1}^{M}\|\mathbf{w}_i(s) - \hat{\mathbf{w}}^s(s)\|^2} + \sqrt{\sum_{i=1}^{M}\|\mathbf{w}_i^* - \hat{\mathbf{w}}^s(s)\|^2}\right) \tag{137}$$

$$\leqslant (\sqrt{M} + 1)L\left(\sum_{i=1}^{M}\|\mathbf{w}_i(s) - \hat{\mathbf{w}}^s(s)\| + \sum_{i=1}^{M}\|\mathbf{w}_i^* - \hat{\mathbf{w}}^s(s)\|\right) \tag{138}$$

$$\leqslant (\sqrt{M} + 1)L\sqrt{M}\sum_{k=1}^{d}\left\|[\mathbf{W}(s)]_k - [\widehat{\mathbf{W}}^{k,s}(s)]_k\right\|$$

$$+ (\sqrt{M} + 1)L\sum_{i=1}^{M}\left(\|\mathbf{w}^* - \hat{\mathbf{w}}^s(s)\| + \|\mathbf{w}^* - \mathbf{w}_i^*\|\right) \tag{139}$$

$$= (\sqrt{M} + 1)L\sqrt{M}\sum_{k=1}^{d}\left\|[\mathbf{W}(s)]_k - [\widehat{\mathbf{W}}^{k,s}(s)]_k\right\| + (\sqrt{M} + 1)LM\|\mathbf{w}^* - \hat{\mathbf{w}}^s(s)\|$$

$$+ (\sqrt{M} + 1)L \sum_{i=1}^{M} \|\mathbf{w}^* - \mathbf{w}_i^*\|. \tag{140}$$

Similarly we get that:

$$\left\| [\overline{\mathbf{T}}(s)]_k - [\mathbf{T}(s)]_k \right\| = \left\| [\nabla \overline{F}(\mathbf{W}(s))]_k - [\nabla F(\mathbf{W}(s))]_k \right\| \leqslant \underbrace{\left\| \frac{\mathbf{1}\mathbf{1}^T}{M} - \mathbf{I} \right\|}_{\leqslant 1} \left\| [\nabla F(\mathbf{W}(s))]_k \right\| \tag{141}$$

$$\leqslant L\sqrt{M} \sum_{k=1}^{d} \left\| [\mathbf{W}(s)]_k - [\widehat{\mathbf{W}}^{k,s}(s)]_k \right\| + LM \|\mathbf{w}^* - \hat{\mathbf{w}}^s(s)\| + L \sum_{i=1}^{M} \|\mathbf{w}^* - \mathbf{w}_i^*\|, \quad \tag{142}$$

which completes the proof. ∎

## D   The RESIST Algorithm as an Inexact Gradient Descent Update

### D.1   Proof of Lemma 4.11

For $f^{k,s}(\cdot) := \sum_{i=1}^{M} [\mathbf{c}_k(s)]_i f_i(\cdot)$, where $\mathbf{c}_k(s)$ is defined in Corollary 4.1 and $0 \leqslant [\mathbf{c}_k(s)]_i \leqslant 1$ for all $i$ with $\sum_{i=1}^{M} [\mathbf{c}_k(s)]_i = 1$, we get that $f^{k,s}$ is $L$-gradient Lipschitz for any $k, s$ by Assumption 4.8. Then, the local vector update at time $s + 1$ defined as $\mathbf{w}_i(s + 1)$ for any node $i$ can be written as:

$$
\begin{bmatrix}
[\mathbf{w}_i(s+1)]_1 \\
[\mathbf{w}_i(s+1)]_2 \\
\vdots \\
\vdots \\
\vdots \\
[\mathbf{w}_i(s+1)]_k \\
\vdots \\
\vdots \\
[\mathbf{w}_i(s+1)]_d
\end{bmatrix}
=
\begin{bmatrix}
\sum_{j=1}^{M} [\mathbf{Q}_1(s)]_{ij} [\mathbf{w}_j(s)]_1 \\
\sum_{j=1}^{M} [\mathbf{Q}_2(s)]_{ij} [\mathbf{w}_j(s)]_2 \\
\vdots \\
\sum_{j=1}^{M} [\mathbf{Q}_k(s)]_{ij} [\mathbf{w}_j(s)]_k \\
\vdots \\
\sum_{j=1}^{M} [\mathbf{Q}_d(s)]_{ij} [\mathbf{w}_j(s)]_d
\end{bmatrix}
- h
\begin{bmatrix}
\nabla_1 f_i(\mathbf{w}_i(s)) \\
\nabla_2 f_i(\mathbf{w}_i(s)) \\
\vdots \\
\vdots \\
\vdots \\
\nabla_k f_i(\mathbf{w}_i(s)) \\
\vdots \\
\vdots \\
\nabla_d f_i(\mathbf{w}_i(s))
\end{bmatrix}. \tag{143}
$$

Applying $\widehat{(\cdot)}^{k,s+1}$ operator or equivalently multiplying $[\mathbf{c}_k(s+1)]$ to both sides of the above equality to average the entries in dimension $k$ and at time $s + 1$, we get the following expression, which is independent of $i$:

$$
\underbrace{
\begin{bmatrix}
\sum_{j=1}^{M} [\mathbf{c}_1(s+1)]_j [\mathbf{w}_j(s+1)]_1 \\
\sum_{j=1}^{M} [\mathbf{c}_2(s+1)]_j [\mathbf{w}_j(s+1)]_2 \\
\vdots \\
\sum_{j=1}^{M} [\mathbf{c}_k(s+1)]_j [\mathbf{w}_j(s+1)]_k \\
\vdots \\
\sum_{j=1}^{M} [\mathbf{c}_d(s+1)]_j [\mathbf{w}_j(s+1)]_d
\end{bmatrix}
}_{\hat{\mathbf{w}}^{s+1}(s+1)}
=
\underbrace{
\begin{bmatrix}
\sum_{j=1}^{M} [\mathbf{c}_1(s)]_j [\mathbf{w}_j(s)]_1 \\
\sum_{j=1}^{M} [\mathbf{c}_2(s)]_j [\mathbf{w}_j(s)]_2 \\
\vdots \\
\sum_{j=1}^{M} [\mathbf{c}_k(s)]_j [\mathbf{w}_j(s)]_k \\
\vdots \\
\sum_{j=1}^{M} [\mathbf{c}_d(s)]_j [\mathbf{w}_j(s)]_d
\end{bmatrix}
}_{\hat{\mathbf{w}}^s(s)}
- h
\begin{bmatrix}
\sum_{j=1}^{M} [\mathbf{c}_1(s+1)]_j \nabla_1 f_j(\mathbf{w}_j(s)) \\
\sum_{j=1}^{M} [\mathbf{c}_2(s+1)]_j \nabla_2 f_j(\mathbf{w}_j(s)) \\
\vdots \\
\sum_{j=1}^{M} [\mathbf{c}_k(s+1)]_j \nabla_k f_j(\mathbf{w}_j(s)) \\
\vdots \\
\sum_{j=1}^{M} [\mathbf{c}_d(s+1)]_j \nabla_d f_j(\mathbf{w}_j(s))
\end{bmatrix}
\tag{144}
$$

$$
\begin{aligned}
=& \begin{bmatrix} \sum_{j=1}^{M} [\mathbf{c}_1(s)]_j [\mathbf{w}_j(s)]_1 \\ \sum_{j=1}^{M} [\mathbf{c}_2(s)]_j [\mathbf{w}_j(s)]_2 \\ \vdots \\ \sum_{j=1}^{M} [\mathbf{c}_k(s)]_j [\mathbf{w}_j(s)]_k \\ \vdots \\ \sum_{j=1}^{M} [\mathbf{c}_d(s)]_j [\mathbf{w}_j(s)]_d \end{bmatrix} - h \begin{bmatrix} \nabla_1 f(\hat{\mathbf{w}}^s(s)) \\ \nabla_2 f(\hat{\mathbf{w}}^s(s)) \\ \vdots \\ \vdots \\ \nabla_k f(\hat{\mathbf{w}}^s(s)) \\ \vdots \\ \vdots \\ \nabla_d f(\hat{\mathbf{w}}^s(s)) \end{bmatrix} + h \underbrace{\left( \begin{bmatrix} \nabla_1 f(\hat{\mathbf{w}}^s(s)) \\ \nabla_2 f(\hat{\mathbf{w}}^s(s)) \\ \vdots \\ \vdots \\ \nabla_k f(\hat{\mathbf{w}}^s(s)) \\ \vdots \\ \vdots \\ \nabla_d f(\hat{\mathbf{w}}^s(s)) \end{bmatrix} - \begin{bmatrix} \nabla_1 f^{1,s+1}(\hat{\mathbf{w}}^s(s)) \\ \nabla_2 f^{2,s+1}(\hat{\mathbf{w}}^s(s)) \\ \vdots \\ \vdots \\ \nabla_k f^{k,s+1}(\hat{\mathbf{w}}^s(s)) \\ \vdots \\ \vdots \\ \nabla_d f^{d,s+1}(\hat{\mathbf{w}}^s(s)) \end{bmatrix} \right)}_{=\mathbf{e}_1(s)} \\
&+ h \underbrace{\left( \begin{bmatrix} \sum_{j=1}^{M} [\mathbf{c}_1(s+1)]_j \nabla_1 f_j(\hat{\mathbf{w}}^s(s)) \\ \sum_{j=1}^{M} [\mathbf{c}_2(s+1)]_j \nabla_2 f_j(\hat{\mathbf{w}}^s(s)) \\ \vdots \\ \vdots \\ \sum_{j=1}^{M} [\mathbf{c}_k(s+1)]_j \nabla_k f_j(\hat{\mathbf{w}}^s(s)) \\ \vdots \\ \vdots \\ \sum_{j=1}^{M} [\mathbf{c}_d(s+1)]_j \nabla_d f_j(\hat{\mathbf{w}}^s(s)) \end{bmatrix} - \begin{bmatrix} \sum_{j=1}^{M} [\mathbf{c}_1(s+1)]_j \nabla_1 f_j(\mathbf{w}_j(s)) \\ \sum_{j=1}^{M} [\mathbf{c}_2(s+1)]_j \nabla_2 f_j(\mathbf{w}_j(s)) \\ \vdots \\ \vdots \\ \sum_{j=1}^{M} [\mathbf{c}_k(s+1)]_j \nabla_k f_j(\mathbf{w}_j(s)) \\ \vdots \\ \vdots \\ \sum_{j=1}^{M} [\mathbf{c}_d(s+1)]_j \nabla_d f_j(\mathbf{w}_j(s)) \end{bmatrix} \right)}_{=\mathbf{e}_2(s)} .
\end{aligned}
\tag{145}
$$

Next, in order to see how the algorithm update (19) is equivalent to the inexact gradient descent update with error terms that are in the form of the above equation, we apply $\widehat{(\cdot)}^{k,s+1}$ operator to (19), substituting $[\mathbf{T}(s)]_k = [\nabla F(\mathbf{W}(s))]_k$ and using Corollary 4.1 to get:

$$
[\widehat{\mathbf{W}}^{k,s+1}(s+1)]_k = \mathbf{Q}_k^\pi(s+1)\mathbf{Q}_k(s)[\mathbf{W}(s)]_k - h[\nabla\widehat{F}^{k,s+1}(\mathbf{W}(s))]_k
\tag{146}
$$

$$
= \mathbf{Q}_k^\pi(s)[\mathbf{W}(s)]_k - h[\nabla\widehat{F}^{k,s+1}(\mathbf{W}(s))]_k
\tag{147}
$$

$$
= [\widehat{\mathbf{W}}^{k,s}(s)]_k - h[\nabla\widehat{F}^{k,s+1}(\widehat{\mathbf{W}}^{k,s}(s))]_k + h([\nabla\widehat{F}^{k,s+1}(\widehat{\mathbf{W}}^{k,s}(s))]_k - [\nabla\widehat{F}^{k,s+1}(\mathbf{W}(s))]_k)
\tag{148}
$$

$$
= [\widehat{\mathbf{W}}^{k,s}(s)]_k - h[\nabla\overline{F}(\widehat{\mathbf{W}}^{k,s}(s))]_k + h([\nabla\overline{F}(\widehat{\mathbf{W}}^{k,s}(s))]_k - [\nabla\widehat{F}^{k,s+1}(\widehat{\mathbf{W}}^{k,s}(s))]_k)
$$
$$
+ h([\nabla\widehat{F}^{k,s+1}(\widehat{\mathbf{W}}^{k,s}(s))]_k - [\nabla\widehat{F}^{k,s+1}(\mathbf{W}(s))]_k).
\tag{149}
$$

Observe that the $k$-th row in the vector equation (145) corresponds to the update (149). Also, notice that the update (149) is in principle a scalar update due to the fact that all the $d$ entries of any given vector on either side of (149) are identical. Then, stacking scalar updates of (149) from $k = 1$ to $d$ and representing the stacked vectors $[\widehat{\mathbf{W}}^{k,s+1}(s+1)]_k$ and $[\widehat{\mathbf{W}}^{k,s}(s)]_k$ as $\hat{\mathbf{w}}^{s+1}(s+1)$ and $\hat{\mathbf{w}}^s(s)$, respectively, yield the exact vector update as (145).

Thus, from (145) we get the following inexact gradient descent update:

$$
\hat{\mathbf{w}}^{s+1}(s+1) = \hat{\mathbf{w}}^s(s) - h\nabla f(\hat{\mathbf{w}}^s(s)) + \mathbf{e}_1(s) + \mathbf{e}_2(s).
\tag{150}
$$

Next, using $L$-gradient Lipschitz continuity of $\nabla_k f_j$ for any $k, j$ from Assumption 4.8, the fact that $0 \leqslant [\mathbf{c}_k(s+1)]_j \leqslant 1$ and a simple application of triangle inequality, we get the following bound on $\mathbf{e}_2(s)$ :

$$\|\mathbf{e}_2(s)\| \leqslant Lh\sqrt{\sum_{k=1}^{d} \left( \sum_{j=1}^{M} \|\hat{\mathbf{w}}^s(s) - \mathbf{w}_j(s)\| \right)^2} \tag{151}$$

$$= Lh\sqrt{d} \sum_{j=1}^{M} \|\hat{\mathbf{w}}^s(s) - \mathbf{w}_j(s)\| . \tag{152}$$

Then using the bound (133) along with (152), we get:

$$\|\mathbf{e}_2(s)\| \leqslant Lh\sqrt{Md} \sum_{k=1}^{d} \left\| [\widehat{\mathbf{W}}^{k,s}(s)]_k - [\mathbf{W}(s)]_k \right\| . \tag{153}$$

This completes the proof. ∎

## E Proofs for Algorithmic Convergence Under Strong Convexity

### E.1 On the non-vacuous nature of Assumption 4.12

The construction below is a restricted non-vacuity example for Assumption 4.12; it relies on additional structural conditions on the objectives, network, attacks, and induced mixing matrices, and should not be read as a verification of the assumption for arbitrary time-varying MITM attack sequences. Suppose the model dimension is 1, i.e., $f_i : \mathbb{R} \to \mathbb{R}$, Assumptions 3.3, 4.8 hold and that $f_i$ is coercive for all $i$, i.e., $\lim_{\|\mathbf{w}\| \to \infty} f_i(\mathbf{w}) = \infty$. Further, suppose the graph induced by the network topology is symmetric and strongly connected, such as a $K$-regular graph with $K = 4b$. Also, assume the Man-in-the-middle attack is such that the mixing matrix $\mathbf{Y}(t)$ is symmetric, simultaneously diagonalizable for all $t$ and the sequence of those simultaneously diagonalizable matrices $\{\mathbf{Q}(s)\}_{s=0}^{\infty}$ is

$$\mathbf{Q}(s) = \prod_{r=J\lfloor \frac{t}{J} \rfloor}^{J\lfloor \frac{t}{J} \rfloor + J - 2} \mathbf{Y}(r), \tag{154}$$

where the matrix $\mathbf{Q}(s)$ matrix is defined from (18) after omitting the subscript $k$ and the sequence also satisfies[11]

$$\mathbf{Q}(0) \leqslant \mathbf{Q}(1) \leqslant \cdots \leqslant \mathbf{Q}(s) \leqslant \cdots . \tag{155}$$

The simultaneous diagonalizable matrices condition will be satisfied by an attack that only changes the graph spectrum (eigenvalues of $\mathbf{Y}(t)$) over time. The condition (155) can be satisfied by an attack that progressively decreases the information mixing rate in the network by increasing the eigenvalues of the mixing matrices.

Next, along similar lines as in Lemma 3, (Zeng & Yin, 2018), for $\mathbf{W} := [\mathbf{w}_1, \cdots, \mathbf{w}_M]^T$ and $F(\mathbf{W}) := \sum_{i=1}^{M} f_i(\mathbf{w}_i)$ we define a Lyapunov function $\mathcal{L}(\cdot; s) : \mathbb{R}^M \to \mathbb{R}$ as follows:

$$\mathcal{L}(\mathbf{W}; s) := F(\mathbf{W}) + \frac{1}{2h} \|\mathbf{W}\|_{\mathbf{I}-\mathbf{Q}(s)}^2 , \tag{156}$$

where[12] $\|\mathbf{W}\|_{\mathbf{I}-\mathbf{Q}(s)}^2 = \langle \mathbf{W}, (\mathbf{I} - \mathbf{Q}(s))\mathbf{W} \rangle$. Note that $\mathcal{L}(\mathbf{W}; s)$ is a Lyapunov function since $F(\cdot)$ is lower bounded and $\mathbf{I} - \mathbf{Q}(s)$ is positive semi-definite due to symmetric mixing matrix $\mathbf{Q}(s)$. Then, the $s$-time scale update for RESIST can be expressed in terms of the Lyapunov function as follows:

$$\mathbf{W}(s + 1) = \mathbf{W}(s) - h\nabla\mathcal{L}(\mathbf{W}(s); s) \tag{157}$$

---

[11]Here, the inequality $\mathbf{A} \leqslant \mathbf{B}$ implies $\mathbf{B} - \mathbf{A}$ is positive semi-definite.

[12]Note that $\|\cdot\|_{\mathbf{I}-\mathbf{Q}(s)}$ is a semi-norm since $(\mathbf{I} - \mathbf{Q}(s))\frac{\mathbf{1}\mathbf{1}^T}{M}\mathbf{W} = \mathbf{0}$ for any $\mathbf{W} \in \mathbb{R}^M$.

[13]due to symmetric $\mathbf{Q}(s)$. Further, the Lyapunov function $\mathcal{L}(\cdot; s)$ is uniformly gradient Lipschitz continuous over all $s \geqslant 0$ where

$$\text{Lip}(\mathcal{L}) \leqslant LM + \sup_{s \geqslant 0} \frac{\|\mathbf{I} - \mathbf{Q}(s)\|_2}{h} = LM + \frac{1 - \inf_{s \geqslant 0} \sigma(\mathbf{Q}(s))}{h}, \tag{158}$$

$\sigma(\mathbf{Q}(s))$ is the smallest eigenvalue of $\mathbf{Q}(s)$ and the eigenvalues of $\mathbf{Q}(s)$ lie in the interval $(0, 1]$.

Next, if $h < \frac{1 + \inf_{s \geqslant 0} \sigma(\mathbf{Q}(s))}{LM}$ then from (158) we have:

$$\text{Lip}(\mathcal{L})h \leqslant LMh + 1 - \inf_{s \geqslant 0} \sigma(\mathbf{Q}(s)) < 2. \tag{159}$$

Then by gradient Lipschitz continuity of $\mathcal{L}(\cdot; s)$ for $h < \frac{1 + \inf_{s \geqslant 0} \sigma(\mathbf{Q}(s))}{LM}$ and (157), (159) we get:

$$\mathcal{L}(\mathbf{W}(s+1); s) \leqslant \mathcal{L}(\mathbf{W}(s); s) + \langle \nabla \mathcal{L}(\mathbf{W}(s); s), \mathbf{W}(s+1) - \mathbf{W}(s) \rangle + \frac{\text{Lip}(\mathcal{L})}{2} \|\mathbf{W}(s+1) - \mathbf{W}(s)\|^2 \tag{160}$$

$$= \mathcal{L}(\mathbf{W}(s); s) - \frac{h}{2} \left( 2 - \text{Lip}(\mathcal{L})h \right) \|\nabla \mathcal{L}(\mathbf{W}(s); s)\|^2 \tag{161}$$

$$\leqslant \mathcal{L}(\mathbf{W}(s); s). \tag{162}$$

From (155) we get that $\|\mathbf{W}(s+1)\|_{\mathbf{I} - \mathbf{Q}(s+1)}^2 \leqslant \|\mathbf{W}(s+1)\|_{\mathbf{I} - \mathbf{Q}(s)}^2$ and then using (162) for $h < \frac{1 + \inf_{s \geqslant 0} \sigma(\mathbf{Q}(s))}{LM}$ we have that:

$$\mathcal{L}(\mathbf{W}(s+1); s+1) \leqslant \mathcal{L}(\mathbf{W}(s); s) \ \ \forall \, s \geqslant 0. \tag{163}$$

Since $f_i$ is coercive, $\mathcal{L}(\cdot; s)$ is coercive for all $s$ and hence $\mathcal{L}(\cdot; s)$ has bounded sublevel sets for all $s$. For an initialization $\mathbf{W}(0)$ of RESIST, let

$$S_{sub}(s) = \left\{ \mathbf{W} \in \mathbb{R}^M : \mathcal{L}(\mathbf{W}; s) \leqslant \mathcal{L}(\mathbf{W}(0); 0) \right\}.$$

Then $S_{sub}(s)$ for any $s \geqslant 0$ is compact. Also, from (155) we get for any $\mathbf{W}$ that $\|\mathbf{W}\|_{\mathbf{I} - \mathbf{Q}(s+1)}^2 \leqslant \|\mathbf{W}\|_{\mathbf{I} - \mathbf{Q}(s)}^2$ for all $s \geqslant 0$ and thus for any $\mathbf{W}$

$$\mathcal{L}(\mathbf{W}; s+1) \leqslant \mathcal{L}(\mathbf{W}; s) \ \ \forall \, s \geqslant 0. \tag{164}$$

Using the inequality (164) we have

$$S_{sub}(\infty) \supseteq \cdots \supseteq S_{sub}(s+1) \supseteq S_{sub}(s) \supseteq \cdots \supseteq S_{sub}(0), \tag{165}$$

with the convention that

$$S_{sub}(\infty) = \left\{ \mathbf{W} \in \mathbb{R}^M : \liminf_{s \to \infty} \mathcal{L}(\mathbf{W}; s) \leqslant \mathcal{L}(\mathbf{W}(0); 0) \right\}.$$

It is important to note that $\liminf_{s \to \infty} \|\mathbf{W}\|_{\mathbf{I} - \mathbf{Q}(s)}^2 \geqslant 0$ for any $\mathbf{W}$ since $\|\mathbf{W}\|_{\mathbf{I} - \mathbf{Q}(s)}^2 \geqslant 0$ for all $s \geqslant 0$ and any $\mathbf{W}$. Then $\liminf_{s \to \infty} \mathcal{L}(\mathbf{W}; s)$ is coercive in $\mathbf{W}$ with compact sub-level sets and hence $S_{sub}(\infty)$ is compact.

Then for $h < \frac{1 + \inf_{s \geqslant 0} \sigma(\mathbf{Q}(s))}{LM}$, from (163), (165) and compactness of $S_{sub}(\infty)$, we have that the sequence $\{\mathbf{W}(s)\}_s$ stays bounded in compact $S_{sub}(\infty)$ for all $s$. This completes the example illustrating Assumption 4.12.

---

[13]Here $\nabla$ is with respect to $\mathbf{W}(s)$.

## E.2 Proof of Lemma 5.3

Since $f := \frac{1}{M}\sum_{i=1}^{M} f_i$ is $\mu$-strongly convex and $L$-gradient Lipschitz, we get that $f$ satisfies Lemma 5.2. Then expanding $\|\hat{\mathbf{w}}^s(s) - h\nabla f(\hat{\mathbf{w}}^s(s)) - \mathbf{w}^*\|^2$ and using (39) we have that:

$$\|\hat{\mathbf{w}}^s(s) - h\nabla f(\hat{\mathbf{w}}^s(s)) - (\mathbf{w}^* - \nabla f(\mathbf{w}^*))\|^2 = \|\hat{\mathbf{w}}^s(s) - \mathbf{w}^*\|^2 + h^2 \|\nabla f(\hat{\mathbf{w}}^s(s)) - \nabla f(\mathbf{w}^*))\|^2$$
$$- 2h\langle \hat{\mathbf{w}}^s(s) - \mathbf{w}^*, \nabla f(\hat{\mathbf{w}}^s(s)) - \nabla f(\mathbf{w}^*))\rangle \tag{166}$$

$$\leqslant \|\hat{\mathbf{w}}^s(s) - \mathbf{w}^*\|^2 + h^2 \|\nabla f(\hat{\mathbf{w}}^s(s)) - \nabla f(\mathbf{w}^*))\|^2 - 2h\left(\frac{\mu L}{\mu + L}\|\hat{\mathbf{w}}^s(s) - \mathbf{w}^*\|^2\right.$$
$$\left. + \frac{1}{\mu + L}\|\nabla f(\hat{\mathbf{w}}^s(s)) - \nabla f(\mathbf{w}^*)\|^2\right) \tag{167}$$

$$\leqslant \left(1 - \frac{2hL\mu}{L + \mu}\right)\|\hat{\mathbf{w}}^s(s) - \mathbf{w}^*\|^2 + \left(h^2 - \frac{2h}{\mu + L}\right)\|\nabla f(\hat{\mathbf{w}}^s(s)) - \nabla f(\mathbf{w}^*)\|^2 \tag{168}$$

$$\leqslant \left(1 - \frac{2hL\mu}{L + \mu}\right)\|\hat{\mathbf{w}}^s(s) - \mathbf{w}^*\|^2 + \mu^2\left(h^2 - \frac{2h}{\mu + L}\right)\|\hat{\mathbf{w}}^s(s) - \mathbf{w}^*\|^2 \tag{169}$$

$$\leqslant (1 - \mu h)^2 \|\hat{\mathbf{w}}^s(s) - \mathbf{w}^*\|^2, \tag{170}$$

where in the second last step we used the fact that $h < \frac{2}{\mu+L}$. Then we get that:

$$\|\hat{\mathbf{w}}^s(s) - h\nabla f(\hat{\mathbf{w}}^s(s)) - \mathbf{w}^*\| \leqslant (1 - \mu h)\|\hat{\mathbf{w}}^s(s) - \mathbf{w}^*\|. \tag{171}$$

Finally subtracting $\mathbf{w}^*$ from both sides of (150) in the proof of Lemma 4.11, taking norm, substituting (171) and (153) we get:

$$\|\hat{\mathbf{w}}^{s+1}(s+1) - \mathbf{w}^*\| \leqslant (1 - \mu h)\|\hat{\mathbf{w}}^s(s) - \mathbf{w}^*\| + \|\mathbf{e}_1(s)\| + Lh\sqrt{Md}\sum_{k=1}^{d}\left\|[\widehat{\mathbf{W}}^{k,s}(s)]_k - [\mathbf{W}(s)]_k\right\|, \tag{172}$$

which completes the proof. ∎

## E.3 Proof of Lemma 5.4

In order to develop rates of convergence for strongly convex functions, using Definition 4.3, we first express $\xi_k^1(s+1), \xi_k^5(s+1)$ for all $k \in \{1, \ldots, d\}$ and $\xi_{\mathbf{w}*}^6(s+1)$ in terms of $\xi_k^1(s), \xi_k^5(s), \xi_{\mathbf{w}*}^6(s)$ and some residual terms corresponding to $\|\mathbf{e}_1(s)\|$ and $\|\mathbf{w}_i^* - \mathbf{w}^*\|$ for $i \in \mathcal{N}$.

Using Lemma 4.7 and Lemma 4.9 we get:

$$\xi_k^1(s+1) \leqslant M^{\frac{3}{2}}(\sqrt{M}+1)(1-\beta^{\tau M})^{\left\lfloor\frac{(J-2)}{\tau M}\right\rfloor}\xi_k^1(s) + h(\sqrt{M}+1)\xi_k^2(s) \tag{173}$$

$$\leqslant a_1\xi_k^1(s) + a_2 h\sqrt{M}\sum_{k=1}^{d}\xi_k^1(s) + a_2 Mh\xi_{\mathbf{w}*}^6(s) + a_2 h\Delta,$$

where $a_1 = M^{\frac{3}{2}}(\sqrt{M}+1)(1-\beta^{\tau M})^{\left\lfloor\frac{(J-2)}{\tau M}\right\rfloor}$, $a_2 = (\sqrt{M}+1)^2 L$ and $\Delta = \sum_{i=1}^{M}\|\mathbf{w}^* - \mathbf{w}_i^*\|$.

Similarly, using Lemma 4.6 and Lemma 4.9 we get:

$$\xi_k^5(s+1) \leqslant M^{\frac{3}{2}}(1-\beta^{\tau M})^{\left\lfloor\frac{(J-2)}{\tau M}\right\rfloor}\xi_k^5(s) + h\left\|[\overline{\mathbf{T}}(s)]_k - [\mathbf{T}(s)]_k\right\| \tag{174}$$

$$\leqslant a_3\xi_k^5(s) + a_4 h\sqrt{M}\sum_{k=1}^{d}\xi_k^1(s) + a_4 Mh\xi_{\mathbf{w}*}^6(s) + a_4 h\Delta, \tag{175}$$

where $a_3 = M^{\frac{3}{2}}(1-\beta^{\tau M})^{\left\lfloor\frac{(J-2)}{\tau M}\right\rfloor}$ and $a_4 = L$.

From the definition of $\mathbf{e}_1(s)$ in Lemma 4.11 and by Jensen's inequality we can write:

$$\|\mathbf{e}_1(s)\| \leqslant h \sum_{k=1}^{d} \underbrace{|\nabla_k f(\widehat{\mathbf{w}}^s(s)) - \nabla_k f^{k,s+1}(\widehat{\mathbf{w}}^s(s))|}_{=\gamma_k(s)} = h\gamma(s). \tag{176}$$

Then using Lemma 5.3 and (176) we get:

$$\xi_{\mathbf{w}*}^6(s+1) \leqslant (1 - \mu h)\xi_{\mathbf{w}*}^6(s) + \|\mathbf{e}_1(s)\| + Lh\sqrt{Md} \sum_{k=1}^{d} \left\|[\widehat{\mathbf{W}}^{k,s}(s)]_k - [\mathbf{W}(s)]_k\right\| \tag{177}$$

$$\leqslant (1 - \mu h)\xi_{\mathbf{w}*}^6(s) + \underbrace{h \sum_{k=1}^{d} \gamma_k(s)}_{=h\gamma(s)} + \underbrace{Lh\sqrt{Md}}_{=a_5 h} \sum_{k=1}^{d} \xi_k^1(s). \tag{178}$$

Let

$$\mathbf{A} = \begin{bmatrix} a_1 + a_2 h\sqrt{M} & 0 \\ a_4 h\sqrt{M} & a_3 \end{bmatrix}, \qquad \mathbf{B} = \begin{bmatrix} a_2 h\sqrt{M} & 0 \\ a_4 h\sqrt{M} & 0 \end{bmatrix}. \tag{179}$$

Stacking $\{\xi_k^1(s)\}_{k=1}^d$, $\{\xi_k^5(s)\}_{k=1}^d$, $\xi_{\mathbf{w}*}^6(s)$ into a vector for any $s$ and invoking the bounds (174), (175), (178) we have the following inexact recursion of the error terms:

$$\underbrace{\begin{bmatrix} \xi_1^1(s+1) \\ \xi_1^5(s+1) \\ \xi_2^1(s+1) \\ \xi_2^5(s+1) \\ \vdots \\ \vdots \\ \vdots \\ \xi_d^1(s+1) \\ \xi_d^5(s+1) \\ \xi_{\mathbf{w}*}^6(s+1) \end{bmatrix}}_{=\mathbf{g}(s+1) \in \mathbb{R}_+^{(2d+1)}} \leqslant \underbrace{\begin{bmatrix} \mathbf{A} & \mathbf{B} & \mathbf{B} & \cdots & \mathbf{B} & \begin{matrix} a_2 Mh \\ a_4 Mh \end{matrix} \\ \mathbf{B} & \mathbf{A} & \mathbf{B} & \cdots & \mathbf{B} & \begin{matrix} a_2 Mh \\ a_4 Mh \end{matrix} \\ \vdots & & \ddots & & \vdots & \vdots \\ \vdots & & & \ddots & \vdots & \vdots \\ \vdots & & & \ddots & \vdots & \vdots \\ \mathbf{B} & \mathbf{B} & \cdots & \mathbf{B} & \mathbf{A} & \begin{matrix} a_2 Mh \\ a_4 Mh \end{matrix} \\ a_5 h \; 0 & a_5 h \; 0 & \cdots & \cdots & a_5 h \; 0 & 1 - \mu h \end{bmatrix}}_{=\mathbf{M}(h,J) \in \mathbb{R}_+^{(2d+1)\times(2d+1)}} \underbrace{\begin{bmatrix} \xi_1^1(s) \\ \xi_1^5(s) \\ \xi_2^1(s) \\ \xi_2^5(s) \\ \vdots \\ \vdots \\ \vdots \\ \xi_d^1(s) \\ \xi_d^5(s) \\ \xi_{\mathbf{w}*}^6(s) \end{bmatrix}}_{=\mathbf{g}(s) \in \mathbb{R}_+^{(2d+1)}} + \underbrace{\begin{bmatrix} a_2 h\Delta \\ a_4 h\Delta \\ a_2 h\Delta \\ a_4 h\Delta \\ \vdots \\ \vdots \\ \vdots \\ a_2 h\Delta \\ a_4 h\Delta \\ h\gamma(s) \end{bmatrix}}_{=\boldsymbol{\epsilon}(s) \in \mathbb{R}_+^{(2d+1)}}. \tag{180}$$

Let us express $\mathbf{M}(h,J) = \mathbf{M}_0 + \mathbf{P}(h,J)$ where

$$\mathbf{M}_0 = \begin{bmatrix} \begin{matrix} a_1 & 0 \\ 0 & a_3 \end{matrix} & \mathbf{0} & \mathbf{0} & \cdots & \mathbf{0} & \begin{matrix} 0 \\ 0 \end{matrix} \\ \mathbf{0} & \begin{matrix} a_1 & 0 \\ 0 & a_3 \end{matrix} & \mathbf{0} & \cdots & \mathbf{0} & \begin{matrix} 0 \\ 0 \end{matrix} \\ \vdots & & \ddots & & \vdots & \vdots \\ \vdots & & & \ddots & \vdots & \vdots \\ \vdots & & & & \ddots & \vdots & \vdots \\ \mathbf{0} & \mathbf{0} & \cdots & \mathbf{0} & \begin{matrix} a_1 & 0 & 0 \\ 0 & a_3 & 0 \end{matrix} \\ 0 & 0 \quad 0 \quad 0 & \cdots & \cdots & \cdots & 0 \quad 0 \quad 1 \end{bmatrix}, \tag{181}$$

$$\mathbf{P}(h, J) = \begin{bmatrix} a_2 h\sqrt{M} & 0 & \mathbf{B} & \mathbf{B} & \cdots & \mathbf{B} & a_2 Mh \\ a_4 h\sqrt{M} & 0 & & & & & a_4 Mh \\ \mathbf{B} & & a_2 h\sqrt{M} & 0 & \mathbf{B} & \cdots & \mathbf{B} & a_2 Mh \\ & & a_4 h\sqrt{M} & 0 & & & & a_4 Mh \\ \vdots & & & \ddots & & & \vdots & \vdots \\ \vdots & & & & \ddots & & \vdots & \vdots \\ \vdots & & & & & \ddots & \vdots & \vdots \\ \mathbf{B} & & \mathbf{B} & \cdots & \mathbf{B} & a_2 h\sqrt{M} & 0 & a_2 Mh \\ & & & & & a_4 h\sqrt{M} & 0 & a_4 Mh \\ a_5 h & 0 & a_5 h & 0 & \cdots & \cdots & \cdots & a_5 h & 0 & -\mu h \end{bmatrix}. \tag{182}$$

Then, from (180) and the above matrix definitions, we get the following recursion

$$\mathbf{g}(s+1) \leqslant \left( \mathbf{M}_0 + \mathbf{P}(h, J) \right) \mathbf{g}(s) + \boldsymbol{\epsilon}(s), \tag{183}$$

where we split the matrix $\mathbf{M}(h, J)$ into the sum of a constant matrix $\mathbf{M}_0$ (constant in $h$) and a perturbation matrix $\mathbf{P}(h, J)$. This completes the proof. ∎

### E.4 Proof of Theorem 5.5

This section consists of three parts of the proof. The first part includes the proof of the geometric rates of $\|\mathbf{g}(S)\|$ as in (45) of Theorem 5.5; the second part consists of the proof of the geometric convergence rate of two error sequences $\xi_k^1(s)$ and $\xi_k^5(s)$ as in (46) and (47) of Theorem 5.5; the last part contains the proof of the geometric convergence rate of the error sequence $\xi_{\mathbf{w}*}^6(s)$ as in (48) of Theorem 5.5.

**Rate analysis for $\|\mathbf{g}(S)\|$ convergence to an $\mathcal{O}(C_0 + \Delta)$ ball as in** (45).

**Theorem E.1.** *(Horn & Johnson, 2012, Theorem 6.3.12) Let $\mathbf{X}, \mathbf{E} \in \mathbb{R}^{n \times n}$ and let $q$ be a simple eigenvalue of $\mathbf{X}$. Let $\mathbf{v}$ and $\mathbf{u}$ be, respectively, the right and left eigenvectors of $\mathbf{X}$ corresponding to the eigenvalue $q$. Then,*

1. *for each $\epsilon > 0$, there exists a $\delta > 0$ such that, $\forall p \in \mathbb{C}$ with $|p| < \delta$, there is a unique eigenvalue $q(p)$ of $\mathbf{X} + p\mathbf{E}$ such that $\left| q(p) - q - p\frac{\mathbf{u}^H \mathbf{E} \mathbf{v}}{\mathbf{u}^H \mathbf{v}} \right| \leqslant |p|\epsilon$,*

2. *$q(p)$ is continuous at $p = 0$, and $\lim_{p \to 0} q(p) = q$,*

3. *$q(p)$ is differentiable at $p = 0$, $\frac{dq(p)}{dp}\Big|_{p=0} = \frac{\mathbf{u}^H \mathbf{E} \mathbf{v}}{\mathbf{u}^H \mathbf{v}}$,*

*where $(\cdot)^H$ is Hermitian operator.*

Observe from Lemma 5.4 that $\mathbf{P}(h, J) = \Theta(h)$ and so we can write $\mathbf{P}(h, J) = h\mathbf{E}$ for some constant matrix $\mathbf{E}$ (constant in terms of $h$). Then for $\mathbf{X} = \mathbf{M}_0$ and $\mathbf{P}(h, J) = h\mathbf{E}$, Theorem E.1 can be readily applied. Note that $\mathbf{u} = [0, 0, \cdots, 0, 1]^T$ is both the left and right eigenvector for $\mathbf{M}_0$ corresponding to the simple eigenvalue 1. Also, we have the following by some simple algebraic manipulation using (182):

$$\frac{\mathbf{u}^H \mathbf{E} \mathbf{u}}{\mathbf{u}^H \mathbf{u}} = -\mu. \tag{184}$$

Then from Theorem E.1 for $\mu > \epsilon > 0$ and any $h$ sufficiently small, $\mathbf{M}(h, J)$ has a unique eigenvalue corresponding to the eigenvalue 1 of $\mathbf{M}_0$ and its absolute value is upper bounded by $1 - (\mu - \epsilon)h$. Since $a_1 > a_3$ we get that $a_3 < a_1 < 0.5$ for any $J > \frac{\tau M \log(2M^{\frac{3}{2}}(\sqrt{M}+1))}{\log(1 - \beta^{\tau M})^{-1}} + \tau M + 2$ from the following bound:

$$M^{\frac{3}{2}}(\sqrt{M} + 1)(1 - \beta^{\tau M})^{\left\lfloor \frac{(J-2)}{\tau M} \right\rfloor} < \frac{1}{2} \tag{185}$$

$$\Longleftarrow \frac{(J-2)}{\tau M} > \frac{\log(2M^{\frac{3}{2}}(\sqrt{M}+1))}{\log(1-\beta^{\tau M})^{-1}} + 1 \tag{186}$$

$$\Longleftarrow J > \frac{\tau M \log(2M^{\frac{3}{2}}(\sqrt{M}+1))}{\log(1-\beta^{\tau M})^{-1}} + \tau M + 2. \tag{187}$$

Also, since $a_3 < a_1 < 0.5$, therefore the spectral radius of $\mathbf{M}_0 = 1$.

Since all the other eigenvalues of $\mathbf{M}_0$ are $a_1, a_3$ with $a_3 < a_1 < 0.5$ and $h$ is sufficiently small, we have that the magnitude of the largest eigenvalue of $\mathbf{M}(h, J)$ is equal to $1 - (\mu - \epsilon)h$, which is strictly smaller than 1 for $\epsilon < \mu$ and greater than 0.5 for sufficiently small $h$. Hence we get that the spectral radius of $\mathbf{M}(h, J)$ satisfies $\rho(\mathbf{M}(h, J)) \leqslant 1 - (\mu - \epsilon)h < 1$. Then we have from Lemma 5.6.10 in Horn & Johnson (2012) that there exists a matrix norm, say $\|\cdot\|_{\mathbf{M}(h,J)}$, such that

$$\|\mathbf{M}(h, J)\|_{\mathbf{M}(h,J)} = \rho(\mathbf{M}(h, J)) < 1.$$

Moreover, from Theorem 5.7.13 in Horn & Johnson (2012), we know that for any matrix norm, $\|\cdot\|_{\mathbf{A}}$, there exists a compatible vector norm, say $\|\cdot\|_{\mathbf{A}}$, such that $\|\mathbf{Bx}\|_{\mathbf{A}} \leqslant \|\mathbf{B}\|_{\mathbf{A}} \|\mathbf{x}\|_{\mathbf{A}}$ for all matrices $\mathbf{B}$ and all vectors $\mathbf{x}$. Hence, taking $\|\cdot\|_{\mathbf{M}(h,J)}$ on both sides of (183), where $\|\cdot\|_{\mathbf{M}(h,J)}$ is a compatible vector norm to the matrix norm $\|\cdot\|_{\mathbf{M}(h,J)}$ associated with $\mathbf{M}(h, J)$, we get that:

$$\|\mathbf{g}(s+1)\|_{\mathbf{M}(h,J)} \leqslant \left\|\left(\mathbf{M}_0 + \mathbf{P}(h, J)\right)\mathbf{g}(s)\right\|_{\mathbf{M}(h,J)} + \|\boldsymbol{\epsilon}(s)\|_{\mathbf{M}(h,J)} \tag{188}$$

$$\leqslant \|\mathbf{M}_0 + \mathbf{P}(h, J)\|_{\mathbf{M}(h,J)} \|\mathbf{g}(s)\|_{\mathbf{M}(h,J)} + \|\boldsymbol{\epsilon}(s)\|_{\mathbf{M}(h,J)} \tag{189}$$

$$= \rho(\mathbf{M}(h, J)) \|\mathbf{g}(s)\|_{\mathbf{M}(h,J)} + \|\boldsymbol{\epsilon}(s)\|_{\mathbf{M}(h,J)} \tag{190}$$

$$\Longrightarrow \|\mathbf{g}(S)\|_{\mathbf{M}(h,J)} \leqslant \left(\rho(\mathbf{M}(h, J))\right)^S \|\mathbf{g}(0)\|_{\mathbf{M}(h,J)} + \sum_{s=0}^{S-1} \left(\rho(\mathbf{M}(h, J))\right)^{(S-s-1)} \|\boldsymbol{\epsilon}(s)\|_{\mathbf{M}(h,J)} \tag{191}$$

$$\lesssim_{\mathbf{M}(h,J)} \left(\rho(\mathbf{M}(h, J))\right)^S \|\mathbf{g}(0)\| + \frac{h(C_0 + \Delta)}{1 - \rho(\mathbf{M}(h, J))}, \tag{192}$$

where in the last step we used the bound[14] $\|\boldsymbol{\epsilon}(s)\|_{\mathbf{M}(h,J)} \lesssim_{\mathbf{M}(h,J)} h\Delta + h\gamma(s)$ followed by the fact that $\sup_{s \geqslant 0} \gamma(s) = \sup_{s \geqslant 0} \sum_{k=1}^{d} |\nabla_k f(\hat{\mathbf{w}}^s(s)) - \nabla_k f^{k,s+1}(\hat{\mathbf{w}}^s(s))| = C_0$ where $C_0$ is finite from (176), Assumption 4.12 and continuity of gradients. This completes the first part of the proof.

**Rate analysis for $\xi_k^1(s)$ and $\xi_k^5(s)$ converging to an $\mathcal{O}(h)$ ball.**

From Assumption 4.12 we have that $\{\sup_s \xi_k^1(s)\}_k, \sup_s \xi_{\mathbf{w}*}^6(s)$ are upper bounded by $C_1 \text{diam}(\mathcal{K})$ for some absolute constant $C_1 > 0$. Then from (174) we have for any $S \geqslant 1$:

$$\xi_k^1(s+1) \leqslant a_1 \xi_k^1(s) + a_2 \sqrt{M}(\sqrt{M}+1)C_1 \text{diam}(\mathcal{K})h + a_2 \Delta h \tag{193}$$

$$\Longrightarrow \xi_k^1(S) \leqslant (a_1)^S \xi_k^1(0) + \frac{h}{1-a_1}\left(a_2\sqrt{M}(\sqrt{M}+1)C_1\text{diam}(\mathcal{K}) + a_2\Delta\right), \tag{194}$$

where $a_1 = M^{\frac{3}{2}}(\sqrt{M}+1)(1-\beta^{\tau M})^{\left\lfloor \frac{(J-2)}{\tau M} \right\rfloor} < 1$.

Along similar lines, from (175) we have for any $S \geqslant 1$:

$$\xi_k^5(s+1) \leqslant a_3 \xi_k^5(s) + a_4\sqrt{M}(\sqrt{M}+1)C_1\text{diam}(\mathcal{K})h + a_4\Delta h \tag{195}$$

$$\Longrightarrow \xi_k^5(S) \leqslant (a_3)^S \xi_k^5(0) + \frac{h}{1-a_3}\left(a_4\sqrt{M}(\sqrt{M}+1)C_1\text{diam}(\mathcal{K}) + a_4\Delta\right), \tag{196}$$

where $a_3 = M^{\frac{3}{2}}(1-\beta^{\tau M})^{\left\lfloor \frac{(J-2)}{\tau M} \right\rfloor} < 1$.

---

[14]The exact constants in $\|\boldsymbol{\epsilon}(s)\|_{\mathbf{M}(h,J)} \lesssim_{\mathbf{M}(h,J)} h\Delta + h\gamma(s)$ will depend on $L, M, d$ but these can be directly absorbed in $\lesssim_{\mathbf{M}(h,J)}$.

**Rate analysis for $\xi^6_{\mathbf{w}*}(s)$ converging to an $\mathcal{O}(C_0 + h)$ ball.**

From (178), (194) and the definition of $C_0$ we have for any $S_0 \geqslant 1$, $S > S_0$:

$$\xi^6_{\mathbf{w}*}(s+1) \leqslant (1 - \mu h)\xi^6_{\mathbf{w}*}(s) + C_0 h + a_5 h \sum_{k=1}^{d} \xi^1_k(s) \tag{197}$$

$$\implies \xi^6_{\mathbf{w}*}(S) \leqslant (1 - \mu h)^{S-S_0}\xi^6_{\mathbf{w}*}(S_0) + \sum_{s=S_0}^{S-1} \left( C_0 h + a_5 h \sum_{k=1}^{d} \xi^1_k(s) \right)(1 - \mu h)^{s-S_0} \tag{198}$$

$$\implies \xi^6_{\mathbf{w}*}(S) \leqslant (1 - \mu h)^{S-S_0}\xi^6_{\mathbf{w}*}(S_0) + \frac{h}{1 - (1-\mu h)}\left( C_0 + a_5 \sup_{s \geqslant S_0} \sum_{k=1}^{d} \xi^1_k(s) \right) \tag{199}$$

$$\leqslant (1 - \mu h)^{S-S_0}\xi^6_{\mathbf{w}*}(S_0)$$
$$+ \frac{1}{\mu}\left( C_0 + a_5 d\left( (a_1)^{S_0}\xi^1_k(0) + \frac{h}{1 - a_1}\left( a_2\sqrt{M}(\sqrt{M} + 1)C_1\mathrm{diam}(\mathcal{K}) + a_2\Delta \right) \right) \right) \tag{200}$$

$$= (1 - \mu h)^{S-S_0}\xi^6_{\mathbf{w}*}(S_0) + \frac{C_0}{\mu}$$
$$+ \frac{L\sqrt{Md}}{\mu}\left( (a_1)^{S_0}\xi^1_k(0) + \frac{h}{1 - a_1}\left( a_2\sqrt{M}(\sqrt{M} + 1)C_1\mathrm{diam}(\mathcal{K}) + a_2\Delta \right) \right), \tag{201}$$

where we substituted $a_5 = L\sqrt{Md}$ in the last step. This completes the third and last part of the proof. ∎

## E.5  Proof of Corollary 5.6

Taking $S \to \infty$ in (192) and substituting $\rho(\mathbf{M}(h, J)) = 1 - (\mu - \epsilon)h$, we get:

$$\limsup_{S \to \infty} \|\mathbf{g}(S)\| \lesssim_{\mathbf{M}(h,J)} \frac{(C_0 + \Delta)}{\mu - \epsilon}. \tag{202}$$

Taking $S \to \infty$ in (194) and (196), we get:

$$\limsup_{S \to \infty} \xi^1_k(S) \leqslant \frac{h}{1 - a_1}\left( a_2\sqrt{M}(\sqrt{M} + 1)C_1\mathrm{diam}(\mathcal{K}) + a_2\Delta \right), \tag{203}$$

$$\limsup_{S \to \infty} \xi^5_k(S) \leqslant \frac{h}{1 - a_3}\left( a_4\sqrt{M}(\sqrt{M} + 1)C_1\mathrm{diam}(\mathcal{K}) + a_4\Delta \right). \tag{204}$$

Finally, taking $S \to \infty$ in (201), we have :

$$\limsup_{S \to \infty} \xi^6_{\mathbf{w}*}(S) \leqslant \frac{C_0}{\mu} + \frac{L\sqrt{Md}}{\mu}(a_1)^{S_0}\xi^1_k(0) + \frac{L\sqrt{Md}}{\mu}\left( \frac{h}{1 - a_1}\left( a_2\sqrt{M}(\sqrt{M} + 1)C_1\mathrm{diam}(\mathcal{K}) + a_2\Delta \right) \right). \tag{205}$$

Since the above bound holds for any $S_0$, taking $S_0 \to \infty$ we have:

$$\limsup_{S \to \infty} \xi^6_{\mathbf{w}*}(S) \leqslant \frac{C_0}{\mu} + \frac{L\sqrt{Md}}{\mu}\left( \frac{h}{1 - a_1}\left( a_2\sqrt{M}(\sqrt{M} + 1)C_1\mathrm{diam}(\mathcal{K}) + a_2\Delta \right) \right). \tag{206}$$

This completes the proof. ∎

## E.6  Proof of Theorem 5.8

This section consists of two parts of the proof. The first part includes the proof of the model parameter of Algorithm RESIST converging at a geometric rate to a $\mathcal{O}(C_0 + \Delta)$ radius ball around $\mathbf{W}^*$ as in (54) of Theorem 5.8; the second part consists of the proof of the model parameter of Algorithm RESIST converging at a geometric rate to a $\mathcal{O}(C_0 + h)$ radius ball around $\mathbf{W}^*$ as in (56) of Theorem 5.8.

**Model parameter of Algorithm RESIST converging to an $\mathcal{O}(C_0 + \Delta)$ ball.**

Recall from (133) that we have the bound :

$$\sum_{j=1}^{M} \|\widehat{\mathbf{w}}^s(s) - \mathbf{w}_j(s)\| \leqslant \sqrt{M} \sum_{k=1}^{d} \left\| [\widehat{\mathbf{W}}^{k,s}(s)]_k - [\mathbf{W}(s)]_k \right\|. \tag{207}$$

Then for $\mathbf{W}^* = \mathbf{1}(\mathbf{w}^*)^T$ and $\widehat{\mathbf{W}}^s(s) = \mathbf{1}(\widehat{\mathbf{w}}^s(s))^T$, using Definition 4.3, inequality (133) and Jensen's inequality we get that:

$$\left\| \mathbf{W}(s) - \overline{\mathbf{W}}(s) \right\|_F^2 = \sum_{k=1}^{d} (\xi_k^5(s))^2 \tag{208}$$

$$\left\| \mathbf{W}^* - \widehat{\mathbf{W}}^s(s) \right\|_F^2 = \sum_{i=1}^{M} (\xi_{\mathbf{w}*}^6(s))^2 = M(\xi_{\mathbf{w}*}^6(s))^2 \tag{209}$$

$$\left\| \mathbf{W}(s) - \widehat{\mathbf{W}}^s(s) \right\|_F^2 = \sum_{j=1}^{M} \|\widehat{\mathbf{w}}^s(s) - \mathbf{w}_j(s)\|^2 \leqslant \left( \sum_{j=1}^{M} \|\widehat{\mathbf{w}}^s(s) - \mathbf{w}_j(s)\| \right)^2$$

$$\leqslant M \left( \sum_{k=1}^{d} \left\| [\widehat{\mathbf{W}}^{k,s}(s)]_k - [\mathbf{W}(s)]_k \right\| \right)^2 \leqslant Md \sum_{k=1}^{d} (\xi_k^1(s))^2. \tag{210}$$

Then summing up (208), (209) and (210), taking square root and using the definition of $\mathbf{g}(s)$ from (180) we have the following bound:

$$\sqrt{\left\| \mathbf{W}(s) - \overline{\mathbf{W}}(s) \right\|_F^2 + \left\| \mathbf{W}^* - \widehat{\mathbf{W}}^s(s) \right\|_F^2 + \left\| \mathbf{W}(s) - \widehat{\mathbf{W}}^s(s) \right\|_F^2} =$$

$$\sqrt{\sum_{k=1}^{d} (\xi_k^5(s))^2 + M(\xi_{\mathbf{w}*}^6(s))^2 + Md \sum_{k=1}^{d} (\xi_k^1(s))^2} \tag{211}$$

$$\leqslant \sqrt{Md} \sqrt{\sum_{k=1}^{d} (\xi_k^5(s))^2 + (\xi_{\mathbf{w}*}^6(s))^2 + \sum_{k=1}^{d} (\xi_k^1(s))^2} \tag{212}$$

$$= \sqrt{Md} \left\| \mathbf{g}(s) \right\|. \tag{213}$$

Next, using Cauchy Schwarz inequality along with (213), Theorem 5.5 and the fact that $\|\mathbf{g}(s)\| \lesssim_{\mathbf{M}(h,J)} \|\mathbf{g}(s)\|_{\mathbf{M}(h,J)}$ we get that:

$$\left\| \mathbf{W}(s) - \overline{\mathbf{W}}(s) \right\|_F + \left\| \mathbf{W}^* - \widehat{\mathbf{W}}^s(s) \right\|_F + \left\| \mathbf{W}(s) - \widehat{\mathbf{W}}^s(s) \right\|_F \lesssim_{\mathbf{M}(h,J)}$$

$$\sqrt{3Md} \Big( \rho(\mathbf{M}(h,J)) \Big)^s \|\mathbf{g}(0)\| + \frac{\sqrt{3Md} h(C_0 + \Delta)}{1 - \rho(\mathbf{M}(h,J))}. \tag{214}$$

We now derive the bounds in (214) in the $t$-time scale. Using the facts that $s = \lfloor \frac{t}{J} \rfloor$, $Js \leqslant t < Js + J - 1$, $\|\mathbf{A}\| \leqslant \sqrt{M} \|\mathbf{A}\|_\infty = \sqrt{M}$ for any row stochastic matrix $\mathbf{A} \in \mathbb{R}^{M \times M}$, $[\overline{\mathbf{W}}(s)]_k$ lies in the null space of $\left( \mathbf{I} - \frac{\mathbf{1}\mathbf{1}^T}{M} \right) \prod_{r=J\lfloor \frac{t}{J} \rfloor}^{t} \mathbf{Y}_k(r)$ and invoking (18) we get:

$$\left\| \mathbf{W}(t) - \overline{\mathbf{W}}(t) \right\|_F^2 = \sum_{k=1}^{d} \left\| [\mathbf{W}(t)]_k - [\overline{\mathbf{W}}(t)]_k \right\|^2 \tag{215}$$

$$= \sum_{k=1}^{d} \left\| \left( \prod_{r=J\lfloor \frac{t}{J} \rfloor}^{t} \mathbf{Y}_k(r)[\mathbf{W}(s)]_k - \frac{\mathbf{1}\mathbf{1}^T}{M} \prod_{r=J\lfloor \frac{t}{J} \rfloor}^{t} \mathbf{Y}_k(r)[\mathbf{W}(s)]_k \right) \right\|^2 \tag{216}$$

$$= \sum_{k=1}^{d} \left\| \left(\mathbf{I} - \frac{\mathbf{1}\mathbf{1}^T}{M}\right) \prod_{r=J\lfloor \frac{t}{J}\rfloor}^{t} \mathbf{Y}_k(r)[\mathbf{W}(s)]_k \right\|^2 \tag{217}$$

$$= \sum_{k=1}^{d} \left\| \left(\mathbf{I} - \frac{\mathbf{1}\mathbf{1}^T}{M}\right) \prod_{r=J\lfloor \frac{t}{J}\rfloor}^{t} \mathbf{Y}_k(r)\left([\mathbf{W}(s)]_k - [\overline{\mathbf{W}}(s)]_k\right) \right\|^2 \tag{218}$$

$$\leqslant \sum_{k=1}^{d} \left\| \left(\mathbf{I} - \frac{\mathbf{1}\mathbf{1}^T}{M}\right) \right\|^2 \left\| \prod_{r=J\lfloor \frac{t}{J}\rfloor}^{t} \mathbf{Y}_k(r)\left([\mathbf{W}(s)]_k - [\overline{\mathbf{W}}(s)]_k\right) \right\|^2 \tag{219}$$

$$= \sum_{k=1}^{d} \left\| \prod_{r=J\lfloor \frac{t}{J}\rfloor}^{t} \mathbf{Y}_k(r)\left([\mathbf{W}(s)]_k - [\overline{\mathbf{W}}(s)]_k\right) \right\|^2 \tag{220}$$

$$\leqslant \sum_{k=1}^{d} \left\| \prod_{r=J\lfloor \frac{t}{J}\rfloor}^{t} \mathbf{Y}_k(r) \right\|^2 \left\| \left([\mathbf{W}(s)]_k - [\overline{\mathbf{W}}(s)]_k\right) \right\|^2 \tag{221}$$

$$\leqslant \sum_{k=1}^{d} M \left\| \left([\mathbf{W}(s)]_k - [\overline{\mathbf{W}}(s)]_k\right) \right\|^2 \tag{222}$$

$$\leqslant M \sum_{k=1}^{d} \left\| \left([\mathbf{W}(s)]_k - [\overline{\mathbf{W}}(s)]_k\right) \right\|^2 \tag{223}$$

$$= M \left\| \mathbf{W}(s) - \overline{\mathbf{W}}(s) \right\|_F^2. \tag{224}$$

Next, from Definition 5.7 we have $\widehat{\mathbf{W}}^s(t) = \mathbf{1}(\widehat{\mathbf{w}}^s(t))^T$. Then using the fact that the vector $[\widehat{\mathbf{W}}^s(s)]_k$ lies in the null space of $\left(\mathbf{I} - \mathbf{Q}_k^\pi(s)\right) \prod_{r=J\lfloor \frac{t}{J}\rfloor}^{t} \mathbf{Y}_k(r)$, $\|\mathbf{A}\| \leqslant \sqrt{M} \|\mathbf{A}\|_\infty = \sqrt{M}$ for any row stochastic matrix $\mathbf{A} \in \mathbb{R}^{M \times M}$ and following the steps leading up to (224) we have that:

$$\left\| \mathbf{W}(t) - \widehat{\mathbf{W}}^s(t) \right\|_F^2 = \sum_{k=1}^{d} \left\| [\mathbf{W}(t)]_k - [\widehat{\mathbf{W}}^s(t)]_k \right\|^2 \tag{225}$$

$$= \sum_{k=1}^{d} \left\| \left( \prod_{r=J\lfloor \frac{t}{J}\rfloor}^{t} \mathbf{Y}_k(r)[\mathbf{W}(s)]_k - \mathbf{Q}_k^\pi(s) \prod_{r=J\lfloor \frac{t}{J}\rfloor}^{t} \mathbf{Y}_k(r)[\mathbf{W}(s)]_k \right) \right\|^2 \tag{226}$$

$$= \sum_{k=1}^{d} \left\| \left(\mathbf{I} - \mathbf{Q}_k^\pi(s)\right) \prod_{r=J\lfloor \frac{t}{J}\rfloor}^{t} \mathbf{Y}_k(r)[\mathbf{W}(s)]_k \right\|^2 \tag{227}$$

$$= \sum_{k=1}^{d} \left\| \left(\mathbf{I} - \mathbf{Q}_k^\pi(s)\right) \prod_{r=J\lfloor \frac{t}{J}\rfloor}^{t} \mathbf{Y}_k(r)\left([\mathbf{W}(s)]_k - [\widehat{\mathbf{W}}^s(s)]_k\right) \right\|^2 \tag{228}$$

$$\leqslant \sum_{k=1}^{d} \left\| \left(\mathbf{I} - \mathbf{Q}_k^\pi(s)\right) \right\|^2 \left\| \prod_{r=J\lfloor \frac{t}{J}\rfloor}^{t} \mathbf{Y}_k(r)\left([\mathbf{W}(s)]_k - [\widehat{\mathbf{W}}^s(s)]_k\right) \right\|^2 \tag{229}$$

$$\leqslant (\sqrt{M} + 1)^2 \sum_{k=1}^{d} \left\| \prod_{r=J\lfloor \frac{t}{J}\rfloor}^{t} \mathbf{Y}_k(r)\left([\mathbf{W}(s)]_k - [\widehat{\mathbf{W}}^s(s)]_k\right) \right\|^2 \tag{230}$$

$$\leqslant (\sqrt{M} + 1)^2 \sum_{k=1}^{d} \left\| \prod_{r=J\lfloor \frac{t}{J} \rfloor}^{t} \mathbf{Y}_k(r) \right\|^2 \left\| \left( [\mathbf{W}(s)]_k - [\widehat{\mathbf{W}}^s(s)]_k \right) \right\|^2 \tag{231}$$

$$\leqslant (\sqrt{M} + 1)^2 \sum_{k=1}^{d} M \left\| \left( [\mathbf{W}(s)]_k - [\overline{\mathbf{W}}(s)]_k \right) \right\|^2 \tag{232}$$

$$\leqslant (\sqrt{M} + 1)^2 M \sum_{k=1}^{d} \left\| \left( [\mathbf{W}(s)]_k - [\widehat{\mathbf{W}}^s(s)]_k \right) \right\|^2 \tag{233}$$

$$= (\sqrt{M} + 1)^2 M \left\| \mathbf{W}(s) - \widehat{\mathbf{W}}^s(s) \right\|_F^2 . \tag{234}$$

Similarly, we also get that

$$\left\| \mathbf{W}^* - \widehat{\mathbf{W}}^s(t) \right\|_F^2 \leqslant (\sqrt{M} + 1)^2 M \left\| \mathbf{W}^* - \widehat{\mathbf{W}}^s(s) \right\|_F^2 . \tag{235}$$

Then combining (214), (224), (234), (235), substituting $s = S$ and using the facts that $\frac{t}{J} - 1 < S \leqslant \frac{t}{J}$, $\rho(\mathbf{M}(h, J)) < 1$ for $0 < \epsilon < \mu$ we get:

$$\left\| \mathbf{W}(t) - \overline{\mathbf{W}}(t) \right\|_F + \left\| \mathbf{W}^* - \widehat{\mathbf{W}}^S(t) \right\|_F + \left\| \mathbf{W}(t) - \widehat{\mathbf{W}}^S(t) \right\|_F \lesssim_{\mathbf{M}(h,J)}$$

$$\sqrt{3d}(\sqrt{M} + 1)M \left( \left( \rho(\mathbf{M}(h, J)) \right)^{\frac{t}{J} - 1} \|\mathbf{g}(0)\| + \frac{h(C_0 + \Delta)}{1 - \rho(\mathbf{M}(h, J))} \right). \tag{236}$$

Last, taking $t \to \infty$ and substituting $\rho(\mathbf{M}(h, J)) = 1 - (\mu - \epsilon)h$ for any $0 < \epsilon < \mu$ from Theorem 5.5 we get that:

$$\limsup_{t \to \infty} \left( \left\| \mathbf{W}(t) - \overline{\mathbf{W}}(t) \right\|_F + \left\| \mathbf{W}^* - \widehat{\mathbf{W}}^S(t) \right\|_F + \left\| \mathbf{W}(t) - \widehat{\mathbf{W}}^S(t) \right\|_F \right) \lesssim_{\mathbf{M}(h,J)}$$

$$\limsup_{t \to \infty} \sqrt{3d}(\sqrt{M} + 1)M \left( \left( \rho(\mathbf{M}(h, J)) \right)^{\frac{t}{J} - 1} \|\mathbf{g}(0)\| + \frac{h(C_0 + \Delta)}{1 - \rho(\mathbf{M}(h, J))} \right) \tag{237}$$

$$= \frac{\sqrt{3d}(\sqrt{M} + 1)M(C_0 + \Delta)}{\mu - \epsilon}. \tag{238}$$

This completes the first part of the proof.

**Model parameter of Algorithm RESIST converging to an $\mathcal{O}(C_0 + h)$ ball.**

Using the bound (212), Jensen's inequality and the second part of Theorem 5.5 for some $S_0 < s$ we can write:

$$\sqrt{\left\| \mathbf{W}(s) - \overline{\mathbf{W}}(s) \right\|_F^2 + \left\| \mathbf{W}^* - \widehat{\mathbf{W}}^s(s) \right\|_F^2 + \left\| \mathbf{W}(s) - \widehat{\mathbf{W}}^s(s) \right\|_F^2} \leqslant \sqrt{M}d \left( \sum_{k=1}^{d} \xi_k^5(s) + \xi_{\mathbf{w}*}^6(s) + \sum_{k=1}^{d} \xi_k^1(s) \right) \tag{239}$$

$$\leqslant \sqrt{M}d \left( \sum_{k=1}^{d} \left( (a_1)^s \xi_k^1(0) + \frac{h}{1 - a_1} \left( a_2 \sqrt{M}(\sqrt{M} + 1)C_1 \mathrm{diam}(\mathcal{K}) + a_2 \Delta \right) + \right. \right.$$

$$(a_3)^s \xi_k^5(0) + \frac{h}{1 - a_3} \left( a_4 \sqrt{M}(\sqrt{M} + 1)C_1 \mathrm{diam}(\mathcal{K}) + a_4 \Delta \right) \right) +$$

$$(1 - \mu h)^{s - S_0} \xi_{\mathbf{w}*}^6(S_0) + \frac{C_0}{\mu} + \frac{L\sqrt{M}d}{\mu} \left( (a_1)^{S_0} \xi_k^1(0) \right.$$

$$\left. \left. + \frac{h}{1 - a_1} \left( a_2 \sqrt{M}(\sqrt{M} + 1)C_1 \mathrm{diam}(\mathcal{K}) + a_2 \Delta \right) \right) \right). \tag{240}$$

Then using Cauchy Schwarz inequality, (224), (234), (235), substituting $s = S$ in (240) and using the facts that $\frac{t}{J} - 1 < S \leqslant \frac{t}{J}$ we get:

$$\left\| \mathbf{W}(t) - \overline{\mathbf{W}}(t) \right\|_F + \left\| \mathbf{W}^* - \widehat{\mathbf{W}}^S(t) \right\|_F + \left\| \mathbf{W}(t) - \widehat{\mathbf{W}}^S(t) \right\|_F \leqslant$$

$$\sqrt{3d}(\sqrt{M} + 1)M\left( d\left( (a_1)^{\frac{t}{J}-1}\xi_k^1(0) + \frac{h}{1-a_1}\left( a_2\sqrt{M}(\sqrt{M}+1)C_1\mathrm{diam}(\mathcal{K}) + a_2\Delta \right) \right.\right.$$

$$+ (a_3)^{\frac{t}{J}-1}\xi_k^5(0) + \frac{h}{1-a_3}\left( a_4\sqrt{M}(\sqrt{M}+1)C_1\mathrm{diam}(\mathcal{K}) + a_4\Delta \right) \bigg) +$$

$$(1-\mu h)^{\frac{t}{J}-1-S_0}\xi_{\mathbf{w}^*}^6(S_0) + \frac{C_0}{\mu} + \frac{L\sqrt{Md}}{\mu}\left( (a_1)^{S_0}\xi_k^1(0) \right.$$

$$\left.\left. + \frac{h}{1-a_1}\left( a_2\sqrt{M}(\sqrt{M}+1)C_1\mathrm{diam}(\mathcal{K}) + a_2\Delta \right) \right) \right), \tag{241}$$

where $S > S_0$, $a_1 = M^{\frac{3}{2}}(\sqrt{M}+1)(1-\beta^{\tau M})^{\left\lfloor \frac{(J-2)}{\tau M} \right\rfloor} < 1$, $a_3 = M^{\frac{3}{2}}(1-\beta^{\tau M})^{\left\lfloor \frac{(J-2)}{\tau M} \right\rfloor} < 1$, $a_2 = (\sqrt{M}+1)^2 L$, $a_4 = L$. Last, taking $t \to \infty$ and $S_0 \to \infty$ in the above inequality we get:

$$\limsup_{t\to\infty}\left( \left\| \mathbf{W}(t) - \overline{\mathbf{W}}(t) \right\|_F + \left\| \mathbf{W}^* - \widehat{\mathbf{W}}^S(t) \right\|_F + \left\| \mathbf{W}(t) - \widehat{\mathbf{W}}^S(t) \right\|_F \right) \leqslant$$

$$\sqrt{3d}(\sqrt{M}+1)M\left( \frac{hd}{1-a_1}\left( a_2\sqrt{M}(\sqrt{M}+1)C_1\mathrm{diam}(\mathcal{K}) + a_2\Delta \right) \right.$$

$$+ \frac{hd}{1-a_3}\left( a_4\sqrt{M}(\sqrt{M}+1)C_1\mathrm{diam}(\mathcal{K}) + a_4\Delta \right) + \frac{C_0}{\mu}$$

$$\left. + \left( \frac{L\sqrt{Md}}{\mu}\frac{h}{1-a_1}\left( a_2\sqrt{M}(\sqrt{M}+1)C_1\mathrm{diam}(\mathcal{K}) + a_2\Delta \right) \right) \right), \tag{242}$$

which completes the proof.

# F   Proofs for Algorithmic Convergence Under Nonconvexity

## F.1   The sum of PŁ functions need not satisfy the PŁ inequality: A counterexample in $\mathbb{R}^2$

Consider the functions

$$f(x,y) = \tfrac{1}{2}\big(y - \sin x\big)^2, \qquad g(x,y) = \tfrac{1}{4}\big(y - 3 - \sin(x-3)\big)^2.$$

The function $f$ satisfies the PŁ inequality (see Apidopoulos et al. (2022)), and its critical set is $\{(x,y) : y = \sin x\}$. The function $g$ is obtained from $f$ by translation and scaling, namely $g(x,y) = \frac{1}{2}f(x-3, y-3)$, and therefore also satisfies the PŁ inequality. However, the sum $f + g$ does not satisfy the PŁ inequality. As illustrated in Fig. 14, the function $f + g$ possesses saddle points. Since any function satisfying the PŁ inequality has the property that every critical point is a global minimizer, the presence of saddle points implies that $f + g$ cannot satisfy the PŁ inequality.

## F.2   Proof of Lemma 6.3

*Proof.* Recall that from the inexact averaged update in Lemma 4.11, we have

$$\widehat{\mathbf{w}}^{s+1}(s+1) = \widehat{\mathbf{w}}^s(s) - h\nabla f(\widehat{\mathbf{w}}^s(s)) + \mathbf{e}_1(s) + \mathbf{e}_2(s), \tag{243}$$

where

$$\left\| \mathbf{e}_2(s) \right\| \leqslant Lh\sqrt{Md}\sum_{k=1}^d \left\| [\widehat{\mathbf{W}}^{k,s}(s)]_k - [\mathbf{W}(s)]_k \right\|. \tag{244}$$

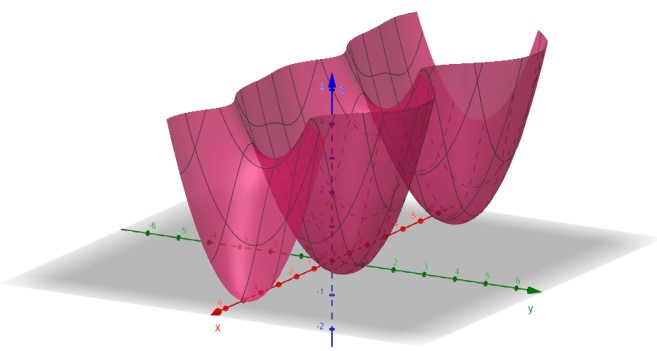

Figure 14: Surface plot of $f(x, y) + g(x, y)$. The landscape exhibits saddle points, showing that the sum fails to satisfy the PŁ inequality even though each term individually does.

Since $f := \frac{1}{M} \sum_{i=1}^{M} f_i$ satisfies the PŁ inequality from Assumption 6.1 and also Assumption 4.8, we get that:

$$f(\hat{\mathbf{w}}^s(s) - h\nabla f(\hat{\mathbf{w}}^s(s))) \leqslant f(\hat{\mathbf{w}}^s(s)) + \langle \nabla f(\hat{\mathbf{w}}^s(s)), -h\nabla f(\hat{\mathbf{w}}^s(s)) \rangle + \frac{L}{2} \|h\nabla f(\hat{\mathbf{w}}^s(s))\|^2 \tag{245}$$

$$= f(\hat{\mathbf{w}}^s(s)) - \frac{h(2 - Lh)}{2} \|\nabla f(\hat{\mathbf{w}}^s(s))\|^2 \tag{246}$$

$$\leqslant f(\hat{\mathbf{w}}^s(s)) - \mu h(2 - Lh)(f(\hat{\mathbf{w}}^s(s)) - f^*). \tag{247}$$

For $0 < h < \frac{2}{L}$, we will have $\mu h(2 - Lh) < 1$ and hence from the last inequality we have

$$f(\hat{\mathbf{w}}^s(s) - h\nabla f(\hat{\mathbf{w}}^s(s))) - f^* \leqslant \left(1 - \mu h(2 - Lh)\right)(f(\hat{\mathbf{w}}^s(s)) - f^*) \tag{248}$$

$$\implies f(\hat{\mathbf{w}}^{s+1}(s+1)) - f^* \leqslant \left(1 - \mu h(2 - Lh)\right)(f(\hat{\mathbf{w}}^s(s)) - f^*) +$$
$$\left(f(\hat{\mathbf{w}}^{s+1}(s+1)) - f(\hat{\mathbf{w}}^s(s) - h\nabla f(\hat{\mathbf{w}}^s(s)))\right). \tag{249}$$

By Assumption 4.8 and for some sufficiently large compact set $\mathcal{K}$ defined in Assumption 4.12, we have that $\sup_{\mathbf{w} \in \mathcal{K}} \|\nabla f(\mathbf{w})\| \leqslant L \, \mathrm{diam}(\mathcal{K})$. Then from the Mean Value Theorem, the function $f$ is locally Lipschitz continuous in $\mathcal{K}$ and for any $\mathbf{w}_1, \mathbf{w}_2 \in \mathcal{K}$ we have:

$$f(\mathbf{w}_1) - f(\mathbf{w}_2) \leqslant L \, \mathrm{diam}(\mathcal{K}) \|\mathbf{w}_1 - \mathbf{w}_2\|. \tag{250}$$

Then using (250) in (249) along with the update (243) and bound on $\|\mathbf{e}_2(s)\|$ we have:

$$f(\hat{\mathbf{w}}^{s+1}(s+1)) - f^* \leqslant \left(1 - \mu h(2 - Lh)\right)(f(\hat{\mathbf{w}}^s(s)) - f^*) +$$
$$L \, \mathrm{diam}(\mathcal{K}) \|\hat{\mathbf{w}}^{s+1}(s+1) - (\hat{\mathbf{w}}^s(s) - h\nabla f(\hat{\mathbf{w}}^s(s)))\| \tag{251}$$

$$\implies f(\hat{\mathbf{w}}^{s+1}(s+1)) - f^* \leqslant \left(1 - \mu h(2 - Lh)\right)(f(\hat{\mathbf{w}}^s(s)) - f^*) + L \, \mathrm{diam}(\mathcal{K})\left(\|\mathbf{e}_1(s)\| + \|\mathbf{e}_2(s)\|\right) \tag{252}$$

$$\leqslant \left(1 - \mu h(2 - Lh)\right)(f(\hat{\mathbf{w}}^s(s)) - f^*) +$$
$$L \, \mathrm{diam}(\mathcal{K})\left(\|\mathbf{e}_1(s)\| + Lh\sqrt{Md} \sum_{k=1}^{d} \left\|[\widehat{\mathbf{W}}^{k,s}(s)]_k - [\mathbf{W}(s)]_k\right\|\right), \tag{253}$$

which completes the proof. ∎

### F.3 Proof of Theorem 6.4

*Proof.* Under Assumption 6.1 suppose $\mathbf{w}_i^* \in \arg\min_{\mathbf{w}} f_i(\mathbf{w})$ for all $i \in \{1, \cdots, M\}$ and without loss of generality $\{\mathbf{w}_i^*\}_{i=1}^M \subset \mathcal{K}$. Then it can be easily checked that the consensus error bounds for the sequences $\{\xi_k^1(s)\}_s, \{\xi_k^5(s)\}_s$ will be exactly the same as in Theorem 5.5 since these bounds were derived without any convexity assumption (see Appendix E.4 for proof of Theorem 5.5). Then recalling the consensus error bounds (194), (196) from proof of Theorem 5.5 we get :

$$\xi_k^1(S) \leqslant (a_1)^S \xi_k^1(0) + \frac{h}{1-a_1}\left(a_2\sqrt{M}(\sqrt{M}+1)C_1\mathrm{diam}(\mathcal{K}) + a_2\Delta\right), \tag{254}$$

$$\xi_k^5(S) \leqslant (a_3)^S \xi_k^5(0) + \frac{h}{1-a_3}\left(a_4\sqrt{M}(\sqrt{M}+1)C_1\mathrm{diam}(\mathcal{K}) + a_4\Delta\right), \tag{255}$$

where $a_1 = M^{\frac{3}{2}}(\sqrt{M}+1)(1-\beta^{\tau M})^{\left\lfloor\frac{(J-2)}{\tau M}\right\rfloor} < 1$, $a_3 = M^{\frac{3}{2}}(1-\beta^{\tau M})^{\left\lfloor\frac{(J-2)}{\tau M}\right\rfloor} < 1$ and $\Delta$ is defined in Lemma 5.4. For deriving the function error sequence rates, we use Lemmas 4.9, 4.7, and 6.3. Using Lemma 4.9 followed by Jensen's inequality and Assumption 4.12 we have that:

$$\left\|[\widehat{\mathbf{T}}^{k,s}(s)]_k - [\mathbf{T}(s)]_k\right\| \leqslant (\sqrt{M}+1)L\sqrt{M}\sum_{k=1}^d \left\|[\mathbf{W}(s)]_k - [\widehat{\mathbf{W}}^{k,s}(s)]_k\right\| +$$

$$(\sqrt{M}+1)LM\|\mathbf{w}^* - \widehat{\mathbf{w}}^s(s)\| + (\sqrt{M}+1)L\sum_{i=1}^M \|\mathbf{w}^* - \mathbf{w}_i^*\| \tag{256}$$

$$\leqslant (\sqrt{M}+1)L\sqrt{M}d\underbrace{\left\|\mathbf{W}(s) - \widehat{\mathbf{W}}^{k,s}(s)\right\|_F}_{=\sqrt{\sum_{i=1}^M \|\mathbf{w}_i(s) - \widehat{\mathbf{w}}^{k,s}(s)\|}} +$$

$$(\sqrt{M}+1)LM\|\mathbf{w}^* - \widehat{\mathbf{w}}^s(s)\| + (\sqrt{M}+1)L\sum_{i=1}^M \|\mathbf{w}^* - \mathbf{w}_i^*\| \tag{257}$$

$$\leqslant (\sqrt{M}+1)LM(\sqrt{d}+2)\,\mathrm{diam}(\mathcal{K}). \tag{258}$$

Then from Lemma 4.7, (258) and Assumption 4.12 we have for any $S > 0$ :

$$\left\|[\widehat{\mathbf{W}}^{k,S}(S)]_k - [\mathbf{W}(S)]_k\right\| \leqslant (a_1)^S \left\|[\widehat{\mathbf{W}}^{k,0}(0)]_k - [\mathbf{W}(0)]_k\right\| + \frac{h(\sqrt{M}+1)}{1-a_1}\sup_{s\geqslant 0}\left\|[\widehat{\mathbf{T}}^{k,s}(s)]_k - [\mathbf{T}(s)]_k\right\| \tag{259}$$

$$\leqslant (a_1)^S \left\|[\widehat{\mathbf{W}}^{k,0}(0)]_k - [\mathbf{W}(0)]_k\right\| + \frac{h(\sqrt{M}+1)^2}{1-a_1}LM(\sqrt{d}+2)\,\mathrm{diam}(\mathcal{K}) \tag{260}$$

$$\leqslant (a_1)^S \left\|\widehat{\mathbf{W}}^{k,0}(0) - \mathbf{W}(0)\right\|_F + \frac{h(\sqrt{M}+1)^2}{1-a_1}LM(\sqrt{d}+2)\,\mathrm{diam}(\mathcal{K}) \tag{261}$$

$$\leqslant (a_1)^S M\mathrm{diam}(\mathcal{K}) + \frac{h(\sqrt{M}+1)^2}{1-a_1}LM(\sqrt{d}+2)\,\mathrm{diam}(\mathcal{K}), \tag{262}$$

where $a_1 < 1$. Substituting the above bound (262) in Lemma 6.3 for $s = S \geqslant 0$ and using the following bound from (176) given by

$$\|\mathbf{e}_1(s)\| \leqslant h\sup_{s\geqslant 0}\gamma(s) = h\sup_{s\geqslant 0}\sum_{k=1}^d |\nabla_k f(\widehat{\mathbf{w}}^s(s)) - \nabla_k f^{k,s+1}(\widehat{\mathbf{w}}^s(s))| = C_0 h,$$

we have:

$$f(\widehat{\mathbf{w}}^{S+1}(S+1)) - f^* \leqslant \left(1 - \mu h(2 - Lh)\right)(f(\widehat{\mathbf{w}}^S(S)) - f^*) +$$

$$L \operatorname{diam}(\mathcal{K}) \bigg( h C_0 + L h d \sqrt{M d} \bigg( (a_1)^S M \operatorname{diam}(\mathcal{K}) + \frac{h(\sqrt{M}+1)^2}{1-a_1} L M (\sqrt{d}+2) \operatorname{diam}(\mathcal{K}) \bigg) \bigg) \tag{263}$$

$$\implies f(\widehat{\mathbf{w}}^{S+1}(S+1)) - f^* \leqslant \bigg( 1 - \mu h(2-Lh) \bigg)^{S+1} (f(\widehat{\mathbf{w}}^0(0)) - f^*) + L \operatorname{diam}(\mathcal{K}) \frac{C_0}{\mu(2-Lh)} +$$
$$L \operatorname{diam}(\mathcal{K}) \bigg( \frac{L h d \sqrt{M d}(\sqrt{M}+1)^2}{(1-a_1)(\mu(2-Lh))} L M (\sqrt{d}+2) \operatorname{diam}(\mathcal{K})$$
$$+ L h d \sqrt{M d} \bigg( \sum_{s=0}^S (a_1)^s \underbrace{(1-\mu h(2-Lh))^{S-s}}_{\leqslant 1} M \operatorname{diam}(\mathcal{K}) \bigg) \bigg) \tag{264}$$

$$\leqslant \bigg( 1 - \mu h(2-Lh) \bigg)^{S+1} (f(\widehat{\mathbf{w}}^0(0)) - f^*) + L \operatorname{diam}(\mathcal{K}) \frac{C_0}{\mu(2-Lh)} +$$
$$L \operatorname{diam}(\mathcal{K}) \bigg( \frac{L h d \sqrt{M d}(\sqrt{M}+1)^2}{(1-a_1)(\mu(2-Lh))} L M (\sqrt{d}+2) \operatorname{diam}(\mathcal{K})$$
$$+ \frac{L h d \sqrt{M d}}{1-a_1} M \operatorname{diam}(\mathcal{K}) \bigg) \tag{265}$$

$$\implies f(\widehat{\mathbf{w}}^S(S)) - f^* \leqslant \bigg( 1 - \mu h(2-Lh) \bigg)^{S} (f(\widehat{\mathbf{w}}^0(0)) - f^*) + L \operatorname{diam}(\mathcal{K}) \frac{C_0}{\mu(2-Lh)} +$$
$$\frac{L^2 h d \sqrt{M d}}{1-a_1} (\operatorname{diam}(\mathcal{K}))^2 \bigg( \frac{(\sqrt{M}+1)^2}{\mu(2-Lh)} L M (\sqrt{d}+2) + M \bigg), \tag{266}$$

which completes the proof. ∎

### F.4 Proof of Theorem 6.6

*Proof.* Recalling the bound (174) from Lemma 4.7 and Lemma 4.9 we have for $h := h(s) = \frac{p}{(s+1)^\omega}$, $p > 0$ that:

$$\xi_k^1(s+1) \leqslant \underbrace{M^{\frac{3}{2}}(\sqrt{M}+1)(1-\beta^{\tau M})^{\left\lfloor \frac{(J-2)}{\tau M} \right\rfloor}}_{=a_1} \xi_k^1(s) + h(s)(\sqrt{M}+1)^2 L \sqrt{M} \sum_{k=1}^d \xi_k^1(s)$$
$$+ h(s)(\sqrt{M}+1)^2 L M \xi_{\mathbf{w}*}^6(s) + h(s)(\sqrt{M}+1)^2 L \underbrace{\sum_{i=1}^M \|\mathbf{w}^* - \mathbf{w}_i^*\|}_{=\Delta}. \tag{267}$$

♣ Using Assumption 4.12 in the last three terms of (267), we can bound

$$\max \bigg\{ \Delta, \sup_{s \geqslant 0} \sum_{k=1}^d \xi_k^1(s), \sup_{s \geqslant 0} \xi_{\mathbf{w}*}^6(s) \bigg\} \leqslant C(M,d) \operatorname{diam}(\mathcal{K})$$

for some sufficiently large constant[15] $C(M,d) = \mathcal{O}(M\sqrt{d})$ to get:

$$\xi_k^1(s+1) \leqslant a_1 \xi_k^1(s) + C(M,d) \operatorname{diam}(\mathcal{K}) h(s), \tag{268}$$

$$\implies \xi_k^1(S) \leqslant (a_1)^S \xi_k^1(0) + C(M,d) \operatorname{diam}(\mathcal{K}) \sum_{s=0}^{S-1} (a_1)^{S-s-1} h(s) \tag{269}$$

$$\implies \limsup_{S \to \infty} \xi_k^1(S) \leqslant \limsup_{S \to \infty} (a_1)^S \xi_k^1(0) + C(M,d) \operatorname{diam}(\mathcal{K}) \limsup_{S \to \infty} \sum_{s=0}^{S-1} (a_1)^{S-s-1} h(s) = 0 \tag{270}$$

---

[15]Observe that $\Delta = \mathcal{O}(M\operatorname{diam}(\mathcal{K}))$, $\xi_{\mathbf{w}*}^6(s) = \mathcal{O}(\operatorname{diam}(\mathcal{K}))$ and $\sum_{k=1}^d \xi_k^1(s) = \mathcal{O}(\sqrt{M d} \operatorname{diam}(\mathcal{K}))$.

$$\implies \xi_k^1(S) \xrightarrow{S \to \infty} 0. \tag{271}$$

Note that in the second last step, we used the fact that $a_1 < 1$ and that the partial sum $\sum_{s=0}^{S-1} (a_1)^{S-s-1} h(s)$ is monotonically decreasing in $S$ after any sufficiently large $S$ from the argument below:

$$\sum_{s=0}^{S-1} (a_1)^{S-s-1} h(s) > \sum_{s=0}^{S} (a_1)^{S+1-s-1} h(s)$$

$$= a_1 \left( \sum_{s=0}^{S-1} (a_1)^{S-s-1} h(s) \right) + (a_1)^{S+1-S-1} h(S) \tag{272}$$

$$\impliedby (1 - a_1) \sum_{s=0}^{S-1} (a_1)^{S-s-1} h(s) > h(S) = \frac{p}{(S+1)^\omega} \tag{273}$$

$$\impliedby \frac{p}{S^\omega} (1 - (a_1)^S) > \frac{p}{(S+1)^\omega} \tag{274}$$

$$\impliedby 1 + \omega S^{-1} + o(S^{-1}) > 1 + (a_1)^S + o((a_1)^S) \text{ for any } \omega > 0 \text{ and } S > 1. \tag{275}$$

Then by Monotone Convergence Theorem[16], taking limit in (272), we get that the partial sum $\sum_{s=0}^{S-1} (a_1)^{S-s-1} h(s)$ converges to 0. In particular, we have a decay rate of $\mathcal{O}(\frac{1}{S^\omega})$ from the following bound:

$$\sum_{s=0}^{S-1} (a_1)^{S-s-1} h(s) = \sum_{s=0}^{\lfloor \frac{S}{2} \rfloor} (a_1)^{S-s-1} h(s) + \sum_{s=\lfloor \frac{S}{2} \rfloor+1}^{S-1} (a_1)^{S-s-1} h(s) \tag{276}$$

$$\leqslant h(0) \sum_{s=0}^{\lfloor \frac{S}{2} \rfloor} (a_1)^{S-s-1} + h\left( \left\lfloor \frac{S}{2} \right\rfloor + 1 \right) \sum_{s=\lfloor \frac{S}{2} \rfloor+1}^{S-1} (a_1)^{S-s-1} \tag{277}$$

$$\leqslant (a_1)^{S-\lfloor \frac{S}{2} \rfloor-1} \frac{p}{1 - a_1} + \frac{p}{(\lfloor \frac{S}{2} \rfloor + 2)^\omega} \frac{1}{1 - a_1} \tag{278}$$

$$\underbrace{\leqslant}_{\text{for any sufficiently large } S} \frac{2p}{(1 - a_1)(\lfloor \frac{S}{2} \rfloor + 2)^\omega} = \frac{C_5}{S^\omega}. \tag{279}$$

Then by (269) and (279) we have that:

$$\xi_k^1(S) = \mathcal{O}\left( \frac{1}{S^\omega} \right). \tag{280}$$

♠

Similarly, recalling the bound (175) from Lemma 4.6 and Lemma 4.9 we get for $h := h(s) = \frac{p}{(s+1)^\omega}$ that :

$$\xi_k^5(s+1) \leqslant \underbrace{M^{\frac{3}{2}} (1 - \beta^{\tau M})^{\left\lfloor \frac{(J-2)}{\tau M} \right\rfloor}}_{=a_3} \xi_k^5(s) + h(s) L \sqrt{M} \sum_{k=1}^{d} \xi_k^1(s) + h(s) LM \xi_{\mathbf{w}*}^6(s) + h(s) L \underbrace{\sum_{i=1}^{M} \| \mathbf{w}^* - \mathbf{w}_i^* \|}_{=\Delta}. \tag{281}$$

Then, following similar steps as before from symbol ♣ to symbol ♠ and using the fact that $a_3 < 1$, we get that

$$\xi_k^5(S) \xrightarrow{S \to \infty} 0. \tag{282}$$

Next, recall from the inexact averaged update of Lemma 4.11 we have for $h := h(s)$ that

$$\hat{\mathbf{w}}^{s+1}(s+1) = \hat{\mathbf{w}}^s(s) - h(s) \nabla f(\hat{\mathbf{w}}^s(s)) + \mathbf{e}_2(s) + \mathbf{e}_1(s), \tag{283}$$

---

[16]The partial sum $\sum_{s=0}^{S-1} (a_1)^{S-s-1} h(s)$ is non-negative and decreasing for large $S$.

where[17]

$$\|\mathbf{e}_2(s)\| \leqslant Lh(s)\sqrt{Md} \sum_{k=1}^{d} \left\|[\widehat{\mathbf{W}}^{k,s}(s)]_k - [\mathbf{W}(s)]_k\right\| \underbrace{=}_{\text{Definition 4.3}} Lh(s)\sqrt{Md} \sum_{k=1}^{d} \xi_k^1(s), \qquad (284)$$

and

$$\|\mathbf{e}_1(s)\| \leqslant h(s)\sup_{s \geqslant 0} \gamma(s) = h(s)\sup_{s \geqslant 0} \sum_{k=1}^{d} |\nabla_k f(\widehat{\mathbf{w}}^s(s)) - \nabla_k f^{k,s+1}(\widehat{\mathbf{w}}^s(s))| = C_0 h(s),$$

from (176) after substituting $h := h(s)$. Using Assumption 4.8 of gradient Lipschitz continuity on $f$ followed by Assumption 4.12 on the update (283) for a compact $\mathcal{K}$ we have that :

$$f(\widehat{\mathbf{w}}^s(s)) - f(\widehat{\mathbf{w}}^{s+1}(s+1)) \geqslant \langle \nabla f(\widehat{\mathbf{w}}^s(s)), \widehat{\mathbf{w}}^s(s) - \widehat{\mathbf{w}}^{s+1}(s+1)\rangle - \frac{L}{2}\left\|\widehat{\mathbf{w}}^s(s) - \widehat{\mathbf{w}}^{s+1}(s+1)\right\|^2 \qquad (285)$$

$$\geqslant h(s)\left\|\nabla f(\widehat{\mathbf{w}}^s(s))\right\|^2 - \underbrace{\left\|\nabla f(\widehat{\mathbf{w}}^s(s))\right\|}_{\leqslant L \ \mathrm{diam}(\mathcal{K})}(\|\mathbf{e}_2(s) + \mathbf{e}_1(s)\|)$$

$$- \frac{2L(h(s))^2}{2}\left\|\nabla f(\widehat{\mathbf{w}}^s(s))\right\|^2 - \frac{2L}{2}(\|\mathbf{e}_2(s) + \mathbf{e}_1(s)\|^2) \qquad (286)$$

$$\geqslant h(s)\Big(1 - Lh(s)\Big)\left\|\nabla f(\widehat{\mathbf{w}}^s(s))\right\|^2 - L\mathrm{diam}(\mathcal{K})h(s)\Big(C_0 + L\sqrt{Md}\sum_{k=1}^{d}\xi_k^1(s)\Big)$$

$$- L(h(s))^2\Big(C_0 + L\sqrt{Md}\sum_{k=1}^{d}\xi_k^1(s)\Big)^2. \qquad (287)$$

Next, for some constant $C_2 = C(L, M, d, \mathrm{diam}(\mathcal{K}))$, using Assumption 4.12 we can bound

$$\sup_{s \geqslant 0} L\Big(C_0 + L\sqrt{Md}\sum_{k=1}^{d}\xi_k^1(s)\Big)^2 \leqslant C(L, M, d, \mathrm{diam}(\mathcal{K})) = C_2 = \mathcal{O}\Big(L^3\Big(Md\ \mathrm{diam}(\mathcal{K})\Big)^2\Big). \qquad (288)$$

We also note that $C_0 = \mathcal{O}(LMd\ \mathrm{diam}(\mathcal{K}))$ from a simple application of gradient Lipschitz continuity. Indeed, recall that

$$C_0 = \sup_{s \geqslant 0} \sum_{k=1}^{d} |\nabla_k f(\widehat{\mathbf{w}}^s(s)) - \nabla_k f^{k,s+1}(\widehat{\mathbf{w}}^s(s))|,$$

and hence

$$C_0 \leqslant \sup_{s \geqslant 0} \sum_{k=1}^{d}\left(|\nabla_k f(\widehat{\mathbf{w}}^s(s)) - \nabla_k f(\mathbf{w}^*)| + \sum_{j=1}^{M} |\nabla_k f_j(\mathbf{w}_j^*) - \nabla_k f_j(\widehat{\mathbf{w}}^s(s))|\right) \leqslant \mathcal{O}(LMd\ \mathrm{diam}(\mathcal{K})) \qquad (289)$$

$$\implies C_0 + L\sqrt{Md}\sum_{k=1}^{d}\xi_k^1(s) \leqslant \mathcal{O}(LMd\ \mathrm{diam}(\mathcal{K})). \qquad (290)$$

Then using the constant $C_2$ from (288) in the last term on right hand side of inequality (287), followed by rearranging, telescoping and finally using $0 < p \leqslant \frac{1}{2L}$ we get:

$$h(s)(1 - Lh(s))\left\|\nabla f(\widehat{\mathbf{w}}^s(s))\right\|^2 \leqslant f(\widehat{\mathbf{w}}^s(s)) - f(\widehat{\mathbf{w}}^{s+1}(s+1)) + C_2(h(s))^2$$

$$+ L\mathrm{diam}(\mathcal{K})h(s)\Big(C_0 + L\sqrt{Md}\sum_{k=1}^{d}\xi_k^1(s)\Big)$$

$$(291)$$

$$\implies \sum_{s=0}^{S-1}\left(h(s)(1 - Lh(s))\left\|\nabla f(\widehat{\mathbf{w}}^s(s))\right\|^2\right) \leqslant f(\widehat{\mathbf{w}}^0(0)) - f(\widehat{\mathbf{w}}^S(S)) + C_2\sum_{s=0}^{S-1}(h(s))^2$$

---

[17]Since the bound on $\|\mathbf{e}_2(s)\|$ from Lemma 4.11 is derived by using just a single update step for $\widehat{\mathbf{w}}^s(s)$, without loss of generality, we can substitute $h := h(s)$ in the right hand side of the bound on $\|\mathbf{e}_2(s)\|$.

$$+ L\mathrm{diam}(\mathcal{K})C_0 \sum_{s=0}^{S-1} h(s)$$

$$+ L^2\mathrm{diam}(\mathcal{K})\sqrt{Md}\bigg( \sum_{k=1}^{d} \sum_{s=0}^{S-1} \xi_k^1(s)h(s) \bigg) \tag{292}$$

$$\implies \min_{0 \leqslant s \leqslant S-1} \|\nabla f(\hat{\mathbf{w}}^s(s))\|^2 \sum_{s=0}^{S-1} \bigg( h(s)\underbrace{(1 - Lh(s))}_{\geqslant \frac{1}{2} \text{ for } p \leqslant \frac{1}{2L}} \bigg) \leqslant f(\hat{\mathbf{w}}^0(0)) - f(\hat{\mathbf{w}}^S(S)) + C_2 \sum_{s=0}^{S-1} (h(s))^2$$

$$+ L\mathrm{diam}(\mathcal{K})C_0 \sum_{s=0}^{S-1} h(s)$$

$$+ L^2\mathrm{diam}(\mathcal{K})\sqrt{Md}\bigg( \sum_{k=1}^{d} \sum_{s=0}^{S-1} \xi_k^1(s)h(s) \bigg) \tag{293}$$

$$\implies \frac{1}{2} \min_{0 \leqslant s \leqslant S-1} \|\nabla f(\hat{\mathbf{w}}^s(s))\|^2 \sum_{s=0}^{S-1} h(s) \leqslant f(\hat{\mathbf{w}}^0(0)) - f(\hat{\mathbf{w}}^S(S)) + C_2 \sum_{s=0}^{S-1} (h(s))^2$$

$$+ L\mathrm{diam}(\mathcal{K})C_0 \sum_{s=0}^{S-1} h(s)$$

$$+ L^2\mathrm{diam}(\mathcal{K})\sqrt{Md}\bigg( \sum_{k=1}^{d} \sum_{s=0}^{S-1} \xi_k^1(s)h(s) \bigg) \tag{294}$$

which, after rearranging yields:

$$\min_{0 \leqslant s \leqslant S-1} \|\nabla f(\hat{\mathbf{w}}^s(s))\|^2 \leqslant \frac{2\bigg( f(\hat{\mathbf{w}}^0(0)) - f(\hat{\mathbf{w}}^S(S)) \bigg)}{\sum_{s=0}^{S-1} h(s)} + 2C_2 \frac{\sum_{s=0}^{S-1}(h(s))^2}{\sum_{s=0}^{S-1} h(s)}$$

$$+ 2L\mathrm{diam}(\mathcal{K})C_0 + 2L^2\mathrm{diam}(\mathcal{K})\sqrt{Md} \underbrace{\frac{\bigg( \sum_{k=1}^{d} \sum_{s=0}^{S-1} \xi_k^1(s)h(s) \bigg)}{\sum_{s=0}^{S-1} h(s)}}_{T_1}. \tag{295}$$

Using the bound on $\xi_k^1(s)$ from (269) and from Lemma 6.3 that $\max_{1 \leqslant k \leqslant d} \xi_k^1(0) \leqslant C_3\mathrm{diam}(\mathcal{K})$ for some constant[18] $C_3$ from Assumption 4.12 followed by Hölder inequality (Lemma 6.5), the term $T_1$ in (295) can be bounded as:

$$T_1 = \sum_{k=1}^{d} \frac{\bigg( \sum_{s=0}^{S-1} \xi_k^1(s)h(s) \bigg)}{\sum_{s=0}^{S-1} h(s)} \leqslant \frac{d\bigg( \sum_{s=0}^{S-1} \bigg( (a_1)^s C_3\mathrm{diam}(\mathcal{K}) + C_2\mathrm{diam}(\mathcal{K}) \sum_{l=0}^{s-1}(a_1)^{s-l-1}h(l) \bigg) h(s) \bigg)}{\sum_{s=0}^{S-1} h(s)} \tag{296}$$

$$= \frac{d\bigg( \sum_{s=0}^{S-1}(a_1)^s h(s)C_3\mathrm{diam}(\mathcal{K}) \bigg)}{\sum_{s=0}^{S-1} h(s)} + \frac{d\bigg( C_2\mathrm{diam}(\mathcal{K}) \sum_{s=0}^{S-1} \bigg( \sum_{l=0}^{s-1}(a_1)^{s-l-1}h(l) \bigg) h(s) \bigg)}{\sum_{s=0}^{S-1} h(s)}$$

$$\underbrace{\leqslant}_{\text{Hölder inequality}} \underbrace{\frac{dC_3\mathrm{diam}(\mathcal{K})\sqrt{\bigg( \sum_{s=0}^{S-1}(a_1)^{2s} \bigg)}\sqrt{\bigg( \sum_{s=0}^{S-1}(h(s))^2 \bigg)}}{\sum_{s=0}^{S-1} h(s)}}_{T_4}$$

---

[18]Note that $C_3 = \mathcal{O}(1)$ provided $\mathcal{K}$ contains some sufficiently large cube in $\mathbb{R}^d$.

$$+ \underbrace{\frac{dC_2\text{diam}(\mathcal{K})\left(\left(\sum_{s=0}^{S-1}\left((h(s))^{1-a}\sum_{l=0}^{s-1}(a_1)^{s-l-1}h(l)\right)^{\frac{q}{q-1}}\right)^{1-\frac{1}{q}}\left(\sum_{s=0}^{S-1}(h(s))^{aq}\right)^{\frac{1}{q}}\right)}{\sum_{s=0}^{S-1}h(s)}}_{T_5},\tag{297}$$

where $a \in (0,1)$ and $q > 1$.

For $h(s) = \frac{p}{(s+1)^\omega}$ with $p \in (0, \frac{1}{2L}]$, we now want to optimize $\omega, a, q$ such that the upper bound in (295) is minimized for any given $S$. Observe that in the first two terms on the right-hand side of (295), we require the partial sum $\sum_{s=0}^{S-1} h(s)$ to diverge and $\sum_{s=0}^{S-1}(h(s))^2$ to converge. But that is only possible for $\omega \in (\frac{1}{2}, 1]$. We also require the numerator of $T_1$ to converge as $S \to \infty$. From the upper bound (297) on term $T_1$, the numerator of term $T_4$ given by $\sqrt{\left(\sum_{s=0}^{S-1}(a_1)^{2s}\right)}\sqrt{\left(\sum_{s=0}^{S-1}(h(s))^2\right)}$ will converge as $S \to \infty$ for any $\omega \in (\frac{1}{2}, 1]$. Next, we simplify the numerator term in $T_5$. Taking the first numerator term $\sum_{s=0}^{S-1}\left((h(s))^{1-a}\sum_{l=0}^{s-1}(a_1)^{s-l-1}h(l)\right)^{\frac{q}{q-1}}$ in $T_5$, using the bound (279) for any fixed large enough $S' \ll S$ and any large enough $S$ we get that:

$$\sum_{s=0}^{S-1}\left((h(s))^{1-a}\sum_{l=0}^{s-1}(a_1)^{s-l-1}h(l)\right)^{\frac{q}{q-1}} \leqslant \underbrace{C(S')}_{\text{constant}} + \underbrace{\sum_{s=S'}^{S-1}\left(\frac{p^{(1-a)}}{s^{\omega(1-a)}}\frac{C_5}{s^\omega}\right)^{\frac{q}{q-1}}}_{\text{tail sum}} \leqslant C_7\sum_{s=S'}^{S-1}\left(\frac{1}{s^{2\omega - a\omega}}\right)^{\frac{q}{q-1}}\tag{298}$$

and hence the partial sum $\sum_{s=0}^{S-1}\left((h(s))^{1-a}\sum_{l=0}^{s-1}(a_1)^{s-l-1}h(l)\right)^{\frac{q}{q-1}}$ converges if $(2\omega - a\omega)\frac{q}{q-1} > 1$ or equivalently

$$aq < \frac{1}{\omega}(2q\omega - q + 1).\tag{299}$$

Also, from (297) the partial sum $\sum_{s=0}^{S-1}(h(s))^{aq}$ of $T_5$ converges if $aq\omega > 1$. Hence, we require the following:

$$\frac{1}{\omega} < aq < \underbrace{\frac{1}{\omega}(2q\omega - q + 1)}_{>\frac{1}{\omega} \text{ for } \omega > \frac{1}{2}},\tag{300}$$

which can be satisfied for any fixed $q > 1$ and a fixed $a \in (0,1)$ that depends on $q$ provided $\omega > \frac{1}{2}$. Hence we get that for any $\omega \in (\frac{1}{2}, 1)$ we can always find some $a, q$ such that the numerator terms of $T_4, T_5$ converge and thus can be uniformly bounded for any $S$. Since $\sum_{s=0}^{S-1}h(s)$ is maximized as $\omega \downarrow \frac{1}{2}$, from (297) we get for $\omega = \frac{1}{2} + \epsilon$ with $0 < \epsilon < 1/2$ that:

$$T_1 \leqslant \frac{dC_4\text{diam}(\mathcal{K})}{S^{\frac{1}{2}-\epsilon}},\tag{301}$$

for some constant[19] $C_4 = \mathcal{O}\left(M^2(1+p)\left(Ld\,\text{diam}(\mathcal{K})\right)^3\right)$ and thus from (295) we get

$$\min_{0\leqslant s\leqslant S-1}\|\nabla f(\hat{\mathbf{w}}^s(s))\|^2 \leqslant \frac{\left(f(\hat{\mathbf{w}}^0(0)) - \inf_{\mathbf{w}} f(\mathbf{w})\right)}{pS^{\frac{1}{2}-\epsilon}} + \frac{C_6}{S^{\frac{1}{2}-\epsilon}}$$
$$+ 2Ld\text{iam}(\mathcal{K})C_0 + \frac{2C_4L^2d\sqrt{Md}(\text{diam}(\mathcal{K}))^2}{S^{\frac{1}{2}-\epsilon}},\tag{302}$$

---

[19]From (295) and (288) we have $C_4 = \mathcal{O}\left(dC_3\text{diam}(\mathcal{K}) + dpC_2\text{diam}(\mathcal{K})\right) = \mathcal{O}\left(M^2(1+p)\left(Ld\,\text{diam}(\mathcal{K})\right)^3\right)$.

$$\implies \limsup_{S \to \infty} \min_{0 \leqslant s \leqslant S-1} \|\nabla f(\hat{\mathbf{w}}^s(s))\|^2 \leqslant 2L\mathrm{diam}(\mathcal{K})C_0 \tag{303}$$

for some constant $C_6 = \mathcal{O}\left(pL^3\left(Md\,\mathrm{diam}(\mathcal{K})\right)^2\right)$. Note that in the first two terms of (302), we used the fact that $f(\hat{\mathbf{w}}^S(S)) \geqslant \inf_{\mathbf{w}} f(\mathbf{w}) > -\infty$ by Assumption 4.8 and the constant $C_6 = \mathcal{O}(pC_2)$ from (295), which completes the proof. ∎

### F.5 Proof of Theorem 6.7

*Proof.* Using (269) from Theorem 6.6's proof for any $0 \leqslant S' \leqslant S$, by substituting $h(s) = \frac{1}{\sqrt{S}}$ for all $0 \leqslant s \leqslant S-1$, we get that:

$$\xi_k^1(S') \leqslant (a_1)^{S'} \xi_k^1(0) + C(M,d)\mathrm{diam}(\mathcal{K}) \sum_{s=0}^{S'-1} (a_1)^{S'-s-1} h(s) \tag{304}$$

$$\implies \xi_k^1(S') \leqslant (a_1)^{S'} \xi_k^1(0) + C(M,d)\mathrm{diam}(\mathcal{K}) \frac{1}{\sqrt{S}(1-a_1)}, \tag{305}$$

where $a_1 = M^{\frac{3}{2}}(\sqrt{M}+1)(1-\beta^{\tau M})^{\left\lfloor \frac{(J-2)}{\tau M} \right\rfloor} < 1$ and $C(M,d) = \mathcal{O}(M\sqrt{d})$. Similarly, using the bound (175) from Lemma 4.6 and Lemma 4.9 we get that

$$\xi_k^5(S') \leqslant (a_3)^{S'} \xi_k^5(0) + C(M,d)\mathrm{diam}(\mathcal{K}) \frac{1}{\sqrt{S}(1-a_3)}, \tag{306}$$

where $a_3 = M^{\frac{3}{2}}(1-\beta^{\tau M})^{\left\lfloor \frac{(J-2)}{\tau M} \right\rfloor} < 1$. This completes the first part of the proof.

For the second part, from (287), for $h(s) = \frac{1}{\sqrt{S}}$, recall that

$$f(\hat{\mathbf{w}}^s(s)) - f(\hat{\mathbf{w}}^{s+1}(s+1)) \geqslant \frac{1}{\sqrt{S}}\left(1 - \frac{L}{\sqrt{S}}\right) \|\nabla f(\hat{\mathbf{w}}^s(s))\|^2 - L\mathrm{diam}(\mathcal{K})\frac{1}{\sqrt{S}}\left(C_0 + L\sqrt{Md}\sum_{k=1}^{d} \xi_k^1(s)\right)$$
$$- L\left(\frac{1}{\sqrt{S}}\right)^2 \left(C_0 + L\sqrt{Md}\sum_{k=1}^{d} \xi_k^1(s)\right)^2, \tag{307}$$

and for some constant $C_2$ as a function of $L, M, d, \mathrm{diam}(\mathcal{K})$ expressed as $C_2 = C(L, M, d, \mathrm{diam}(\mathcal{K}))$, using Assumption 4.12 and (288) we have the bound

$$\sup_{s \geqslant 0} L\left(C_0 + L\sqrt{Md}\sum_{k=1}^{d} \xi_k^1(s)\right)^2 \leqslant C(L, M, d, \mathrm{diam}(\mathcal{K})) = C_2 = \mathcal{O}\left(L^3\left(Md\,\mathrm{diam}(\mathcal{K})\right)^2\right).$$

Then summing (307) from $s = 0$ to $S-1$, dividing both sides by $\sqrt{S}$ and using the above bound followed by (305) we get:

$$f(\hat{\mathbf{w}}^0(0)) - f(\hat{\mathbf{w}}^S(S)) \geqslant \frac{1}{\sqrt{S}}\left(1 - \frac{L}{\sqrt{S}}\right) \sum_{s=0}^{S-1} \|\nabla f(\hat{\mathbf{w}}^s(s))\|^2$$
$$- L\mathrm{diam}(\mathcal{K})\frac{1}{\sqrt{S}} \sum_{s=0}^{S-1}\left(C_0 + L\sqrt{Md}\sum_{k=1}^{d} \xi_k^1(s)\right)$$
$$- L\left(\frac{1}{\sqrt{S}}\right)^2 \sum_{s=0}^{S-1}\left(C_0 + L\sqrt{Md}\sum_{k=1}^{d} \xi_k^1(s)\right)^2 \tag{308}$$

$$\implies \frac{f(\hat{\mathbf{w}}^0(0)) - f(\hat{\mathbf{w}}^S(S))}{\sqrt{S}} \geqslant \frac{1}{S}\left(1 - \frac{L}{\sqrt{S}}\right) \sum_{s=0}^{S-1} \|\nabla f(\hat{\mathbf{w}}^s(s))\|^2$$

$$- L\mathrm{diam}(\mathcal{K})\frac{1}{S}\sum_{s=0}^{S-1}\left(C_0 + L\sqrt{Md}\sum_{k=1}^{d}\xi_k^1(s)\right) - \frac{1}{\sqrt{S}}\left(\frac{1}{\sqrt{S}}\right)^2 SC_2 \tag{309}$$

$$\implies \frac{1}{S}\left(1 - \frac{L}{\sqrt{S}}\right)\sum_{s=0}^{S-1}\|\nabla f(\hat{\mathbf{w}}^s(s))\|^2 \leqslant \frac{f(\hat{\mathbf{w}}^0(0)) - f(\hat{\mathbf{w}}^S(S))}{\sqrt{S}} + L\,\mathrm{diam}(\mathcal{K})C_0 + \frac{C_2}{\sqrt{S}}$$

$$+ L^2\,\mathrm{diam}(\mathcal{K})\sqrt{Md}\frac{d}{S}\sum_{s=0}^{S-1}\left((a_1)^s\xi_k^1(0) + C(M,d)\mathrm{diam}(\mathcal{K})\frac{1}{\sqrt{S}(1-a_1)}\right) \tag{310}$$

$$\implies \frac{1}{S}\left(1 - \frac{L}{\sqrt{S}}\right)\sum_{s=0}^{S-1}\|\nabla f(\hat{\mathbf{w}}^s(s))\|^2 \leqslant \frac{f(\hat{\mathbf{w}}^0(0)) - f(\hat{\mathbf{w}}^S(S))}{\sqrt{S}} + L\,\mathrm{diam}(\mathcal{K})C_0 + \frac{C_2}{\sqrt{S}}$$

$$+ L^2\,\mathrm{diam}(\mathcal{K})\sqrt{Md}\frac{d}{S(1-a_1)}\xi_k^1(0) + (L\,\mathrm{diam}(\mathcal{K}))^2\sqrt{Md}\frac{C(M,d)d}{\sqrt{S}(1-a_1)} \tag{311}$$

$$\implies \frac{1}{S}\sum_{s=0}^{S-1}\|\nabla f(\hat{\mathbf{w}}^s(s))\|^2 \leqslant \left(1 - \frac{L}{\sqrt{S}}\right)^{-1}\frac{f(\hat{\mathbf{w}}^0(0)) - \inf_{\mathbf{w}} f(\mathbf{w})}{\sqrt{S}} + \frac{C_9}{\sqrt{S}} + \left(1 - \frac{L}{\sqrt{S}}\right)^{-1}L\,\mathrm{diam}(\mathcal{K})C_0, \tag{312}$$

where $C_9 = \mathcal{O}(C_2) = \mathcal{O}\left(L^3\left(Md\,\mathrm{diam}(\mathcal{K})\right)^2\right)$ is a constant that depends on $L, M, d, \mathrm{diam}(\mathcal{K})$ and we used the fact that $f(\hat{\mathbf{w}}^S(S)) \geqslant \inf_{\mathbf{w}} f(\mathbf{w}) > -\infty$ from Assumption 4.8. Finally, $S > L^6(Md\,\mathrm{diam}(\mathcal{K}))^4$ so that $\frac{C_9}{\sqrt{S}} < 1$ for any large $S$. This completes the proof. ∎

## G   Proofs for Statistical Learning Rates and Sample Complexity

Note that by data homogeneity (i.e., $\mathbf{z}_{jn} \overset{\text{i.i.d.}}{\sim} \mathbb{P}$ across all nodes $j$ and samples $n$) and linearity of expectation, for any fixed deterministic model $\mathbf{w} \in \mathbb{R}^d$ and any fixed weighting vector $\mathbf{q} \in \mathbb{R}^M$ with $\sum_{j=1}^{M} q_j = 1$, we have for every coordinate $k$:

$$\mathbb{E}\left[\frac{1}{MN}\sum_{j=1}^{M}\sum_{n=1}^{N}\nabla_k\ell(\mathbf{w};\mathbf{z}_{jn})\right] = \mathbb{E}\left[\frac{1}{N}\sum_{n=1}^{N}\sum_{j=1}^{M}q_j\,\nabla_k\ell(\mathbf{w};\mathbf{z}_{jn})\right] = \nabla_k\mathcal{R}(\mathbf{w}), \tag{313}$$

where the first equality uses $\sum_{j=1}^{M} q_j = 1$ together with the fact that the distributions of $\{\mathbf{z}_{jn}\}_{n=1}^{N}$ are identical across $j$, and the second equality follows from the identity $\nabla\mathcal{R}(\mathbf{w}) = \mathbb{E}[\nabla\ell(\mathbf{w};\mathbf{z})]$ established in Sec. 8. Because the algorithmic iterates $\hat{\mathbf{w}}^s(s)$ and consensus weights $\mathbf{c}_k(s+1)$ depend on the random data samples, the proofs in the sequel will leverage this deterministic identity by establishing uniform convergence bounds over a compact set.

The proofs in this section will be divided into three parts: the first part includes the proof of the sample complexity of the parameter $C_0$ defined in Theorem 5.5; the second part includes the proof of the sample complexity of the parameter $\Delta$ defined in Lemma 5.4 along with the proof of Theorem 8.2; the last part includes the proof of Theorem 8.3. Finally, we provide a supplementary discussion giving a restricted non-vacuity construction for Assumption 8.1.

### G.1 $C_0$ sample complexity

**Lemma G.1.** *Under Assumptions 3.3, 4.8, and 8.1 with $N$ i.i.d. samples at each node, for any $\epsilon' \in (0,1)$, and for any large enough $N \gg \left(\frac{d}{\epsilon'}\right)^2$ with $d > \epsilon'$, we have*

$$C_0 < \mathcal{O}\left(\sqrt{\frac{L'^2 d^2 \|\boldsymbol{\alpha}\|^2 \log\frac{4}{\delta}}{N}}\right) \tag{314}$$

*with probability at least $1 - \delta$, where*

$$\delta = 2\exp\left(-\frac{4MN(\epsilon')^2}{16(L')^2 Md^2\|\boldsymbol{\alpha}\|^2 + (\epsilon')^2} + M\log\left(\frac{12L'd\sqrt{M}}{\epsilon'}\right) + d\log\left(\frac{12L'\Gamma_0 d}{\epsilon'}\right)\right)$$
$$+ 2d\exp\left(-\frac{(\epsilon')^2 MN}{4(L'd)^2}\right), \tag{315}$$

*and $\boldsymbol{\alpha}$ denotes the effective mixing weight vector defined in Theorem 8.2.*

*Proof.* The gradient samples $\{\nabla\ell(\mathbf{w}; \mathbf{z}_{jn})\}_{n=1}^N$ at each node $j$ for any given $\mathbf{w}$ are i.i.d. since $\{\mathbf{z}_{jn}\}_{n=1}^N$ are i.i.d.; consequently, $\{[\nabla\ell(\mathbf{w}; \mathbf{z}_{jn})]_k\}_{n=1}^N$ are i.i.d. for any coordinate $k$. Since $\hat{\mathbf{w}}^s(s) \in \mathcal{K}$ for all $s$ by Assumption 8.1, it suffices to bound $\sup_{\mathbf{w}\in\mathcal{K}}|\nabla_k f(\mathbf{w}) - \nabla_k\mathcal{R}(\mathbf{w})|$. Moreover, assuming without loss of generality that the origin $\mathbf{0} \in \mathcal{K}$, Assumption 8.1 implies that there exist constants $L' > 0$ and $\Gamma_0 := \text{diam}(\mathcal{K})$ such that

$$\max\left\{\sup_{\mathbf{w}\in\mathcal{K}}|\nabla_k\ell(\mathbf{w}; \mathbf{z}_{jn})|, \sup_{\mathbf{w}\in\mathcal{K}}|\ell(\mathbf{w}; \mathbf{z}_{jn})|\right\} \leqslant \frac{L'}{2}, \qquad \sup_{\mathbf{w}\in\mathcal{K}}\|\mathbf{w}\| \leqslant \Gamma_0, \tag{316}$$

for all $k \in \{1, \dots, d\}$, all $j \in \{1, \dots, M\}$, all sample collections $\{\mathbf{z}_{jn}\}_{n=1}^N \overset{\text{i.i.d.}}{\sim} \mathbb{P}$, and all $N \geqslant 1$. In particular, one may take $L' = \max\{\mathcal{O}(Ld\,\text{diam}(\mathcal{K})), \mathcal{O}(L(\text{diam}(\mathcal{K}))^2)\}$ by applying the fundamental theorem of calculus to $\ell(\cdot; \mathbf{z})$ as a function of $\mathbf{w}$.

Next, apply a union bound over coordinates, together with a covering argument for $\mathcal{K}$. Let $\{\mathbf{w}_\ell\}_{\ell=1}^{m_\nu}$ be a $\nu$-net of $\mathcal{K}$ with covering number $m_\nu$. Using the triangle inequality for $\|\mathbf{w} - \mathbf{w}_\ell\| \leqslant \nu$ and $L$-smoothness,

$$\left|\frac{1}{MN}\sum_{j=1}^M\sum_{n=1}^N \nabla_k\ell(\mathbf{w}_\ell; \mathbf{z}_{jn}) - \nabla_k\mathcal{R}(\mathbf{w}_\ell)\right| + 2L\nu \geqslant \sup_{\|\mathbf{w}-\mathbf{w}_\ell\|\leqslant\nu}\left|\frac{1}{MN}\sum_{j=1}^M\sum_{n=1}^N \nabla_k\ell(\mathbf{w}; \mathbf{z}_{jn}) - \nabla_k\mathcal{R}(\mathbf{w})\right|. \tag{317}$$

Then, for any $\epsilon_0 \in (0,1)$, Hoeffding's inequality (Hoeffding, 1963) yields

$$\mathbb{P}\left(\sum_{k=1}^d \sup_{\mathbf{w}\in\mathcal{K}}\left|\frac{1}{MN}\sum_{j=1}^M\sum_{n=1}^N \nabla_k\ell(\mathbf{w}; \mathbf{z}_{jn}) - \nabla_k\mathcal{R}(\mathbf{w})\right| \geqslant \epsilon_0\right)$$

$$\leqslant \sum_{k=1}^d\sum_{\ell=1}^{m_\nu}\mathbb{P}\left(\left|\frac{1}{MN}\sum_{j=1}^M\sum_{n=1}^N \nabla_k\ell(\mathbf{w}_\ell; \mathbf{z}_{jn}) - \nabla_k\mathcal{R}(\mathbf{w}_\ell)\right| + 2L\nu \geqslant \frac{\epsilon_0}{d}\right)$$

$$\leqslant \sum_{k=1}^d\sum_{\ell=1}^{m_\nu}\mathbb{P}\left(\left|\frac{1}{MN}\sum_{j=1}^M\sum_{n=1}^N \nabla_k\ell(\mathbf{w}_\ell; \mathbf{z}_{jn}) - \nabla_k\mathcal{R}(\mathbf{w}_\ell)\right| \geqslant \frac{\epsilon_0}{2d}\right) \leqslant 2m_\nu\sum_{k=1}^d\exp\left(-\frac{2\epsilon_0^2 MN}{4(L'd)^2}\right), \tag{318}$$

where in the second-to-last inequality we set $\nu = \frac{\epsilon_0}{4Ld}$. Using the covering number bound $m_\nu \leqslant \left(\frac{3\Gamma_0\sqrt{d}}{\nu}\right)^d$ (see Fang et al. (2022), supplementary material) gives

$$\mathbb{P}\left(\sum_{k=1}^d \sup_{\mathbf{w}\in\mathcal{K}}\left|\frac{1}{MN}\sum_{j=1}^M\sum_{n=1}^N \nabla_k\ell(\mathbf{w}; \mathbf{z}_{jn}) - \nabla_k\mathcal{R}(\mathbf{w})\right| < \epsilon_0\right) > 1 - \delta_0, \tag{319}$$

where

$$\delta_0 := 2d\exp\left(-\frac{2\epsilon_0^2 MN}{4(L'd)^2} + d\log\left(\frac{12L\Gamma_0 d\sqrt{d}}{\epsilon_0}\right)\right). \tag{320}$$

Hence, with probability at least $1 - \delta_0$,

$$\sup_{s \geqslant 0} \sum_{k=1}^{d} \left| \nabla_k f(\hat{\mathbf{w}}^s(s)) - \nabla_k \mathcal{R}(\hat{\mathbf{w}}^s(s)) \right| < \epsilon_0 < 2L'd\sqrt{\frac{\log(\frac{2d}{\delta_0})}{MN}}, \tag{321}$$

where the last step uses $\log(\frac{2d}{\delta_0}) = \frac{2\epsilon_0^2 MN}{4(L'd)^2} - d\log(\frac{12L\Gamma_0 d\sqrt{d}}{\epsilon_0}) > \frac{2\epsilon_0^2 MN}{8(L'd)^2}$ for $N \gg \frac{d^2}{\epsilon_0^2}$.

Next, let $\mathcal{S}_{\mathbf{c}} = \{\mathbf{c}_k(s)\}_{s,k=1}^{\infty,d}$ and let $\boldsymbol{\alpha} \in \arg\max_{\mathbf{q} \in \mathcal{S}_{\mathbf{c}}} \|\mathbf{q}\|$.[20] Define

$$T_5(s) := \sum_{k=1}^{d} \left| \frac{1}{N} \sum_{j=1}^{M} \sum_{n=1}^{N} [\mathbf{c}_k(s+1)]_j \nabla_k \ell(\hat{\mathbf{w}}^s(s); \mathbf{z}_{jn}) - \nabla_k \mathcal{R}(\hat{\mathbf{w}}^s(s)) \right|,$$

$$T_6(s) := \sqrt{\sum_{k=1}^{d} \left| \frac{1}{N} \sum_{j=1}^{M} \sum_{n=1}^{N} [\mathbf{c}_k(s+1)]_j \nabla_k \ell(\hat{\mathbf{w}}^s(s); \mathbf{z}_{jn}) - \nabla_k \mathcal{R}(\hat{\mathbf{w}}^s(s)) \right|^2}.$$

The remainder follows from (S.17)–(S.18) in Fang et al. (2022) (supplementary material). In particular, for any $\epsilon_1 \in (0,1)$,

$$\mathbb{P}\left( \sup_s T_5(s) \geqslant \epsilon_1 \right) \leqslant 2\exp\left( -\frac{4MN\epsilon_1^2}{16(L')^2 M d^2 \|\boldsymbol{\alpha}\|^2 + \epsilon_1^2} + M\log\left(\frac{12L'd\sqrt{M}}{\epsilon_1}\right) + d\log\left(\frac{12L'\Gamma_0 d}{\epsilon_1}\right) \right). \tag{322}$$

Equivalently, with probability at least $1 - \delta_1$,

$$\sup_s T_5(s) < \mathcal{O}\left( \sqrt{\frac{L'^2 d^2 \|\boldsymbol{\alpha}\|^2 \log\frac{2}{\delta_1}}{N}} \right), \tag{323}$$

where $\delta_1$ equals the right-hand side above.

Using a union bound over (321) and (323), with probability at least $1 - (\delta_0 + \delta_1)$ we obtain

$$C_0 := \sup_{s \geqslant 0} \sum_{k=1}^{d} \left| \frac{1}{N} \sum_{j=1}^{M} \sum_{n=1}^{N} [\mathbf{c}_k(s+1)]_j \nabla_k \ell(\hat{\mathbf{w}}^s(s); \mathbf{z}_{jn}) - \nabla_k f(\hat{\mathbf{w}}^s(s)) \right|$$

$$< 2\max\left\{ \sqrt{\log\left(\tfrac{2d}{\delta_0}\right)} \frac{2L'd}{\sqrt{MN}}, \ \mathcal{O}\left( \sqrt{\frac{L'^2 d^2 \|\boldsymbol{\alpha}\|^2 \log\frac{2}{\delta_1}}{N}} \right) \right\}. \tag{324}$$

Finally, set $\epsilon_0 = \epsilon_1 = \epsilon'$. Since $N \gg \left(\frac{d}{\epsilon'}\right)^2$, the linear term in $MN$ dominates the covering term in $\delta_0$, and hence

$$\delta_0 = 2d\exp\left( -\frac{2(\epsilon')^2 MN}{4(L'd)^2} + d\log\left(\frac{12L\Gamma_0 d\sqrt{d}}{\epsilon'}\right) \right) \leqslant 2d\exp\left( -\frac{(\epsilon')^2 MN}{4(L'd)^2} \right).$$

Define

$$\delta := \delta_1 + 2d\exp\left( -\frac{(\epsilon')^2 MN}{4(L'd)^2} \right),$$

where $\delta_1$ is given by the right-hand side of the bound in (323). Then the union bound yields the claimed estimate

$$C_0 < \mathcal{O}\left( \sqrt{\frac{L'^2 d^2 \|\boldsymbol{\alpha}\|^2 \log\frac{4}{\delta}}{N}} \right)$$

with probability at least $1 - \delta$, completing the proof. ∎

[20] Although $\boldsymbol{\alpha}$ depends on the i.i.d. sample draw $\{\mathbf{z}_{jn}\}_{j,n}$ and the adversary's specific actions, and is therefore a random variable, this does not affect the bound. Indeed, $\boldsymbol{\alpha}$ is a probability vector (its entries are nonnegative and sum to one), which implies $\|\boldsymbol{\alpha}\|^{-2} \in [1, M]$ and hence $\boldsymbol{\alpha}$ is uniformly bounded independently of $N$. Moreover, this bound can be decoupled from both the data and the adversary. While the data and adversarial strategy determine the *specific sequence* of mixing matrices used over time, the CWTM algorithm guarantees that every selected matrix belongs to the finite, deterministic set of filtered graph topologies $\mathcal{T}_{\mathcal{F}}$ (cf. Definition 3.2). Taking the supremum of the norm over the closed set of consensus vectors generated by arbitrary sequences from $\mathcal{T}_{\mathcal{F}}$ yields a deterministic worst-case structural constant. Substituting this constant for $\|\boldsymbol{\alpha}\|^2$ gives a rigorous, data-independent sample complexity bound that naturally interpolates between the fully centralized rate $\mathcal{O}(1/\sqrt{MN})$ and the purely local rate $\mathcal{O}(1/\sqrt{N})$.

### G.2 Proof of Theorem 8.2

*Proof.* To obtain statistical convergence rates for RESIST in the strongly convex setting, we bound the residual term in (54) from Theorem 5.8. We split the residual into $C_0$- and $\Delta$-dependent components so that their sample-complexity bounds can be invoked separately. Recall that

$$C_0 := \sup_{s \geqslant 0} \sum_{k=1}^{d} \left| \nabla_k f(\hat{\mathbf{w}}^s(s)) - \nabla_k f^{k,s+1}(\hat{\mathbf{w}}^s(s)) \right|, \qquad \Delta := \sum_{j=1}^{M} \|\mathbf{w}^* - \mathbf{w}_j^*\|,$$

where $C_0 < \infty$. The sample complexity of $C_0$ is established in Lemma G.1; it remains to bound $\Delta$.

**Sample complexity for $\Delta$.**

Recall that

$$\Delta = \sum_{j=1}^{M} \|\mathbf{w}^* - \mathbf{w}_j^*\| \leqslant \sum_{j=1}^{M} \left( \|\mathbf{w}^* - \mathbf{w}_{\mathrm{SR}}^*\| + \|\mathbf{w}_{\mathrm{SR}}^* - \mathbf{w}_j^*\| \right). \tag{325}$$

By $\mu$-strong convexity of $f$ and each $f_j$ and using (91), we have

$$\mu \|\mathbf{w}^* - \mathbf{w}_{\mathrm{SR}}^*\| \leqslant \|\nabla f(\mathbf{w}^*) - \nabla f(\mathbf{w}_{\mathrm{SR}}^*)\| = \|\nabla f(\mathbf{w}_{\mathrm{SR}}^*)\| = \left\| \nabla f(\mathbf{w}_{\mathrm{SR}}^*) - \nabla \mathcal{R}(\mathbf{w}_{\mathrm{SR}}^*) \right\|, \tag{326}$$

$$\mu \|\mathbf{w}_{\mathrm{SR}}^* - \mathbf{w}_j^*\| \leqslant \|\nabla f_j(\mathbf{w}_{\mathrm{SR}}^*) - \nabla f_j(\mathbf{w}_j^*)\| = \|\nabla f_j(\mathbf{w}_{\mathrm{SR}}^*)\| = \left\| \nabla f_j(\mathbf{w}_{\mathrm{SR}}^*) - \nabla \mathcal{R}(\mathbf{w}_{\mathrm{SR}}^*) \right\|. \tag{327}$$

Using $\nabla f(\mathbf{w}) = \frac{1}{MN} \sum_{j=1}^{M} \sum_{n=1}^{N} \nabla \ell(\mathbf{w}; \mathbf{z}_{jn})$ and $\nabla f_j(\mathbf{w}) = \frac{1}{N} \sum_{n=1}^{N} \nabla \ell(\mathbf{w}; \mathbf{z}_{jn})$, and applying Jensen's inequality together with the bound $\|\mathbf{v}\| \leqslant \sum_{k=1}^{d} |v_k|$, followed by a union bound over $k$ and Hoeffding's inequality (Hoeffding, 1963), we obtain for any $\epsilon_2 \in (0,1)$:

$$\mathbb{P}\left( \left\| \frac{1}{MN} \sum_{j=1}^{M} \sum_{n=1}^{N} \nabla \ell(\mathbf{w}_{\mathrm{SR}}^*; \mathbf{z}_{jn}) - \nabla \mathcal{R}(\mathbf{w}_{\mathrm{SR}}^*) \right\| \geqslant \epsilon_2 \right) \leqslant \sum_{k=1}^{d} \mathbb{P}\left( \left| \frac{1}{MN} \sum_{j=1}^{M} \sum_{n=1}^{N} \nabla_k \ell(\mathbf{w}_{\mathrm{SR}}^*; \mathbf{z}_{jn}) - \nabla_k \mathcal{R}(\mathbf{w}_{\mathrm{SR}}^*) \right| \geqslant \frac{\epsilon_2}{d} \right)$$

$$\leqslant 2d \exp\left( -\frac{2\epsilon_2^2 MN}{(L'd)^2} \right). \tag{328}$$

Equivalently, with probability at least $1 - \delta_2$,

$$\left\| \frac{1}{MN} \sum_{j=1}^{M} \sum_{n=1}^{N} \nabla \ell(\mathbf{w}_{\mathrm{SR}}^*; \mathbf{z}_{jn}) - \nabla \mathcal{R}(\mathbf{w}_{\mathrm{SR}}^*) \right\| < \sqrt{\log\left(\frac{2d}{\delta_2}\right)} \frac{L'd}{\sqrt{2MN}}, \quad \delta_2 := 2d \exp\left( -\frac{2\epsilon_2^2 MN}{(L'd)^2} \right). \tag{329}$$

Similarly, for any fixed node $j$ and any $\epsilon_3 \in (0,1)$,

$$\mathbb{P}\left( \left\| \frac{1}{N} \sum_{n=1}^{N} \nabla \ell(\mathbf{w}_{\mathrm{SR}}^*; \mathbf{z}_{jn}) - \nabla \mathcal{R}(\mathbf{w}_{\mathrm{SR}}^*) \right\| \geqslant \epsilon_3 \right) \leqslant 2d \exp\left( -\frac{2\epsilon_3^2 N}{(L'd)^2} \right), \tag{330}$$

so with probability at least $1 - \delta_3$,

$$\left\| \frac{1}{N} \sum_{n=1}^{N} \nabla \ell(\mathbf{w}_{\mathrm{SR}}^*; \mathbf{z}_{jn}) - \nabla \mathcal{R}(\mathbf{w}_{\mathrm{SR}}^*) \right\| < \sqrt{\log\left(\frac{2d}{\delta_3}\right)} \frac{L'd}{\sqrt{2N}}, \quad \delta_3 := 2d \exp\left( -\frac{2\epsilon_3^2 N}{(L'd)^2} \right). \tag{331}$$

Applying a union bound to (329) and (331), and combining with (325)–(327), we obtain that with probability at least $1 - (\delta_2 + \delta_3)$,

$$\Delta < \frac{2M}{\mu} \max\left\{ \sqrt{\log\left(\frac{2d}{\delta_2}\right)} \frac{L'd}{\sqrt{2MN}}, \ \sqrt{\log\left(\frac{2d}{\delta_3}\right)} \frac{L'd}{\sqrt{2N}} \right\}. \tag{332}$$

Finally, set $\epsilon_2 = \epsilon_3 = \epsilon'$ and define

$$\delta := \underbrace{2d \exp\left(-\frac{2(\epsilon')^2 MN}{(L'd)^2}\right)}_{=\delta_2} + \underbrace{2d \exp\left(-\frac{2(\epsilon')^2 N}{(L'd)^2}\right)}_{=\delta_3}.$$

Then $\delta_2 < \delta_3$ and hence $\delta < 2\delta_3$, yielding for $N$ large enough:

$$\Delta < \frac{2M}{\mu}\sqrt{\log\left(\frac{2d}{\delta_3}\right)}\frac{L'd}{\sqrt{2N}} < \frac{2M}{\mu}\sqrt{\log\left(\frac{4d}{\delta}\right)}\frac{L'd}{\sqrt{2N}}, \tag{333}$$

with probability at least $1 - \delta$.

Substituting (333) into Corollary 5.6 gives, with probability at least $1 - \delta$,

$$\limsup_{s\to\infty} \xi_k^1(s) \leqslant \mathcal{O}\big(hM \operatorname{diam}(\mathcal{K})\big) + \mathcal{O}\left(\frac{2Mh}{\mu}\sqrt{\log\left(\frac{4d}{\delta}\right)}\frac{L'd}{\sqrt{2N}}\right), \tag{334}$$

$$\limsup_{s\to\infty} \xi_k^5(s) \leqslant \mathcal{O}\big(hM \operatorname{diam}(\mathcal{K})\big) + \mathcal{O}\left(\frac{2Mh}{\mu}\sqrt{\log\left(\frac{4d}{\delta}\right)}\frac{L'd}{\sqrt{2N}}\right), \tag{335}$$

where $\delta$ is as defined above.

Next, recalling the asymptotics of $\xi_{\mathbf{w}^*}^6(s)$ from Corollary 5.6, using $\mathbf{w}_{\text{ERM}}^* = \mathbf{w}^*$ and the triangle inequality, we have that the averaged iterate error satisfies (with probability at least $1 - \delta$):

$$\limsup_{s\to\infty} \|\mathbf{w}_{\text{SR}}^* - \widehat{\mathbf{w}}^s(s)\| \leqslant \frac{C_0}{\mu} + \frac{L\sqrt{Md}}{\mu}\left(\frac{h}{1-a_1}\left(a_2\sqrt{M}(\sqrt{M}+1)C_1 \operatorname{diam}(\mathcal{K}) + a_2\Delta\right)\right) + \|\mathbf{w}_{\text{SR}}^* - \mathbf{w}_{\text{ERM}}^*\|. \tag{336}$$

**Combining concentration bounds.** Choose a common $\epsilon'$ across the three probability bounds for $C_0$ (Lemma G.1), (329), and (333). Denote their corresponding failure probabilities by $\delta_0$, $\delta_1$, and $\delta_2$, respectively, i.e.,

$$\delta_0 = 2\exp\left(-\frac{4MN(\epsilon')^2}{16(L')^2 Md^2\|\boldsymbol{\alpha}\|^2 + (\epsilon')^2} + M\log\left(\frac{12L'd\sqrt{M}}{\epsilon'}\right) + d\log\left(\frac{12L'\Gamma_0 d}{\epsilon'}\right)\right) + 2d\exp\left(-\frac{(\epsilon')^2 MN}{4(L'd)^2}\right),$$

$$\delta_1 = 2d\exp\left(-\frac{2(\epsilon')^2 MN}{(L'd)^2}\right) + 2d\exp\left(-\frac{2(\epsilon')^2 N}{(L'd)^2}\right),$$

$$\delta_2 = 2d\exp\left(-\frac{2(\epsilon')^2 MN}{(L'd)^2}\right).$$

For $N$ sufficiently large (and $d > \epsilon'$), we have $\delta_2 < \delta_1 < \delta_0$. Applying a union bound over the three events corresponding to (314), (329), and (333), and defining $\delta := \delta_0 + \delta_1 + \delta_2 < 3\delta_0$, we obtain that, under the stepsize condition $h < \frac{1}{M^2\sqrt{d}}$,

$$\frac{C_0}{\mu} + \frac{hLa_2\sqrt{Md}}{\mu(1-a_1)}\Delta + \|\mathbf{w}_{\text{SR}}^* - \mathbf{w}_{\text{ERM}}^*\| \leqslant 3\max\left\{\mathcal{O}\left(\frac{1}{\mu}\sqrt{\frac{L'^2 d^2\|\boldsymbol{\alpha}\|^2\log\frac{4}{\delta_0}}{N}}\right), \frac{2MhLa_2\sqrt{Md}}{\mu^2(1-a_1)}\sqrt{\log\left(\frac{4d}{\delta_1}\right)}\frac{L'd}{\sqrt{2N}}, \right.$$
$$\left. \frac{1}{\mu}\sqrt{\log\left(\frac{2d}{\delta_2}\right)}\frac{L'd}{\sqrt{2MN}}\right\} = \mathcal{O}\left(\frac{6}{\mu}\sqrt{\frac{L'^2 d^2\|\boldsymbol{\alpha}\|^2\log\frac{12}{\delta}}{N}}\right), \tag{337}$$

with probability at least $1 - \delta$, where the last equality uses $h < \frac{1}{M^2\sqrt{d}}$ so that the second term is absorbed into the leading statistical term. Consequently,

$$\limsup_{s\to\infty} \|\mathbf{w}_{\text{SR}}^* - \widehat{\mathbf{w}}^s(s)\| \leqslant \mathcal{O}\left(\frac{6}{\mu}\sqrt{\frac{L'^2 d^2\|\boldsymbol{\alpha}\|^2\log\frac{12}{\delta}}{N}}\right) + \mathcal{O}\big(hM\sqrt{Md}\operatorname{diam}(\mathcal{K})\big), \tag{338}$$

with probability at least $1 - \delta$. This completes the first part of the proof of Theorem 8.2.

**Second part (infinite-sample regime).** Recall from (214) that, letting $s \to \infty$,

$$\limsup_{s \to \infty} \Big( \|\mathbf{W}(s) - \overline{\mathbf{W}}(s)\|_F + \|\mathbf{W}^* - \widehat{\mathbf{W}}^s(s)\|_F + \|\mathbf{W}(s) - \widehat{\mathbf{W}}^s(s)\|_F \Big) \lesssim_{\mathbf{M}(h,J)} \mathcal{O}(C_0 + \Delta), \qquad (339)$$

with probability at least $1 - \delta$. Using $\mathbf{W}^*_{\text{ERM}} = \mathbf{W}^*$ and the triangle inequality,

$$\limsup_{s \to \infty} \Big( \|\mathbf{W}(s) - \overline{\mathbf{W}}(s)\|_F + \|\mathbf{W}^*_{\text{SR}} - \widehat{\mathbf{W}}^s(s)\|_F + \|\mathbf{W}(s) - \widehat{\mathbf{W}}^s(s)\|_F \Big) \lesssim_{\mathbf{M}(h,J)} \mathcal{O}\Big(C_0 + \Delta + \|\mathbf{W}^*_{\text{SR}} - \mathbf{W}^*_{\text{ERM}}\|_F\Big),$$
$$(340)$$

with probability at least $1 - \delta$. Since $C_0 + \Delta + \|\mathbf{W}^*_{\text{SR}} - \mathbf{W}^*_{\text{ERM}}\|_F \xrightarrow{P} 0$ as $N \to \infty$ by (337), it follows that

$$\lim_{N \to \infty} \limsup_{s \to \infty} \Big( \|\mathbf{W}(s) - \overline{\mathbf{W}}(s)\|_F + \|\mathbf{W}^*_{\text{SR}} - \widehat{\mathbf{W}}^s(s)\|_F + \|\mathbf{W}(s) - \widehat{\mathbf{W}}^s(s)\|_F \Big) \xrightarrow{P} 0, \qquad (341)$$

where $X_N \xrightarrow{P} 0$ denotes convergence in probability. This completes the proof of Theorem 8.2. ∎

### G.3 Proof of Theorem 8.3

*Proof.* From Lemma G.1, we have

$$C_0 < \mathcal{O}\Bigg( \sqrt{\frac{L'^2 d^2 \|\boldsymbol{\alpha}\|^2 \log \frac{4}{\delta_0}}{N}} \Bigg) \qquad (342)$$

with probability at least $1 - \delta_0$, where

$$\delta_0 = 2 \exp\Bigg( -\frac{4MN(\epsilon')^2}{16(L')^2 M d^2 \|\boldsymbol{\alpha}\|^2 + (\epsilon')^2} + M \log\Bigg(\frac{12L'd\sqrt{M}}{\epsilon'}\Bigg) + d \log\Bigg(\frac{12L'\Gamma_0 d}{\epsilon'}\Bigg) \Bigg)$$
$$+ 2d \exp\Bigg( -\frac{(\epsilon')^2 MN}{4(L'd)^2} \Bigg).$$

Taking $\limsup_{s \to \infty}$ on both sides of (74) from Theorem 6.4, we obtain

$$\limsup_{s \to \infty} \big( f(\widehat{\mathbf{w}}^s(s)) - f^* \big) \leqslant \frac{L \operatorname{diam}(\mathcal{K})}{\mu(2 - Lh)} C_0 + \frac{L^2 h d \sqrt{Md}}{1 - a_1} (\operatorname{diam}(\mathcal{K}))^2 \Bigg( \frac{(\sqrt{M} + 1)^2}{\mu(2 - Lh)} LM(\sqrt{d} + 2) + M \Bigg), \qquad (343)$$

and therefore

$$\limsup_{s \to \infty} \big| f(\widehat{\mathbf{w}}^s(s)) - \mathcal{R}^*_{\text{SR}} \big| \leqslant \frac{L \operatorname{diam}(\mathcal{K})}{\mu(2 - Lh)} C_0 + \mathcal{O}\Bigg( \frac{hL^3 M^{\frac{5}{2}} (d \operatorname{diam}(\mathcal{K}))^2}{\mu} \Bigg) + |f^* - \mathcal{R}^*_{\text{SR}}|. \qquad (344)$$

Next, note that $f^* \equiv f^*_{\text{ERM}} = \frac{1}{MN} \sum_{j=1}^{M} \sum_{n=1}^{N} \ell(\mathbf{w}^*_{\text{ERM}}; \mathbf{z}_{jn})$, while $\mathbf{w}^*_{\text{SR}}$ is deterministic with respect to the probability law $\mathbb{P}$ and satisfies

$$\mathcal{R}(\mathbf{w}^*_{\text{SR}}) = \mathcal{R}^*_{\text{SR}}, \qquad \nabla \mathcal{R}(\mathbf{w}^*_{\text{SR}}) = \mathbf{0}.$$

By the triangle inequality and Assumption 6.1,

$$|f^* - \mathcal{R}^*_{\text{SR}}| \leqslant \Bigg| \frac{1}{MN} \sum_{j=1}^{M} \sum_{n=1}^{N} \ell(\mathbf{w}^*_{\text{SR}}; \mathbf{z}_{jn}) - \mathcal{R}^*_{\text{SR}} \Bigg| + |f^* - f(\mathbf{w}^*_{\text{SR}})|$$

$$\leqslant \Bigg| \frac{1}{MN} \sum_{j=1}^{M} \sum_{n=1}^{N} \ell(\mathbf{w}^*_{\text{SR}}; \mathbf{z}_{jn}) - \mathcal{R}(\mathbf{w}^*_{\text{SR}}) \Bigg| + \frac{1}{2\mu} \|\nabla f(\mathbf{w}^*_{\text{SR}})\|^2$$

$$= \underbrace{\left| \frac{1}{MN} \sum_{j=1}^{M} \sum_{n=1}^{N} \ell(\mathbf{w}_{\text{SR}}^*; \mathbf{z}_{jn}) - \mathcal{R}(\mathbf{w}_{\text{SR}}^*) \right|}_{=:T_1} + \underbrace{\frac{1}{2\mu} \left\| \frac{1}{MN} \sum_{j=1}^{M} \sum_{n=1}^{N} \nabla\ell(\mathbf{w}_{\text{SR}}^*; \mathbf{z}_{jn}) - \nabla\mathcal{R}(\mathbf{w}_{\text{SR}}^*) \right\|^2}_{=:T_2}. \quad (345)$$

From Assumption 8.1, we have the following uniform bounds, as in the proof of Lemma G.1:

$$\max\left\{ \sup_{\mathbf{w}\in\mathcal{K}} \left|\nabla_k \ell(\mathbf{w}; \mathbf{z}_{jn})\right|, \ \sup_{\mathbf{w}\in\mathcal{K}} \left|\ell(\mathbf{w}; \mathbf{z}_{jn})\right| \right\} \leqslant \frac{L'}{2}, \qquad \sup_{\mathbf{w}\in\mathcal{K}} \|\mathbf{w}\| \leqslant \Gamma_0 = \operatorname{diam}(\mathcal{K}), \quad (346)$$

for any coordinate $k$, any node $j$, any i.i.d. realization $\{\mathbf{z}_{jn}\}_{n=1}^{N} \sim \mathbb{P}$, and any $N \geqslant 1$, where

$$L' = \max\left\{ \mathcal{O}(Ld\operatorname{diam}(\mathcal{K})), \ \mathcal{O}(L(\operatorname{diam}(\mathcal{K}))^2) \right\}.$$

Applying Hoeffding's inequality to the term $T_1$ in (345), for any $\epsilon' \in (0,1)$ we obtain

$$\mathbb{P}(T_1 \geqslant \epsilon') \leqslant 2\exp\left( -\frac{2(\epsilon')^2 MN}{(L')^2} \right). \quad (347)$$

Equivalently,

$$T_1 < \sqrt{\log\left(\frac{2}{\delta_1}\right)} \frac{L'}{\sqrt{2MN}} \qquad \text{with probability at least } 1 - \delta_1, \quad (348)$$

where

$$\delta_1 = 2\exp\left( -\frac{2(\epsilon')^2 MN}{(L')^2} \right).$$

Next, using a union bound over coordinates followed by Hoeffding's inequality for the gradient deviation term in $T_2$, we obtain

$$\mathbb{P}\left(\sqrt{2\mu T_2} \geqslant \epsilon'\right) = \mathbb{P}\left( \left\| \frac{1}{MN} \sum_{j=1}^{M} \sum_{n=1}^{N} \nabla\ell(\mathbf{w}_{\text{SR}}^*; \mathbf{z}_{jn}) - \nabla\mathcal{R}(\mathbf{w}_{\text{SR}}^*) \right\| \geqslant \epsilon' \right)$$

$$\leqslant \mathbb{P}\left( \sum_{k=1}^{d} \left| \frac{1}{MN} \sum_{j=1}^{M} \sum_{n=1}^{N} \nabla_k\ell(\mathbf{w}_{\text{SR}}^*; \mathbf{z}_{jn}) - \nabla_k\mathcal{R}(\mathbf{w}_{\text{SR}}^*) \right| \geqslant \epsilon' \right)$$

$$\leqslant \sum_{k=1}^{d} \mathbb{P}\left( \left| \frac{1}{MN} \sum_{j=1}^{M} \sum_{n=1}^{N} \nabla_k\ell(\mathbf{w}_{\text{SR}}^*; \mathbf{z}_{jn}) - \nabla_k\mathcal{R}(\mathbf{w}_{\text{SR}}^*) \right| \geqslant \frac{\epsilon'}{d} \right)$$

$$\leqslant 2d\exp\left( -\frac{2(\epsilon')^2 MN}{(L'd)^2} \right). \quad (349)$$

Hence

$$\sqrt{2\mu T_2} < \sqrt{\log\left(\frac{2d}{\delta_2}\right)} \frac{L'd}{\sqrt{2MN}} \qquad \text{with probability at least } 1 - \delta_2, \quad (350)$$

where

$$\delta_2 = 2d\exp\left( -\frac{2(\epsilon')^2 MN}{(L'd)^2} \right).$$

Now choose the same $\epsilon'$ in (348) and (350). For sufficiently large $N$ (and $d > \epsilon'$), we have $\max\{\delta_1, \delta_2\} < \delta_0$. Applying a union bound over the three events corresponding to the bound on $C_0$, (348), and (350), and defining $\delta := \delta_0 + \delta_1 + \delta_2 < 3\delta_0$, we obtain with probability at least $1 - \delta$ that

$$\frac{L\operatorname{diam}(\mathcal{K})}{\mu(2 - Lh)} C_0 + |f^* - \mathcal{R}_{\text{SR}}^*| \leqslant 3\max\left\{ \mathcal{O}\left( \frac{4L\operatorname{diam}(\mathcal{K})}{\mu(2 - Lh)} \sqrt{\frac{L'^2 d^2 \|\boldsymbol{\alpha}\|^2 \log\frac{4}{\delta_0}}{N}} \right), \right.$$

$$\sqrt{\log\left(\frac{2}{\delta_1}\right)}\frac{L'}{\sqrt{2MN}}, \ \log\left(\frac{2d}{\delta_2}\right)\frac{(L'd)^2}{4MN\mu}\Bigg\}. \tag{351}$$

Since $\sqrt{M} > \mu$ by assumption, the above implies

$$\frac{L\operatorname{diam}(\mathcal{K})}{\mu(2-Lh)}\,C_0 + |f^* - \mathcal{R}_{\mathrm{SR}}^*| \leqslant \mathcal{O}\left(\frac{L\operatorname{diam}(\mathcal{K})}{\mu(2-Lh)}\sqrt{\frac{L'^2d^2\|\boldsymbol{\alpha}\|^2\log\frac{12}{\delta}}{N}}\right). \tag{352}$$

Moreover,

$$\delta = \delta_0 + \delta_1 + \delta_2$$
$$\leqslant 2\exp\left(-\frac{4MN(\epsilon')^2}{16(L')^2Md^2\|\boldsymbol{\alpha}\|^2 + (\epsilon')^2} + M\log\left(\frac{12L'd\sqrt{M}}{\epsilon'}\right) + d\log\left(\frac{12L'\Gamma_0 d}{\epsilon'}\right)\right)$$
$$+ 4d\exp\left(-\frac{(\epsilon')^2MN}{4(L'd)^2}\right) + 2\exp\left(-\frac{2(\epsilon')^2MN}{(L')^2}\right).$$

Finally, substituting (352) into (344) yields

$$\limsup_{s\to\infty}\left|f(\hat{\mathbf{w}}^s(s)) - \mathcal{R}_{\mathrm{SR}}^*\right| \leqslant \mathcal{O}\left(\frac{L\operatorname{diam}(\mathcal{K})}{\mu(2-Lh)}\sqrt{\frac{L'^2d^2\|\boldsymbol{\alpha}\|^2\log\frac{12}{\delta}}{N}}\right) + \mathcal{O}\left(\frac{hL^3M^{\frac{5}{2}}(d\operatorname{diam}(\mathcal{K}))^2}{\mu}\right) \tag{353}$$

with probability at least $1 - \delta$. This completes the proof of Theorem 8.3. ∎

Observe that, in Theorem 8.3 for PŁ functions, unlike Theorem 8.2 for strongly convex functions, we do not provide statistical rates for the consensus error terms $\xi_k^1(s)$ and $\xi_k^5(s)$. To understand this distinction, note first that after any sufficiently large $S$, the consensus errors $\xi_k^1(S)$ and $\xi_k^5(S)$ in the ERM problem (3) are upper bounded by an $\mathcal{O}(h\Delta)$ term irrespective of the function class (see Theorems 5.5 and 6.4), where

$$\Delta = \sum_{j=1}^{M}\|\mathbf{w}_j^* - \mathbf{w}^*\| \leqslant M\operatorname{diam}(\mathcal{K}).$$

In the strongly convex case, we can further upper bound the distance $\|\mathbf{w}_j^* - \mathbf{w}^*\|$ by the corresponding gradient difference, namely

$$\|\mathbf{w}_j^* - \mathbf{w}^*\| \leqslant \frac{1}{\mu}\|\nabla f(\mathbf{w}_j^*) - \nabla f(\mathbf{w}^*)\|,$$

which allows us to derive statistical bounds for $\Delta$ in terms of empirical gradient deviations. By contrast, in the PŁ setting, although the PŁ inequality controls function suboptimality, it does not directly control distances between minimizers. Since PŁ functions may admit multiple minima, we do not derive statistical convergence rates for the consensus error terms $\xi_k^1(s)$ and $\xi_k^5(s)$ in this case.

### G.4    On the non-vacuous nature of Assumption 8.1

We provide a restricted construction showing that Assumption 8.1 is not vacuous under additional structural conditions. We follow the setup of Appendix E.1 with mild modifications to incorporate random data samples. As in Appendix E.1, this construction should not be read as a verification of the assumption for arbitrary time-varying MITM attack sequences.

**Setup.**    For simplicity, assume the model dimension is $d = 1$. Let the data samples satisfy $\mathbf{z}_{jn} \overset{\text{i.i.d.}}{\sim} \mathbb{P}$ and $\operatorname{supp}(\mathbb{P}) \subseteq \mathcal{U}$, where $\mathcal{U}$ is a compact set (e.g., a closed ball) independent of $N$. Assume the loss $\ell(\mathbf{w}; \mathbf{z})$ is nonnegative, jointly continuous in $(\mathbf{w}, \mathbf{z})$, and uniformly coercive in $\mathbf{w}$ over $\mathcal{U}$, i.e.,

$$\lim_{\|\mathbf{w}\|\to\infty}\min_{\mathbf{z}\in\mathcal{U}}\ell(\mathbf{w}; \mathbf{z}) = \infty.$$

Assume further that the network and adversary satisfy the same structural conditions as in Appendix E.1: the mixing matrices are symmetric, simultaneously diagonalizable, and the corresponding products are monotone in the Loewner order. Concretely, letting

$$\mathbf{Q}(s; N) := \prod_{r=J\lfloor t/J \rfloor}^{J\lfloor t/J \rfloor + J - 2} \mathbf{Y}(r; N), \tag{354}$$

(with the subscript $k$ omitted as in Appendix E.1), assume

$$\mathbf{Q}(0; N) \preccurlyeq \mathbf{Q}(1; N) \preccurlyeq \cdots \preccurlyeq \mathbf{Q}(s; N) \preccurlyeq \cdots \preccurlyeq \mathbf{Q}(\infty; N). \tag{355}$$

We emphasize that $\mathbf{Q}(s; N)$ may depend on $N$ and on the realized sample draw $\{\mathbf{z}_{jn}\}$, but we suppress this dependence in the notation.

**A realization-dependent Lyapunov function.** For $\mathbf{W} = [\mathbf{w}_1, \ldots, \mathbf{w}_M]^\top$ and the empirical objective

$$F(\mathbf{W}; N) := \frac{1}{N} \sum_{n=1}^{N} \sum_{j=1}^{M} \ell(\mathbf{w}_j; \mathbf{z}_{jn}),$$

define, for each $s \geqslant 0$ and each $N$,

$$\mathcal{L}(\mathbf{W}; s, N) := F(\mathbf{W}; N) + \frac{1}{2h} \|\mathbf{W}\|^2_{\mathbf{I} - \mathbf{Q}(s;N)}, \qquad \|\mathbf{W}\|^2_{\mathbf{I} - \mathbf{Q}(s;N)} := \langle \mathbf{W}, (\mathbf{I} - \mathbf{Q}(s; N))\mathbf{W} \rangle \geqslant 0. \tag{356}$$

As in Appendix E.1, $\mathcal{L}(\cdot; s, N)$ is coercive in $\mathbf{W}$, and for a sufficiently small stepsize $h$ (e.g., $h < 1/(LM)$) the RESIST updates guarantee that $\mathcal{L}$ is monotonically non-increasing along the iterates:

$$\mathcal{L}(\mathbf{W}(s); s, N) \leqslant \mathcal{L}(\mathbf{W}(0); 0, N), \qquad \forall s \geqslant 0. \tag{357}$$

**A deterministic bound on the initial Lyapunov value.** Assumption 8.1 requires the initialization to be uniformly bounded across nodes, i.e.,

$$\max_{1 \leqslant j \leqslant M} \|\mathbf{w}_j(0)\| \leqslant B_0$$

for some deterministic constant $B_0$ independent of $N$ and the sample realization. By continuity of $\ell(\mathbf{w}; \mathbf{z})$ in $\mathbf{z}$ and compactness of $\mathcal{U}$, the quantity

$$\bar{\ell}_0 := \max_{1 \leqslant j \leqslant M} \sup_{\mathbf{z} \in \mathcal{U}} \ell(\mathbf{w}_j(0); \mathbf{z})$$

is finite and deterministic. Moreover, since $\mathbf{I} - \mathbf{Q}(0; N) \preccurlyeq \mathbf{I}$, we have

$$\|\mathbf{W}(0)\|^2_{\mathbf{I} - \mathbf{Q}(0;N)} \leqslant \|\mathbf{W}(0)\|^2 \leqslant MB_0^2.$$

Therefore

$$\mathcal{L}(\mathbf{W}(0); 0, N) = \frac{1}{N} \sum_{n=1}^{N} \sum_{j=1}^{M} \ell(\mathbf{w}_j(0); \mathbf{z}_{jn}) + \frac{1}{2h} \|\mathbf{W}(0)\|^2_{\mathbf{I} - \mathbf{Q}(0;N)}$$

$$\leqslant M\bar{\ell}_0 + \frac{MB_0^2}{2h} =: C_{\text{init}} < \infty, \tag{358}$$

where $C_{\text{init}}$ is deterministic and independent of $N$ and the sample realization.

**Realization-dependent compact sets and a deterministic envelope.** Fix any $N$ and any realized datasets $\{\mathcal{Z}_j\}_{j=1}^M$ (equivalently, a realization $\{\mathbf{z}_{jn}\}$). Define the realization-dependent set

$$\mathcal{K}_N(\{\mathcal{Z}_j\}_{j=1}^M) := \left\{ \mathbf{w} \in \mathbb{R} : \min_{\mathbf{z} \in \cup_{j=1}^M \mathcal{Z}_j} \ell(\mathbf{w}; \mathbf{z}) \leqslant C_{\text{init}} \right\}. \tag{359}$$

Since $\bigcup_{j=1}^M \mathcal{Z}_j$ is finite and $\ell(\cdot; \mathbf{z})$ is coercive for each $\mathbf{z} \in \mathcal{U}$, the set $\mathcal{K}_N(\{\mathcal{Z}_j\}_{j=1}^M)$ is compact. From (357) and (358), for all $s \geqslant 0$,

$$\mathcal{L}(\mathbf{W}(s); s, N) \leqslant C_{\text{init}}.$$

Using nonnegativity of the quadratic penalty term yields

$$F(\mathbf{W}(s); N) = \frac{1}{N} \sum_{n=1}^N \sum_{j=1}^M \ell(\mathbf{w}_j(s); \mathbf{z}_{jn}) \leqslant C_{\text{init}}. \tag{360}$$

In particular, for each node $j$,

$$\frac{1}{N} \sum_{n=1}^N \ell(\mathbf{w}_j(s); \mathbf{z}_{jn}) \leqslant C_{\text{init}}.$$

Hence there exists at least one sample index $n_j$ such that

$$\ell(\mathbf{w}_j(s); \mathbf{z}_{jn_j}) \leqslant C_{\text{init}}.$$

Since $\mathbf{z}_{jn_j} \in \mathcal{Z}_j$, this implies

$$\min_{\mathbf{z} \in \bigcup_{j=1}^M \mathcal{Z}_j} \ell(\mathbf{w}_j(s); \mathbf{z}) \leqslant C_{\text{init}},$$

and therefore

$$\mathbf{w}_j(s) \in \mathcal{K}_N(\{\mathcal{Z}_j\}_{j=1}^M), \qquad \forall j, \ \forall s \geqslant 0.$$

Finally define the deterministic compact set

$$\mathcal{K} := \left\{ \mathbf{w} \in \mathbb{R} : \min_{\mathbf{z} \in \mathcal{U}} \ell(\mathbf{w}; \mathbf{z}) \leqslant C_{\text{init}} \right\}. \tag{361}$$

Since $\bigcup_{j=1}^M \mathcal{Z}_j \subseteq \mathcal{U}$ almost surely, we have

$$\mathcal{K}_N(\{\mathcal{Z}_j\}_{j=1}^M) \subseteq \mathcal{K}.$$

Thus the RESIST iterates satisfy $\mathbf{w}_j(s) \in \mathcal{K}_N(\{\mathcal{Z}_j\}_{j=1}^M) \subseteq \mathcal{K}$ for all $j$ and $s$, establishing Assumption 8.1.

