# OpenReview forum: "RESIST: Resilient Decentralized Learning Using Consensus Gradient Descent"
_TMLR — Under review for TMLR_

### Review · Reviewer_PVhS · 2026-03-31

**Summary Of Contributions:**

This paper introduces RESIST, a decentralized machine learning algorithm designed to solve empirical risk minimization (ERM) problems while defending against dynamic man-in-the-middle (MITM) attacks on communication networks. To mitigate adversarial interference, the algorithm employs a multistep consensus gradient descent framework combined with robust statistics-based screening methods. The primary contribution of this work is that RESIST achieves both algorithmic convergence and statistical consistency across multiple settings. Furthermore, it overcomes the limitations of prior robust decentralized methods by guaranteeing linear (geometric) convergence rates to the exact global minimum for strongly convex problems.

**Audience:**

Yes

**Audience Explanation:**

The paper addresses Empirical Risk Minimization (ERM), which is fundamental in learning theory. Furthermore, ensuring robustness and safety against adversarial attacks in distributed networks is currently a highly active and critical area of interest for the machine learning community.

**Broader Impact Concerns:**

No specific ethical concerns.

**Claims And Evidence:**

Yes

**Claims Explanation:**

Yes, the paper provides comprehensive and rigorous proofs, including both algorithmic convergence rate and statistical rate.

**Requested Changes:**

**(Heterogeneity).** While the paper’s algorithmic convergence analysis allows heterogeneous local objectives, the statistical learning-rate results still rely on i.i.d. data across nodes, and the manuscript explicitly leaves non-i.i.d. statistical theory for future work. Since decentralized learning is often motivated by heterogeneous data, it would strengthen the paper to at least discuss more clearly what is expected to carry over to non-i.i.d. settings, what breaks in the current proof, and how the new non-i.i.d. experiments should be interpreted relative to the theory.

**(Scaling).** The paper argues that RESIST can be more computationally efficient than BRIDGE by using multiple consensus rounds per gradient step, and it highlights advantages over methods like ByRDiE for high-dimensional models. However, there is still little explicit discussion of how the method scales with model dimension: per-round communication costs, the cost of coordinate-wise screening, and whether communication or memory becomes the bottleneck for large neural networks. Even if new experiments are out of scope, a short discussion of these scaling tradeoffs would make the practical applicability much clearer.

---

### Review · Reviewer_LFj8 · 2026-05-01

**Summary Of Contributions:**

This paper develops a decentralized optimization algorithm that is robust to man-in-the-middle attacks on communication links. The technique of coordinate-wise trimmed mean appears to be interesting. Theoretically, it establishes various convergence guarantees, covering both algorithmic and statistical convergence for strongly convex, Polyak–Łojasiewicz, and general smooth nonconvex loss functions. Empirically, it presents extensive numerical experiments to validate its practical performance.

**Audience:**

Yes

**Audience Explanation:**

This application scenario of robust decentralized learning under man-in-the-middle attacks appears to be useful in practice.

**Claims And Evidence:**

Yes

**Claims Explanation:**

The description of the setup for decentralized learning under a man-in-the-middle attack model appears to be valid. The analysis provides convergence guarantees, covering both algorithmic and statistical convergence for strongly convex, Polyak–Łojasiewicz, and general smooth nonconvex loss functions.

**Requested Changes:**

1. This paper provides solid theoretical analysis and a thorough discussion of the algorithm. However, I believe there is significant room for improvement in terms of presentation clarity and reader-friendliness. The paper would benefit from a careful reconsideration of how the material is presented.

In particular, for a paper of this length, insufficient attention to presentation can greatly affect its quality. For example, the contributions and abstract sections should be simplified, as they are intended to convey key information concisely and are often the first parts readers consult. In addition, I recommend separating the presentation of mathematical content from the discussion of related work (especially Section 2), rather than interweaving them. The literature review could be placed in a dedicated section. Presenting the mathematical arguments in a clear and concise manner would further enhance the overall quality of the paper.



2. In the dynamic MITM model, the adversary may alter the set of compromised links Eb(t) at each iteration. Does the theoretical analysis impose any implicit assumptions on the adversarial sequence, such as limits on how quickly it can vary, or is it completely arbitrary, constrained only by the cardinality condition |Njb(t)| < b.

If the adversarial sequence changes too frequently, would it cause problems for convergence?​

3. The analysis of the paper relies on Assumption 4.12, which imposes that the iterate sequences {wj(t)} at all nodes remain uniformly bounded. In an adversarial setting, an attacker can inject arbitrary values through compromised communication links, creating a theoretical risk of causing the iterates to diverge. While this assumption is common in non-adversarial distributed optimization, its justification requires more thorough argumentation or safeguarding mechanisms within the considered MITM attack model, where information can be arbitrarily altered.

---

### Review · Reviewer_Z1us · 2026-05-17

**Summary Of Contributions:**

This paper studies decentralized empirical risk minimization (ERM) under the Man-in-the-Middle (MITM) attack model, in which an adversary can compromise a bounded number of communication links per iteration and may dynamically shift the attacked subset across iterations. To address this threat, the authors propose RESIST (Resilient dEcentralized learning using conSensus gradIent deScenT), a first-order decentralized algorithm that interleaves (i) $J-1$ rounds of coordinate-wise trimmed-mean (CWTM) consensus updates with (ii) one local gradient step. The CWTM screening guarantees that, in each coordinate, the local update is a convex combination of uncompromised values.

Contributions can be summarized as follows:

* The authors propose a dynamic MITM attack model for decentralized learning, distinct from static MITM models in the distributed literature and from node-level Byzantine attacks. They further argue  that the classical decentralized Byzantine model can be embedded as a static special case of MITM.
* This work introduces many theoretical results, inclduing geometric convergence in the $s$-time scale (algorithmic iterations) and $t$-time scale (slot index) for (a) $\mu$-strongly convex losses, (b) loss functions whose average satisfies the PL inequality, and sublinear convergence for (c) smooth nonconvex losses, both with diminishing and constant stepsizes.
* They also conduct fairly comprehensive empirical study on MNIST ($\ell_2$-regularized linear classifier, strongly convex) and CIFAR-10 (CNN, smooth nonconvex).

**Audience:**

Yes

**Audience Explanation:**

The paper addresses a question that is genuinely of interest to the decentralized-learning and adversarial-ML communities served by TMLR. But, I still want to say that the paper is too long to verify the correctness of proofs, and densely technical where its likely audience is researchers already working on related topics.

**Broader Impact Concerns:**

This paper is mainly theoretical; thus it does not have broader impact concerns.

**Claims And Evidence:**

Yes

**Claims Explanation:**

The proofs are generally rigorous; the assumptions, when invoked, are stated correctly; and the experimental evidence is consistent with the theory. However, several claims would benefit from more careful qualification, and a few statements are stronger than what the proofs actually establish. See **Requested Changes** below.

**Requested Changes:**

I have the following requested changes and minor suggestions for this paper.

1. **Recalibrate the framing of contributions relative to BRIDGE and REDGRAF.** The introduction and Sec. 1.2 claim that RESIST achieves exact convergence to the global minimizer $w^*$, in contrast to REDGRAF. The actual theorems only guarantee neighborhood convergence under heterogeneity for finite $N$; exact convergence holds only in the homogeneous case or in the $N \to \infty$ limit. The same applies to the comparison with BRIDGE: when $J = 2$, RESIST coincides with BRIDGE up to the stepsize choice.
2. **Clarify what the dynamic part of the MITM model buys.** The reduction in Sec. 7 only uses a static MITM construction to embed Byzantine attacks. If dynamic MITM is strictly more powerful than static MITM, the paper should exhibit a concrete (even synthetic) attack pattern for which a static MITM model gives weaker convergence guarantees. Otherwise the dynamic adjective is decorative.
3. **Tighten the discussion of Assumption 4.12 and Assumption 8.1.** These assumptions are invoked throughout but are very strong: Appendix E.1 and G.4 verify them only for adversaries whose induced mixing matrices are symmetric, simultaneously diagonalizable, and monotone in the Loewner order. This is not the dynamic adversary that the rest of the paper considers. The reader is likely to be misled into thinking these assumptions are mild.
4. **Provide a quantitative comparison with REDGRAF or BRIDGE on the same task.** The paper's most striking algorithmic claim is geometric convergence to a meaningfully smaller neighborhood than REDGRAF achieves, but no head-to-head numerical comparison is provided. The redundancy-requirement objection can be sidestepped by choosing graphs and $b$ values that satisfy both algorithms' connectivity requirements.
5. **Move some assumption discussion and the residual-ball scaling into the main text.** Corollary 5.10, i.e., Eq. (59), hides poor $d$- and $M$-dependence inside a controlled statement. Therefore, the residual ball is not small in general for high-dimensional problems.

Below, there are some minor suggestions that would strengthen the work.

* Definition 4.3 introduces six error sequences $(\xi^1_k,\, \ldots,\, \xi^6_{w^*})$ at once, before the reader has any intuition for which ones drive the analysis. The proof then jumps between them via Lemmas 4.6 to 4.11. This is hard to follow on a first reading.
* The proof of Lemma 5.9 uses $\|\nabla (f - g)\|\_{L^{\infty}(\Omega)}$ where $\Omega$ must contain both $y_f$ and $y_g$; this is not quite the same $\Omega$ (any comparator) as used later in Lemma 5.12. Give some discussions.